# Improving the Robustness-Utility Trade-off in Decentralized Learning over Sparse Networks

Yangnan Li [1]   Xuanyu Cao [2]   Shenghui Song [1]

## Abstract

Resilience against Byzantine attackers and faster convergence on sparse networks are critical for decentralized optimization, yet existing methods fail to achieve both simultaneously. Existing DSGD-based Byzantine-resilient methods suffer from high transient complexity of $\mathcal{O}((1-\lambda)^{-6})$, where $1 - \lambda$ denotes the spectral gap of the network. While bias-correction methods such as Exact Diffusion can improve topology dependence, directly combining them with robust aggregators can lead to error accumulation. To address this issue, we introduce the *scaled dual ascent* (SDA) within the augmented Lagrangian framework for decentralized optimization, which mitigates error accumulation by scaling the dual update steps. Based on this, we propose `BRED`, which integrates Byzantine-robust Exact Diffusion with the SDA framework. We prove that `BRED` attains linear speedup, and achieves transient complexity of $\mathcal{O}((1-\lambda)^{-2})$ when the Byzantine fraction $\delta$ is small. We further propose the momentum variant `BRED-M`, which reduces the Byzantine-affected transient complexity from $\mathcal{O}(\delta^2(1-\lambda)^{-6})$ to $\mathcal{O}(\delta^2(1-\lambda)^{-4})$. Empirical results on benchmark datasets demonstrate the efficacy of the proposed methods across diverse network topologies.

## 1. Introduction

In machine learning applications, training data is often distributed across multiple sources, such as individual users or separate organizations. The privacy-sensitive nature of the data, particularly in domains like healthcare and finance, often renders centralized data collection impractical or undesirable (Han et al., 2021; Long et al., 2020). This has

[1] The Hong Kong University of Science and Technology, Hong Kong SAR, China  [2] Washington State University, Pullman, WA, USA . Correspondence to: Shenghui Song <eeshsong@ust.hk>.

*Proceedings of the 43rd International Conference on Machine Learning*, Seoul, South Korea. PMLR 306, 2026. Copyright 2026 by the author(s).

led to the emergence of federated and decentralized learning (McMahan et al., 2017; Lian et al., 2017; Alghunaim & Yuan, 2022; Koloskova et al., 2020), in which training is performed directly on the data-holding entities without sharing raw data. Compared to federated learning, decentralized learning further eliminates the need for a central server, allowing clients to exchange model updates directly with their neighbors over a communication network (Che et al., 2022; Shi et al., 2015; Tang et al., 2018). The objective of decentralized learning can be formulated as minimizing the global loss across $n$ clients:

$$\min_{x \in \mathbb{R}^d} f(x), \quad f(x) := \frac{1}{n} \sum_{i=1}^{n} f_i(x),$$

where $x \in \mathbb{R}^d$ denotes the model parameters. Each client $i$ has access only to its local data $\mathcal{D}_i$ and objective $f_i(x) = \mathbb{E}_{\varphi_i \sim \mathcal{D}_i}[F_i(x; \varphi_i)]$, where the data distributions $\{\mathcal{D}_i\}_{i=1}^{n}$ are heterogeneous across clients.

Despite its effectiveness, decentralized learning is vulnerable to faulty or adversarial participants (Armstrong, 2003; Zhang et al., 2020). Such misbehaving workers may stem from hardware malfunctions, data poisoning, or malicious actors controlling part of the network. These misbehaving workers are referred to as *Byzantine workers* (Xie et al., 2020; Baruch et al., 2019), which may constitute a fraction $\delta$ of all participants. Given the growing deployment of decentralized learning in critical applications, such as autonomous driving, healthcare systems, and financial fraud detection (Chen et al., 2021; Kasyap et al., 2024), Byzantine-resilient learning has received significant attention (Blanchard et al., 2017; Yin et al., 2018).

The general idea for mitigating the impact of Byzantine attackers is to replace the standard averaging with robust aggregation rules (Gorbunov et al., 2022; El-Mhamdi et al., 2021; Farhadkhani et al., 2023). Despite their effectiveness, most prior works only consider the simplified decentralized learning setting with a fully connected communication graph, where each client communicates directly with all other participants (El-Mhamdi et al., 2021; Farhadkhani et al., 2023). Recent studies have explored sparse communication topologies (Wu et al., 2023), and typically rely on Decentralized Stochastic Gradient Descent (DSGD) com-

*Table 1.* Comparison of convergence rates and transient times for decentralized stochastic optimization. (*) For the convergence rates, we omit the asymptotic error induced by Byzantine attackers. The quantity $\lambda = \rho(\mathbf{W} - \frac{1}{n}\mathbf{1}\mathbf{1}^\top)$ denotes the second-largest eigenvalue magnitude of the mixing matrix $\mathbf{W}$, with $p := 1 - \lambda$ representing the spectral gap. $\kappa$ is the robustness coefficient, chosen as $\kappa = \mathcal{O}(\delta)$. $n_h := (1 - \delta)n$ is the number of honest workers, and $\sigma^2, \zeta^2$ are the constants defined in Assumptions 3 and 4.

| Method | Work | Byzantine? | Convergence Rate[(*)] | Transient Time |
|--------|------|-----------|----------------------|----------------|
| DSGD | (Koloskova et al., 2020) | *NO* | $\mathcal{O}\left(\frac{1}{\sqrt{nK}} + \frac{\lambda^{2/3}\sigma^2}{p^{1/3}K^{2/3}} + \frac{\lambda^{2/3}\zeta^{2/3}}{p^{2/3}K^{2/3}}\right)$ | $\mathcal{O}\left(\frac{n^3}{(1-\lambda)^4}\right)$ |
| ED/$D^2$ | (Alghunaim & Yuan, 2022) | *NO* | $\mathcal{O}\left(\frac{1}{\sqrt{nK}} + \frac{n\lambda^2\sigma^2}{pK} + \frac{n\lambda^2\sigma^2}{p^3K^2} + \frac{n\lambda^2\zeta^2}{p^2K^2}\right)$ | $\mathcal{O}\left(\frac{n^3}{(1-\lambda)^2}\right)$ |
| ATC-GT | (Alghunaim & Yuan, 2022) | *NO* | $\mathcal{O}\left(\frac{1}{\sqrt{nK}} + \frac{n\lambda^4\sigma^2}{pK} + \frac{n\lambda^4\sigma^2}{p^4K^2} + \frac{n\lambda^4\zeta^2}{p^3K^2}\right)$ | $\mathcal{O}\left(\max\left\{\frac{n^3}{(1-\lambda)^2}, \frac{n}{(1-\lambda)^{8/3}}\right\}\right)$ |
| BR-DSGD | (Wu et al., 2023) | *YES* | $\mathcal{O}\left(\frac{1}{\sqrt{n_hK}} + \frac{n_h\left(\lambda+8\delta\sqrt{n_h}\right)(\sigma^2+\zeta^2)}{(p-8\delta\sqrt{n_h})^3K}\right)$ | $\mathcal{O}\left(\frac{n_h^3}{(1-\lambda-8\delta\sqrt{n_h})^6}\right)$ |
| BRED | **This work** | *YES* | $\mathcal{O}\left(\frac{1}{\sqrt{n_hK}} + \frac{n_h\lambda^2\sigma^2}{pqK} + \frac{n_h\lambda^2\sigma^2}{p^3qK^2} + \frac{n_h\lambda^2\zeta^2}{p^2qK^2} + \frac{\kappa n_h}{p^3qK}\right)$ | $\mathcal{O}\left(\max\left\{\frac{n_h^3}{(1-\lambda)^2}, \frac{\delta^2 n_h^3}{(1-\lambda)^6}\right\}\right)$ |
| BRED-M | **This work** | *YES* | $\mathcal{O}\left(\frac{1}{\sqrt{n_hK}} + \frac{n_h^{3/2}\lambda^2\sigma^2}{pqK^{3/2}} + \frac{n_h\lambda^2\zeta^2}{p^3qK^2} + \frac{n_h^2\lambda^2\zeta^2}{p^2qK^3} + \frac{\kappa n_h}{p^2qK}\right)$ | $\mathcal{O}\left(\max\left\{\frac{n_h^2}{1-\lambda}, \frac{n_h}{(1-\lambda)^2}, \frac{\delta^2 n_h^3}{(1-\lambda)^4}\right\}\right)$ |

bined with robust aggregators. However, such a DSGD-based method is highly sensitive to network sparsity, suffering from a transient time of $\mathcal{O}((1 - \lambda)^{-6})$. While bias-correction methods, such as EXTRA (Shi et al., 2015), Exact Diffusion (ED/$D^2$) (Yuan et al., 2018a; Tang et al., 2018), and gradient tracking (GT) (Di Lorenzo & Scutari, 2016; Nedic et al., 2017), have demonstrated superior convergence on sparse networks, it remains uncertain whether their performance superiority persists when combined with robust aggregation in the presence of Byzantine attackers.

**Key results.** In this work, we show that directly combining bias-correction algorithms with existing Byzantine-resilient robust aggregators is non-trivial. As shown in (Alghunaim & Yuan, 2022), bias-correction methods can be formulated within the augmented Lagrangian framework, where dual variables are introduced to track and correct the heterogeneity bias. Under Byzantine attacks, the integration of historical errors by the dual variables leads to the asymptotic error term that accumulates with the number of iterations $K$. To address this fundamental challenge, we propose a *scaled dual ascent* (SDA) based approach that introduces a fractional scaling factor to the dual update. This scaling eliminates the dependence of the asymptotic error on $K$. Based on this framework, we develop *Byzantine-Resilient Exact Diffusion* (BRED) and its momentum variant BRED-M.

Our main contributions are summarized as follows.

- **Augmented Lagrangian method with scaled dual ascent.** We demonstrate that directly combining existing bias-correction methods with robust aggregators causes the Byzantine-induced error to accumulate with the number of iterations $K$. To address this, we introduce SDA within the augmented Lagrangian framework for decentralized optimization with bias-correction, which bounds the dual variables and eliminates error accumulation.

- **Linear speedup and transient iteration complexity.** Based on the SDA framework, we propose BRED. As shown in Table 1, BRED attains linear speedup with respect to the number of *honest clients* $n_h$. Furthermore, our analysis demonstrates that the proposed method achieves superior transient iteration complexity compared to DSGD-based approaches. In the absence of attackers (i.e., $\delta = 0$), our results recover the transient complexity of Exact Diffusion $\mathcal{O}(n^3/(1 - \lambda)^2)$.

- **Achievable optimal Byzantine resilience.** The momentum variant, BRED-M, further reduces the transient time of BRED. Specifically, it reduces the Byzantine-related transient time from $\mathcal{O}(\delta^2 n_h^3/(1 - \lambda)^6)$ to $\mathcal{O}(\delta^2 n_h^3/(1 - \lambda)^4)$. Moreover, the asymptotic error of BRED-M matches the lower bound established in (Karimireddy et al., 2020), demonstrating it achieves optimal Byzantine resilience.

- **Experimental validation.** Extensive experiments across diverse attack scenarios and network topologies demonstrate the superior performance of the proposed algorithms over baseline methods.

### 1.1. Related Work

**Byzantine-resilient decentralized learning.** Byzantine-resilient decentralized optimization has garnered significant research interest (Peng et al., 2021; He et al., 2022; Wu et al., 2023; Farhadkhani et al., 2023; El-Mhamdi et al., 2021). To mitigate the impact of Byzantine attacks, various robust aggregation mechanisms have been proposed. For instance, (Peng et al., 2021) extended Robust Stochastic Aggregation (RSA) to decentralized settings, while (He et al., 2022) adapted Centered Clipping (CC) to develop its decentralized variant, Self-Centered Clipping (SCC). More recently, (Wu et al., 2023) introduced Iterative Outlier Scoring (IOS), a filtering-based aggregation scheme with provable convergence guarantees. From a theoretical perspective,

Karimireddy et al. (2020) established a fundamental lower bound on the asymptotic error of $\Omega(\delta\zeta^2)$, where $\delta$ denotes the Byzantine fraction and $\zeta^2$ quantifies data heterogeneity. Subsequently, (Farhadkhani et al., 2023) showed that incorporating Polyak's momentum achieves the asymptotic error that matches the lower bound of $\Omega(\delta\zeta^2)$. Despite these advances, most existing approaches assume fully connected communication topologies, leaving Byzantine-resilient optimization over sparse networks largely unexplored.

**Decentralized optimization with bias-correction.** Numerous studies have explored decentralized stochastic optimization methods for non-convex loss functions (Alghunaim & Yuan, 2022; Lian et al., 2017; Assran et al., 2019; Bianchi & Jakubowicz, 2012; Swenson et al., 2022). Among these, DSGD remains the most extensively studied approach (Lian et al., 2017; Assran et al., 2019; Bianchi & Jakubowicz, 2012). However, DSGD is highly sensitive to network topology, exhibiting a transient time of $\mathcal{O}(n^3/(1-\lambda)^4)$ (Yuan et al., 2020; Koloskova et al., 2020), where $1-\lambda \in (0,1)$ denotes the spectral gap that measures network connectivity. To mitigate this dependence, significant efforts have focused on correcting the bias induced by data heterogeneity. Prominent bias-correction techniques include EXTRA (Shi et al., 2015), Exact Diffusion (ED/$D^2$) (Yuan et al., 2018a; Li et al., 2019; Tang et al., 2018), and gradient tracking (GT) (Xu et al., 2015; Di Lorenzo & Scutari, 2016; Nedic et al., 2017), all of which achieve an improved transient time over DSGD (Alghunaim & Yuan, 2022). Despite these advances, the presence of Byzantine attackers introduces critical challenges that can significantly impede convergence, and the impact of network topology under such adversarial conditions remains largely unexplored.

**Notations.** Throughout this paper, boldface uppercase letters (e.g., $\mathbf{W}$) denote matrices. Non-bold letters (e.g., $x_i \in \mathbb{R}^d$) represent local vectors, where $\bar{x} = \frac{1}{n_h}\sum_{i\in\mathcal{H}} x_i$ denotes the average of honest models with $n_h := (1-\delta)n$ being the number of honest workers. Boldface lowercase letters (e.g., $\mathbf{x} \in \mathbb{R}^{dn_h}$) denote the corresponding stacked vectors across all $n_h$ honest clients. We use $\|\cdot\|$ to denote the $\ell_2$ norm operator and $\mathbb{E}[\cdot]$ for expectation.

## 2. Preliminaries

**Problem statement.** We consider a decentralized system comprising $n$ clients, indexed by $[n] = \{1,\ldots,n\}$, where at most $\delta < 1/2$ fraction of them are Byzantine attackers. An attacker can access honest gradients and transmit arbitrary, well-crafted updates to their neighbors. The goal of the honest clients is to collaboratively minimize the global objective function while reaching consensus on the model parameters. Formally, this is formulated as the following constrained

optimization problem over the honest set $\mathcal{H} \subseteq [n]$:

$$\min_{\mathbf{x}\in\mathbb{R}^{dn_h}} f_{\mathcal{H}}(\mathbf{x}) = \frac{1}{n_h}\sum_{i\in\mathcal{H}} f_i(x_i) = \frac{1}{n_h}\sum_{i\in\mathcal{H}}\mathbb{E}\left[F_i(x_i;\varphi_i)\right],$$
$$\text{s.t. } x_i = x_j, \quad \forall i,j \in \mathcal{H}, \tag{1}$$

where $x_i \in \mathbb{R}^d$ denotes the local model parameters of client $i$, and $\mathbf{x} := \mathrm{col}\{x_i\}_{i\in\mathcal{H}}$ represents the stacked vector of parameters across all $n_h$ honest clients.

**Robust decentralized learning.** The goal of robust decentralized learning is to design an algorithm that can tolerate up to a $\delta$-fraction of adversarial workers with unknown identities (Yin et al., 2018; Allouah et al., 2023). Adversarial workers does not follow the prescribed algorithm and can send arbitrary information to their neighbors (Baruch et al., 2019; Xie et al., 2020). The objective of honest workers is to minimize the honest global loss and achieve an $\varepsilon$-stationary point. We introduce the definition of $(\delta,\varepsilon)$-Byzantine-resilient (Liu et al., 2021) as follows.

**Definition 1** ($(\delta,\varepsilon)$-Byzantine-resilient). *A decentralized algorithm is $(\delta,\varepsilon)$-Byzantine-resilient if, despite the presence of a $\delta$-fraction of Byzantine attackers, it outputs $\hat{x}$ with*

$$\mathbb{E}\left[\|\nabla f_{\mathcal{H}}(\hat{x})\|^2\right] \leq \varepsilon, \tag{2}$$

*where $f_{\mathcal{H}}(x) := \frac{1}{n_h}\sum_{i\in\mathcal{H}} f_i(x)$ denotes the global honest loss.*

Then, we introduce the topology for decentralized networks, following prior work (Alghunaim & Yuan, 2022; Koloskova et al., 2020). In particular, we represent the peer-to-peer connectivity among the $n_h$ workers by an undirected graph and encode the resulting network topology through a matrix.

**Assumption 1** (communication matrix). *Let $W = [w_{ij}] \in [0,1]^{n_h \times n_h}$ be the matrix encoding the network topology. We assume $W$ is symmetric and doubly stochastic:*

$$W = W^\top, \qquad W\mathbf{1} = \mathbf{1}, \qquad \mathbf{1}^\top W = \mathbf{1}^\top, \tag{3}$$

*and respect the graph sparsity ($w_{ij} = 0$ if nodes $i$ and $j$ are not connected) with positive diagonal entries ($w_{ii} > 0$). We define the extended communication matrix by $\mathbf{W} := W \otimes \mathbf{I}_d \in \mathbb{R}^{dn_h \times dn_h}$, where $\otimes$ denotes the Kronecker product.*

Under Assumption 1, the connectivity of the communication graph ensures that $W$ has a unique eigenvalue of 1, with remaining eigenvalues $\{\lambda_i\}_{i=2}^{n_h}$ satisfying $|\lambda_i| < 1$ (Sayed et al., 2014). We define $\lambda := \max_{i\geq 2}|\lambda_i|$ as the second-largest eigenvalue magnitude and denote the minimum nonzero eigenvalue by $\underline{\lambda}$. The spectral gap $1 - \lambda \in (0,1]$ quantifies network connectivity: $1 - \lambda = 1$ for fully connected graphs, and $1 - \lambda = \Theta(1/n_h^2)$ for ring topologies.

The key to achieving Byzantine resilience lies in replacing the averaging operation in classic decentralized learning

with a robust aggregator, such as coordinate-wise median (CWMed) (Yin et al., 2018), Krum (Blanchard et al., 2017), and Iterative Outlier Scissor (IOS) (Wu et al., 2023). To evaluate the effectiveness of these robust aggregators, we adopt the $(\delta, \kappa)$-robustness criterion, as proposed by Allouah et al. (2023).

**Definition 2** ( $(\delta, \kappa)$-robustness). *Let $\delta < \frac{1}{2}$ and $\kappa \geq 0$. A robust aggregation rule $\mathcal{A}(\cdot) : \mathbb{R}^{d \times n} \to \mathbb{R}^d$ is said to be $(\delta, \kappa)$-robustness if, for node $i$ and its neighbors $\mathcal{N}_i$, and any collection of vectors $\{x_i\} \cup \{x_m\}_{m \in \mathcal{N}_i} \subset \mathbb{R}^d$, we have:*

$$\left\| \mathcal{A}(x_i, \{x_m\}_{m \in \mathcal{N}_i}) - \bar{x}_i \right\|^2 \leq \kappa \sum_{j \in \mathcal{H}_i} w_{ij} \left\| x_j - \bar{x}_i \right\|^2,$$

*where $\bar{x}_i = \sum_{j \in \mathcal{H}_i} w_{ij} x_j$ and $\sum_{j \in \mathcal{H}_i} w_{ij} = 1$. Here, $\mathcal{H}_i$ denotes honest agents connected to agent $i$ (including $i$).*

This criterion provides a tight convergence guarantee and is satisfied by most robust aggregation methods. Notably, $\kappa$ has a theoretical lower bound given by $\kappa > \frac{\delta}{1-2\delta}$ (Allouah et al., 2023), which implies that no robust aggregator can achieve arbitrarily close proximity to the average output of the honest workers. Although the value of $\kappa$ varies across different robust aggregation methods, it is possible to achieve $\kappa = \mathcal{O}(\delta)$ when combining aggregation methods with pre-aggregation techniques such as NNM (Allouah et al., 2023).

## 3. Methodology

### 3.1. Primal-Dual Formulation

The Byzantine-robust decentralized optimization problem in (1) seeks to achieve two objectives: minimizing the combination of honest local objective functions and reaching consensus across all participants. To transform the consensus constraint, we introduce matrices $\widehat{\mathbf{A}}, \widehat{\mathbf{B}}, \widehat{\mathbf{D}} \in \mathbb{R}^{dn_h \times dn_h}$ that satisfy

$$\begin{aligned} x_i = x_j, \\ \forall i, j \in \mathcal{H}, \end{aligned} \iff \widehat{\mathbf{A}}\mathbf{x} = \mathbf{x} \Leftrightarrow \widehat{\mathbf{B}}\mathbf{x} = \mathbf{0} \Leftrightarrow \widehat{\mathbf{D}}\mathbf{x} = \mathbf{0}, \quad (4)$$

where $\widehat{\mathbf{A}}$ is a doubly stochastic matrix and $\widehat{\mathbf{B}}, \widehat{\mathbf{D}}$ satisfy $\mathrm{Null}(\widehat{\mathbf{B}}) = \mathrm{Null}(\widehat{\mathbf{D}}) = \mathrm{Span}\{\mathbf{1}_{n_h}\}$. Consequently, the decentralized problem in (1) can be rewritten as follows

$$\min_{\mathbf{x} \in \mathbb{R}^{dn_h}} \mathbf{f}_{\mathcal{H}}(\widehat{\mathbf{A}}\mathbf{x}), \quad \text{s.t.} \quad \widehat{\mathbf{B}}\mathbf{x} = \mathbf{0}. \quad (5)$$

To solve this constrained problem, we employ the augmented Lagrangian method (ALM) (Hestenes, 1969) and construct the augmented Lagrangian function as follows

$$\mathcal{L}_\rho(\mathbf{x}, \hat{\mathbf{d}}) = \mathbf{f}_{\mathcal{H}}(\widehat{\mathbf{A}}\mathbf{x}) + \langle \hat{\mathbf{d}}, \widehat{\mathbf{B}}\mathbf{x} \rangle + \frac{\rho}{2} \|\widehat{\mathbf{D}}\mathbf{x}\|^2, \quad (6)$$

where $\mathbf{x}$ denotes the primal variable, $\hat{\mathbf{d}}$ represents the dual variable, and $\rho > 0$ is the penalty coefficient.

The solution can be obtained by solving the saddle-point problem $\max_{\hat{\mathbf{d}}} \min_{\mathbf{x}} \mathcal{L}_\rho(\mathbf{x}, \hat{\mathbf{d}})$. Following classical primal-dual methods, and using $\widehat{\mathbf{A}}\mathbf{x} = \mathbf{x}$ as in (4), we update the primal variable with gradient descent as follows

$$\begin{aligned} \mathbf{x}^{k+1} &= \mathbf{x}^k - \eta \nabla_{\mathbf{x}} \mathcal{L}_\rho(\mathbf{x}^k, \hat{\mathbf{d}}^k), \\ &= \left( \mathbf{I} - \eta \rho \widehat{\mathbf{D}}^2 \right) \mathbf{x}^k - \eta \widehat{\mathbf{A}} \mathbf{f}_{\mathcal{H}}(\mathbf{x}^k) - \eta \widehat{\mathbf{B}} \hat{\mathbf{d}}^k, \end{aligned} \quad (7)$$

where $\eta > 0$ denotes the step size. Similarly, the dual variable $\mathbf{d}^k$ is updated with the dual ascent as follows

$$\hat{\mathbf{d}}^{k+1} = \hat{\mathbf{d}}^k + \rho \widehat{\mathbf{B}} \mathbf{x}^{k+1}, \quad (8)$$

where the learning rate for the dual variable is chosen to be the same as the penalty coefficient $\rho$.

### 3.2. Scaled Dual Ascent with Modified Lagrangian

However, a fundamental limitation arises in Byzantine settings. In particular, since the Lagrangian $\mathcal{L}_\rho(\mathbf{x}, \hat{\mathbf{d}})$ is linear in the dual variable $\hat{\mathbf{d}}$, it is potentially unbounded. When Byzantine adversaries inject corrupted gradients, the standard dual update can accumulate these errors over $K$ iterations, leading to an asymptotic error term that grows with $K$.

To address this issue, we regularize the dual update by introducing a quadratic penalty term that bounds the dual variables (Cao & Liu, 2018; Sun & Sun, 2024). Such regularization techniques have been used in primal–dual optimization and online learning (Mahdavi et al., 2012; Li et al., 2020). Specifically, we introduce the modified Lagrangian function as follows

$$\mathcal{P}(\mathbf{x}, \hat{\mathbf{d}}) := \mathcal{L}_\rho(\mathbf{x}, \hat{\mathbf{d}}) - \frac{\tau}{2\rho} \|\hat{\mathbf{d}}\|^2, \quad (9)$$

where $\tau > 0$ is the regularization parameter. Then, the dual variable is updated by maximizing the modified Lagrangian function coupled with a proximal term as follows

$$\max_{\hat{\mathbf{d}}} \mathcal{P}(\mathbf{x}^{k+1}, \hat{\mathbf{d}}) - \frac{1}{2\rho} \|\hat{\mathbf{d}} - \hat{\mathbf{d}}^k\|^2. \quad (10)$$

Here, the penalty $\|\hat{\mathbf{d}}\|^2$ in $\mathcal{P}$ controls the magnitude of the dual iterate, while the proximal term $\|\hat{\mathbf{d}} - \hat{\mathbf{d}}^k\|^2$ anchors the new iterate to $\hat{\mathbf{d}}^k$, formulating the step as an iterative update. Then, the closed-form solution to (10) can be given by:

$$\hat{\mathbf{d}}^{k+1} = \frac{1}{\tau + 1} \left( \hat{\mathbf{d}}^k + \rho \widehat{\mathbf{B}} \mathbf{x}^{k+1} \right), \quad (11)$$

which is referred to as *scaled dual ascent* (SDA). For simplicity, let $q := \frac{1}{\tau+1} \in (0, 1)$ denote the scaling factor. Through explicit contraction of the dual update, we prove in Section 4.3 that SDA mitigates error accumulation, thereby rendering the asymptotic error independent of $K$.

*Table 2.* The framework facilitates different decentralized algorithms by specifying matrices $\mathbf{A}$, $\mathbf{B}$, and $\mathbf{C}$.

| Algorithms | $\mathbf{A}$ | $\mathbf{B}$ | $\mathbf{C}$ |
|---|---|---|---|
| ED / EXTRA | $\mathbf{W}$ / $\mathbf{I}$ | $(\mathbf{I} - \mathbf{W})^{1/2}$ | $\mathbf{W}$ |
| {ATC / Semi-ATC / Non-ATC}-GT | $\mathbf{W}^2$ / $\mathbf{W}$ / $\mathbf{I}$ | $\mathbf{I} - \mathbf{W}$ | $\mathbf{W}^2$ |

By applying the variable substitutions $\mathbf{A} = \widehat{\mathbf{A}}$, $\mathbf{B} = \sqrt{\eta\rho}\widehat{\mathbf{B}}$, and $\mathbf{C} = \mathbf{I} - \eta\rho\widehat{\mathbf{D}}^2$, and rescaling the dual variable as $\mathbf{d}^k = \sqrt{\eta/\rho}\widehat{\mathbf{d}}^k$, we merge the primal update (7) with the scaled dual update (11). This yields the following decentralized learning framework with SDA:

$$
\begin{aligned}
\mathbf{x}^{k+1} &= \mathbf{C}\mathbf{x}^k - \eta\mathbf{A}\nabla\mathbf{f}_{\mathcal{H}}(\mathbf{x}^k) - \mathbf{B}\mathbf{d}^k, \\
\mathbf{d}^{k+1} &= q\left(\mathbf{d}^k + \mathbf{B}\mathbf{x}^{k+1}\right).
\end{aligned}
\tag{12}
$$

By specifying matrices $\mathbf{A}$, $\mathbf{B}$, and $\mathbf{C}$, the framework in (12) unifies a wide range of decentralized optimization algorithms. Table 2 details how specific parameterizations recover established methods, including Exact Diffusion (ED), EXTRA, and Gradient Tracking (GT) variants.

### 3.3. The BRED Algorithm

While the framework accommodates various algorithmic instantiations, we adopt Exact Diffusion (ED) as the backbone by setting $\mathbf{A} = \mathbf{W}$, $\mathbf{B} = (\mathbf{I} - \mathbf{W})^{1/2}$, and $\mathbf{C} = \mathbf{W}$. We prioritize ED over other methods for the following reasons. Unlike gradient-tracking methods, which necessitate multiple communication rounds per iteration (corresponding to $\mathbf{C} = \mathbf{W}^2$ and $\mathbf{B}^2 = (\mathbf{I} - \mathbf{W})^2$), ED features a single-round communication structure, thereby reducing the system's exposure to Byzantine attacks. Furthermore, compared with EXTRA, ED offers better stability while maintaining equivalent convergence guarantees (Yuan et al., 2018b).

Substituting matrices $\mathbf{A}$, $\mathbf{B}$, and $\mathbf{C}$ corresponding to ED from Table 2 into the decentralized framework (12) yields:

$$
\begin{aligned}
\mathbf{x}^{k+1} &= \mathbf{W}\mathbf{x}^k - \eta\mathbf{W}\nabla\mathbf{f}_{\mathcal{H}}(\mathbf{x}^k) - (\mathbf{I} - \mathbf{W})^{1/2}\mathbf{d}^k, \\
\mathbf{d}^{k+1} &= q\left(\mathbf{d}^k + (\mathbf{I} - \mathbf{W})^{1/2}\mathbf{x}^{k+1}\right).
\end{aligned}
\tag{13}
$$

Then, by computing $\mathbf{x}^{k+1} - q\mathbf{x}^k$ to eliminate the dual variable $\mathbf{d}^k$, we obtain the following recursion:

$$
\begin{aligned}
\mathbf{x}^{k+1} = \mathbf{W}\Big(&\mathbf{x}^k + q\left(\mathbf{x}^k - \mathbf{x}^{k-1}\right) \\
&- \eta\left(\nabla\mathbf{f}_{\mathcal{H}}(\mathbf{x}^k) - q\nabla\mathbf{f}_{\mathcal{H}}(\mathbf{x}^{k-1})\right)\Big).
\end{aligned}
\tag{14}
$$

Based on these foundations, we propose *Byzantine-Resilient Exact Diffusion* (BRED) and its momentum-enhanced variant, BRED-M, which integrates Polyak's momentum.

**Description of BRED/BRED-M.** As presented in Algorithm 1, each honest worker $i \in \mathcal{H}$ starts by initializing model $x_i^0$ and momentum $m_i^{-1} = 0$. At each iteration $k \in \{0, \dots, K-1\}$, the algorithm proceeds in

---

**Algorithm 1** BRED / BRED-M Algorithm

1: **Input:** Learning rate $\eta > 0$, scaling coefficient $q \in (0, 1)$, momentum $\beta \in [0, 1)$, robust aggregator $\mathcal{A}(\cdot)$.
2: **Initialize:** Honest worker $i \in \mathcal{H}$ sets initial model $x_i^0 \in \mathbb{R}^d$ and initial momentum $m_i^{-1} = 0$.
3: **for** $k = 0, 1, \dots, K - 1$ **do**
4:     **for each honest worker** $i$ **in parallel do**
5:         Compute stochastic gradient $g_i^k = \nabla F_i(x_i^k; \xi_i^k)$.
6:         Update momentum: $m_i^k = \beta m_i^{k-1} + (1 - \beta)g_i^k$.
        *// $\beta = 0$ for BRED.*
7:         **if** $k = 0$ **then**
8:             $x_i^{k+\frac{1}{2}} = x_i^k - \eta m_i^k$.
9:         **else**
10:            $x_i^{k+\frac{1}{2}} = x_i^k + q(x_i^k - x_i^{k-1}) - \eta(m_i^k - q m_i^{k-1})$.
11:         **end if**
12:         Send $x_i^{k+\frac{1}{2}}$ to **neighbors** of $i$.    *// Byzantine workers may send arbitrary vectors.*
13:         Receive $\{x_m^{k+\frac{1}{2}}\}_{m \in \mathcal{N}_i}$ from **neighbors** $\mathcal{N}_i$.
14:         $R_i^{k+\frac{1}{2}} = \mathcal{A}\left(x_i^{k+\frac{1}{2}}, \{x_m^{k+\frac{1}{2}}\}_{m \in \mathcal{N}_i}\right)$.
15:         $x_i^{k+1} = R_i^{k+\frac{1}{2}}$.
16:     **end for**
17: **end for**

---

three phases. *(i) Local computation (steps 5-11):* Each honest worker computes a stochastic gradient, updates the momentum buffer, and forms an intermediate iterate via exact-diffusion correction with the scaling factor as in (14) $x_i^{k+\frac{1}{2}} = x_i^k + q(x_i^k - x_i^{k-1}) - \eta(m_i^k - q m_i^{k-1})$. *(ii) Communication (steps 12-13):* Workers exchange intermediate iterates $x_i^{k+\frac{1}{2}}$ with their neighbors, where Byzantine workers may send arbitrary vectors. *(iii) Aggregation (step 14-15):* Each worker applies the robust aggregator $\mathcal{A}(\cdot)$ to filter malicious updates and computes the next iterate. In Algorithm 1, setting the momentum parameter $\beta = 0$ yields BRED, while $\beta > 0$ yields the momentum variant BRED-M.

## 4. Theoretical Analysis

### 4.1. Assumptions

We establish the assumptions adopted throughout our analysis, which are standard in decentralized learning (Alghunaim & Yuan, 2022; Farhadkhani et al., 2023). We first assume the smoothness of the local functions.

**Assumption 2** (*L*-smoothness)**.** *Each local function $f_i$, for $i \in \mathcal{H}$, is differentiable, and there exists a constant $L \geq 0$ such that for all $x, y \in \mathbb{R}^d$:*

$$
\|\nabla f_i(x) - \nabla f_i(y)\| \leq L\|x - y\|.
\tag{15}
$$

Then, we make the following standard assumption on the variance of the stochastic gradient.

**Assumption 3** (Bounded Variance). *For all $i \in \mathcal{H}$, we assume $\nabla F_i(x, \varphi_i)$ to be an unbiased estimate of $\nabla f_i(x)$ with bounded variance $\sigma^2$.*

We also adopt the bounded gradient heterogeneity assumption, which is widely used in the decentralized optimization literature to characterize non-IID data distributions.

**Assumption 4** (Bounded Gradient Dissimilarity). *There exists a constant $\zeta \geq 0$ such that for all $x \in \mathbb{R}^d$ and each honest client $i \in \mathcal{H}$, we have*

$$\frac{1}{|\mathcal{H}|} \sum_{i \in \mathcal{H}} \|\nabla f_i(x) - \nabla f_{\mathcal{H}}(x)\|^2 \leq \zeta^2,$$
$$\sum_{j \in \mathcal{H}_i} w_{ij} \left\| \nabla f_j(x) - \overline{\nabla f_i}(x) \right\|^2 \leq \zeta^2, \tag{16}$$

*with $\overline{\nabla f_i}(x) := \sum_{c \in \mathcal{H}_i} w_{ic} \nabla f_c(x)$, and $\sum_{c \in \mathcal{H}_i} w_{ic} = 1$.*

### 4.2. Convergence Results

This section details the convergence analysis of the proposed algorithms. Detailed proofs are provided in Appendices D and E. Consistent with Definition 1, we evaluate convergence using the average stationarity gap over $K$ iterations:

$$\frac{1}{K} \sum_{k=0}^{K-1} \mathbb{E}\left[\|\nabla f_{\mathcal{H}}(\bar{x}^k)\|^2\right] \leq \varepsilon_K,$$

where $\bar{x}^k := \frac{1}{n_h} \sum_{i \in \mathcal{H}} x_i^k$ denotes the average model over honest agents.

The following result establishes the convergence of BRED and BRED-M under general non-convex loss functions.

**Theorem 1** (Convergence of BRED). *Suppose Assumptions 1–4 hold. Let $\kappa$ be the robustness parameter. For step size $\eta = \Theta(\sqrt{n_h/K})$, BRED in Algorithm 1 is $(\delta, \varepsilon_K)$-Byzantine-resilient with $\varepsilon_K$ as follows*

$$\varepsilon_K = \underbrace{\mathcal{O}\left(\frac{\Delta_0 + \sigma^2}{\sqrt{n_h K}}\right)}_{\text{dominant terms}} + \underbrace{\mathcal{O}\left(\frac{n_h \lambda^2 \sigma^2}{p \cdot q\underline{\lambda}K} + \frac{n_h \lambda^2 \sigma^2}{p^3 q\underline{\lambda}K^2} + \frac{n_h L^2 \lambda^2 \zeta^2}{p^2 q\underline{\lambda}K^2}\right)}_{\text{topology-related terms}}$$
$$+ \underbrace{\mathcal{O}\left(\frac{\kappa n_h \sigma^2}{p^3 q\underline{\lambda}K} + \frac{\kappa n_h \zeta^2}{p^2 q\underline{\lambda}K}\right)}_{\text{Byzantine-induced terms}} + \underbrace{\mathcal{O}\left(\frac{\kappa(\sigma^2 + \zeta^2)}{(1-q)^2}\right)}_{\text{asymptotic error}},$$

*where $\Delta_0 := f_{\mathcal{H}}(\bar{x}^0) - f_{\mathcal{H}}^\star$ denotes the initial optimality gap, and $p := 1 - \lambda$ represents the spectral gap.*

**Remark 1.** The dominant term $\mathcal{O}(1/\sqrt{n_h K})$ in Theorem 1 indicates that BRED achieves linear speedup with respect to the number of *honest clients* $n_h$. Furthermore, when $\kappa = \mathcal{O}(\delta)$, the asymptotic error of BRED becomes $\mathcal{O}(\delta(\sigma^2 + \zeta^2))$. Compared to the lower bound $\Omega(\delta\zeta^2)$ established in (Karimireddy et al., 2020), this error bound contains an additional variance term $\mathcal{O}(\delta\sigma^2)$.

**Remark 2** (Trade-off in coefficient $q$). Theorem 1 reveals a critical trade-off in the selection of the coefficient $q$. Specifically, as $q \to 1$, the asymptotic error induced by Byzantine attackers increases significantly, scaling with $\mathcal{O}((1-q)^{-2})$. Conversely, selecting a small $q$ mitigates this error floor but slows down convergence, as the topology-related and Byzantine-induced transient terms are inversely proportional to $q$. Therefore, $q$ should be selected in an intermediate range to balance convergence speed and robustness.

**Corollary 1** (Transient time of BRED). *Under the conditions of Theorem 1, suppose $\mathcal{A}(\cdot)$ satisfies the $(\delta, \kappa)$-robustness criterion with $\kappa = \mathcal{O}(\delta)$, then the transient time $\tau$ satisfies*

$$\tau = \mathcal{O}\left(\max\left\{\frac{n_h^3}{(1-\lambda)^2}, \frac{\delta^2 n_h^3}{(1-\lambda)^6}\right\}\right). \tag{17}$$

**Remark 3.** As shown in Corollary 1, while DSGD-based methods (Wu et al., 2023) suffer from a transient time of $\mathcal{O}(n_h^3/(1-\lambda-8\delta\sqrt{n_h})^6)$, BRED improves it to $\mathcal{O}(n_h^3/(1-\lambda)^2)$ when $\delta$ is small. This matches the standard transient time of Exact Diffusion (Alghunaim & Yuan, 2022) in environments without Byzantine attackers ($\delta = 0$).

**Theorem 2** (Convergence of BRED-M). *Suppose Assumptions 1–4 hold. Let the momentum coefficient satisfy $1-\beta = \Theta(\eta L)$. For step size $\eta = \Theta(\sqrt{n_h/K})$, BRED-M in Algorithm 1 is $(\delta, \varepsilon_K)$-Byzantine-resilient with $\varepsilon_K$ as follows*

$$\varepsilon_K = \underbrace{\mathcal{O}\left(\frac{a_\kappa}{\sqrt{n_h K}}\right)}_{\text{dominant terms}} + \underbrace{\mathcal{O}\left(\frac{n_h^{3/2}\lambda^2 \sigma^2}{p \cdot q\underline{\lambda}K^{3/2}} + \frac{n_h \lambda^2 \zeta^2}{p^3 q\underline{\lambda}K^2} + \frac{n_h^2 L^4 \lambda^2 \zeta^2}{p^2 q\underline{\lambda}K^3}\right)}_{\text{topology-related terms}}$$
$$+ \underbrace{\mathcal{O}\left(\frac{\kappa n_h^{3/2} \sigma^2}{p^2 q\underline{\lambda}K^{3/2}} + \frac{\kappa n_h \zeta^2}{p^2 q\underline{\lambda}K}\right)}_{\text{Byzantine-induced terms}} + \underbrace{\mathcal{O}\left(\frac{\kappa\zeta^2}{(1-q)^2}\right)}_{\text{asymptotic error}},$$

*where $\Delta_0 := f_{\mathcal{H}}(\bar{x}^0) - f_{\mathcal{H}}^\star$ denotes the initial optimality gap, $p := 1 - \lambda$ represents the spectral gap and $a_\kappa := \Delta_0 + \sigma^2 + \kappa\sigma^2 n_h$.*

**Remark 4.** Comparing Theorem 1 and Theorem 2, BRED-M achieves two key improvements: (i) variance-dependent transient terms decay at a faster rate of $K^{-3/2}$ instead of $K^{-1}$, and (ii) the asymptotic error reduces from $\mathcal{O}(\kappa(\sigma^2 + \zeta^2)/(1-q)^2)$ to $\mathcal{O}(\kappa\zeta^2/(1-q)^2)$, eliminating the term related to stochastic noise $\mathcal{O}(\delta\sigma^2)$. Consequently, when $\kappa = \mathcal{O}(\delta)$, the latter matches the lower bound $\Omega(\delta\zeta^2)$ for Byzantine-resilient learning (Karimireddy et al., 2020).

**Corollary 2** (Transient time of BRED-M). *Under the conditions of Theorem 2, suppose $\mathcal{A}(\cdot)$ satisfies the $(\delta, \kappa)$-robustness criterion with $\kappa = \mathcal{O}(\delta)$. Then the transient time $\tau$ satisfies*

$$\tau = \mathcal{O}\left(\max\left\{\frac{n_h^2}{1-\lambda}, \frac{n_h}{(1-\lambda)^2}, \frac{\delta^2 n_h^3}{(1-\lambda)^4}\right\}\right). \tag{18}$$

**Remark 5.** Comparing Corollaries 1 and 2, `BRED-M` further reduces the topology dependence from $\mathcal{O}(n_h^3/(1-\lambda)^2)$ to $\mathcal{O}(n_h^2/(1-\lambda)^2)$ and the Byzantine-affected term from $\mathcal{O}(\delta^2 n_h^3/(1-\lambda)^6)$ to $\mathcal{O}(\delta^2 n_h^3/(1-\lambda)^4)$. This improvement is particularly significant when the Byzantine proportion $\delta$ is high, as the $\delta^2$-dependent term becomes dominant.

**Remark 6** (Step-size selection)**.** We remark that in Theorems 1 and 2, the step size is chosen as $\eta = \Theta(\sqrt{n_h/K})$ to simplify the expressions. This choice is not optimal and tighter rates can be obtained by carefully selecting the step size, similar to (Koloskova et al., 2020). However, such choices do not affect the transient time order in terms of network quantities, which is the main conclusion of our results.

### 4.3. The Necessity of Scaled Dual Ascent

As discussed in Section 3, the conventional dual update in bias-correction methods accumulates corrupted gradients from Byzantine adversaries, leading to the asymptotic error term that grows with the number of iterations $K$. This subsection provides theoretical evidence for this phenomenon and demonstrates how SDA resolves this issue. All proofs are deferred to Appendix F.

**Corollary 3.** *Suppose Assumptions 1–4 hold. Let $\delta < \frac{1}{2}$ and let $\mathcal{A}(\cdot)$ be $(\delta, \kappa)$-robust with $\kappa = \mathcal{O}(\delta)$. When $q = 1$ (no scaling), the asymptotic error of* `BRED` *satisfies*

$$\varepsilon_{K\to\infty} = \mathcal{O}\left(K^2\delta(\sigma^2 + \zeta^2)\right). \tag{19}$$

**Remark 7** (Accumulating error without scaling)**.** Corollary 3 reveals the limitation of directly combining bias-correction methods with robust aggregators. Specifically, the asymptotic error grows linearly with $K^2$, implying that the aggregation error accumulates over iterations and prevents convergence to a fixed neighborhood of the optimum.

**Corollary 4.** *Suppose Assumptions 1–4 hold. Let $\delta < \frac{1}{2}$ and let $\mathcal{A}(\cdot)$ be $(\delta, \kappa)$-robust with $\kappa = \mathcal{O}(\delta)$. When $q \in (0, 1)$ (with scaling), the asymptotic error of* `BRED` *satisfies*

$$\varepsilon_{K\to\infty} = \mathcal{O}\left(\frac{\delta(\sigma^2 + \zeta^2)}{(1-q)^2}\right). \tag{20}$$

**Remark 8** (Bounded error with scaling)**.** Corollary 4 demonstrates that SDA eliminates the dependence on $K$. With an appropriate choice of $q$, the asymptotic error reduces to $\mathcal{O}(\delta(\sigma^2 + \zeta^2))$, matching the order achieved by DSGD-based methods (Wu et al., 2023).

**Remark 9** (Momentum variant)**.** The above corollaries analyze the non-momentum case. When momentum is employed, as in `BRED-M`, the asymptotic error further reduces by eliminating the variance term $\mathcal{O}(\delta\sigma^2)$, yielding $\mathcal{O}(\delta\zeta^2/(1-q)^2)$ as established in Theorem 2.

## 5. Experiments

In this section, we evaluate the performance of the `BRED` and `BRED-M` methods under various settings on two standard image classification benchmarks. Detailed implementation specifics and extensive experimental results are available in Appendix B.

### 5.1. Experimental Setup

**Datasets and models.** We evaluate the proposed `BRED` and `BRED-M` methods on two standard image classification datasets: MNIST (LeCun, 1998) and CIFAR-10 (Krizhevsky et al., 2009). For MNIST, we employ a convolutional neural network (CNN) with two convolutional layers followed by fully connected layers. For CIFAR-10, we use a ResNet-18 architecture. Data heterogeneity among clients is simulated using the Dirichlet distribution with concentration parameter $\alpha$, where a smaller $\alpha$ corresponds to more skewed class distributions across clients.

**Baselines and training configuration.** We compare the proposed methods against BR-DSGD, BR-EXTRA, and BR-GT, as well as their momentum variants. All methods were trained using SGD with learning rate $\eta = 0.01$ for MNIST and $\eta = 0.005$ for CIFAR-10. We set the dual scaling coefficient $q = 0.7$. For the momentum variants, we additionally use the momentum coefficient $\beta = 0.9$.

**Byzantine attacks and robust aggregators.** We evaluate robustness under four attack strategies: ALIE (Baruch et al., 2019), Sign Flipping (SF), Fall of Empires (FOE) (Xie et al., 2020), and Label Flipping (LF). We integrate our methods with four robust aggregation mechanisms: Krum (Blanchard et al., 2017), Median, Trimmed Mean (Fang et al., 2022), and iterative outlier scissor (IOS) (Wu et al., 2023).

**Network configurations.** We consider $n = 32$ clients arranged in a $k$-Ring topology, where each node connects to its $2k$ nearest neighbors. We systematically vary two factors: (i) *network sparsity* via $k \in \{4, 8, 12\}$, where smaller $k$ yields sparser graphs with slower mixing; and (ii) *Byzantine proportion* with $f \in \{4, 8\}$ Byzantine attackers, corresponding to $\delta \in \{0.125, 0.25\}$ fraction of Byzantine neighbors per honest client. Detailed topology visualizations and spectral properties are provided in Appendix B.1.

### 5.2. Main Results

**Results on MNIST.** Table 3 presents the test accuracy of all methods under the ALIE attack with Median aggregation across six network configurations with different combinations of network sparsity and Byzantine proportion. The spectral gap $1 - \lambda$ characterizes network connectivity, where $1 - \lambda = 1$ corresponds to a fully-connected topology. As shown in Table 3, `BRED` and `BRED-M` consistently achieve

*Table 3.* Accuracy comparisons (mean $\pm$ std on 3 trials, %) on MNIST under various network configurations with $n = 32$ nodes. Results are reported under the ALIE attack with the Median aggregator. The spectral gap $1 - \lambda$ indicates the mixing rate, and $\delta$ represents the fraction of Byzantine neighbors. **Bold** and underline denote the best and second-best performance, respectively.

| METHOD | $\lambda = 0.81$ | | $\lambda = 0.54$ | | $\lambda = 0.31$ | | $\lambda = 0$ | |
| | $\delta = 0.125$ | $\delta = 0.25$ | $\delta = 0.125$ | $\delta = 0.25$ | $\delta = 0.125$ | $\delta = 0.25$ | $\delta = 0.125$ | $\delta = 0.25$ |
|---|---|---|---|---|---|---|---|---|
| BR-DSGD | 44.69$\pm$8.47 | 49.06$\pm$5.70 | 88.12$\pm$1.54 | 68.55$\pm$14.40 | 88.75$\pm$0.59 | 70.94$\pm$10.60 | 88.45$\pm$0.70 | 65.96$\pm$12.21 |
| BR-DSGD-M | 39.97$\pm$9.57 | 45.35$\pm$7.36 | 87.94$\pm$1.64 | 66.95$\pm$14.27 | 88.44$\pm$0.99 | 70.40$\pm$11.79 | 88.26$\pm$0.96 | 65.34$\pm$12.91 |
| BRGT | 26.26$\pm$3.72 | 32.63$\pm$8.80 | 43.58$\pm$6.03 | 27.78$\pm$10.59 | 52.37$\pm$8.92 | 44.22$\pm$8.72 | 51.29$\pm$8.89 | 31.70$\pm$7.35 |
| BRGT-M | 36.55$\pm$4.34 | 41.79$\pm$7.36 | 43.07$\pm$10.27 | 25.16$\pm$7.42 | 32.44$\pm$2.02 | 15.22$\pm$2.75 | 33.73$\pm$3.90 | 19.54$\pm$1.98 |
| BR-EXTRA | 41.21$\pm$1.88 | 48.13$\pm$1.27 | 49.91$\pm$0.97 | 48.16$\pm$3.46 | 53.36$\pm$2.16 | 47.79$\pm$2.64 | 52.46$\pm$1.34 | 47.26$\pm$2.45 |
| BR-EXTRA-M | 64.57$\pm$3.44 | 69.85$\pm$3.47 | 80.15$\pm$1.86 | 76.82$\pm$0.86 | 79.81$\pm$1.64 | 78.44$\pm$1.06 | 79.97$\pm$1.77 | 78.23$\pm$2.76 |
| BRED | **81.88$\pm$5.41** | **75.02$\pm$5.62** | 89.46$\pm$0.36 | **84.92$\pm$2.25** | 89.75$\pm$0.40 | 85.55$\pm$1.45 | 89.43$\pm$0.40 | 83.23$\pm$2.31 |
| BRED-M | 72.84$\pm$7.66 | 74.33$\pm$5.76 | **89.58$\pm$0.92** | 84.89$\pm$3.94 | **89.89$\pm$0.39** | **87.16$\pm$1.29** | **89.83$\pm$0.33** | **85.59$\pm$2.07** |

*Table 4.* Accuracy comparisons (mean $\pm$ std on 3 trials, %) on CIFAR-10 under various network configurations with $n = 32$ nodes. Results are reported under the ALIE attack with the Median aggregator. The spectral gap $1 - \lambda$ indicates the mixing rate, and $\delta$ represents the fraction of Byzantine neighbors. **Bold** and underline denote the best and second-best performance, respectively.

| METHOD | $\lambda = 0.81$ | | $\lambda = 0.54$ | | $\lambda = 0.31$ | | $\lambda = 0$ | |
| | $\delta = 0.125$ | $\delta = 0.25$ | $\delta = 0.125$ | $\delta = 0.25$ | $\delta = 0.125$ | $\delta = 0.25$ | $\delta = 0.125$ | $\delta = 0.25$ |
|---|---|---|---|---|---|---|---|---|
| BR-DSGD | 52.25$\pm$1.02 | 28.98$\pm$2.23 | 57.83$\pm$1.91 | 39.52$\pm$3.49 | 58.19$\pm$0.98 | 23.53$\pm$6.26 | 58.47$\pm$1.31 | 40.03$\pm$3.72 |
| BR-DSGD-M | 33.92$\pm$2.38 | 23.53$\pm$6.26 | 42.67$\pm$4.12 | 21.50$\pm$6.41 | 43.22$\pm$4.61 | 28.25$\pm$5.83 | 43.99$\pm$3.59 | 29.65$\pm$4.87 |
| BR-EXTRA | 35.44$\pm$0.09 | 28.96$\pm$1.65 | 33.80$\pm$0.20 | 30.48$\pm$1.49 | 33.61$\pm$0.39 | 28.96$\pm$1.65 | 33.77$\pm$0.29 | 29.88$\pm$1.20 |
| BR-EXTRA-M | 36.39$\pm$2.79 | 23.66$\pm$1.44 | 34.53$\pm$0.23 | 25.22$\pm$1.82 | 33.26$\pm$3.07 | 23.66$\pm$1.440 | 34.97$\pm$2.88 | 25.40$\pm$3.09 |
| BRED | 58.09$\pm$1.16 | 40.25$\pm$1.49 | 60.47$\pm$0.38 | 44.21$\pm$4.99 | 61.53$\pm$0.75 | 48.25$\pm$2.18 | 46.23$\pm$4.65 | 46.04$\pm$1.32 |
| BRED-M | **59.62$\pm$0.76** | **46.23$\pm$4.65** | **64.91$\pm$0.31** | **56.54$\pm$0.49** | **64.73$\pm$1.09** | **58.99$\pm$0.84** | **61.52$\pm$1.43** | **48.38$\pm$2.38** |

the highest test accuracy, surpassing all baseline methods by significant margins. When the proportion of Byzantine neighbors increases from $12.5\%$ to $25\%$, all methods experience performance degradation; however, BRED exhibits greater resilience, maintaining substantially higher accuracy under stronger attacks. Furthermore, as networks become sparser, BR-DSGD suffers significant accuracy drops due to amplified heterogeneity bias, whereas BRED effectively mitigates this topology-induced degradation.

**Results on CIFAR-10.** Table 4 summarizes the performance on CIFAR-10 under different network configurations. We omit BR-GT as this method fails to converge despite extensive hyperparameter tuning, revealing its vulnerability under Byzantine attackers. The results corroborate our findings on MNIST: BRED and BRED-M consistently achieve superior accuracy across all settings. Notably, BRED-M outperforms BRED by a larger margin on CIFAR-10, demonstrating that momentum-based variance reduction is particularly beneficial for more complex learning tasks.

### 5.3. Ablation Studies

We conduct ablation studies to investigate the impact of various factors on algorithm performance. The detailed results of the ablation studies are provided in Appendix B.4.

**Impact of data heterogeneity.** We vary the Dirichlet concentration parameter $\alpha \in \{0.5, 1.0, 5.0\}$, where smaller values indicate more heterogeneous data distributions. The results show that higher data heterogeneity amplifies the challenges posed by Byzantine attacks, yet BRED and BRED-M

maintain superior robustness across all heterogeneity levels.

**Impact of number of clients.** We evaluate scalability by varying the number of clients $n \in \{16, 24, 32\}$ under fully connected topologies. Empirical results confirm that BRED and BRED-M scale effectively with the number of participating clients and outperform existing approaches.

**Impact of local epochs.** We evaluate the effect of varying the number of local SGD updates $E \in \{1, 5, 10\}$ in each communication round. BRED and BRED-M consistently outperform baseline methods across all local epoch settings, demonstrating robustness to different communication frequencies under Byzantine attacks.

**Sensitivity to the scaling factor $q$.** We further analyze the impact of the scaling factor $q$ on performance. Our results show that the optimal $q^\star$ lies within the range $[0.6, 0.8]$, suggesting that setting $q = 0.7$ is sufficient for most scenarios. Notably, when $q \to 1$, performance degrades rapidly, which is consistent with our theoretical results.

### 6. Conclusion

This paper studied Byzantine-resilient decentralized optimization over sparse networks. Existing DSGD-based approaches suffer from high sensitivity to network topology, while bias-correction methods, though achieving improved topology dependence, cause the asymptotic error to grow linearly with the number of iterations. To overcome this limitation, we proposed scaled dual ascent (SDA) within the augmented Lagrangian framework for decentralized learning, which introduces a fractional scaling factor that effectively

stabilizes the dual dynamics. Based on this framework, we developed `BRED` and its momentum variant `BRED-M`. Theoretically, we proved that `BRED` ensures linear speedup with reduced topology dependence, while `BRED-M` accelerates variance decay and achieves the optimal asymptotic error by eliminating stochastic noise effects. Extensive experiments validated the effectiveness of the proposed methods across diverse network topologies and attack scenarios.

## Impact Statement

The objective of this paper is to advance the field of machine learning. There are many potential societal consequences of our work, none of which we feel must be specifically highlighted here.

## Acknowledgements

This work was supported in part by a grant from the NSFC/RGC Joint Research Scheme sponsored by the Research Grants Council of the Hong Kong Special Administrative Region, China and National Natural Science Foundation of China (Project No. N_HKUST656/22).

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

## Organization

The appendices are organized as follows:

## A. More Related Works

**Byzantine-resilient distributed/decentralized learning.** Numerous approaches to Byzantine-resilient machine learning have been proposed in recent years. A majority of these methods focus predominantly on server-based coordination mechanisms (Blanchard et al., 2017; Yin et al., 2018; Pillutla et al., 2022; Acharya et al., 2022; Murata et al., 2024), whereas relatively fewer studies address the decentralized setting (Wu et al., 2023; Allouah et al., 2024; El-Mhamdi et al., 2021). Under decentralized frameworks, works such as (Su & Vaidya, 2015; 2020; Fang et al., 2022; Yang & Bajwa, 2019) investigated Byzantine-resilient decentralized deterministic optimization based on trimmed mean aggregation. The convergence behavior of Byzantine-resilient distributed learning has been extensively studied over the years (Yin et al., 2018; Karimireddy et al., 2021; Pillutla et al., 2022; Farhadkhani et al., 2023; Allouah et al., 2024). More recently, research has focused on analyzing the convergence behavior in heterogeneous settings (Karimireddy et al., 2020; Allouah et al., 2023; Murata et al., 2024). In this line of work, Karimireddy et al. (2020) was the first to establish the lower bound on the asymptotic error, given by $\Omega\left(\delta\zeta^2\right)$. Building on this foundation, Allouah et al. (2023) proposed Nearest-Neighbor Mixing (NNM) algorithm, showing that most robust aggregation methods, when combined with NNM, are able to achieve an asymptotic error of $\mathcal{O}\left(\delta\zeta^2\right)$. Despite these advancements, existing decentralized approaches predominantly assume fully connected communication topologies, leaving Byzantine-resilient optimization over sparse networks an open research direction.

**Decentralized optimization with bias-correction.** Numerous studies have explored decentralized stochastic optimization methods tailored to nonconvex loss functions (Alghunaim & Yuan, 2022; Lian et al., 2017; Assran et al., 2019; Bianchi & Jakubowicz, 2012; Swenson et al., 2022). Among these methods, Decentralized Stochastic Gradient Descent (DSGD) remains the most extensively studied and widely adopted approach (Lian et al., 2017; Assran et al., 2019; Koloskova et al., 2020). Nevertheless, DSGD is known to be particularly sensitive to network topology, a sensitivity primarily arising from heterogeneity in local data distributions across participating agents (Yuan et al., 2020; Koloskova et al., 2020). To mitigate the dependence of DSGD on topology, significant efforts have been made toward effectively correcting biases induced by data heterogeneity. Prominent bias-correction techniques proposed in this context include EXTRA (Shi et al., 2015), Exact Diffusion (ED) (Yuan et al., 2018a; Li et al., 2019; Tang et al., 2018), and gradient-tracking-based algorithms (Xu et al., 2015; Di Lorenzo & Scutari, 2016; Nedic et al., 2017). In Byzantine-robust decentralized optimization, real-world network topologies tend to be inherently sparse. Furthermore, replacing the standard mean aggregator with robust aggregation functions typically leads to even greater sparsification of the communication topology. Consequently, there remains a critical need for decentralized stochastic optimization methods with reduced sensitivity to network topology while maintaining robustness and effective convergence performance.

**Augmented Lagrangian Methods.** A standard approach for decentralized optimization is to formulate it as a constrained optimization problem, as in (5). Many methods achieving linear convergence (Jakovetić et al., 2014; Li & Lin, 2020; Schmidt et al., 2011; Shi et al., 2014; Arjevani et al., 2020) can be unified under a general framework based on the augmented Lagrangian method, as discussed in Section 3. The primary distinction among these methods lies in how the primal subproblem (5) is solved. Schmidt et al. (2011) employed an alternating direction method of multipliers (ADMM); Shi et al. (2014) proposed the EXTRA algorithm, which performs a single gradient descent step to approximately solve the primal subproblem; Jakovetić et al. (2014) adopted multi-step iterative schemes such as Jacobi or Gauss-Seidel methods. More recently, Arjevani et al. (2020) introduced a novel primal algorithm that achieves an optimal convergence rate, matching the

established lower bounds. Sun & Sun (2024) proposed a primal-dual algorithm with scaled dual descent (SDD), termed SDD-ADMM, and demonstrated that it finds an $\epsilon$-stationary solution in $\mathcal{O}(\epsilon^{-4})$ iterations. Also, adding a regularization term to the Lagrangian function is widely used in primal–dual optimization and online learning (Cao & Liu, 2018; Mahdavi et al., 2012; Li et al., 2020). Our work follows a similar approach, leveraging regularization to scale dual ascent and mitigate the aggregation noise induced by decentralized primal–dual methods.

# B. Experiments

## B.1. Implementation details.

### B.1.1. ENVIRONMENTS

All methods were implemented using the PyTorch framework (Paszke et al., 2019) and executed on NVIDIA A100 GPUs with 40GB of memory or V100 GPUs with 32GB of memory.

### B.1.2. INTRODUCTION OF THE DATASET

*Table 5.* Dataset introductions.

| Dataset | Training Data | Test Data | Class | Size |
|---|---|---|---|---|
| Synthetic Logistic | $n \times 2000$ | – | 2 | $d = 20$ |
| MNIST | 60,000 | 10,000 | 10 | 1×28×28 |
| CIFAR-10 | 50,000 | 10,000 | 10 | 3×32×32 |

**Synthetic Logistic.** We evaluate the proposed algorithms on a synthetic dataset designed for distributed non-convex logistic regression. The dataset considers a network of $n$ agents, where each agent $i$ possesses a local dataset $\mathcal{D}_i = \{(u_{ij}, v_{ij})\}_{j=1}^m$ consisting of $m$ i.i.d. samples. The feature vectors $u_{ij} \in \mathbb{R}^d$ are drawn independently from the standard Gaussian distribution $\mathcal{N}(0, I_d)$, and the binary labels $v_{ij} \in \{-1, +1\}$ are generated according to the logistic model with probability $\mathbb{P}(v_{ij} = +1) = \sigma(u_{ij}^\top x_i)$, where $\sigma(z) = 1/(1 + e^{-z})$ denotes the sigmoid function. Each agent minimizes a local objective function comprising the logistic loss and a non-convex regularization term $r(x) = \sum_{j=1}^d \frac{x_j^2}{1+x_j^2}$, which promotes sparsity while preserving smoothness. In our experiments, we set the feature dimension $d = 20$ and the number of local samples $m = 2000$ per agent.

**MNIST/CIFAR-10.** MNIST and CIFAR-10 are fundamental datasets in computer vision. While CIFAR-10 consists of color images, MNIST comprises grayscale digits; both feature a low resolution of $32 \times 32$ pixels (or $28 \times 28$ for MNIST). Federated Learning (FL) and Decentralized FL (DFL) often prioritize such smaller-scale datasets because they effectively simulate the constraints of real-world privacy-preserving scenarios, such as medical imaging. In these contexts, data is often isolated, resolutions are moderate, and sample sizes per class are limited due to high labeling costs.

### B.1.3. HETEROGENEITY LEVEL

**Synthetic Logistic.** To model statistical heterogeneity across agents, we adopt a hierarchical parameter generation scheme. Specifically, we first define a global ground-truth parameter $x_0 = \mathbf{1}_d \in \mathbb{R}^d$. Each agent $i$ then has its own local true parameter $x_i = x_0 + \epsilon_i$, where $\epsilon_i \sim \mathcal{N}(0, \sigma_h^2 I_d)$ is an agent-specific perturbation. The hyperparameter $\sigma_h^2$ controls the degree of data heterogeneity: when $\sigma_h^2 = 0$, all agents share identical data distributions (homogeneous setting); as $\sigma_h^2$ increases, the local data distributions diverge, introducing statistical heterogeneity that reflects realistic federated learning scenarios where different clients may have non-i.i.d. data. In our experiments, we evaluate algorithm performance under both homogeneous ($\sigma_h^2 = 0.0$) and heterogeneous ($\sigma_h^2 = 0.2$) settings to comprehensively assess the impact of data heterogeneity on convergence behavior.

**MNIST/CIFAR-10.** For MNIST and CIFAR-10, we partition data across clients using the Dirichlet distribution $\text{Dir}(\alpha)$, where the concentration parameter $\alpha$ controls heterogeneity. Figure 1 illustrates the data distribution with $\alpha = 0.3$ across 10 clients and 10 classes. The heatmaps show the number of samples each client holds per class: darker regions indicate higher sample counts, while lighter regions represent fewer samples. As observed, each client exhibits a skewed class distribution—some clients predominantly hold samples from specific classes (e.g., Client 3 on MNIST is dominated by class 0), reflecting realistic non-i.i.d. federated scenarios.

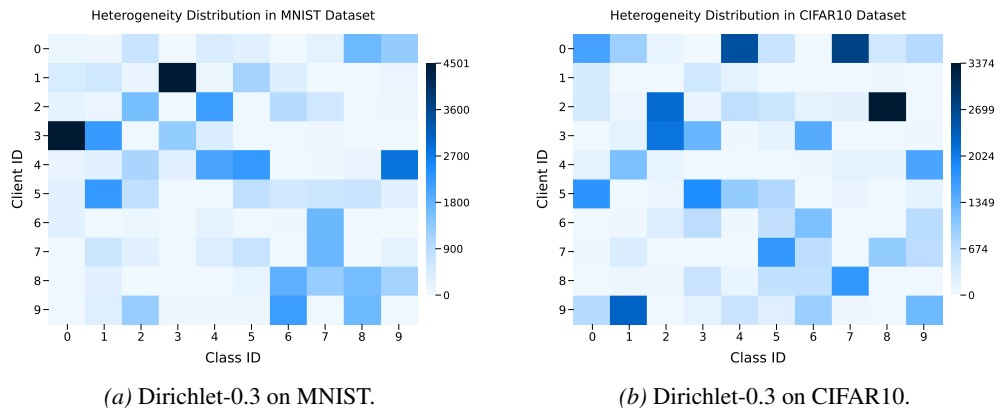

*(a)* Dirichlet-0.3 on MNIST.  *(b)* Dirichlet-0.3 on CIFAR10.

*Figure 1.* Heat-map of the Dirichlet split on different datasets.

### B.1.4. MODEL ARCHITECTURE AND DATASET

Table 6 summarizes the model architectures and training configurations used in our experiments across three datasets. For the Synthetic Logistic dataset, we employ a linear logistic regression model with feature dimension $d = 20$, augmented with a non-convex regularization term to evaluate algorithm performance on non-convex optimization problems. We use full-batch gradient descent with learning rate $\eta = 0.1$ and $m = 2000$ samples per client. For MNIST, we adopt a convolutional neural network (CNN) consisting of two convolutional layers (Conv1: $1 \times 32 \times 5$, Conv2: $32 \times 64 \times 5$) followed by fully connected layers ($64 \times 7 \times 7 \rightarrow 256 \rightarrow 10$). The model is trained using SGD with learning rate $\eta = 0.01$ and batch size $b = 128$. For CIFAR-10, we utilize ResNet-18, a deeper architecture with four convolutional blocks progressively increasing channel dimensions from 64 to 512, followed by a fully connected layer for 10-class classification. We train with SGD using learning rate $\eta = 0.005$ and batch size $b = 128$.

*Table 6.* Model Architectures and Dataset Information

| Details | **Synthetic Logistic** | **MNIST** | **CIFAR-10** |
|---|---|---|---|
| Model type | Logistic Regression | CNN | ResNet-18 |
| Model architecture | Linear: $d = 20$ 
 Non-convex regularizer: $r(x) = \sum_j \frac{x_j^2}{1+x_j^2}$ | Conv1: $1 \times 32 \times 5$, Conv2: $32 \times 64 \times 5$ 
 FC: $64 \times 7 \times 7 \rightarrow 256 \rightarrow 10$ | Conv1: $3 \times 64$, Conv2: $64 \times 128$ 
 Conv3: $128 \times 256$, Conv4: $256 \times 512$ 
 FC: $512 \rightarrow 10$ |
| Optimizer | SGD | SGD | SGD |
| Learning rate ($\eta$) | $\eta = 0.1$ | $\eta = 0.01$ | $\eta = 0.005$ |
| Batch size ($b$) | $b = 2000$ (full batch) | $b = 128$ | $b = 128$ |
| Feature dimension | $d = 20$ | $1 \times 28 \times 28$ | $3 \times 32 \times 32$ |

### B.1.5. ATTACKS.

In our experiments, the Byzantine workers execute four state-of-the-art gradient attacks, namely A Little is Enough (ALIE) (Baruch et al., 2019), Fall of Empires (Xie et al., 2020), Sign-flipping (SF), and Label-flipping (LF). Let $a^k$ denote the attack direction at iteration $k$, and let $\eta \geq 0$ be a fixed scalar. At each step $k$, every Byzantine worker sends to the server a manipulated vector $B^k = \bar{s}^k + \eta a^k$, where $\bar{s}^k$ is an estimate of the true gradient (or momentum). In our gradient descent (GD) experiments, we compute this estimate as $\bar{s}^k = \frac{1}{|\mathcal{H}|} \sum_{i \in \mathcal{H}} g_i^k$, where $g_i^k$ is the gradient computed by honest worker $w_i$ at iteration $k$.

- **ALIE:** The adversarial direction is chosen as $a^k = \sigma^k$, where $\sigma^k$ is the coordinate-wise standard deviation of $\bar{s}^k$.
- **IPM:** The adversary sets $a^k = -\bar{s}^k$. In this case, each Byzantine worker sends $(1 - \eta)\bar{s}^k$ to the server.
- **SF:** This attack also uses $a^k = -\bar{s}^k$, but fixes $\eta = 2$. Thus, Byzantine workers send $B^k = a^k = -\bar{s}^k$ at every step.
- **LF:** Byzantine workers perform a data poisoning attack by flipping their local labels. For MNIST classification with $C = 10$ classes, each label $y$ is replaced by $C - 1 - y$ (i.e., $0 \rightarrow 9$, $1 \rightarrow 8$, ..., $9 \rightarrow 0$). The Byzantine workers then

compute gradients using these corrupted labels: $B_i^k = \nabla f_i(x^k; \mathcal{D}_i^{\text{flipped}})$, where $\mathcal{D}_i^{\text{flipped}} = \{(x_j, C - 1 - y_j)\}$ denotes the dataset with flipped labels. This attack causes Byzantine gradients to push the model toward incorrect decision boundaries.

### B.1.6. ROBUST AGGREGATORS

Furthermore, we evaluate our method's compatibility and performance when integrated with four established robust aggregation mechanisms: Krum (Blanchard et al., 2017), Median, Trimmed Mean (Yin et al., 2018), and iterative outlier scissor (IOS) (Wu et al., 2023).

- **Krum.** Given input vectors $x_1, \ldots, x_n$, Krum selects one of them based on its proximity to its closest neighbors while ignoring the $f$ farthest vectors. For each $i \in [n]$, let $i_1, \ldots, i_{n-1}$ be the indices of the other vectors sorted by distance from $x_i$, and define the set of its $n - f - 1$ nearest neighbors as $C_i = \{i_1, \ldots, i_{n-f-1}\}$. Krum then chooses the vector whose neighbors are collectively the closest:

$$\text{Krum}(x_1, \ldots, x_n) = x_{i^*}, \quad i^* \in \arg\min_{i \in [n]} \sum_{j \in C_i} \|x_i - x_j\|^2.$$

- **Median.** Given vectors $x_1, \ldots, x_n \in \mathbb{R}^d$, the multivariate median is defined as the point $\text{Med}(x_1, \ldots, x_n) := \arg\min_{x \in \mathbb{R}^d} \sum_{i=1}^n \|x - x_i\|_2$. This estimator minimizes the sum of Euclidean distances to all input vectors. Although robust, it is computationally expensive in high-dimensional settings.

- **Trimmed Mean.** Given $x_1, \ldots, x_n \in \mathbb{R}^d$ and a trimming parameter $f$, let $x_{(1)}, \ldots, x_{(n)}$ denote the vectors sorted by their distance to the mean, with $\|x_{(1)} - \bar{x}\|_2 \leq \cdots \leq \|x_{(n)} - \bar{x}\|_2$, where $\bar{x} = \frac{1}{n} \sum_{i=1}^n x_i$. The trimmed mean removes the $f$ farthest and $f$ closest vectors and averages the remaining:

$$\text{TM}(x_1, \ldots, x_n) = \frac{1}{n - 2f} \sum_{j=f+1}^{n-f} x_{(j)}.$$

- **Iterative Outlier Scissor (IOS).** Given vectors $x_1, \ldots, x_n$ with weights $\{w_m\}$ and an estimate $q$ of Byzantine inputs, IOS iteratively prunes outliers. Starting from the full set $\mathcal{U}^{(0)} = [n]$, at each iteration $i = 0, \ldots, q - 1$, it computes the weighted average $x_{\text{avg}}^{(i)}$ and removes the farthest vector:

$$m^{(i)} = \arg\max_{m \in \mathcal{U}^{(i)}} \|x_m - x_{\text{avg}}^{(i)}\|_2, \quad \mathcal{U}^{(i+1)} = \mathcal{U}^{(i)} \setminus \{m^{(i)}\}.$$

After $q$ removals, it returns the weighted average of the remaining trusted set $\mathcal{U}^{(q)}$.

### B.1.7. NETWORK TOPOLOGY

We consider decentralized networks with $n = 32$ nodes arranged in a $k$-Ring topology, which is widely used in decentralized optimization due to its regularity and tunable connectivity (Lian et al., 2017; Koloskova et al., 2020). In a $k$-Ring topology, each node $i$ is connected to its $2k$ nearest neighbors, i.e., nodes $\{(i - k) \mod n, \ldots, (i - 1) \mod n, (i + 1) \mod n, \ldots, (i + k) \mod n\}$.

**Network Sparsity.** The sparsity of the network is controlled by the parameter $k$. A smaller $k$ yields a sparser graph with fewer edges, while a larger $k$ results in a denser, more connected network. Specifically, the total number of edges in a $k$-Ring is $nk$, and each node has degree $2k$. The extreme cases are $k = 1$ (standard ring, sparsest) and $k = \lfloor n/2 \rfloor$ (fully connected, densest).

**Spectral Gap.** The mixing rate of gossip-based algorithms depend on the spectral gap $1 - \lambda(W)$, where $\lambda(W)$ is the second-largest eigenvalue of the mixing matrix $W$. For $k$-Ring topologies, a larger $k$ leads to a larger spectral gap and thus faster information mixing across the network. Table 7 summarizes the spectral properties of our configurations.

**Byzantine Attack Model.** We place $f$ Byzantine nodes uniformly around the ring to ensure balanced attack exposure. This placement guarantees that each honest node has exactly $b = \lfloor 2kf/n \rfloor$ Byzantine neighbors, enabling a fair comparison across different configurations. The ratio $b/(2k)$ represents the fraction of corrupted neighbors, which directly impacts the effectiveness of robust aggregation methods.

*Table 7.* Network topology configurations with $n = 32$ nodes. The spectral gap $1 - \lambda$ indicates the mixing rate, and $\delta = b/(2k)$ represents the fraction of Byzantine neighbors.

| Setting | $k$ | Degree | Attackers ($f$) | Byz. Neighbors ($b$) | $\delta$ | $1 - \lambda$ |
|---------|-----|--------|-----------------|----------------------|----------|---------------|
| 1 | 4 | 8 | 4 | 1 | 12.5% | 0.1910 |
| 2 | 4 | 8 | 8 | 2 | 25.0% | 0.1910 |
| 3 | 8 | 16 | 4 | 2 | 12.5% | 0.4609 |
| 4 | 8 | 16 | 8 | 4 | 25.0% | 0.4609 |
| 5 | 12 | 24 | 4 | 3 | 12.5% | 0.6910 |
| 6 | 12 | 24 | 8 | 6 | 25.0% | 0.6910 |

Our six configurations systematically vary two key factors:

- **Network connectivity ($k \in \{4, 8, 12\}$):** Controls the sparsity and mixing rate of the network. Sparser networks ($k = 4$) have slower convergence but lower communication cost, while denser networks ($k = 12$) converge faster but require more communication per round.

- **Attack intensity ($f \in \{4, 8\}$):** Determines the fraction of Byzantine nodes ($f/n \in \{12.5\%, 25\%\}$) and the number of Byzantine neighbors per honest node ($b$).

These configurations span a wide range of scenarios: Settings 1 and 2 represent sparse networks with limited connectivity, Settings 3 and 4 represent moderate connectivity, and Settings 5 and 6 represent dense networks approaching full connectivity. The ratio $b/2k$ represents the fraction of Byzantine neighbors among all neighbors, ranging from $12.5\%$ (Setting 1, 3, 5) to $25\%$ (Settings 2, 4, 6).

**Topology Visualization.** Figure 2 visualizes the six topology configurations. Blue nodes represent honest participants, while red nodes indicate Byzantine attackers. The highlighted edges show connections to Byzantine nodes, illustrating how each honest node is exposed to exactly $b$ attackers. Comparing across configurations, we observe that increasing $k$ leads to denser connectivity, while increasing $f$ results in more Byzantine neighbors per honest node.

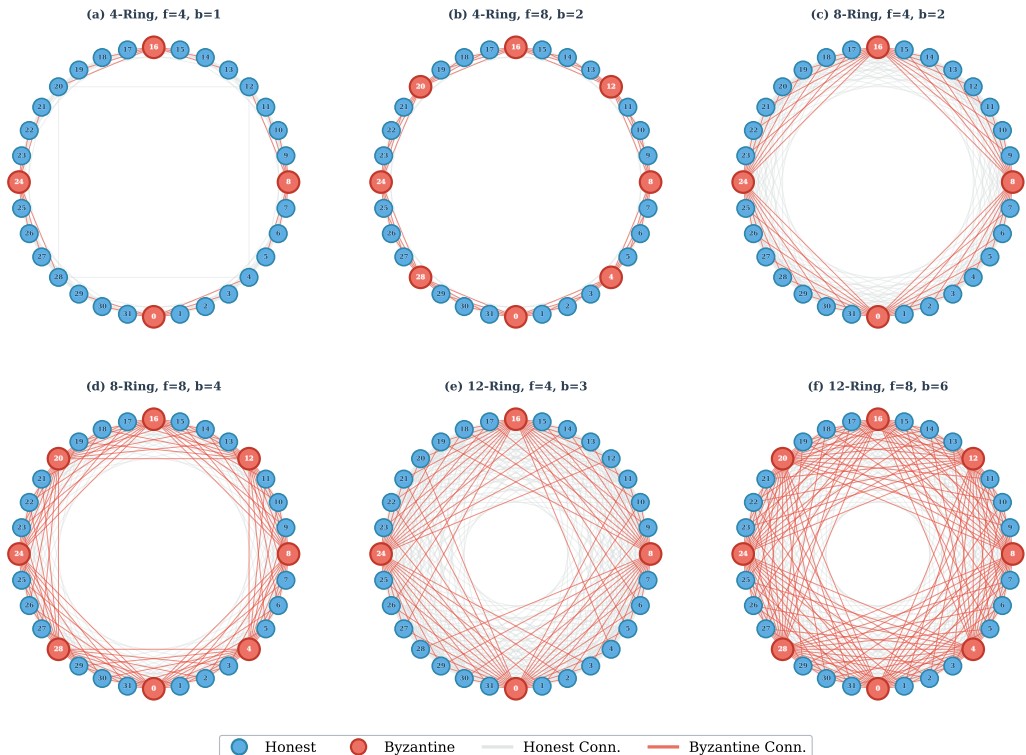

*Figure 2.* Network topologies for decentralized learning with Byzantine attackers. Each configuration shows a k-Ring topology with n=32 nodes, where f denotes the number of Byzantine nodes (red) and b denotes the number of Byzantine neighbors per honest node (blue).

## B.2. Additional empirical performance on synthetic dataset

In this subsection, we present additional experiments on synthetic quadratic objectives to validate the theoretical analysis. We systematically investigate the influence of three key factors on convergence: (i) data heterogeneity $\sigma_h^2 \in \{0, 0.1, 0.2\}$, (ii) network sparsity $k \in \{4, 8, 12\}$, and (iii) fraction of Byzantine attackers $f \in \{4, 8\}$. All experiments are conducted on $k$-Ring topologies with $n = 32$ nodes, evaluating the squared gradient norm $\|\nabla f(\bar{\mathbf{x}})\|^2$ over communication rounds under various attack strategies (ALIE, Sign-Flipping, Label-Flipping).

**Data heterogeneity.** Figures 3–8 illustrate the influence of data heterogeneity $\sigma_h^2$ on convergence under various Byzantine attacks. In the i.i.d. setting ($\sigma_h^2 = 0$), small $q$ values achieve the best performance due to efficient information aggregation. As heterogeneity increases, the optimal $q$ shifts toward larger values to compensate for the gradient bias induced by non-i.i.d. data distributions. Notably, $q = 1$ (pure ED) consistently underperforms, confirming that a balanced trade-off between local correction and global mixing is essential. These empirical observations are well-aligned with the theoretical analysis, which predicts that higher heterogeneity requires stronger local gradient correction.

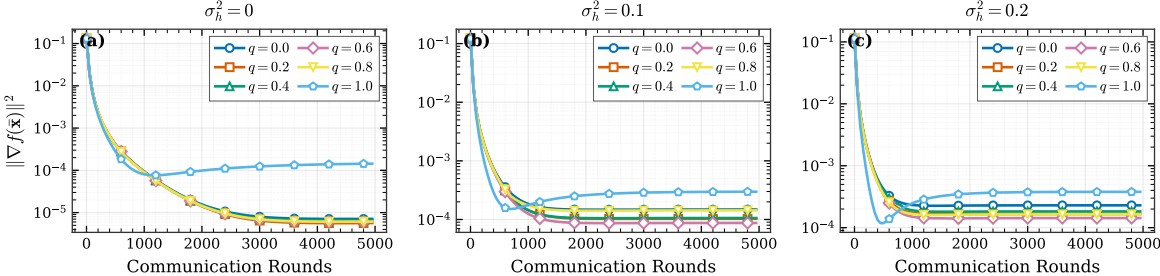

*Figure 3.* Convergence comparison of the BRED algorithm under Sign-Flipping (SF) attack with Setting 1 in Table 7. The three subplots show the squared gradient norm $\|\nabla f(\bar{\mathbf{x}})\|^2$ versus communication rounds under different data heterogeneity levels: (a) $\sigma_h^2 = 0$, (b) $\sigma_h^2 = 0.1$, and (c) $\sigma_h^2 = 0.2$.

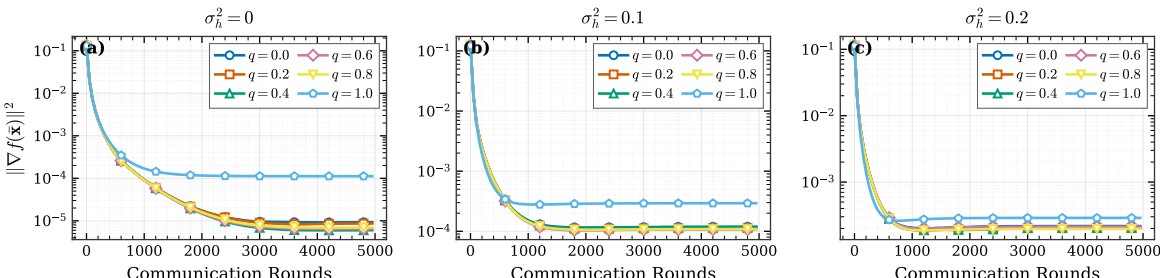

*Figure 4.* Convergence comparison of the BRED algorithm under Sign-Flipping (SF) attack with Setting 3 in Table 7. The three subplots show the squared gradient norm $\|\nabla f(\bar{\mathbf{x}})\|^2$ versus communication rounds under different data heterogeneity levels: (a) $\sigma_h^2 = 0$, (b) $\sigma_h^2 = 0.1$, and (c) $\sigma_h^2 = 0.2$.

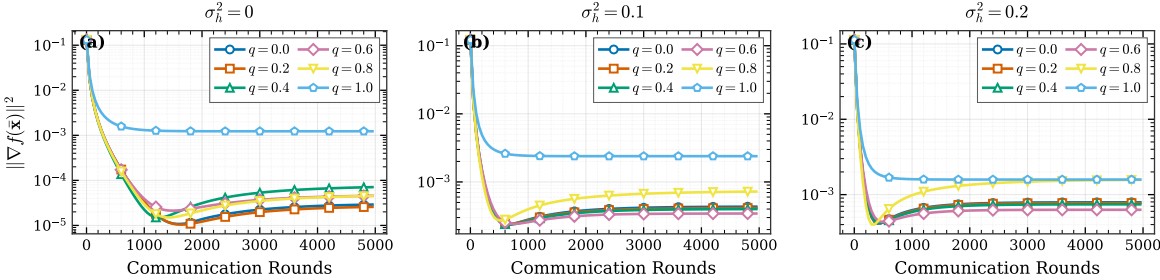

*Figure 5.* Convergence comparison of the BRED algorithm under A Little Is Enough (ALIE) attack with Setting 2 in Table 7. The three subplots show the squared gradient norm $\|\nabla f(\bar{\mathbf{x}})\|^2$ versus communication rounds under different data heterogeneity levels: (a) $\sigma_h^2 = 0$, (b) $\sigma_h^2 = 0.1$, and (c) $\sigma_h^2 = 0.2$.

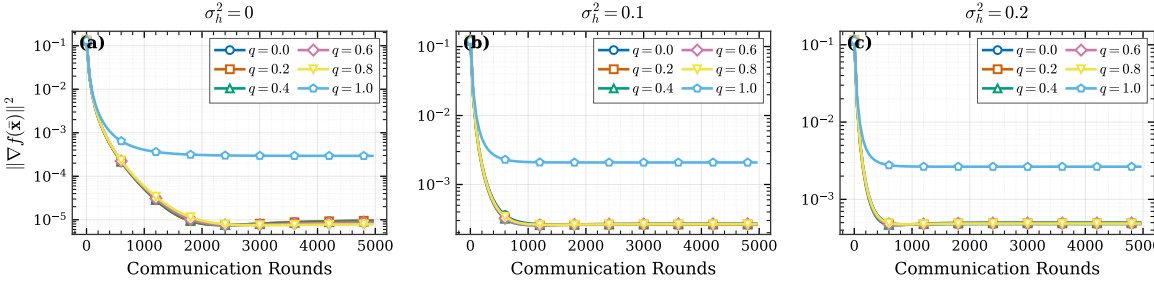

*Figure 6.* Convergence comparison of the BRED algorithm under A Little Is Enough (ALIE) attack with Setting 4. The three subplots show the squared gradient norm $\|\nabla f(\bar{\mathbf{x}})\|^2$ versus communication rounds under different data heterogeneity levels: (a) $\sigma_h^2 = 0$, (b) $\sigma_h^2 = 0.1$, and (c) $\sigma_h^2 = 0.2$.

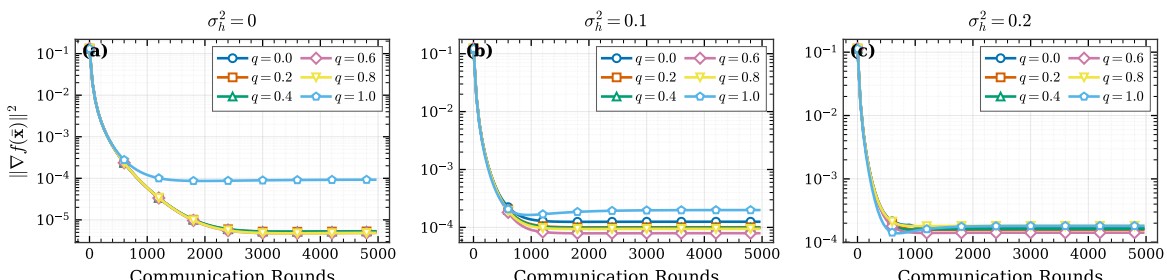

*Figure 7.* Convergence comparison of the BRED algorithm under Label-Flipping (LF) attack with Setting 1 in Table 7. The three subplots show the squared gradient norm $\|\nabla f(\bar{\mathbf{x}})\|^2$ versus communication rounds under different data heterogeneity levels: (a) $\sigma_h^2 = 0$, (b) $\sigma_h^2 = 0.1$, and (c) $\sigma_h^2 = 0.2$.

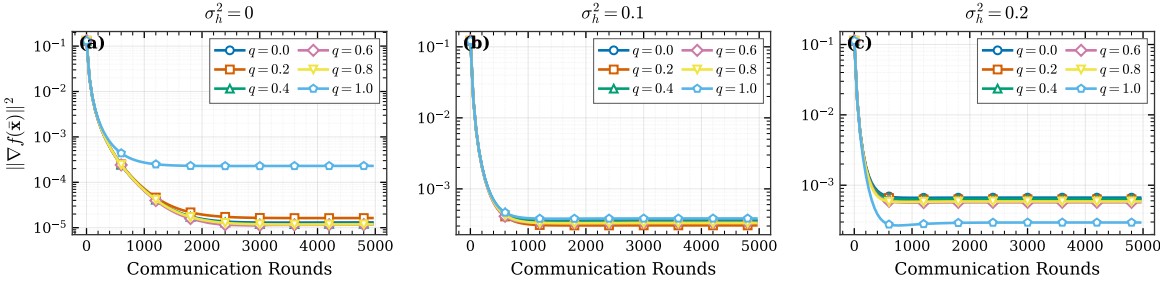

*Figure 8.* Convergence comparison of the BRED algorithm under Label-Flipping (LF) attack with Setting 2 in Table 7. The three subplots show the squared gradient norm $\|\nabla f(\bar{\mathbf{x}})\|^2$ versus communication rounds under different data heterogeneity levels: (a) $\sigma_h^2 = 0$, (b) $\sigma_h^2 = 0.1$, and (c) $\sigma_h^2 = 0.2$.

**Network sparsity.** Figures 9–14 illustrate the effect of network connectivity $k$ on convergence, where larger $k$ corresponds to denser topologies. In sparse networks ($k = 4$), convergence is more sensitive to the choice of $q$, with larger $q$ values achieving better performance due to enhanced global information exchange. As networks become denser ($k = 8, 12$), the performance gap among different $q$ values diminishes, and the algorithm becomes less sensitive to the mixing parameter. Notably, the optimal $q$ increases as network sparsity increases. These observations align with the theoretical predictions that sparser networks necessitate smaller $q$ to maintain convergence guarantees.

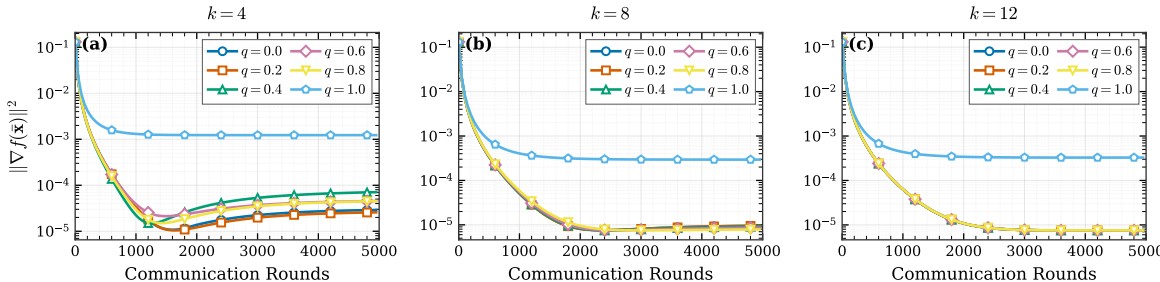

*Figure 9.* Effect of network sparsity on convergence under A Little Is Enough (ALIE) attack (8 attackers, $\sigma_h^2 = 0$). The three subplots show the squared gradient norm $\|\nabla f(\bar{\mathbf{x}})\|^2$ versus communication rounds under different network sparsity levels: (a) $k = 4$, (b) $k = 8$, (c) $k = 12$.

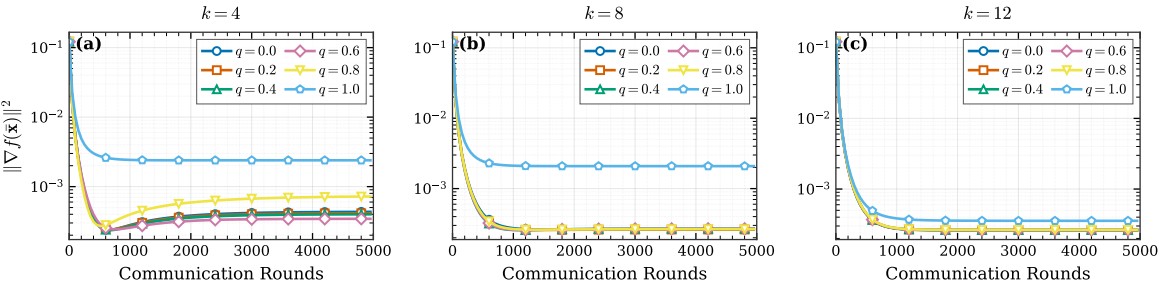

*Figure 10.* Effect of network sparsity on convergence under A Little Is Enough (ALIE) attack (8 attackers, $\sigma_h^2 = 0.1$). The three subplots show the squared gradient norm $\|\nabla f(\bar{\mathbf{x}})\|^2$ versus communication rounds under different network sparsity levels: (a) $k = 4$, (b) $k = 8$, (c) $k = 12$.

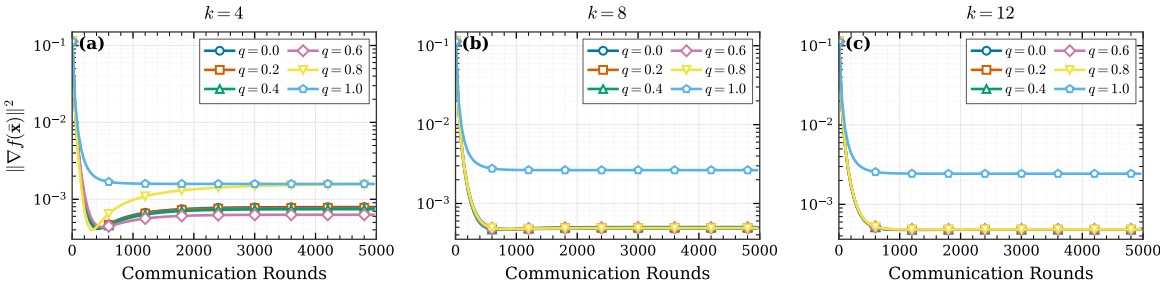

*Figure 11.* Effect of network sparsity on convergence under A Little Is Enough (ALIE) attack (8 attackers, $\sigma_h^2 = 0.2$). The three subplots show the squared gradient norm $\|\nabla f(\bar{\mathbf{x}})\|^2$ versus communication rounds under different network sparsity levels: (a) $k = 4$, (b) $k = 8$, (c) $k = 12$.

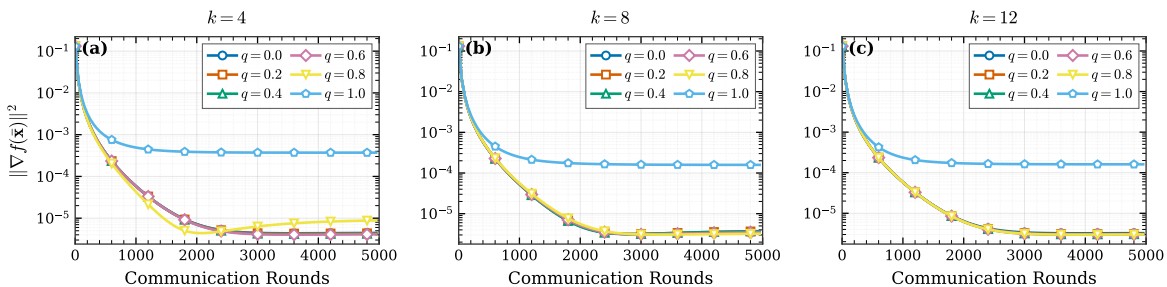

*Figure 12.* Effect of network sparsity on convergence under A Little Is Enough (ALIE) attack (4 attackers, $\sigma_h^2 = 0$). The three subplots show the squared gradient norm $\|\nabla f(\bar{\mathbf{x}})\|^2$ versus communication rounds under different network sparsity levels: (a) $k = 4$, (b) $k = 8$, (c) $k = 12$.

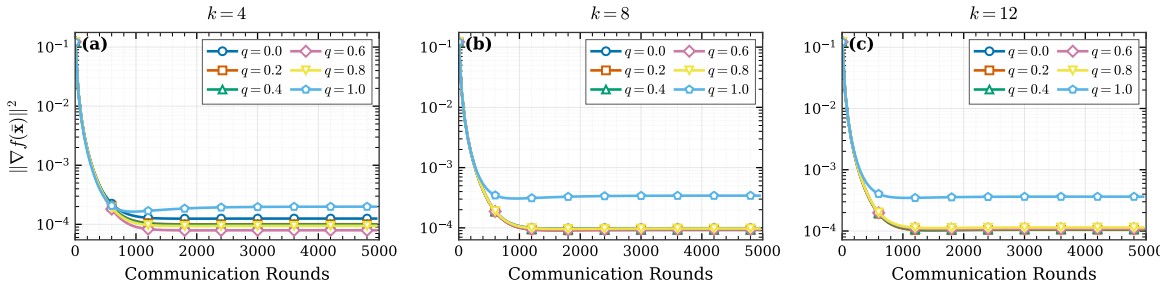

*Figure 13.* Effect of network sparsity on convergence under Label-Flipping (LF) attack (4 attackers, $\sigma_h^2 = 0.1$). The three subplots show the squared gradient norm $\|\nabla f(\bar{\mathbf{x}})\|^2$ versus communication rounds under different network sparsity levels: (a) $k = 4$, (b) $k = 8$, (c) $k = 12$.

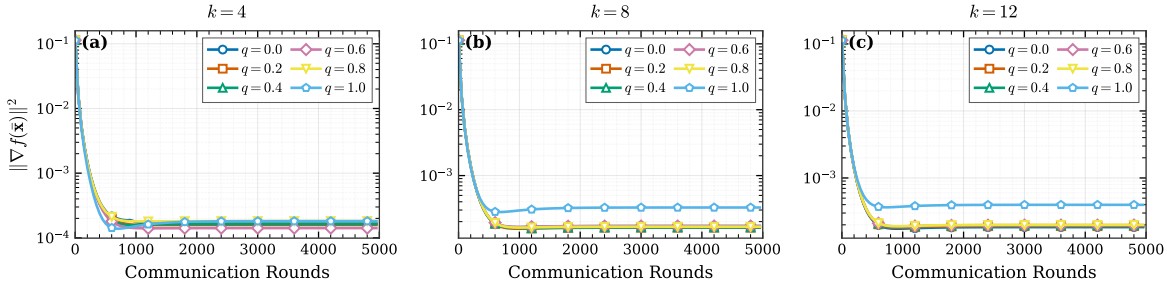

*Figure 14.* Effect of network sparsity on convergence under Label-Flipping (LF) attack (4 attackers, $\sigma_h^2 = 0.2$). The three subplots show the squared gradient norm $\|\nabla f(\bar{\mathbf{x}})\|^2$ versus communication rounds under different network sparsity levels: (a) $k = 4$, (b) $k = 8$, (c) $k = 12$.

**Fraction of Byzantine attackers.** Figures 15–19 compare the convergence under different Byzantine fractions $f \in \{4, 8\}$. As the number of attackers increases, the overall convergence performance degrades, and the optimal $q$ shifts toward smaller values to enhance global averaging and dilute malicious influence. Notably, $q = 1$ consistently underperforms regardless of the attack intensity, while moderate $q$ values (e.g., $q = 0.2$–$0.4$) achieve robust performance across different Byzantine fractions. These results suggest that practitioners should choose smaller $q$ values when facing higher attack intensities to maintain convergence guarantees.

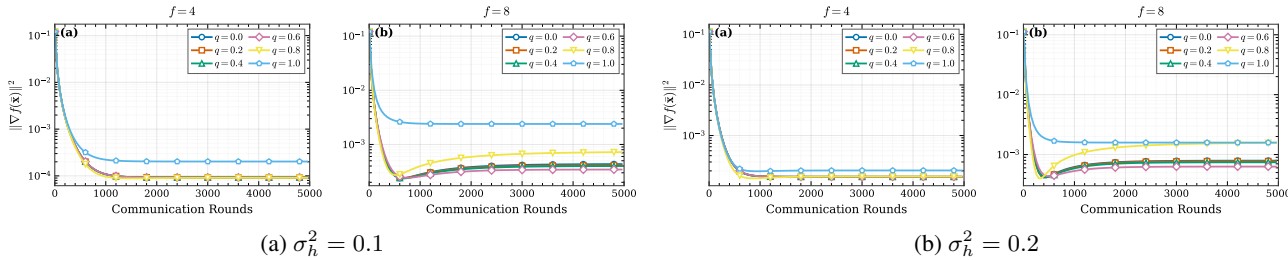

*Figure 15.* Effect of Byzantine number $f$ on convergence under A Little Is Enough (ALIE) attack ($k = 4$). Each panel compares $f = 4$ (left subplot) and $f = 8$ (right subplot) Byzantine attackers.

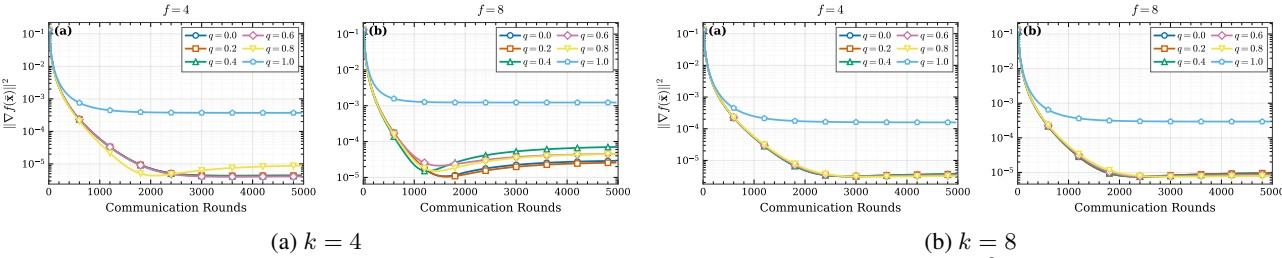

*Figure 16.* Effect of Byzantine number $f$ on convergence under A Little Is Enough (ALIE) attack ( $\sigma_h^2 = 0.0$). Each panel compares $f = 4$ (left subplot) and $f = 8$ (right subplot) Byzantine attackers.

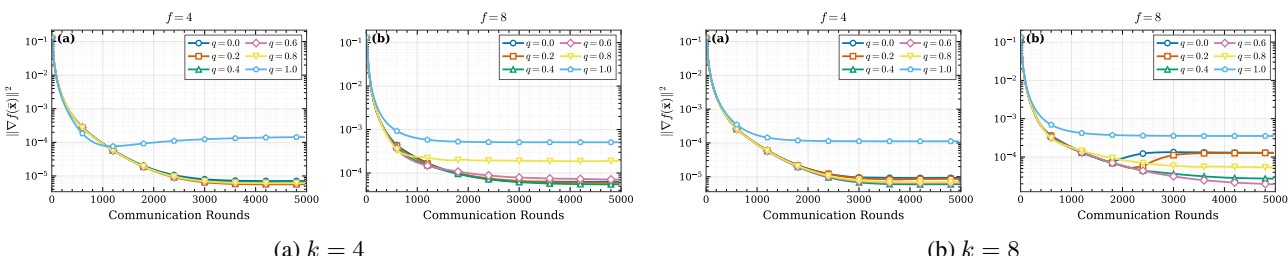

*Figure 17.* Effect of Byzantine number $f$ on convergence under Sign-Flipping (SF) attack ( $\sigma_h^2 = 0.0$). Each panel compares $f = 4$ (left subplot) and $f = 8$ (right subplot) Byzantine attackers.

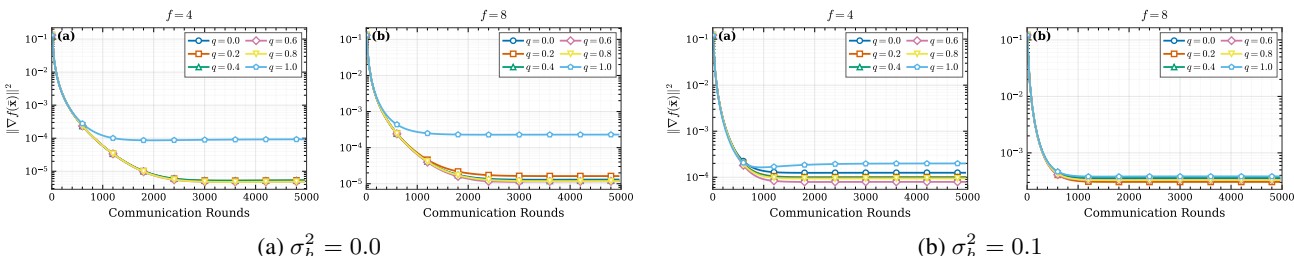

*Figure 18.* Effect of Byzantine number $f$ on convergence under Sign-Flipping (SF) attack ($k = 4$). Each panel compares $f = 4$ (left subplot) and $f = 8$ (right subplot) Byzantine attackers.

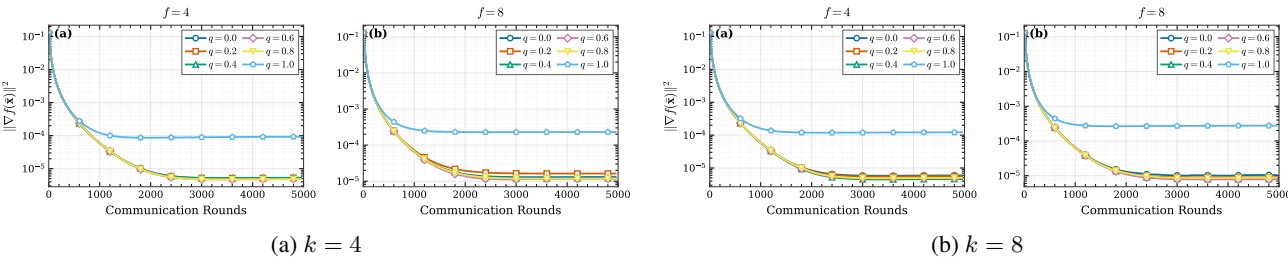

*Figure 19.* Effect of Byzantine number $f$ on convergence under Sign-Flipping (SF) attack ( $\sigma_h^2 = 0.1$). Each panel compares $f = 4$ (left subplot) and $f = 8$ (right subplot) Byzantine attackers.

### B.3. Additional empirical performance on MNIST dataset

In this sub-section, we present the training curves corresponding to Table 3 in the main paper. All experiments are conducted on MNIST with a CNN architecture, trained using SGD with learning rate $\eta = 0.01$, batch size $b = 128$, and $K = 1000$ communication rounds. We evaluate $n = 32$ clients arranged in $k$-Ring topologies with $k \in \{4, 8, 12\}$, where each honest node has either $b = 1$ or $b = 2$ Byzantine neighbors (corresponding to $f \in \{4, 8\}$ total Byzantine nodes). Data heterogeneity is simulated using Dirichlet distribution with $\alpha = 0.3$. For BRED and BRED-M, we set $q = 0.7$ and $\beta = 0.9$. Figures 20–23

show the test accuracy curves under ALIE attack with four robust aggregators (Krum, Median, Trimmed Mean, IOS) across all six network configurations. Shaded regions indicate standard deviation over three independent runs.

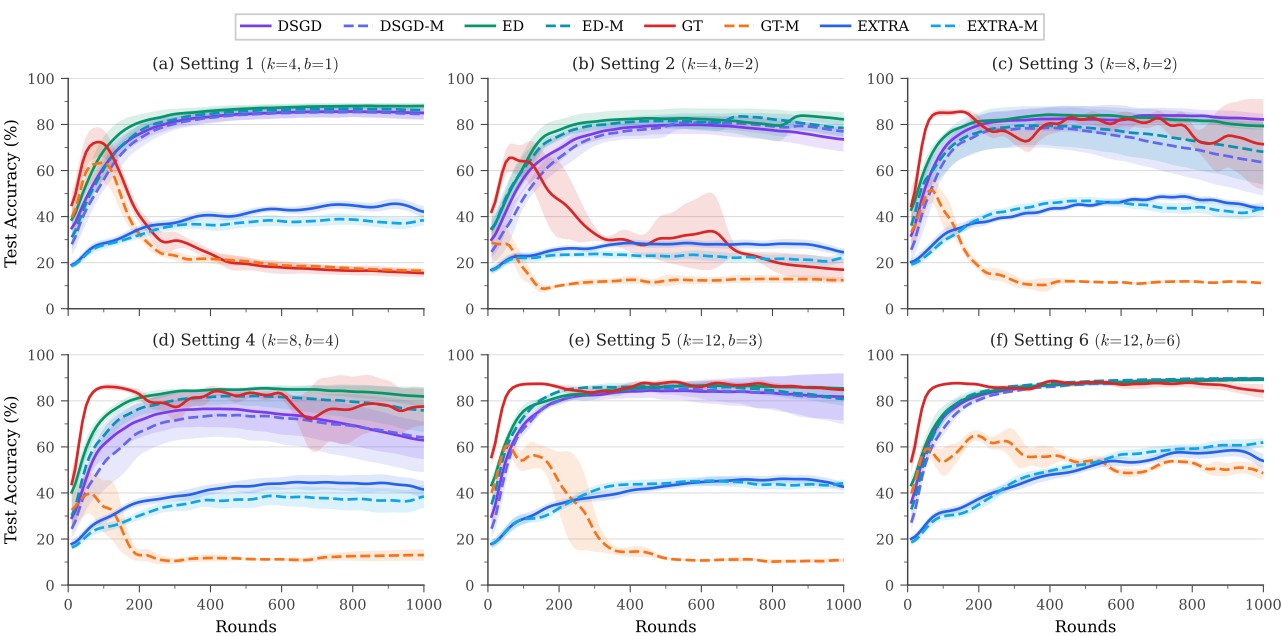

*Figure 20.* Test accuracy under ALIE attack with Krum aggregation on MNIST. Six network settings with varying topology ($k$) and Byzantine nodes ($b$) are evaluated. Shaded regions represent the standard deviation.

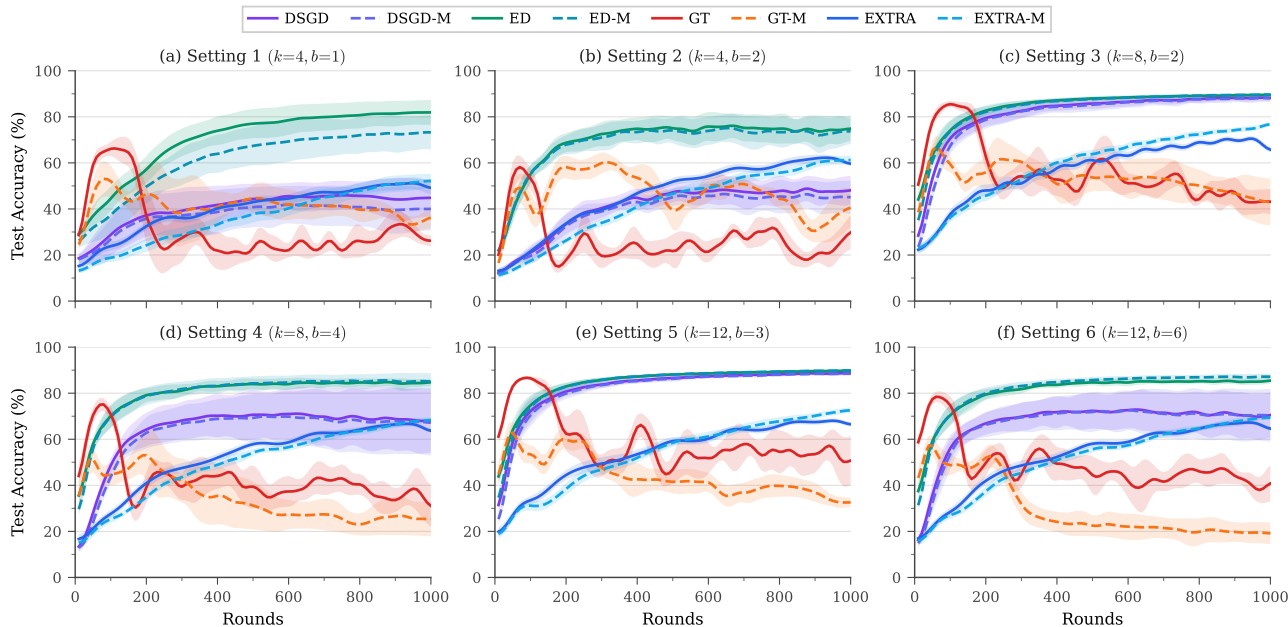

*Figure 21.* Test accuracy under ALIE attack with Median aggregation on MNIST. Six network settings with varying topology ($k$) and Byzantine nodes ($b$) are evaluated. Shaded regions represent the standard deviation.

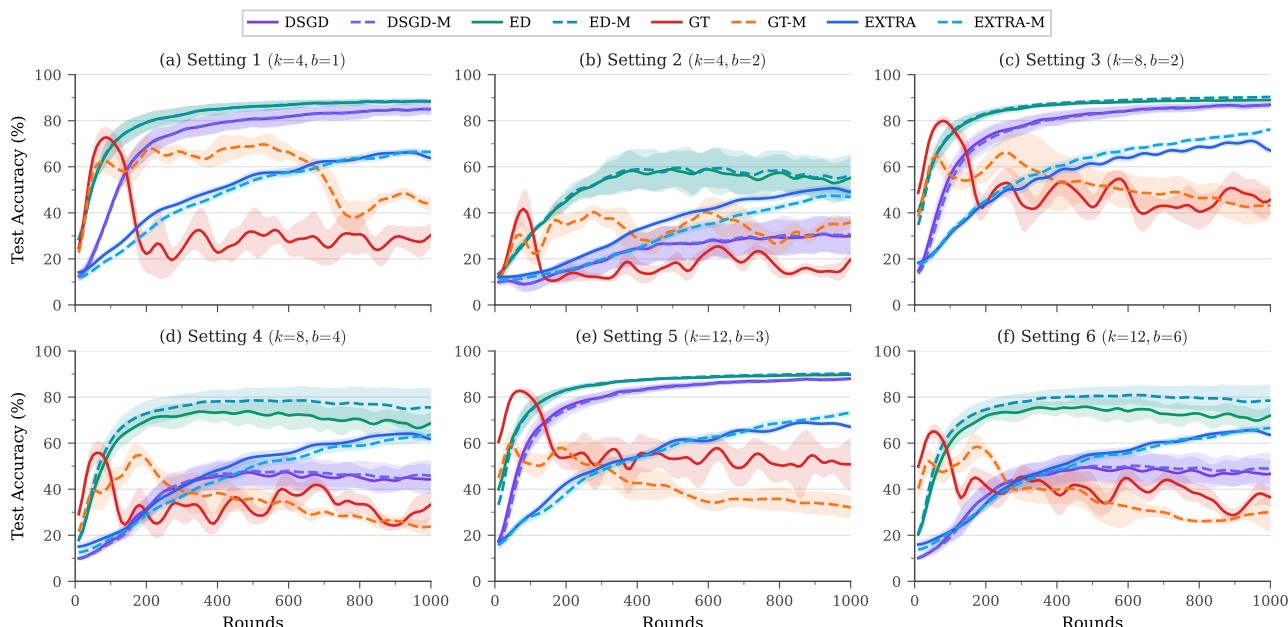

*Figure 22.* Test accuracy under ALIE attack with Trimmed Mean aggregation on MNIST. Six network settings with varying topology ($k$) and Byzantine nodes ($b$) are evaluated. Shaded regions represent the standard deviation.

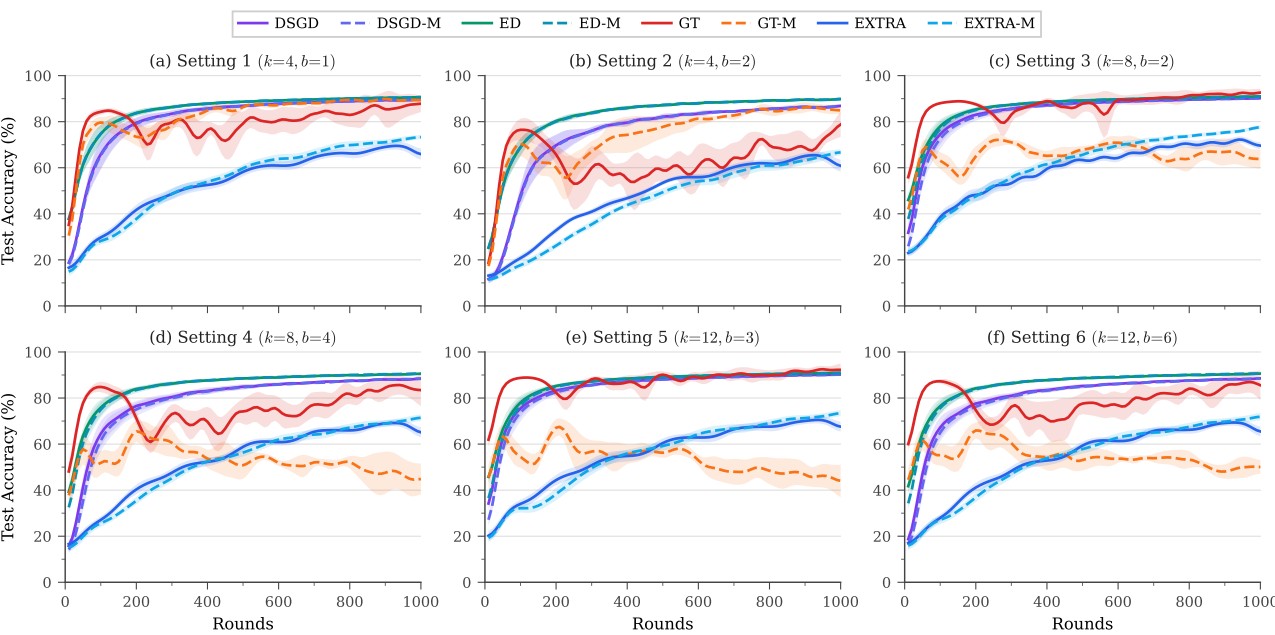

*Figure 23.* Test accuracy under ALIE attack with IOS aggregation on MNIST. Six network settings with varying topology ($k$) and Byzantine nodes ($b$) are evaluated. Shaded regions represent the standard deviation.

## B.4. Ablation Studies

### B.4.1. EXPERIMENT WITH DIFFERENT LEVELS OF DATA HETEROGENEITY

We investigate the impact of data heterogeneity on algorithm robustness under Byzantine attacks by varying the Dirichlet concentration parameter $\alpha \in \{0.5, 1.0, 5.0\}$, where smaller values indicate more heterogeneous data distributions. As shown in Figures 24-29, ED and ED-M consistently achieve the highest test accuracy across all heterogeneity levels, reaching approximately 90% at $\alpha = 5.0$. In contrast, gradient tracking methods (GT, GT-M) exhibit poor robustness under

coordinate-wise median aggregation, with accuracy remaining below 41% even under mild heterogeneity. These results confirm that higher data heterogeneity amplifies the challenges posed by Byzantine attacks, while ED-based methods maintain superior robustness across varying conditions.

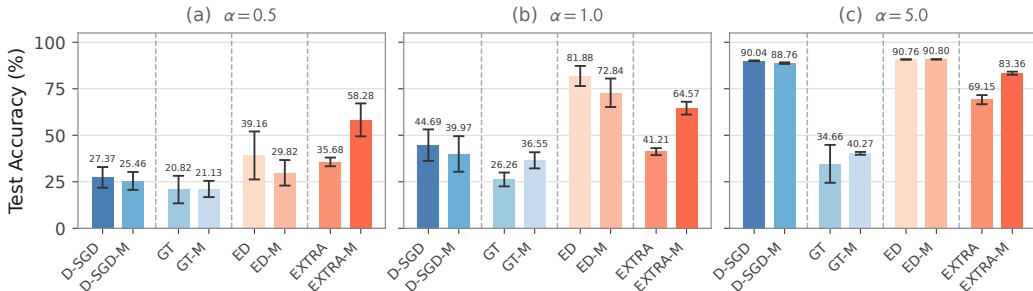

*Figure 24.* Test accuracy (%) under varying data heterogeneity levels for Setting 1 ($k = 4, b = 1$) with coordinate-wise median aggregation under ALIE attack. Error bars represent standard deviation across 3 independent runs.

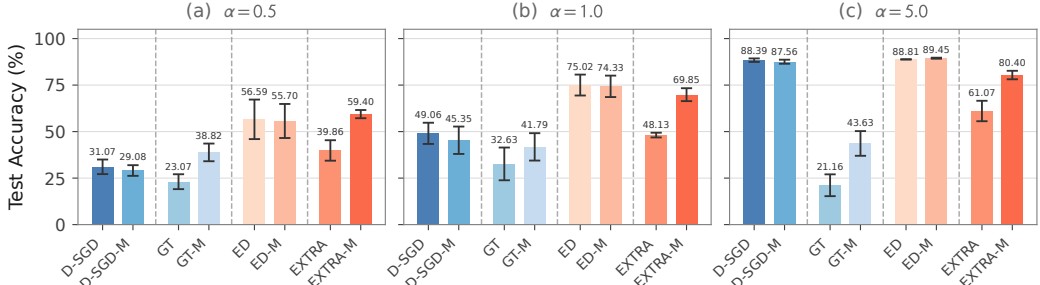

*Figure 25.* Test accuracy (%) under varying data heterogeneity levels for Setting 2 ($k = 4, b = 2$) with coordinate-wise median aggregation under ALIE attack. Error bars represent standard deviation across 3 independent runs.

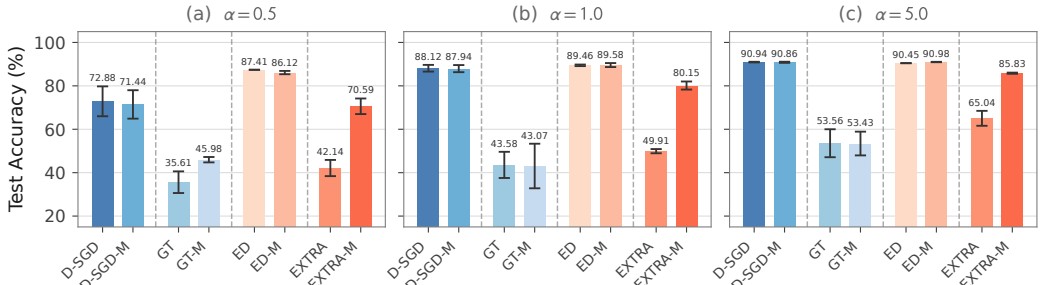

*Figure 26.* Test accuracy (%) under varying data heterogeneity levels for Setting 3 ($k = 8, b = 2$) with coordinate-wise median aggregation under ALIE attack. Error bars represent standard deviation across 3 independent runs.

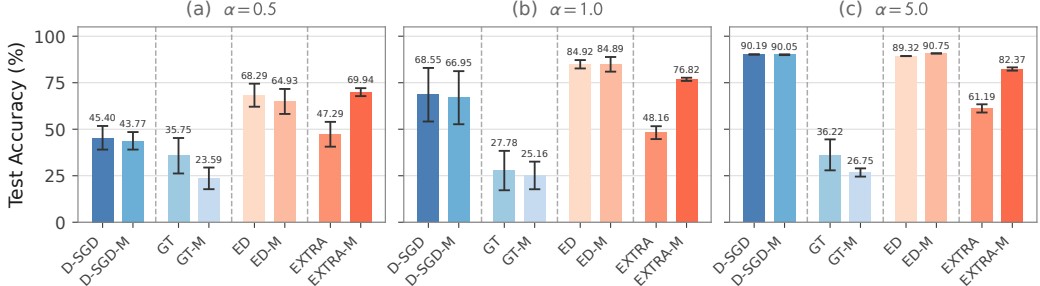

*Figure 27.* Test accuracy (%) under varying data heterogeneity levels for Setting 4 ($k = 8, b = 4$) with coordinate-wise median aggregation under ALIE attack. Error bars represent standard deviation across 3 independent runs.

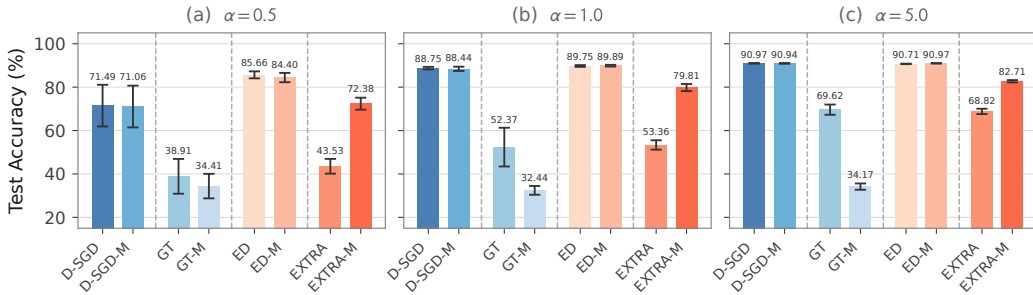

*Figure 28.* Test accuracy (%) under varying data heterogeneity levels for Setting 5 ($k = 12$, $b = 3$) with coordinate-wise median aggregation under ALIE attack. Error bars represent standard deviation across 3 independent runs.

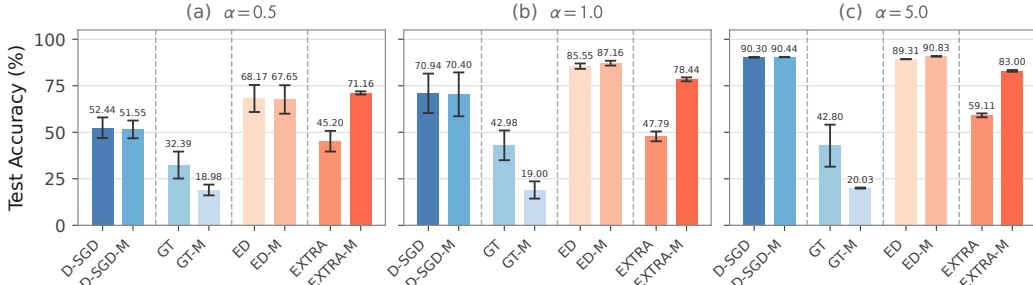

*Figure 29.* Test accuracy (%) under varying data heterogeneity levels for Setting 6 ($k = 12$, $b = 6$) with coordinate-wise median aggregation under ALIE attack. Error bars represent standard deviation across 3 independent runs.

### B.4.2. EXPERIMENT WITH DIFFERENT NUMBERS OF CLIENTS

In this subsection, we evaluate the effect of varying number of participating clients on algorithm performance. We test under a fully connected network topology with $n \in \{16, 24, 32\}$ clients. For each configuration, we maintain a fixed ratio of Byzantine attackers: Figure 30 shows performance with a lower attacker ratio ($\delta = 0.125$), while Figure 31 shows performance with a higher attacker ratio ($\delta = 0.25$). It can be observed that ED and ED-M consistently outperform other methods across all client configurations and attacker ratios, demonstrating strong scalability and robustness.

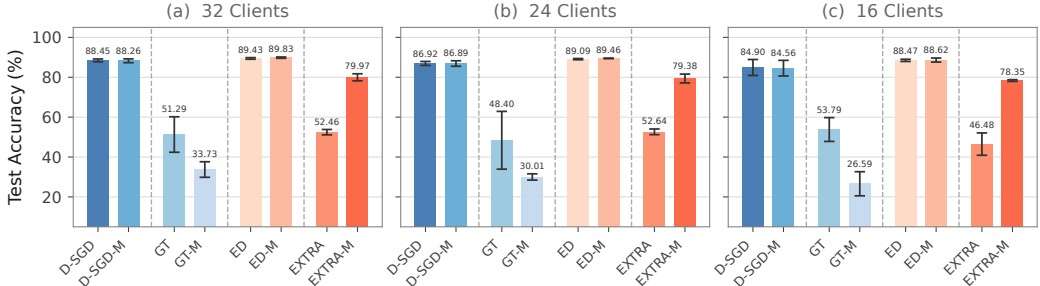

*Figure 30.* Test accuracy (%) under varying number of clients ($n = 16, 24, 32$) with low attacker ratio ($\delta = 0.125$) using coordinate-wise median aggregation under ALIE attack. Error bars represent standard deviation across 3 independent runs.

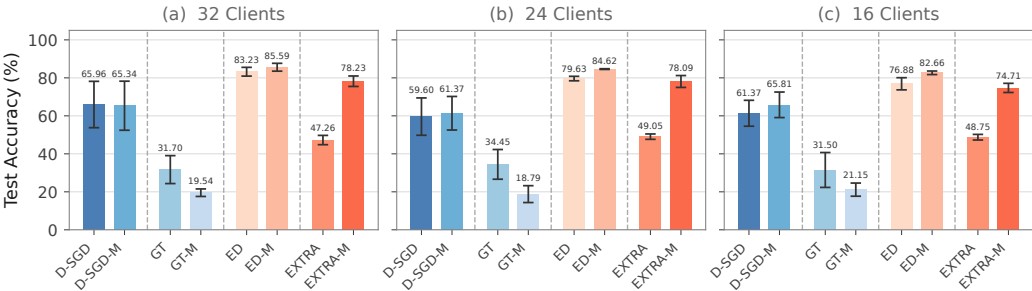

*Figure 31.* Test accuracy (%) under varying number of clients ($n = 16, 24, 32$) with high attacker ratio ($\delta = 0.25$) using coordinate-wise median aggregation under ALIE attack. Error bars represent standard deviation across 3 independent runs.

### B.4.3. EXPERIMENT WITH DIFFERENT NUMBERS OF LOCAL EPOCHS

In this subsection, we evaluate the effect of varying the number of local epochs on algorithm performance. Local epochs refer to the number of local SGD updates each node performs before communicating with neighbors. We test three configurations: $E \in \{1, 5, 10\}$, where $E = 1$ corresponds to standard decentralized SGD with communication after every gradient step. As shown in Figures 32-37, ED and ED-M consistently outperform other algorithms across all local epoch settings, demonstrating their robustness to different communication frequencies under Byzantine attacks.

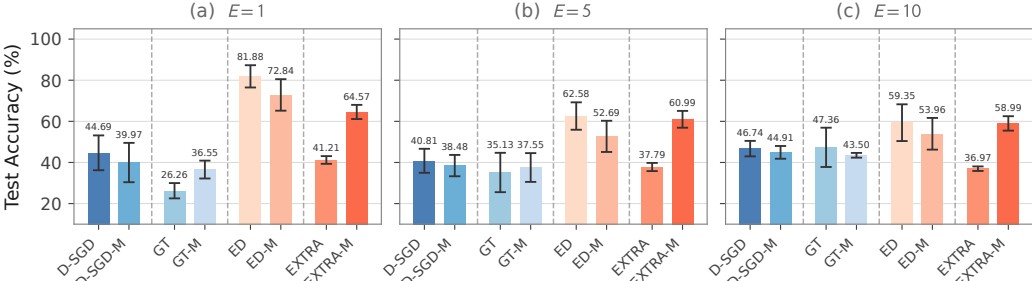

*Figure 32.* Test accuracy (%) under varying local epochs ($E = 1, 5, 10$) for Setting 1 ($k = 4$, $b = 1$) with median aggregation under ALIE attack. Error bars show standard deviation across 3 runs.

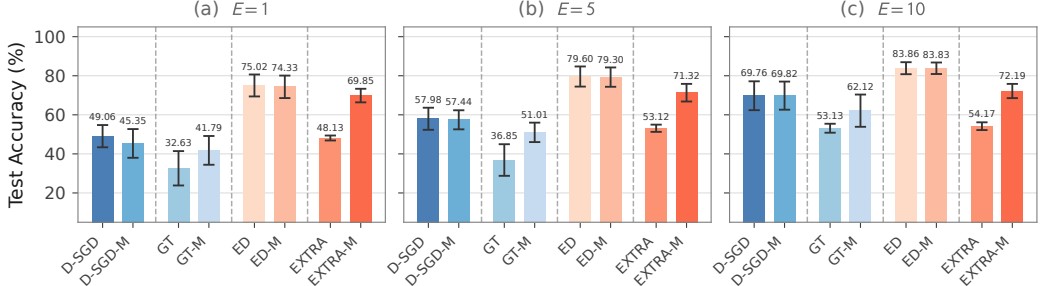

*Figure 33.* Test accuracy (%) under varying local epochs ($E = 1, 5, 10$) for Setting 2 ($k = 4$, $b = 2$) with median aggregation under ALIE attack. Error bars show standard deviation across 3 runs.

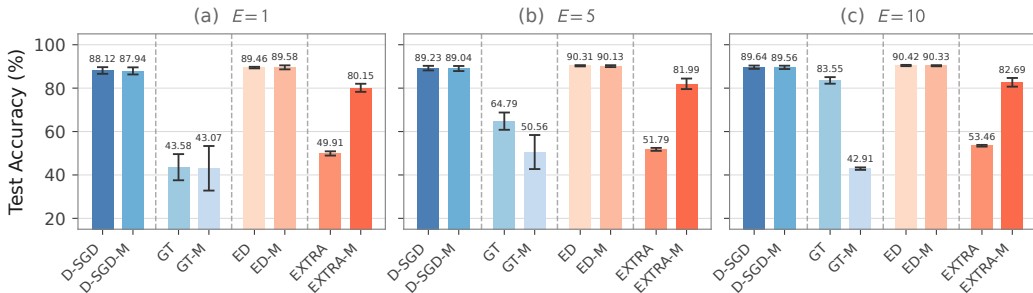

*Figure 34.* Test accuracy (%) under varying local epochs ($E = 1, 5, 10$) for Setting 3 ($k = 8$, $b = 2$) with median aggregation under ALIE attack. Error bars show standard deviation across 3 runs.

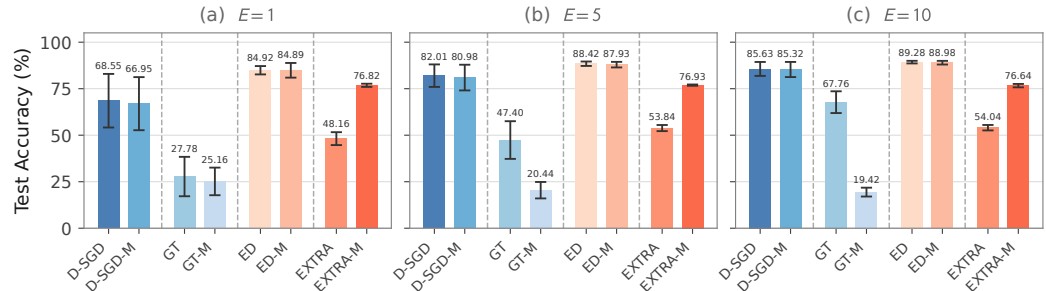

*Figure 35.* Test accuracy (%) under varying local epochs ($E = 1, 5, 10$) for Setting 4 ($k = 8$, $b = 4$) with median aggregation under ALIE attack. Error bars show standard deviation across 3 runs.

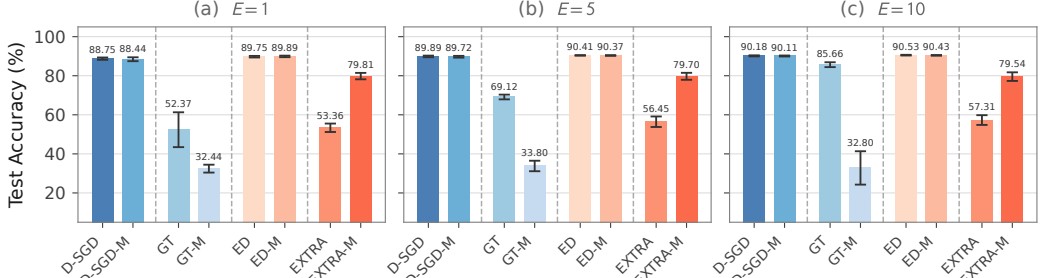

*Figure 36.* Test accuracy (%) under varying local epochs ($E = 1, 5, 10$) for Setting 5 ($k = 12$, $b = 3$) with median aggregation under ALIE attack. Error bars show standard deviation across 3 runs.

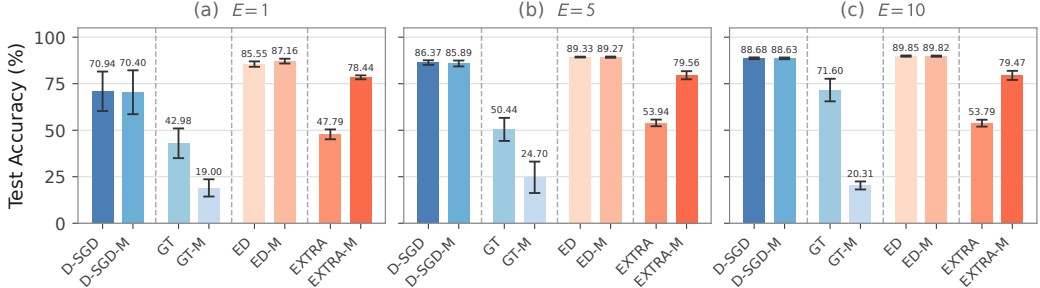

*Figure 37.* Test accuracy (%) under varying local epochs ($E = 1, 5, 10$) for Setting 6 ($k = 12$, $b = 6$) with median aggregation under ALIE attack. Error bars show standard deviation across 3 runs.

B.4.4. DISCUSSION ON THE INFLUENCE OF COEFFICIENT $q$

This subsection examines the impact of the coefficient $q$. Our theoretical analysis demonstrates that setting $q < 1$ is necessary to ensure robustness against Byzantine adversaries. Consequently, we investigate the optimal selection of $q$ under varying conditions, including different levels of data heterogeneity, attacker fractions, and network sparsity.

Figures 38-39 illustrate the test accuracy with respect to the parameter $q$ across six different network topologies under ALIE attack with Median aggregation on MNIST. We observe that for all settings, the optimal $q^\star$ consistently lies within the range $[0.6, 0.8]$ (highlighted region), regardless of the network connectivity $k$ or the number of Byzantine neighbors $b$ per honest client. Notably, the accuracy curves remain relatively flat within this interval, indicating that the performance is not highly sensitive to the precise choice of $q$. This robustness is particularly valuable in practice, as it eliminates the need for extensive hyperparameter tuning. Based on these results, we recommend a universal choice of $q = 0.7$, which achieves near-optimal performance across all tested configurations.

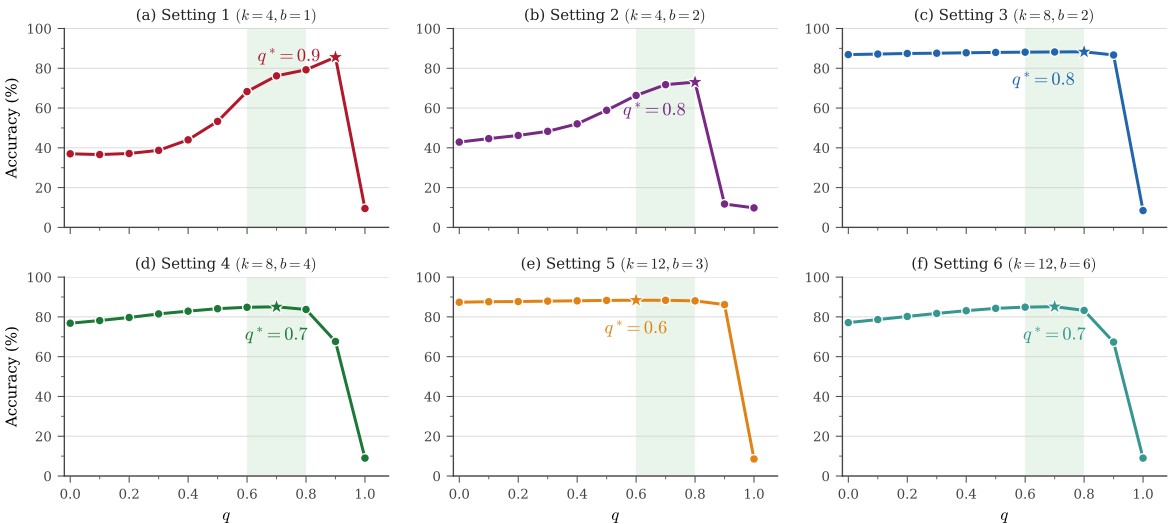

*Figure 38.* Sensitivity of $q$ selection on MNIST (ALIE attack, Median aggregation). The optimal $q^\star$ lies within $[0.6, 0.8]$ (shaded) across most network settings, indicating low sensitivity to hyperparameter tuning. Settings vary by connectivity $k$ and Byzantine neighbors $b$ per honest client.

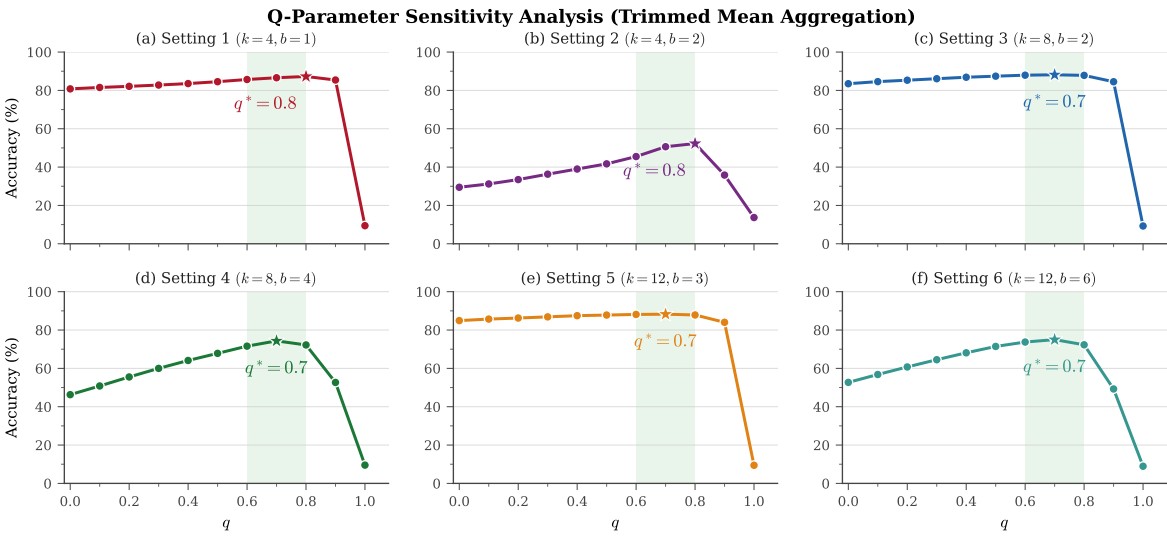

*Figure 39.* Sensitivity of $q$ selection on MNIST (ALIE attack, Trimmed-mean aggregation). The optimal $q^\star$ lies within $[0.6, 0.8]$ (shaded) across most network settings, indicating low sensitivity to hyperparameter tuning. Settings vary by connectivity $k$ and Byzantine neighbors $b$ per honest client.

# C. Preliminaries

In this section, we establish the unified primal-dual framework underlying Algorithm 1 and derive the key mathematical tools for our convergence analysis. We begin by introducing the notation conventions used throughout the appendix. We then present the decentralized optimization framework with scaled dual ascent, demonstrating how it generalizes existing bias-correction methods such as Exact Diffusion, EXTRA, and gradient tracking variants. Next, we derive the matrix transformations that convert the update equations into a canonical form amenable to analysis. Finally, we establish a fundamental factorization lemma for the transition matrix, which provides explicit bounds on the contraction rate and serves as the foundation for the convergence proofs in subsequent sections.

**Notation convention.** Throughout this section, boldface lowercase letters represent stacked vectors across all $n_h$ honest agents, while boldface uppercase letters denote block matrices operating on these stacked vectors. Specifically:

- Agents: We denote the set of honest agents by $\mathcal{H} \subseteq [n]$ with cardinality $n_h = |\mathcal{H}| = (1 - \delta)n$. The set of Byzantine agents is denoted by $\mathcal{B} := [n] \setminus \mathcal{H}$.
- Stacked Vectors: For local variables $x_i^k \in \mathbb{R}^d$ (where $i \in \mathcal{H}$), we define the stacked vector $\mathbf{x}^k := \text{col}\{x_i^k\}_{i \in \mathcal{H}} \in \mathbb{R}^{dn_h}$.
- Consensus Vector: The stacked average vector is denoted by $\bar{\mathbf{x}}^k := \mathbf{1}_{n_h} \otimes \bar{x}^k \in \mathbb{R}^{dn_h}$, where $\bar{x}^k := \frac{1}{n_h} \sum_{i \in \mathcal{H}} x_i^k$ and $\mathbf{1}_{n_h}$ is the all-ones vector in $\mathbb{R}^{n_h}$.
- Gradients: The stacked gradient vector is defined as $\nabla \mathbf{f}_{\mathcal{H}}(\mathbf{x}^k) := \text{col}\{\nabla f_i(x_i^k)\}_{i \in \mathcal{H}} \in \mathbb{R}^{dn_h}$.
- Matrices: We denote the Kronecker product expansion of the mixing matrix $W \in \mathbb{R}^{n_h \times n_h}$ as $\mathbf{W} := W \otimes I_d \in \mathbb{R}^{dn_h \times dn_h}$.
- Norms: We use $\|\cdot\|$ to denote the $\ell_2$-norm for vectors and the spectral norm for matrices.

## C.1. Decentralized Optimization with Scaled Dual Ascent

The decentralized framework with scaled dual ascent (12) takes the form:

$$
\begin{aligned}
\mathbf{x}^{k+1} &= \mathbf{C}\mathbf{x}^k - \eta \mathbf{A} \nabla \mathbf{f}_{\mathcal{H}}\left(\mathbf{x}^k\right) - \mathbf{B}\mathbf{d}^k, \\
\mathbf{d}^{k+1} &= q\left(\mathbf{d}^k + \mathbf{B}\mathbf{x}^{k+1}\right),
\end{aligned}
\tag{21}
$$

where $\eta > 0$ denotes the learning rate, and $q := \frac{1}{1+\tau} \in (0, 1)$ is the dual scaling coefficient.

Then, we show how the decentralized framework with scaled dual ascent extends to existing bias-correction methods. First, from (21), we obtain that

$$
\begin{aligned}
\mathbf{x}^{k+1} - q\mathbf{x}^k &= \mathbf{C}(\mathbf{x}^k - q\mathbf{x}^{k-1}) - \eta \mathbf{A}(\nabla \mathbf{f}_{\mathcal{H}}\left(\mathbf{x}^k\right) - q\nabla \mathbf{f}_{\mathcal{H}}(\mathbf{x}^{k-1})) - \mathbf{B}\mathbf{d}^k - q\mathbf{B}\mathbf{d}^{k-1} \\
&= (\mathbf{C} - q\mathbf{B}^2)\mathbf{x}^k - q\mathbf{C}\mathbf{x}^{k-1} - \eta \mathbf{A}(\nabla \mathbf{f}_{\mathcal{H}}\left(\mathbf{x}^k\right) - q\nabla \mathbf{f}_{\mathcal{H}}(\mathbf{x}^{k-1})).
\end{aligned}
\tag{22}
$$

Thus, for $k \geq 1$, we have

$$
\mathbf{x}^{k+1} = (\mathbf{C} - q\mathbf{B}^2 + q\mathbf{I})\mathbf{x}^k - q\mathbf{C}\mathbf{x}^{k-1} - \eta \mathbf{A}(\nabla \mathbf{f}_{\mathcal{H}}\left(\mathbf{x}^k\right) - q\nabla \mathbf{f}_{\mathcal{H}}(\mathbf{x}^{k-1})),
\tag{23}
$$

with $\mathbf{x}^1 = \mathbf{C}\mathbf{x}^0 - \eta \mathbf{A} \nabla \mathbf{f}_{\mathcal{H}}(\mathbf{x}^0)$.

**Relationship with previous decentralized algorithms.** As summarized in Table 2, this framework unifies various bias-correction methods. By selecting specific matrices $\mathbf{A}$, $\mathbf{B}$, $\mathbf{C}$ and setting $q = 1$, we recover ED (Tang et al., 2018), EXTRA (Shi et al., 2015), ATC-GT, and other GT variants (Xu et al., 2015; Di Lorenzo & Scutari, 2016; Nedic et al., 2017).

- **Exact Diffusion:** Applying $\mathbf{A} = \mathbf{W}$, $\mathbf{B} = (\mathbf{I} - \mathbf{W})^{1/2}$ and $\mathbf{C} = \mathbf{W}$, we obtain that:

$$
\mathbf{x}^{k+1} = \mathbf{W}\left(2\mathbf{x}^k - \mathbf{x}^{k-1} - \eta(\nabla \mathbf{f}_{\mathcal{H}}\left(\mathbf{x}^k\right) - \nabla \mathbf{f}_{\mathcal{H}}(\mathbf{x}^{k-1}))\right),
\tag{24}
$$

with $\mathbf{x}^1 = \mathbf{W}(\mathbf{x}^0 - \eta \nabla \mathbf{f}_{\mathcal{H}}(\mathbf{x}^0))$.

- **EXTRA:** Applying $\mathbf{A} = \mathbf{I}$, $\mathbf{B} = (\mathbf{I} - \mathbf{W})^{1/2}$ and $\mathbf{C} = \mathbf{W}$, we obtain that:

$$
\mathbf{x}^{k+1} = \mathbf{W}\left(2\mathbf{x}^k - \mathbf{x}^{k-1}\right) - \eta\left(\nabla \mathbf{f}_{\mathcal{H}}\left(\mathbf{x}^k\right) - \nabla \mathbf{f}_{\mathcal{H}}(\mathbf{x}^{k-1})\right),
\tag{25}
$$

with $\mathbf{x}^1 = \mathbf{W}\mathbf{x}^0 - \eta \nabla \mathbf{f}_{\mathcal{H}}(\mathbf{x}^0)$.

- **ATC-GT:** Applying $\mathbf{A} = \mathbf{W}^2$, $\mathbf{B} = \mathbf{I} - \mathbf{W}$ and $\mathbf{C} = \mathbf{W}^2$, we obtain that:

$$\mathbf{x}^{k+1} = 2\mathbf{W}\mathbf{x}^k - \mathbf{W}^2\mathbf{x}^{k-1} - \eta\mathbf{W}^2\big(\nabla\mathbf{f}_{\mathcal{H}}\left(\mathbf{x}^k\right) - \nabla\mathbf{f}_{\mathcal{H}}(\mathbf{x}^{k-1})\big), \tag{26}$$

with $\mathbf{x}^1 = \mathbf{W}^2\mathbf{x}^0 - \eta\mathbf{W}^2\nabla\mathbf{f}_{\mathcal{H}}(\mathbf{x}^0)$.

- **Semi-ATC-GT:** Applying $\mathbf{A} = \mathbf{W}$, $\mathbf{B} = \mathbf{I} - \mathbf{W}$ and $\mathbf{C} = \mathbf{W}^2$, we obtain that:

$$\mathbf{x}^{k+1} = 2\mathbf{W}\mathbf{x}^k - \mathbf{W}^2\mathbf{x}^{k-1} - \eta\mathbf{W}\big(\nabla\mathbf{f}_{\mathcal{H}}\left(\mathbf{x}^k\right) - \nabla\mathbf{f}_{\mathcal{H}}(\mathbf{x}^{k-1})\big), \tag{27}$$

with $\mathbf{x}^1 = \mathbf{W}^2\mathbf{x}^0 - \eta\mathbf{W}\nabla\mathbf{f}_{\mathcal{H}}(\mathbf{x}^0)$.

- **Non-ATC-GT:** Applying $\mathbf{A} = \mathbf{W}$, $\mathbf{B} = \mathbf{I} - \mathbf{W}$ and $\mathbf{C} = \mathbf{W}^2$, we obtain that:

$$\mathbf{x}^{k+1} = 2\mathbf{W}\mathbf{x}^k - \mathbf{W}^2\mathbf{x}^{k-1} - \eta\big(\nabla\mathbf{f}_{\mathcal{H}}\left(\mathbf{x}^k\right) - \nabla\mathbf{f}_{\mathcal{H}}(\mathbf{x}^{k-1})\big), \tag{28}$$

with $\mathbf{x}^1 = \mathbf{W}^2\mathbf{x}^0 - \eta\nabla\mathbf{f}_{\mathcal{H}}(\mathbf{x}^0)$.

### C.2. Transformation

This subsection derives the key transformations that convert the update equations into a canonical form suitable for convergence analysis. We first introduce auxiliary variables to reformulate the primal-dual recursion, then show how BRED or BRED-M arises as a specific instantiation. We subsequently extend the framework to the Byzantine-resilient setting by incorporating the aggregation error, and finally exploit the spectral structure of the combination matrices to obtain a compact representation.

We denote the stochastic gradient as $\mathbf{g}^k := \nabla\mathbf{F}_{\mathcal{H}}(\mathbf{x}^k; \boldsymbol{\varphi}^k) \in \mathbb{R}^{dn_h}$. For the momentum-based variant (BRED-M), we instead utilize the momentum estimator $\mathbf{m}^k$, for $k \geq 0$ and $\mathbf{m}^{-1} = \mathbf{0}$, defined recursively as:

$$\mathbf{m}^k = \beta\mathbf{m}^{k-1} + (1 - \beta)\mathbf{g}^k. \tag{29}$$

Note that for the remainder of the analysis, we focus on the formulation using $\mathbf{g}^k$ for simplicity. The analysis can be extended to the momentum case following a similar logic, but we omit it for brevity.

Substituting $\mathbf{g}^k$ (or $\mathbf{m}^k$ for BRED-M) into (21), we obtain the following update equations:

$$\mathbf{x}^{k+1} = \mathbf{C}\,\mathbf{x}^k - \eta\,\mathbf{A}\,\mathbf{g}^k - \mathbf{B}\,\mathbf{d}^k, \tag{30a}$$

$$\mathbf{d}^{k+1} = q\left(\mathbf{d}^k + \mathbf{B}\,\mathbf{x}^{k+1}\right), \tag{30b}$$

where $\mathbf{w}^k := \mathbf{g}^k - \nabla\mathbf{f}_{\mathcal{H}}(\mathbf{x}^k) \in \mathbb{R}^{dn_h}$ represents the stochastic gradient noise.

To facilitate the analysis, we introduce an auxiliary variable $\mathbf{s}^k \in \mathbb{R}^{dn_h}$ defined as:

$$\mathbf{s}^k := \mathbf{B}\left(\mathbf{d}^k - q\,\mathbf{B}\,\mathbf{x}^k\right) + \eta\,\mathbf{A}\,\mathbf{N}^k. \tag{31}$$

Let $\{\mathbf{N}^k\}_{k \geq 0} \subset \mathbb{R}^{dn_h}$ be a sequence satisfying the initialization condition $\mathbf{N}^0 = \nabla\mathbf{f}_{\mathcal{H}}(\bar{\mathbf{x}}^0)$, or alternatively $\mathbf{N}^0 = (1 - \beta)\nabla\mathbf{f}_{\mathcal{H}}(\bar{\mathbf{x}}^0)$ for momentum-based schemes. Substituting $\mathbf{s}^k$ into (30), we derive the following recursion for all $k \geq 0$:

$$\mathbf{x}^{k+1} = (\mathbf{C} - q\,\mathbf{B}^2)\,\mathbf{x}^k - \eta\,\mathbf{A}\left(\mathbf{g}^k - \mathbf{N}^k\right) - \mathbf{s}^k, \tag{32a}$$

$$\mathbf{s}^{k+1} = q\,\mathbf{s}^k + q^2\,\mathbf{B}^2\,\mathbf{x}^k + \eta\,\mathbf{A}\left(\mathbf{N}^{k+1} - q\,\mathbf{N}^k\right). \tag{32b}$$

Leveraging the unbiasedness property $\nabla\mathbf{f}_{\mathcal{H}}(\mathbf{x}^k) = \mathbb{E}[\mathbf{g}^k]$, we decompose the stochastic noise term in (32) to obtain:

$$\mathbf{x}^{k+1} = (\mathbf{C} - q\,\mathbf{B}^2)\,\mathbf{x}^k - \eta\,\mathbf{A}\left(\nabla\mathbf{f}_{\mathcal{H}}(\mathbf{x}^k) - \mathbf{N}^k\right) - \mathbf{s}^k - \eta\,\mathbf{A}\,\mathbf{w}^k, \tag{33a}$$

$$\mathbf{s}^{k+1} = q\,\mathbf{s}^k + q^2\,\mathbf{B}^2\,\mathbf{x}^k + \eta\,\mathbf{A}\left(\mathbf{N}^{k+1} - q\,\mathbf{N}^k\right). \tag{33b}$$

The proposed algorithm BRED (Algorithm 1) can be viewed as a specific instantiation of framework (33). By setting $\mathbf{A} = \mathbf{W}$, $\mathbf{C} = \mathbf{W}$, and $\mathbf{B} = (\mathbf{I} - \mathbf{W})^{1/2}$, we recover the update dynamics of the proposed algorithm:

$$\mathbf{x}^{k+1} = (\mathbf{W} + q\,(\mathbf{W} - \mathbf{I}))\,\mathbf{x}^k - \eta\,\mathbf{W}\left(\nabla\mathbf{f}_{\mathcal{H}}(\mathbf{x}^k) - \mathbf{N}^k\right) - \mathbf{s}^k - \eta\,\mathbf{W}\,\mathbf{w}^k, \tag{34a}$$

$$\mathbf{s}^{k+1} = q\,\mathbf{s}^k + q^2\,(\mathbf{I} - \mathbf{W})\,\mathbf{x}^k + \eta\,\mathbf{W}\left(\mathbf{N}^{k+1} - q\,\mathbf{N}^k\right). \tag{34b}$$

**Extension to Byzantine-resilient setting.** The preceding analysis establishes the scaled dual ascent framework under the assumption that all agents are honest and faithfully follow the prescribed update rules. In this idealized setting, the mixing matrix $\mathbf{W}$ aggregates neighboring iterates through a simple weighted average. However, in the presence of Byzantine adversaries, i.e., agents that may send arbitrary or malicious messages, this averaging operation must be replaced by a robust aggregation rule $\mathcal{A}(\cdot)$ that can tolerate adversarial inputs.

To quantify the impact of Byzantine agents on the convergence analysis for BRED in Algorithm 1, we introduce the aggregation error term $\boldsymbol{\xi}^k \in \mathbb{R}^{dn_h}$, which captures the discrepancy between the robust aggregator output and the ideal honest-only aggregation. Here, $x_i^{k+\frac{1}{2}}$ denotes the pre-aggregation iterate computed by each agent $i$ before the communication and aggregation step:

$$x_i^{k+\frac{1}{2}} := (1+q)\, x_i^k - q\, x_i^{k-1} - \eta \left( \nabla F_i(x_i^k; \varphi_i^k) - q\, \nabla F_i(x_i^{k-1}; \varphi_i^{k-1}) \right). \tag{35}$$

The aggregation error is then defined as:

$$\boldsymbol{\xi}^k := col\{\xi_i^k\}_{i \in \mathcal{H}}, \qquad \xi_i^k := \frac{1}{\eta} \left( R_i^{k+\frac{1}{2}} - \bar{x}_i^{k+\frac{1}{2}} \right) = \frac{1}{\eta} \left( \mathcal{A}\left( x_i^{k+\frac{1}{2}}, \{x_m^{k+\frac{1}{2}}\}_{m \in \mathcal{N}_i} \right) - \bar{x}_i^{k+\frac{1}{2}} \right), \tag{36}$$

where $\mathcal{N}_i$ denotes the neighbors of agent $i$ and $\bar{x}_i^{k+\frac{1}{2}} := \sum_{j \in \mathcal{H}_i} w_{ij} x_j^{k+\frac{1}{2}}$ denotes the average over honest agents connected to $i$ (including $i$). Thus, for

$$x_i^{k+1} = R_i^{k+\frac{1}{2}} = \bar{x}_i^{k+\frac{1}{2}} + \eta \xi_i^k, \quad \mathbf{x}^{k+1} = col\{x_i^{k+1}\} = \bar{\mathbf{x}}^{k+\frac{1}{2}} + \eta \boldsymbol{\xi}^k. \tag{37}$$

Incorporating this aggregation error into the update equations (34) yields:

$$\mathbf{x}^{k+1} = \underbrace{(\mathbf{W} + q\,(\mathbf{W} - \mathbf{I}))\, \mathbf{x}^k - \eta\, \mathbf{W}\left( \nabla \mathbf{f}_{\mathcal{H}}(\mathbf{x}^k) - \mathbf{N}^k \right) - \mathbf{s}^k - \eta\, \mathbf{W}\, \mathbf{w}^k}_{=\bar{\mathbf{x}}^{k+\frac{1}{2}}} + \eta\, \boldsymbol{\xi}^k, \tag{38a}$$

$$\mathbf{s}^{k+1} = q\, \mathbf{s}^k + q^2\,(\mathbf{I} - \mathbf{W})\, \mathbf{x}^k + \eta\, \mathbf{W}\left( \mathbf{N}^{k+1} - q\, \mathbf{N}^k \right). \tag{38b}$$

The average iterate update can be decomposed into the update without attackers and aggregation error contributions:

$$\bar{\mathbf{x}}^{k+1} = \bar{\mathbf{x}}^k + \underbrace{q\left( \bar{\mathbf{x}}^k - \bar{\mathbf{x}}^{k-1} \right) - \eta\left( \bar{\mathbf{g}}^k - q\bar{\mathbf{g}}^{k-1} \right)}_{:=-\eta\bar{\mathbf{v}}^k} + \eta\, \bar{\boldsymbol{\xi}}^k$$

$$= \bar{\mathbf{x}}^k - \eta\, \bar{\mathbf{v}}^k + \eta\, \bar{\boldsymbol{\xi}}^k, \tag{39}$$

where $\bar{\mathbf{x}}^k := \frac{1}{n_h}(\mathbf{1}_{n_h}^\top \otimes I_d)\, \mathbf{x}^k \in \mathbb{R}^d$ denotes the average iterate and $\bar{\mathbf{v}}^k$ is the effective update direction.

We now derive the spectral structure of the matrices $\mathbf{A}$, $\mathbf{B}$, and $\mathbf{C}$ to transform recursion (33) into a more analytically tractable form. Under Assumption 1, the combination matrix $\mathbf{W} = W \otimes I_d \in \mathbb{R}^{dn_h \times dn_h}$ admits the following eigenvalue decomposition:

$$\mathbf{W} = \mathbf{U}\, \boldsymbol{\Lambda}\, \mathbf{U}^\top = \begin{bmatrix} \frac{1}{\sqrt{n_h}} \mathbf{1}_{n_h} \otimes I_d & \hat{\mathbf{U}} \end{bmatrix} \begin{bmatrix} I_d & 0 \\ 0 & \hat{\boldsymbol{\Lambda}} \end{bmatrix} \begin{bmatrix} \frac{1}{\sqrt{n_h}} \mathbf{1}_{n_h}^\top \otimes I_d \\ \hat{\mathbf{U}}^\top \end{bmatrix}, \tag{40}$$

where $\hat{\boldsymbol{\Lambda}} := \mathrm{diag}\{\lambda_i\}_{i=2}^{n_h} \otimes I_d \in \mathbb{R}^{d(n_h-1) \times d(n_h-1)}$ contains the non-unity eigenvalues, $\mathbf{U} \in \mathbb{R}^{dn_h \times dn_h}$ is an orthogonal matrix, and $\hat{\mathbf{U}} := \hat{U} \otimes I_d \in \mathbb{R}^{dn_h \times d(n_h-1)}$ satisfies:

$$\hat{\mathbf{U}}^\top \hat{\mathbf{U}} = \mathbf{I}_{d(n_h-1)}, \quad \hat{\mathbf{U}}\hat{\mathbf{U}}^\top = \mathbf{I}_{dn_h} - \frac{1}{n_h}\mathbf{1}_{n_h}\mathbf{1}_{n_h}^\top \otimes I_d, \quad (\mathbf{1}_{n_h}^\top \otimes I_d)\hat{\mathbf{U}} = 0.$$

Since $\mathbf{A}, \mathbf{B}^2, \mathbf{C}$ are constructed as polynomial functions of $\mathbf{W}$, they share the same eigenvector structure:

$$\mathbf{A} = \mathbf{U}\, \boldsymbol{\Lambda}_a\, \mathbf{U}^\top = \begin{bmatrix} \frac{1}{\sqrt{n_h}} \mathbf{1}_{n_h} \otimes I_d & \hat{\mathbf{U}} \end{bmatrix} \begin{bmatrix} I_d & 0 \\ 0 & \hat{\boldsymbol{\Lambda}}_a \end{bmatrix} \begin{bmatrix} \frac{1}{\sqrt{n_h}} \mathbf{1}_{n_h}^\top \otimes I_d \\ \hat{\mathbf{U}}^\top \end{bmatrix}, \tag{41a}$$

$$\mathbf{C} = \mathbf{U}\,\boldsymbol{\Lambda}_c\,\mathbf{U}^\top = \begin{bmatrix} \frac{1}{\sqrt{n_h}}\mathbf{1}_{n_h} \otimes I_d & \hat{\mathbf{U}} \end{bmatrix} \begin{bmatrix} I_d & 0 \\ 0 & \hat{\boldsymbol{\Lambda}}_c \end{bmatrix} \begin{bmatrix} \frac{1}{\sqrt{n_h}}\mathbf{1}_{n_h}^\top \otimes I_d \\ \hat{\mathbf{U}}^\top \end{bmatrix}, \tag{41b}$$

$$\mathbf{B}^2 = \mathbf{U}\,\boldsymbol{\Lambda}_b^2\,\mathbf{U}^\top = \begin{bmatrix} \frac{1}{\sqrt{n_h}}\mathbf{1}_{n_h} \otimes I_d & \hat{\mathbf{U}} \end{bmatrix} \begin{bmatrix} 0 & 0 \\ 0 & \hat{\boldsymbol{\Lambda}}_b^2 \end{bmatrix} \begin{bmatrix} \frac{1}{\sqrt{n_h}}\mathbf{1}_{n_h}^\top \otimes I_d \\ \hat{\mathbf{U}}^\top \end{bmatrix}, \tag{41c}$$

where the diagonal eigenvalue matrices are:

$$\hat{\boldsymbol{\Lambda}}_a = \mathrm{diag}\{\lambda_{a,i}\}_{i=2}^{n_h} \otimes I_d, \quad \hat{\boldsymbol{\Lambda}}_b^2 = \mathrm{diag}\{\lambda_{b,i}^2\}_{i=2}^{n_h} \otimes I_d, \quad \hat{\boldsymbol{\Lambda}}_c = \mathrm{diag}\{\lambda_{c,i}\}_{i=2}^{n_h} \otimes I_d.$$

Pre-multiplying both sides of (38) by $\mathbf{U}^\top$ and exploiting the structure in (41), we obtain the transformed recursion:

$$\mathbf{U}^\top\mathbf{x}^{k+1} = \mathbf{U}^\top(\mathbf{C} - q\mathbf{B}^2)\mathbf{x}^k - \eta\mathbf{U}^\top\mathbf{A}\left(\nabla\mathbf{f}_{\mathcal{H}}(\mathbf{x}^k) - \mathbf{N}^k\right) - \mathbf{U}^\top\mathbf{s}^k - \eta\mathbf{U}^\top\mathbf{A}\mathbf{w}^k + \eta\mathbf{U}^\top\boldsymbol{\xi}^k, \tag{42a}$$

$$\mathbf{U}^\top\mathbf{s}^{k+1} = q\mathbf{U}^\top\mathbf{s}^k + q^2\mathbf{U}^\top\mathbf{B}^2\mathbf{x}^k + \eta\mathbf{U}^\top\mathbf{A}\left(\mathbf{N}^{k+1} - q\mathbf{N}^k\right), \tag{42b}$$

with $\mathbf{A} = \mathbf{W}$, $\mathbf{B}^2 = \mathbf{I} - \mathbf{W}$, and $\mathbf{C} = \mathbf{W}$. Leveraging the block structure of $\mathbf{U}$, the projected variables admit the following decompositions:

$$\mathbf{U}^\top\mathbf{x}^k = \begin{bmatrix} \sqrt{n_h}\bar{x}^k \\ \hat{\mathbf{U}}^\top\mathbf{x}^k \end{bmatrix}, \quad \mathbf{U}^\top\mathbf{s}^k = \begin{bmatrix} \sqrt{n_h}\eta\bar{N}^k \\ \hat{\mathbf{U}}^\top\mathbf{s}^k \end{bmatrix}, \tag{43a}$$

$$\mathbf{U}^\top\nabla\mathbf{f}_{\mathcal{H}}(\mathbf{x}^k) = \begin{bmatrix} \sqrt{n_h}\overline{\nabla f_{\mathcal{H}}}(x^k) \\ \hat{\mathbf{U}}^\top\nabla\mathbf{f}_{\mathcal{H}}(\mathbf{x}^k) \end{bmatrix}, \quad \mathbf{U}^\top\mathbf{w}^k = \begin{bmatrix} \sqrt{n_h}\overline{\mathbf{w}}^k \\ \hat{\mathbf{U}}^\top\mathbf{w}^k \end{bmatrix}. \tag{43b}$$

Combining the structural relations from (41), (42), and (43), we obtain the dynamics for the average iterate and the deviation from consensus. Then, the update dynamics become

$$\bar{\mathbf{x}}^{k+1} = \bar{\mathbf{x}}^k - \eta\bar{\mathbf{v}}^k + \eta\bar{\boldsymbol{\xi}}^k, \tag{44a}$$

$$\hat{\mathbf{U}}^\top\mathbf{x}^{k+1} = \hat{\mathbf{U}}^\top(\mathbf{C} - q\mathbf{B}^2)\mathbf{x}^k - \eta\hat{\mathbf{U}}^\top\mathbf{A}\left(\nabla\mathbf{f}_{\mathcal{H}}(\mathbf{x}^k) - \mathbf{N}^k\right) - \hat{\mathbf{U}}^\top\mathbf{s}^k - \eta\hat{\mathbf{U}}^\top\mathbf{A}\mathbf{w}^k + \eta\hat{\mathbf{U}}^\top\boldsymbol{\xi}^k, \tag{44b}$$

$$\hat{\mathbf{U}}^\top\mathbf{s}^{k+1} = q\hat{\mathbf{U}}^\top\mathbf{s}^k + q^2\hat{\mathbf{U}}^\top\mathbf{B}^2\mathbf{x}^k + \eta\hat{\mathbf{U}}^\top\mathbf{A}\left(\mathbf{N}^{k+1} - q\mathbf{N}^k\right). \tag{44c}$$

Pre-multiplying the third equation of (44c) by $\hat{\boldsymbol{\Lambda}}_b^{-1}$ and applying the identity $\hat{\mathbf{U}}^\top\hat{\mathbf{U}} = \mathbf{I}_{d(n_h-1)}$, we consolidate the recursion into compact matrix notation as follows:

$$\bar{\mathbf{x}}^{k+1} = \bar{\mathbf{x}}^k - \eta\bar{\mathbf{v}}^k + \eta\bar{\boldsymbol{\xi}}^k, \tag{45a}$$

$$\begin{bmatrix} \hat{\mathbf{U}}^\top\mathbf{x}^{k+1} \\ \hat{\boldsymbol{\Lambda}}_b^{-1}\hat{\mathbf{U}}^\top\mathbf{s}^{k+1} \end{bmatrix} = \begin{bmatrix} \hat{\boldsymbol{\Lambda}}_c - q\hat{\boldsymbol{\Lambda}}_b^2 & -\hat{\boldsymbol{\Lambda}}_b \\ q^2\hat{\boldsymbol{\Lambda}}_b & q\mathbf{I} \end{bmatrix} \begin{bmatrix} \hat{\mathbf{U}}^\top\mathbf{x}^k \\ \hat{\boldsymbol{\Lambda}}_b^{-1}\hat{\mathbf{U}}^\top\mathbf{s}^k \end{bmatrix}$$
$$- \eta \begin{bmatrix} \hat{\boldsymbol{\Lambda}}_a\hat{\mathbf{U}}^\top\left(\nabla\mathbf{f}_{\mathcal{H}}(\mathbf{x}^k) - \mathbf{N}^k + \mathbf{w}^k\right) - \hat{\mathbf{U}}^\top\boldsymbol{\xi}^k \\ \hat{\boldsymbol{\Lambda}}_b^{-1}\hat{\boldsymbol{\Lambda}}_a\hat{\mathbf{U}}^\top\left(\mathbf{N}^{k+1} - q\mathbf{N}^k\right) \end{bmatrix}. \tag{45b}$$

The convergence behavior of recursion (45) is governed by the transition matrix $\mathbf{G}$:

$$\mathbf{G} := \begin{bmatrix} \hat{\boldsymbol{\Lambda}}_c - q\hat{\boldsymbol{\Lambda}}_b^2 & -\hat{\boldsymbol{\Lambda}}_b \\ q^2\hat{\boldsymbol{\Lambda}}_b & q\mathbf{I} \end{bmatrix} \in \mathbb{R}^{2d(n_h-1)\times 2d(n_h-1)}. \tag{46}$$

To obtain the final canonical form, we pre-multiply both sides of (45) by $\frac{1}{v}\hat{\mathbf{V}}^{-1}$. Under Assumption 1, there exists an invertible matrix $\hat{\mathbf{V}}$ such that recursion (45) transforms into:

$$\bar{\mathbf{x}}^{k+1} = \bar{\mathbf{x}}^k - \eta\bar{\mathbf{v}}^k + \eta\bar{\boldsymbol{\xi}}^k, \tag{47a}$$

$$\hat{\mathbf{e}}^{k+1} = \boldsymbol{\Gamma}\hat{\mathbf{e}}^k - \frac{\eta}{v}\hat{\mathbf{V}}^{-1} \left[ \begin{array}{c} \hat{\boldsymbol{\Lambda}}_a \hat{\mathbf{U}}^\top \left( \nabla \mathbf{f}_{\mathcal{H}}(\mathbf{x}^k) - \mathbf{N}^k + \mathbf{w}^k \right) - \hat{\mathbf{U}}^\top \boldsymbol{\xi}^k \\ \hat{\boldsymbol{\Lambda}}_b^{-1}\hat{\boldsymbol{\Lambda}}_a \hat{\mathbf{U}}^\top \left( \mathbf{N}^{k+1} - q\mathbf{N}^k \right) \end{array} \right], \tag{47b}$$

where $v > 0$ is an arbitrary, strictly positive scaling constant, and the transformed error variable $\hat{\mathbf{e}}^k$ is defined as:

$$\hat{\mathbf{e}}^k := \frac{1}{v}\hat{\mathbf{V}}^{-1} \left[ \begin{array}{c} \hat{\mathbf{U}}^\top \mathbf{x}^k \\ \hat{\boldsymbol{\Lambda}}_b^{-1}\hat{\mathbf{U}}^\top \mathbf{s}^k \end{array} \right].$$

We now establish the relationship between the transformed variable $\hat{\mathbf{e}}^k$ and the deviation from consensus. Recall that the average iterate is defined as $\bar{x}^k = \frac{1}{n_h}\sum_{i\in\mathcal{H}} x_i^k$, and denote $\bar{\mathbf{x}}^k = \mathbf{1} \otimes \bar{x}^k$. Since $\hat{\mathbf{U}}^\top\hat{\mathbf{U}} = \mathbf{I}$, the following identity holds:

$$\left\|\hat{\mathbf{U}}^\top\mathbf{x}\right\|^2 = \mathbf{x}^\top\hat{\mathbf{U}}\hat{\mathbf{U}}^\top\hat{\mathbf{U}}\hat{\mathbf{U}}^\top\mathbf{x} = \left\|\hat{\mathbf{U}}\hat{\mathbf{U}}^\top\mathbf{x}^k\right\|^2 = \left\|\mathbf{x}^k - \bar{\mathbf{x}}^k\right\|^2. \tag{48}$$

From the definition of $\hat{\mathbf{e}}^k$, we obtain:

$$\left\|v\hat{\mathbf{V}}\hat{\mathbf{e}}^k\right\|^2 = \left\|\hat{\mathbf{U}}^\top\mathbf{x}^k\right\|^2 + \left\|\hat{\boldsymbol{\Lambda}}_b^{-1}\hat{\mathbf{U}}^\top\mathbf{s}^k\right\|^2. \tag{49}$$

Consequently, the consensus deviation $\left\|\mathbf{x}^k - \bar{\mathbf{x}}^k\right\|^2$ can be bounded as:

$$\left\|\mathbf{x}^k - \bar{\mathbf{x}}^k\right\|^2 = \left\|\hat{\mathbf{U}}^\top\mathbf{x}^k\right\|^2 = \left\|v\hat{\mathbf{V}}\hat{\mathbf{e}}^k\right\|^2 - \left\|\hat{\boldsymbol{\Lambda}}_b^{-1}\hat{\mathbf{U}}^\top\mathbf{s}^k\right\|^2 \le \left\|v\hat{\mathbf{V}}\hat{\mathbf{e}}^k\right\|^2. \tag{50}$$

## C.3. Fundamental Factorization

This subsection establishes a key factorization of the transition matrix $\mathbf{G}$, which is pivotal to the convergence analysis. Adopting the methodology of (Alghunaim & Yuan, 2022), we present the following lemma. Its proof adapts the construction of Lemma 1 in (Alghunaim & Yuan, 2022) to accommodate the specific structure of our transition matrix.

**Lemma 1** (Fundamental factorization). *Suppose that all eigenvalues of $\mathbf{G}$ are strictly less than one in magnitude. Then, there exists an invertible matrix $\hat{\mathbf{V}}$ such that*

$$\mathbf{G} = \hat{\mathbf{V}}\,\boldsymbol{\Gamma}\,\hat{\mathbf{V}}^{-1}, \tag{51}$$

*where $\boldsymbol{\Gamma}$ satisfies $\|\boldsymbol{\Gamma}\| < 1$.*

*Proof.* The proof proceeds by exploiting the block structure of $\mathbf{G}$ and constructing similarity transformations for each block separately.

Let $\hat{\boldsymbol{\Lambda}}_b = \mathrm{diag}\{\lambda_{b,i}\}_{i=2}^{n_h} \otimes I_d$ and $\hat{\boldsymbol{\Lambda}}_c = \mathrm{diag}\{\lambda_{c,i}\}_{i=2}^{n_h} \otimes I_d$. The matrix $\mathbf{G}$ defined in (46) can be rewritten as

$$\mathbf{G} = \left[ \begin{array}{cc} \mathrm{diag}\{\lambda_{c,i} - q\,\lambda_{b,i}^2\}_{i=2}^{n_h} & -\mathrm{diag}\{\lambda_{b,i}\}_{i=2}^{n_h} \\ \mathrm{diag}\{q^2\lambda_{b,i}\}_{i=2}^{n_h} & q\,I_{n_h-1} \end{array} \right] \otimes I_d. \tag{52}$$

Exploiting this block-diagonal structure, there exists a permutation matrix $\mathbf{P} \in \mathbb{R}^{2d(n_h-1)\times 2d(n_h-1)}$ such that

$$\mathbf{P}\,\mathbf{G}\,\mathbf{P}^\top = \mathrm{blkdiag}\{G_i\}_{i=2}^{n_h} \otimes I_d, \tag{53}$$

where each $2 \times 2$ block takes the form

$$G_i := \left[ \begin{array}{cc} \lambda_{c,i} - q\,\lambda_{b,i}^2 & -\lambda_{b,i} \\ q^2\lambda_{b,i} & q \end{array} \right] \in \mathbb{R}^{2\times2}. \tag{54}$$

Denote the eigenvalues of $G_i$ by $\gamma_{1,i}$ and $\gamma_{2,i}$. Writing $G_i = \begin{bmatrix} a_1 & a_2 \\ a_3 & a_4 \end{bmatrix}$ with $a_1 = \lambda_{c,i} - q\lambda_{b,i}^2$, $a_2 = -\lambda_{b,i}$, $a_3 = q^2\lambda_{b,i}$, and $a_4 = q$, the characteristic equation yields

$$\gamma_{1,2,i} = \frac{(a_1 + a_4) \pm \sqrt{(a_1 + a_4)^2 - 4(a_1 a_4 - a_2 a_3)}}{2} = \frac{q + \lambda_{c,i} - q\lambda_{b,i}^2 \pm \sqrt{\left(q + \lambda_{c,i} - q\lambda_{b,i}^2\right)^2 - 4q\,\lambda_{c,i}}}{2}. \quad (55)$$

We proceed by considering two cases based on the multiplicity of these eigenvalues.

*Case 1: distinct eigenvalues.* Suppose $\gamma_{1,i} \neq \gamma_{2,i}$. Then $G_i$ is diagonalizable, and there exists an invertible matrix $V_i \in \mathbb{R}^{2\times 2}$ such that

$$G_i = V_i \Gamma_i V_i^{-1}, \qquad \Gamma_i := \mathrm{diag}\{\gamma_{1,i}, \gamma_{2,i}\}. \quad (56)$$

When $a_2 = -\lambda_{b,i} \neq 0$, a valid choice of eigenvectors is given by

$$v_{1,2} = \frac{1}{r}\mathrm{col}\{\, a_2,\ \gamma_{1,2,i} - a_1 \,\}, \quad V_i = [v_1\ v_2],\ \ r \neq 0. \quad (57)$$

Consequently, $\mathbf{G}$ admits the similarity transformation $\mathbf{G} = \hat{\mathbf{V}}\mathbf{\Gamma}\hat{\mathbf{V}}^{-1}$, where

$$\hat{\mathbf{V}} := \mathbf{P}^\top \mathbf{V}, \quad \mathbf{V} := \mathrm{blkdiag}\left\{V_i\right\}_{i=2}^{n_h} \otimes I_d, \quad \mathbf{\Gamma} := \mathrm{blkdiag}\left\{\Gamma_i\right\} \otimes I_d, \quad (58)$$

and $\|\mathbf{\Gamma}\| = \max_{i\in\{2,\ldots,n_h\}}\|\Gamma_i\| = \max_{i\in\{2,\ldots,n_h\}}\{|\gamma_{1,i}|, |\gamma_{2,i}|\} < 1$.

*Case 2: repeated eigenvalue.* Suppose $\gamma_{1,i} = \gamma_{2,i} = \gamma_i$. In this case, $G_i$ may be defective, and we appeal to the Jordan canonical form. There exists an invertible matrix $T_i$ such that

$$G_i = T_i J_i T_i^{-1}, \qquad J_i = \begin{bmatrix} \gamma_i & 1 \\ 0 & \gamma_i \end{bmatrix}. \quad (59)$$

To achieve a contraction, we introduce a scaling matrix $E_i = \mathrm{diag}\{1, \epsilon_i\}$ for any $\epsilon_i > 0$. A direct computation shows that

$$G_i = V_i \Gamma_i V_i^{-1}, \qquad V_i = T_i E_i, \qquad \Gamma_i = E_i^{-1} J_i E_i = \begin{bmatrix} \gamma_i & \epsilon_i \\ 0 & \gamma_i \end{bmatrix}. \quad (60)$$

Since the spectral radius is bounded by any matrix norm, we have

$$\|\Gamma_i\|^2 = \rho(\Gamma_i \Gamma_i^*) \leq \|\Gamma_i \Gamma_i^*\|_1 = |\gamma_i|^2 + \epsilon_i|\gamma_i| + \epsilon_i^2 \leq (|\gamma_i| + \epsilon_i)^2. \quad (61)$$

Because $|\gamma_i| < 1$ by assumption, choosing $\epsilon_i < 1 - |\gamma_i|$ ensures $\|\Gamma_i\| < 1$.

Combining the constructions from both cases, we define

$$\mathbf{V} = \mathrm{blkdiag}\{V_i\}_{i=2}^{n_h} \otimes I_d, \qquad \mathbf{\Gamma} = \mathrm{blkdiag}\{\Gamma_i\}_{i=2}^{n_h} \otimes I_d, \qquad \hat{\mathbf{V}} = \mathbf{P}^\top \mathbf{V}. \quad (62)$$

Then the global similarity transformation takes the form

$$\mathbf{G} = \hat{\mathbf{V}}\,\mathbf{\Gamma}\,\hat{\mathbf{V}}^{-1}, \qquad \|\mathbf{\Gamma}\| = \max_{i=2,\ldots,n_h}\|\Gamma_i\| < 1, \quad (63)$$

which establishes the desired factorization. $\qquad\square$

### C.4. A Special Case of Lemma 1

We now specialize Lemma 1 to the specific matrix choices corresponding to Algorithm 1. This specialization yields explicit bounds on the key quantities $\gamma$, $v_1^2$, $v_2^2$, $\lambda_a$, and $\underline{\lambda}_b$. For notational convenience, we define:

$$v_1 := \|\hat{\mathbf{V}}\|, \quad v_2 := \left\|\hat{\mathbf{V}}^{-1}\right\|, \quad \lambda_a := \left\|\hat{\mathbf{\Lambda}}_a\right\|, \quad \gamma := \|\mathbf{\Gamma}\| < 1, \quad \underline{\lambda}_b := \frac{1}{\left\|\hat{\mathbf{\Lambda}}_b^{-1}\right\|}. \quad (64)$$

**Lemma 2.** *Suppose that all eigenvalues of $\mathbf{G}$ are strictly less than one in magnitude. Then, there exists an invertible matrix $\hat{\mathbf{V}}$ such that:*

$$\mathbf{G} = \hat{\mathbf{V}}\mathbf{\Gamma}\hat{\mathbf{V}}^{-1},$$

*with the following bounds:*

$$\|\hat{\mathbf{V}}\|^2 \leq 2(1 + q^2), \quad \|\hat{\mathbf{V}}^{-1}\|^2 \leq \frac{1 + q^2}{q\underline{\lambda}} \cdot \max\left\{1, \frac{1}{2\chi(q)}\right\}, \quad \text{and} \quad \|\mathbf{\Gamma}\| = \sqrt{q\lambda}, \tag{65}$$

*where $\lambda = \max_{i \in \{2,\dots,n_h\}} \lambda_i$ denotes the second largest eigenvalue magnitude, $\underline{\lambda}$ is the minimum non-zero eigenvalue of $W$, and $\chi(q) := 1 - \frac{(1+q)^2}{4q}\lambda$.*

*Proof.* We first recall the standard eigenvalue formula for $2 \times 2$ matrices. Consider a general matrix

$$G = \begin{bmatrix} a_1 & a_2 \\ a_3 & a_4 \end{bmatrix} \in \mathbb{R}^{2 \times 2}. \tag{66}$$

The eigenvalues of $G$ are given by the roots of the characteristic polynomial:

$$\gamma_{1,2} = \frac{(a_1 + a_4) \pm \sqrt{(a_1 + a_4)^2 - 4(a_1 a_4 - a_2 a_3)}}{2}. \tag{67}$$

When the eigenvalues are distinct, i.e., $\gamma_1 \neq \gamma_2$, the matrix $G$ is diagonalizable and admits the factorization $G = V \operatorname{diag}\{\gamma_1, \gamma_2\} V^{-1}$ with $V = [v_1 \quad v_2]$, where $v_{1,2} = \frac{1}{r} \operatorname{col}\{a_2, \gamma_{1,2} - a_1\}$ for any nonzero scalar $r$ when $a_2 \neq 0$.

We now specialize to the matrix structure arising from Algorithm 1. The combination matrices are given by $\mathbf{A} = \mathbf{W}$, $\mathbf{B} = (\mathbf{I} - \mathbf{W})^{1/2}$, and $\mathbf{C} = \mathbf{W}$, which yield $\lambda_{a,i} = \lambda_i$, $\lambda_{b,i} = \sqrt{1 - \lambda_i}$, and $\lambda_{c,i} = \lambda_i$, where $\{\lambda_1, \cdots, \lambda_{n_h}\}$ denote the eigenvalues of $\mathbf{W}$. Substituting these expressions into the block structure, for each $i = 2, \cdots, n_h$, we obtain

$$G_i = \begin{bmatrix} (1 + q)\lambda_i - q & -\sqrt{1 - \lambda_i} \\ q^2 \sqrt{1 - \lambda_i} & q \end{bmatrix} \in \mathbb{R}^{2 \times 2}. \tag{68}$$

Applying the quadratic formula (67) to $G_i$, the eigenvalues are given by

$$\gamma_{(1,2),i} = \frac{(1 + q)\lambda_i \pm \sqrt{(1 + q)^2 \lambda_i^2 - 4q\lambda_i}}{2}. \tag{69}$$

Define the critical threshold $q_c(\lambda_i) := \frac{\left(1 - \sqrt{1 - \lambda_i}\right)^2}{\lambda_i}$. Under the condition $q_c(\lambda_i) \leq q < 1$, the discriminant satisfies $(1 + q)^2 \lambda_i^2 - 4q\lambda_i \leq 0$, indicating that the eigenvalues are either complex conjugates or coincide.

We proceed by analyzing the two possible scenarios for $\lambda_i$.

First, suppose $0 < \lambda_i < 1$. For $q \in (q_c(\lambda_i), 1)$, the discriminant is negative, and the eigenvalues form a complex conjugate pair:

$$\gamma_{(1,2),i} = \frac{(1 + q)\lambda_i}{2} \pm j\frac{\sqrt{4q\lambda_i - (1 + q)^2 \lambda_i^2}}{2}, \quad \left|\gamma_{(1,2),i}\right| = \sqrt{q\lambda_i} < 1. \tag{70}$$

Since $a_2 = -\sqrt{1 - \lambda_i} \neq 0$, the matrix $G_i$ is diagonalizable over $\mathbb{C}$, and we can write $G_i = V_i \operatorname{diag}\{\gamma_{1,i}, \gamma_{2,i}\} V_i^{-1}$ with

$$V_i = \begin{bmatrix} -1 & -1 \\ \beta_i & \beta_i^* \end{bmatrix}, \quad \text{where} \quad \beta_i = \frac{q^2 \sqrt{1 - \lambda_i}}{q - \gamma_{1,i}}. \tag{71}$$

In the complex eigenvalue regime, direct computation shows that $|\beta_i| = q$ and $\Im\beta_i = \frac{1}{2}\sqrt{\frac{\lambda_i(4q - (1+q)^2\lambda_i)}{1 - \lambda_i}}$. The inverse of $V_i$ is given by

$$V_i^{-1} = \frac{1}{2j\Im\beta_i} \begin{bmatrix} \beta_i^* & 1 \\ -\beta_i & -1 \end{bmatrix}. \tag{72}$$

To bound the norm of $V_i$, we compute the Gram matrix

$$V_i V_i^* = \begin{bmatrix} 2 & -2\Re\beta_i \\ -2\Re\beta_i & 2|\beta_i|^2 \end{bmatrix} = \begin{bmatrix} 2 & -2a_i \\ -2a_i & 2q^2 \end{bmatrix}, \quad a_i := \Re\beta_i. \tag{73}$$

The spectral norm satisfies $\|V_i\|^2 = \rho(V_i V_i^*) = (1 + q^2) + \sqrt{(1 - q^2)^2 + 4a_i^2} \le 2(1 + q^2)$. For the inverse, we have

$$V_i^{-1}(V_i^{-1})^* = \frac{1}{4(\Im\beta_i)^2} \begin{bmatrix} |\beta_i|^2 + 1 & -(\beta_i^2)^* - 1 \\ -\beta_i^2 - 1 & |\beta_i|^2 + 1 \end{bmatrix}. \tag{74}$$

Using the identities $|\beta_i| = q$ and $|\beta_i^2 + 1| \le |\beta_i|^2 + 1$, we obtain

$$\|V_i^{-1}\|^2 \le \frac{2(1 + q^2)}{4(\Im\beta_i)^2} = \frac{2(1 + q^2)(1 - \lambda_i)}{\lambda_i(4q - (1 + q)^2\lambda_i)}, \tag{75}$$

where the denominator is strictly positive since $q > q_c(\lambda)$.

To derive a uniform bound over all indices $i$, we introduce $\chi(q) := 1 - \frac{(1+q)^2}{4q}\lambda \in (0, 1]$. Using the bounds $\underline{\lambda} \le \lambda_i \le \lambda$ for all $i$, we obtain the chain of inequalities

$$\frac{1 - \lambda_i}{\lambda_i(4q - (1 + q)^2\lambda_i)} \le \frac{1}{\underline{\lambda}(4q - (1 + q)^2\lambda_i)} \le \frac{1}{\underline{\lambda}(4q - (1 + q)^2\lambda)} = \frac{1}{4q\underline{\lambda}\chi(q)}.$$

It follows that

$$\|\mathbf{V}^{-1}\|^2 \le 2(1 + q^2) \cdot \max_i \frac{1 - \lambda_i}{\lambda_i(4q - (1 + q)^2\lambda_i)} \le \frac{1 + q^2}{2q\underline{\lambda}\chi(q)}.$$

Next, suppose $\lambda_i = 0$. In this degenerate case, both eigenvalues vanish ($\gamma_{(1,2),i} = 0$), and the block matrix reduces to the nilpotent form

$$G_i = \begin{bmatrix} -q & -1 \\ q^2 & q \end{bmatrix}, \quad \text{with} \quad G_i^2 = \mathbf{0}. \tag{76}$$

Since $G_i$ is defective, we employ the Jordan canonical form. There exists an invertible matrix $T_i$ such that

$$T_i^{-1}G_iT_i = \begin{bmatrix} 0 & 1 \\ 0 & 0 \end{bmatrix}, \quad T_i = \begin{bmatrix} 1 & -\frac{q}{1+q^2} \\ -q & -\frac{1}{1+q^2} \end{bmatrix}, \quad \|T_i\|^2 = \|T_i^{-1}\|^2 = 1 + q^2. \tag{77}$$

For any $0 < \varepsilon < 1$, we introduce a scaling to obtain

$$V_i = T_i \operatorname{diag}(1, \varepsilon), \quad \Gamma_i = \begin{bmatrix} 0 & \varepsilon \\ 0 & 0 \end{bmatrix}, \tag{78}$$

which yields the norm bounds $\|\Gamma_i\| = \varepsilon$, $\|V_i\|^2 \le 1 + q^2$, and $\|V_i^{-1}\|^2 \le \frac{1+q^2}{\varepsilon^2}$. Selecting $\varepsilon^2 = q\lambda = q\max_{i\in\{2,\dots,n_h\}}\lambda_i$ ensures that the contraction rate is consistent with the complex eigenvalue case.

Combining the results from both scenarios, we define the block-diagonal matrices $\mathbf{V} = \operatorname{blkdiag}(V_2, \dots, V_{n_h})$ and $\Gamma = \operatorname{blkdiag}(\Gamma_2, \dots, \Gamma_{n_h})$, and set $\hat{\mathbf{V}} = \mathbf{P}^\top\mathbf{V}$ where $\mathbf{P}$ is the orthonormal permutation matrix from the block decomposition. The consolidated bounds are

$$\|\hat{\mathbf{V}}\|^2 \le 2(1 + q^2),$$
$$\|\hat{\mathbf{V}}^{-1}\|^2 \le \max\left\{\frac{1 + q^2}{2q\underline{\lambda}\chi(q)}, \frac{1 + q^2}{q\lambda}\right\} = \frac{1 + q^2}{q\underline{\lambda}} \cdot \max\left\{1, \frac{1}{2\chi(q)}\right\}, \tag{79}$$
$$\gamma = \|\Gamma\| = \sqrt{q\lambda}.$$

Finally, for the specific choices $\mathbf{A} = \mathbf{W}$ and $\mathbf{B} = (\mathbf{I} - \mathbf{W})^{1/2}$ prescribed by Algorithm 1, the spectral quantities evaluate to

$$\lambda_a = \left\| \hat{\mathbf{\Lambda}}_a \right\| = \lambda, \quad \underline{\lambda}_b = \frac{1}{\left\| \hat{\mathbf{\Lambda}}_b^{-1} \right\|} = \sqrt{1 - \lambda}, \tag{80}$$

where $\lambda = \max_{i \in \{2, \ldots, n_h\}} \lambda_i$ and $\underline{\lambda}$ denotes the minimum non-zero eigenvalue of $W$. This completes the proof.

$\square$

## D. Convergence Analysis: Proof of Theorem 1 and Corollary 1

In this section, we establish the convergence guarantees for BRED (Algorithm 1) by proving Theorem 1 and Corollary 1. We first present a proof outline in Section D.1, where we introduce four key lemmas: (i) the aggregation error bound characterizing the discrepancy introduced by robust aggregation, (ii) the consensus error bound measuring deviation from the global average, (iii) the descent lemma quantifying per-iteration progress, and (iv) the aggregation bias bound capturing error accumulation from historical gradients. In Section D.2, we combine these lemmas via a Lyapunov argument to prove the main convergence theorem and derive the transient time.

### D.1. Proof Outline

We begin by recalling the update rule of BRED in Algorithm 1. At each iteration $k$, every honest agent $i \in \mathcal{H}$ computes a stochastic gradient, performs a local update with scaled dual ascent, and then aggregates information from its neighbors using a robust aggregation rule:

$$\begin{aligned}
g_i^k &= \nabla F_i(x_i^k; \varphi_i^k), \\
x_i^{k+\frac{1}{2}} &= x_i^k + q(x_i^k - x_i^{k-1}) - \eta(g_i^k - q g_i^{k-1}), \\
x_i^{k+1} &= R_i^{k+\frac{1}{2}} = \mathcal{A}\big(x_i^{k+\frac{1}{2}}, \{x_m^{k+\frac{1}{2}}\}_{m \in \mathcal{N}_i}\big),
\end{aligned} \tag{81}$$

where $R_i^{k+\frac{1}{2}}$ denotes the output of the robust aggregator, $q \in (0, 1)$ is the scaling factor for dual ascent, and $\eta > 0$ is the step size. To facilitate the analysis, we introduce the following auxiliary quantities for each agent $i \in \mathcal{H}$:

$$\bar{x}_i^{k+\frac{1}{2}} := \sum_{j \in \mathcal{H}_i} w_{ij} x_j^{k+\frac{1}{2}}, \tag{82}$$

where $\bar{x}_i^{k+\frac{1}{2}}$ denotes the average over honest neighbors with $\sum_{j \in \mathcal{H}_i} w_{ij} = 1$ and $\mathcal{H}_i$ denotes honest agents connected to agent $i$ (including $i$).

With these definitions in place, the convergence analysis relies on bounding four key quantities: the *aggregation error* induced by robust aggregation, the *consensus error* measuring deviation from the global average, the *descent* in objective value per iteration, and the *aggregation bias* arising from historical gradient information.

**Aggregation error.**    To defend against Byzantine adversaries, each client employs a robust aggregation rule in place of simple averaging. This introduces the aggregation error, defined as the discrepancy between the robust aggregator $R_i^{k+\frac{1}{2}}$ and the honest average $\bar{x}_i^{k+\frac{1}{2}} := \sum_{j \in \mathcal{H}_i} w_{ij} x_j^{k+\frac{1}{2}}$. Specifically, we define the aggregation error as

$$\xi_i^k := \frac{1}{\eta} \left( R_i^{k+\frac{1}{2}} - \bar{x}_i^{k+\frac{1}{2}} \right), \tag{83}$$

and $\boldsymbol{\xi}^k = col\{\xi_i^k\}_{i \in \mathcal{H}}$ denotes the stacked vector of aggregation error across the honest clients. The following lemma establishes a bound on this error, with the detailed proof deferred to Appendix D.3.1.

**Lemma 3.** *Suppose that Assumptions 2, 3, and 4 hold. For BRED in Algorithm 1, the following holds for all $k \in \{0, \cdots, K-1\}$:*

$$\frac{1}{n_h} \sum_{k=0}^{K-1} \mathbb{E}\left[ \left\| \boldsymbol{\xi}^k \right\|^2 \right] \le \sum_{k=0}^{K-1} \frac{15\kappa v_2^2 v_1^2}{\eta^2} \mathbb{E}\left[ \left\| \hat{\mathbf{e}}^k \right\|^2 \right] + 24\kappa K \sigma^2 + 12\kappa K \zeta^2. \tag{84}$$

**Consensus error.** A key quantity in decentralized optimization is the consensus error, which measures how far each client's model parameter $x_i^k$ deviates from the global average $\bar{x}^k$ at iteration $k$. According to (50), the consensus error satisfies

$$\left\| \mathbf{x}^k - \bar{\mathbf{x}}^k \right\|^2 = \left\| \hat{\mathbf{U}}^\top \mathbf{x}^k \right\|^2 = \left\| v\hat{\mathbf{V}}\hat{\mathbf{e}}^k \right\|^2 - \left\| \hat{\mathbf{\Lambda}}_b^{-1}\hat{\mathbf{U}}^\top \mathbf{s}^k \right\|^2 \leq \left\| v\hat{\mathbf{V}}\hat{\mathbf{e}}^k \right\|^2. \tag{85}$$

Consequently, bounding $\|\hat{\mathbf{e}}^k\|^2$ suffices to bound the consensus error. The subsequent lemma presents this bound, with the proof provided in Appendix D.3.2.

**Lemma 4.** *Suppose that Assumptions 1, 3, and 4 hold. For* BRED *in Algorithm 1, the following holds for all $k \geq 0$:*

$$\sum_{k=0}^{K-1} \mathbb{E}\left[\left\|\hat{\mathbf{e}}^{k+1}\right\|^2\right] \leq \sum_{k=0}^{K-1} \frac{\gamma+1}{2}\mathbb{E}\left[\left\|\hat{\mathbf{e}}^k\right\|^2\right] + \sum_{k=0}^{K-1} \frac{4\eta^4\lambda_a^2 L^2}{(1-\gamma)\underline{\lambda_b}^2}\mathbb{E}\left[\left\|\bar{v}^k\right\|^2\right] + \eta^2\lambda_a^2\sigma^2 K$$

$$+ \sum_{k=0}^{K-1} \frac{2\eta^2\lambda_a^2(1-q)^2}{(1-\gamma)\underline{\lambda_b}^2}\mathbb{E}\left[\left\|\nabla f_{\mathcal{H}}(\bar{x}^k)\right\|^2\right] + \frac{56\eta^2\kappa}{1-\gamma}\sigma^2 K + \frac{28\eta^2\kappa}{1-\gamma}\zeta^2 K, \tag{86}$$

*where $v_1 := \|\hat{\mathbf{V}}\|$, $v_2 := \left\|\hat{\mathbf{V}}^{-1}\right\|$, $\lambda_a := \|\mathbf{\Lambda}_a\|$, $\gamma := \|\mathbf{\Gamma}\| < 1$, and $\underline{\lambda_b} := \frac{1}{\|\mathbf{\Lambda}_b^{-1}\|}$.*

**Descent lemma.** Finally, we quantify the per-iteration progress in minimizing the objective $f_{\mathcal{H}}$. From (44c), the averaged iterate evolves according to $\bar{x}^{k+1} = \bar{x}^k - \eta\bar{v}^k + \eta\bar{\xi}^k$. Decomposing the update in terms of the stochastic gradient yields

$$\bar{x}^{k+1} = \bar{x}^k - \eta\bar{g}^k + \eta\left(\bar{g}^k - \bar{v}^k\right) + \eta\bar{\xi}^k. \tag{87}$$

We also define the noise-free counterpart of the local update direction:

$$\tilde{v}_i^k := -\frac{q}{\eta}(x_i^k - x_i^{k-1}) + (\nabla f_i(x_i^k) - qg_i^{k-1}), \quad \bar{\tilde{v}}^k := \frac{1}{n_h}\sum_{i\in\mathcal{H}}\tilde{v}_i^k. \tag{88}$$

Note that when $q = 0$, the noise-free update direction reduces to $\tilde{v}_i^k = \nabla f_i(x_i^k)$, which recovers the standard gradient descent update. The following lemma establishes the descent bound, with the proof deferred to Appendix D.3.3.

**Lemma 5.** *Suppose that Assumptions 1 and 2 hold. For* BRED *in Algorithm 1, the following holds for all $k \geq 0$:*

$$f_{\mathcal{H}}(\bar{x}^K) - f_{\mathcal{H}}(\bar{x}^0) \leq -\frac{\eta}{4}\sum_{k=0}^{K-1}\mathbb{E}\left[\left\|\nabla f_{\mathcal{H}}(\bar{x}^k)\right\|^2\right] + \sum_{k=0}^{K-1}\eta L^2 v_2^2 v_1^2\mathbb{E}\left[\left\|\hat{\mathbf{e}}^k\right\|^2\right] + \frac{\eta^2 L\sigma^2 K}{n_h}$$

$$+ \sum_{k=0}^{K-1}\left(L\eta^2 - \frac{\eta}{2}\right)\mathbb{E}\left[\left\|\bar{\tilde{v}}^k\right\|^2\right] + \frac{\eta}{(1-q)^2}\sum_{k=0}^{K-1}\mathbb{E}\left[\left\|\bar{\xi}^k\right\|^2\right] + \sum_{k=0}^{K-1}\left(L\eta^2 + \eta\right)\mathbb{E}\left[\left\|\bar{\xi}^k\right\|^2\right]. \tag{89}$$

**Aggregation bias.** When historical gradient information is incorporated into the updates, aggregation errors can accumulate and induce a bias. Specifically, the aggregation bias quantifies the discrepancy between the averaged model update $\bar{\mathbf{v}}^k$ and the true averaged gradient $\overline{\nabla f_{\mathcal{H}}}(\mathbf{x}^k)$. The following lemma characterizes this bias as a function of the aggregation error, the detailed proof is provided in Appendix D.3.4.

**Lemma 6.** *Suppose that Assumptions 1, 2, 3, and 4 hold. For* BRED *in Algorithm 1, the following holds for all $k \geq 0$:*

$$\sum_{k=0}^{K-1}\mathbb{E}\left[\left\|\bar{\tilde{v}}^k - \overline{\nabla f_{\mathcal{H}}}(x^k)\right\|^2\right] \leq \frac{1}{(1-q)^2}\sum_{k=0}^{K-1}\mathbb{E}\left[\left\|\bar{\xi}^k\right\|^2\right]. \tag{90}$$

### D.2. Proof of Theorem 1

**Theorem 1.** *Suppose that Assumptions 1, 2, 3, and 4 hold. Let $\kappa$ be the robustness parameter. For step size $\eta = \Theta(\sqrt{n_h/K})$, BRED in Algorithm 1 is $(\delta, \varepsilon_K)$-Byzantine-resilient with $\varepsilon_K$ as follows*

$$
\frac{1}{K} \sum_{k=0}^{K-1} \mathbb{E}\big[\|\nabla f_{\mathcal{H}}(\bar{x}^k)\|^2\big] \leq \underbrace{\mathcal{O}\bigg( \frac{f_{\mathcal{H}}(x^0) - f_{\mathcal{H}}^{\star}}{\sqrt{n_h K}} + \frac{\sigma^2}{\sqrt{n_h K}} \bigg)}_{\text{dominant terms}}
$$

$$
+ \underbrace{\mathcal{O}\bigg( \frac{n_h \lambda^2 \sigma^2}{(1-\lambda)q\underline{\lambda}K} + \frac{n_h \lambda^2 \sigma^2}{(1-\lambda)^3 q\underline{\lambda}K^2} + \frac{n_h L^2 \lambda^2 \zeta^2}{(1-\lambda)^2 q\underline{\lambda}K^2} \bigg)}_{\text{topology-related terms}} \tag{91}
$$

$$
+ \underbrace{\mathcal{O}\bigg( \frac{\kappa n_h \sigma^2}{(1-\lambda)^3 q\underline{\lambda}K} + \frac{\kappa n_h \zeta^2}{(1-\lambda)^2 q\underline{\lambda}K} \bigg)}_{\text{Byzantine-induced terms}} + \underbrace{\mathcal{O}\bigg( \frac{\kappa \sigma^2}{(1-q)^2} + \frac{\kappa \zeta^2}{(1-q)^2} \bigg)}_{\text{asymptotic error}}.
$$

*Proof.* Our convergence analysis proceeds via a Lyapunov argument. We define the Lyapunov function $\mathcal{L}^k$ as follows

$$
\mathcal{L}^k := \mathbb{E}\left[ f_{\mathcal{H}}(\bar{x}^k) - f_{\mathcal{H}}^{\star} + z_\eta \|\hat{\mathbf{e}}^k\|^2 \right], \tag{92}
$$

where the weighting parameter $z_\eta$ and the auxiliary constants are

$$
z_\eta := z_1 \eta + \frac{\kappa z_2}{\eta}, \; z_1 := \frac{2L^2 v_2^2 v_1^2}{1-\gamma}, \; z_2 := \frac{30 v_2^2 v_1^2 c_1}{1-\gamma}, \; c_1 := 2 + \frac{1}{(1-q)^2}. \tag{93}
$$

By applying Lemma 4, we bound the evolution of the weighted consensus error over $K$ iterations as:

$$
\sum_{k=0}^{K-1} \mathbb{E}\left[ z_\eta \|\hat{\mathbf{e}}^{k+1}\|^2 - z_\eta \|\hat{\mathbf{e}}^k\|^2 \right] \leq \sum_{k=0}^{K-1} \frac{z_\eta(\gamma-1)}{2} \mathbb{E}\left[ \|\hat{\mathbf{e}}^k\|^2 \right] + \sum_{k=0}^{K-1} \frac{4 z_\eta \eta^4 \lambda_a^2 L^2}{(1-\gamma)\underline{\lambda_b}^2} \mathbb{E}\left[ \|\bar{\bar{v}}^k\|^2 \right]
$$

$$
+ \sum_{k=0}^{K-1} \frac{2 z_\eta \eta^2 \lambda_a^2 (1-q)^2}{(1-\gamma)\underline{\lambda_b}^2} \mathbb{E}\left[ \|\nabla f_{\mathcal{H}}(\bar{x}^k)\|^2 \right] + z_\eta \eta^2 \lambda_a^2 \sigma^2 K \tag{94}
$$

$$
+ \frac{56 z_\eta \eta^2 \kappa}{1-\gamma} \sigma^2 K + \frac{28 z_\eta \eta^2 \kappa}{1-\gamma} \zeta^2 K + \frac{4 z_\eta \eta^4 \lambda_a^2 L^2 \sigma^2}{(1-\gamma)\underline{\lambda_b}^2 n_h}.
$$

Applying Lemma 5 yields the following descent inequality for the global objective:

$$
f_{\mathcal{H}}(\bar{x}^K) - f_{\mathcal{H}}(\bar{x}^0) \leq -\frac{\eta}{4} \sum_{k=0}^{K-1} \mathbb{E}\left[ \|\nabla f_{\mathcal{H}}(\bar{x}^k)\|^2 \right] + \sum_{k=0}^{K-1} \eta L^2 v_2^2 v_1^2 \mathbb{E}\left[ \|\hat{\mathbf{e}}^k\|^2 \right] + \frac{\eta^2 L \sigma^2 K}{n_h}
$$

$$
+ \sum_{k=0}^{K-1} \left( L\eta^2 - \frac{\eta}{2} \right) \mathbb{E}\left[ \|\bar{\bar{v}}^k\|^2 \right] + \frac{\eta}{(1-q)^2} \sum_{k=0}^{K-1} \mathbb{E}\left[ \|\bar{\xi}^k\|^2 \right] + \sum_{k=0}^{K-1} \left( L\eta^2 + \eta \right) \mathbb{E}\left[ \|\bar{\xi}^k\|^2 \right]. \tag{95}
$$

Summing the consensus error bound (94) and the descent inequality (95) yields the cumulative drift of the Lyapunov function over $K$ iterations

$$
\sum_{k=0}^{K-1} \mathbb{E}\left[ \mathcal{L}^{k+1} - \mathcal{L}^k \right] \leq \underbrace{\left( -\frac{\eta}{4} + \frac{2 z_\eta \eta^2 \lambda_a^2 (1-q)^2}{(1-\gamma)\underline{\lambda_b}^2} \right)}_{=: \alpha_1} \sum_{k=0}^{K-1} \mathbb{E}\left[ \|\nabla f_{\mathcal{H}}(\bar{x}^k)\|^2 \right] + \underbrace{\left( \frac{z_\eta(\gamma-1)}{2} + \eta L^2 v_2^2 v_1^2 \right)}_{=: \alpha_2} \sum_{k=0}^{K-1} \mathbb{E}\left[ \|\hat{\mathbf{e}}^k\|^2 \right]
$$

$$
+ \underbrace{\left( L\eta^2 - \frac{\eta}{2} + \frac{4 z_\eta \eta^4 \lambda_a^2 L^2}{(1-\gamma)\underline{\lambda_b}^2} \right)}_{=: \alpha_3} \sum_{k=0}^{K-1} \mathbb{E}\left[ \|\bar{\bar{v}}^k\|^2 \right] + \underbrace{\left( L\eta^2 + \eta + \frac{\eta}{(1-q)^2} \right)}_{=: \alpha_4} \sum_{k=0}^{K-1} \mathbb{E}\left[ \|\bar{\xi}^k\|^2 \right] + R_K, \tag{96}
$$

where $R_K$ collects the residual terms:

$$R_K := \frac{\eta^2 L \sigma^2 K}{n_h} + \frac{4 z_\eta \eta^4 \lambda_a^2 L^2 \sigma^2}{(1-\gamma)\underline{\lambda_b}^2 n_h} + z_\eta \eta^2 \lambda_a^2 \sigma^2 K + \frac{56 z_\eta \eta^2 \kappa K}{1-\gamma}\sigma^2 + \frac{28 z_\eta \eta^2 \kappa K}{1-\gamma}\zeta^2.$$

We now analyze the coefficients $\alpha_1, \alpha_3$, and $\alpha_4$ to guarantee convergence. First, for the gradient coefficient $\alpha_1$, substituting $z_\eta = z_1\eta + \kappa z_2/\eta$ yields:

$$\alpha_1 = -\frac{\eta}{4} + \frac{2\eta^3 z_1 \lambda_a^2 (1-q)^2}{(1-\gamma)\underline{\lambda_b}^2} + \frac{2\kappa z_2 \eta \lambda_a^2 (1-q)^2}{(1-\gamma)\underline{\lambda_b}^2}. \tag{97}$$

Under the conditions $\eta \le \frac{(1-\gamma)\underline{\lambda_b}}{8\lambda_a v_1 v_2 L(1-q)}$ and $\kappa \le \frac{(1-\gamma)\underline{\lambda_b}^2}{32 z_2 \lambda_a^2 (1-q)^2}$, the positive terms are each bounded by $\eta/16$, resulting in:

$$\alpha_1 \le -\frac{\eta}{4} + \frac{\eta}{16} + \frac{\eta}{16} = -\frac{\eta}{8}. \tag{98}$$

Next, for the coefficient $\alpha_3$ (associated with the noise-free update direction), we have:

$$\alpha_3 = L\eta^2 - \frac{\eta}{2} + \frac{4 z_\eta \eta^4 \lambda_a^2 L^2}{(1-\gamma)\underline{\lambda_b}^2} = -\frac{\eta}{2} + L\eta^2 + \frac{4\eta^5 z_1 \lambda_a^2 L^2}{(1-\gamma)\underline{\lambda_b}^2} + \frac{4\eta^3 \kappa z_2 \lambda_a^2 L^2}{(1-\gamma)\underline{\lambda_b}^2}. \tag{99}$$

Provided that $\kappa \le \frac{(1-\gamma)^2}{v_1^2 v_2^2}$ and the step size satisfies

$$\eta \le \min\left\{ \frac{1}{20L}, \frac{\sqrt{(1-\gamma)\underline{\lambda_b}}}{4L\sqrt{\lambda_a v_1 v_2}}, \frac{\lambda_b}{24\lambda_a L\sqrt{c_1}} \right\},$$

we can bound the terms as follows:

$$\alpha_3 \le -\frac{\eta}{2} + \frac{\eta}{20} + \frac{\eta}{16} + \frac{\eta}{4} = -\frac{11\eta}{80} \le 0. \tag{100}$$

Finally, we bound the aggregation error coefficient $\alpha_4$. Provided that $\eta \le \frac{1}{20L}$, and defining the constant $c_1 := 2 + \frac{1}{(1-q)^2}$, we have:

$$\alpha_4 = L\eta^2 + \eta + \frac{\eta}{(1-q)^2} \le \left(\frac{1}{20}+1\right)\eta + \frac{\eta}{(1-q)^2} \le 2\eta + \frac{\eta}{(1-q)^2} = c_1\eta. \tag{101}$$

Since $\alpha_1 \le -\eta/8$ and $\alpha_3 \le 0$, substituting these bounds into (96) yields

$$\sum_{k=0}^{K-1} \mathbb{E}\left[\mathcal{L}^{k+1} - \mathcal{L}^k\right] \le -\frac{\eta}{8}\sum_{k=0}^{K-1}\mathbb{E}\left[\left\|\nabla f_{\mathcal{H}}(\bar{x}^k)\right\|^2\right] + \alpha_2 \sum_{k=0}^{K-1}\mathbb{E}\left[\left\|\hat{\mathbf{e}}^k\right\|^2\right] + c_1\eta \sum_{k=0}^{K-1}\mathbb{E}\left[\left\|\bar{\xi}^k\right\|^2\right] + R_K. \tag{102}$$

Substituting the aggregation error bound established in Lemma 3 into (102), we obtain

$$\sum_{k=0}^{K-1} \mathbb{E}\left[\mathcal{L}^{k+1} - \mathcal{L}^k\right] \le -\frac{\eta}{8}\sum_{k=0}^{K-1}\mathbb{E}\left[\left\|\nabla f_{\mathcal{H}}(\bar{x}^k)\right\|^2\right] + \underbrace{\left(\alpha_2 + \frac{15\kappa v_2^2 v_1^2 c_1}{\eta}\right)}_{=: \alpha_2'}\sum_{k=0}^{K-1}\mathbb{E}\left[\left\|\hat{\mathbf{e}}^k\right\|^2\right] + R_K', \tag{103}$$

where $R_K'$ collects all residual terms depending on $\sigma^2, \zeta^2$, and $K$:

$$R_K' := \frac{\eta^2 L \sigma^2 K}{n_h} + \frac{4 z_\eta \eta^4 \lambda_a^2 L^2 \sigma^2}{(1-\gamma)\underline{\lambda_b}^2 n_h} + z_\eta \eta^2 \lambda_a^2 \sigma^2 K + \frac{56 z_\eta \eta^2 \kappa}{1-\gamma}K\sigma^2 + \frac{28 z_\eta \eta^2 \kappa}{1-\gamma}K\zeta^2$$
$$+ 24 c_1 \eta \kappa K \sigma^2 + 12 c_1 \eta \kappa K \zeta^2.$$

It remains to bound the consensus error coefficient $\alpha_2'$. Substituting the definition of $z_\eta$ into the expression for $\alpha_2'$, we obtain:

$$\alpha_2' = \frac{z_\eta(\gamma-1)}{2} + \eta L^2 v_2^2 v_1^2 + \frac{15\kappa v_2^2 v_1^2 c_1}{\eta} = z_1\eta\frac{(\gamma-1)}{2} + \eta L^2 v_2^2 v_1^2 \le 0, \tag{104}$$

where the inequality follows from $\gamma < 1$. Since $\alpha_2' \leq 0$, we can drop the consensus error term. Telescoping the Lyapunov sum from $k = 0$ to $K - 1$ yields the convergence bound:

$$\mathcal{L}^K - \mathcal{L}^0 \leq -\frac{\eta}{8} \sum_{k=0}^{K-1} \mathbb{E}\left[|\nabla f_{\mathcal{H}}(\bar{x}^k)|^2\right] + R_K'. \tag{105}$$

Since $\mathcal{L}^K \geq 0$ (by the non-negativity of the potential function), rearranging (105) gives

$$\frac{\eta}{8} \sum_{k=0}^{K-1} \mathbb{E}\left[\left\|\nabla f_{\mathcal{H}}(\bar{x}^k)\right\|^2\right] \leq \mathcal{L}^0 + R_K'. \tag{106}$$

It remains to bound $\mathcal{L}^0$. Assuming identical initialization across all agents, i.e., $\mathbf{x}^0 = \mathbf{1} \otimes x^0$, we have $\|\hat{\mathbf{U}}^\top \mathbf{x}^0\|^2 = 0$ and $\mathbf{x}^0 = \bar{\mathbf{x}}^0$. Hence, for $\mathbf{A} = \mathbf{W}$, we have

$$
\begin{aligned}
\left\|\hat{\mathbf{e}}^0\right\|^2 &\leq \frac{1}{\underline{\lambda_b}^2 n_h} \left\|\hat{\mathbf{U}}^\top \mathbf{s}^0\right\|^2 = \frac{\eta^2}{\underline{\lambda_b}^2 n_h} \left\|\hat{\mathbf{U}}^\top \mathbf{W} \mathbf{N}^0\right\|^2 = \frac{\eta^2}{\underline{\lambda_b}^2 n_h} \left\|\hat{\mathbf{U}}^\top \mathbf{W} \nabla \mathbf{f}\left(\bar{\mathbf{x}}^0\right)\right\|^2 \\
&= \frac{\eta^2}{\underline{\lambda_b}^2 n_h} \left\|\left(\mathbf{W} - \frac{1}{n_h} \mathbf{1}^\top \mathbf{1} \otimes I_d\right) \nabla \mathbf{f}\left(\mathbf{x}^0\right)\right\|^2 = \frac{\eta^2 \varsigma_0^2}{\underline{\lambda_b}^2} \leq \frac{\eta^2 \lambda^2 \zeta^2}{\underline{\lambda_b}^2},
\end{aligned}
\tag{107}
$$

where $\varsigma_0^2 := \frac{1}{n_h} \sum_{i \in \mathcal{H}} \|\sum_{j \in \mathcal{H}_i} w_{ij} \nabla f_j(x^0) - \nabla f_{\mathcal{H}}(x^0)\|^2 \leq \lambda^2 \zeta^2$. Substituting (107) into the definition of $\mathcal{L}^0$ yields

$$\mathcal{L}^0 \leq \left(f_{\mathcal{H}}(\bar{x}^0) - f_{\mathcal{H}}^\star\right) + \frac{2\eta^3 L^2 v_1^2 v_2^2 \lambda^2 \zeta^2}{\underline{\lambda_b}^2 (1 - \gamma)} + \kappa z_2 \frac{\eta \lambda^2 \zeta^2}{\underline{\lambda_b}^2}. \tag{108}$$

For notational convenience, we collect the following constants:

$$
\begin{aligned}
a_1 &:= 12c_1 + \frac{28\kappa z_2}{1 - \gamma}, \; a_2 := z_2 \lambda_a^2 + \frac{56\kappa z_2}{1 - \gamma} + 24c_1, \; a_3 := \frac{56 L^2 v_1^2 v_2^2}{(1 - \gamma)^2}, \\
a_4 &:= \frac{112 v_1^2 v_2^2}{(1 - \gamma)^2} + \frac{120 \lambda_a^2 L^2 v_1^2 v_2^2}{(1 - \gamma)^2 \underline{\lambda_b}^2 n_h}, \; a_5 := \frac{2\lambda_a^2 L^2 v_1^2 v_2^2}{1 - \gamma}, \; a_6 := \frac{8 L^4 \lambda_a^2 v_1^2 v_2^2}{(1 - \gamma)^2 \underline{\lambda_b}^2 n_h}.
\end{aligned}
\tag{109}
$$

Combining the bound on $\mathcal{L}^0$ from (108) with the residual terms and substituting it into (106), we obtain

$$
\begin{aligned}
\frac{\eta}{8} \sum_{k=0}^{K-1} \mathbb{E}\left[\left\|\nabla f_{\mathcal{H}}(\bar{x}^k)\right\|^2\right] &\leq \left(f_{\mathcal{H}}(\bar{x}^0) - f_{\mathcal{H}}^\star\right) + \frac{2\eta^3 L^2 v_1^2 v_2^2 \lambda^2 \zeta^2}{\underline{\lambda_b}^2 (1 - \gamma)} + \kappa z_2 \frac{\eta \lambda^2 \zeta^2}{\underline{\lambda_b}^2} + a_1 \kappa \eta \zeta^2 K + a_2 \kappa \eta \sigma^2 K \\
&\quad + \frac{\eta^2 L \sigma^2 K}{n_h} + a_3 \kappa \zeta^2 \eta^3 K + a_4 \kappa \eta^3 \sigma^2 K + a_5 \eta^3 \sigma^2 K + a_6 \eta^5 \sigma^2 K.
\end{aligned}
\tag{110}
$$

Dividing both sides by $\eta K / 8$ and rearranging yields

$$
\begin{aligned}
\frac{1}{K} \sum_{k=0}^{K-1} \mathbb{E}\left[\left\|\nabla f_{\mathcal{H}}(\bar{x}^k)\right\|^2\right] &\leq \frac{8}{\eta K} \left(f_{\mathcal{H}}(\bar{x}^0) - f_{\mathcal{H}}^\star + \frac{2\eta^3 L^2 v_1^2 v_2^2 \lambda^2 \zeta^2}{\underline{\lambda_b}^2 (1 - \gamma)} + \kappa z_2 \frac{\eta \lambda^2 \zeta^2}{\underline{\lambda_b}^2}\right) + 8a_1 \kappa \zeta^2 + 8a_2 \kappa \sigma^2 \\
&\quad + \frac{8\eta L \sigma^2}{n_h} + 8a_3 \eta^2 \kappa \zeta^2 + 8a_4 \kappa \eta^2 \sigma^2 + 8a_5 \eta^2 \sigma^2 + 8a_6 \eta^4 \sigma^2.
\end{aligned}
\tag{111}
$$

Substituting (65) into (111) via the spectral bounds from Lemma 2, and choosing the learning rate $\eta = \Theta\left(\sqrt{n_h/K}\right)$, we

derive the final convergence bound for sufficiently large $K$:

$$
\frac{1}{K} \sum_{k=0}^{K-1} \mathbb{E}\left[\left\|\nabla f_{\mathcal{H}}(\bar{x}^k)\right\|^2\right] \leq \underbrace{\mathcal{O}\left(\frac{f_{\mathcal{H}}(x^0) - f_{\mathcal{H}}^\star}{\sqrt{n_h K}} + \frac{\sigma^2}{\sqrt{n_h K}}\right)}_{\text{dominant terms}}
$$

$$
+ \underbrace{\mathcal{O}\left(\frac{n_h \lambda^2 \sigma^2}{(1-\lambda)q\underline{\lambda}K} + \frac{n_h \lambda^2 \sigma^2}{(1-\lambda)^3 q\underline{\lambda}K^2} + \frac{n_h L^2 \lambda^2 \zeta^2}{(1-\lambda)^2 q\underline{\lambda}K^2}\right)}_{\text{topology-related terms}} \tag{112}
$$

$$
+ \underbrace{\mathcal{O}\left(\frac{\kappa n_h \sigma^2}{(1-\lambda)^3 q\underline{\lambda}K} + \frac{\kappa n_h \zeta^2}{(1-\lambda)^2 q\underline{\lambda}K}\right)}_{\text{Byzantine-induced terms}} + \underbrace{\mathcal{O}\left(\frac{\kappa \sigma^2}{(1-q)^2} + \frac{\kappa \zeta^2}{(1-q)^2}\right)}_{\text{asymptotic error}}.
$$

This completes the proof.

$\square$

**Corollary 1.** *Suppose that Assumptions 1, 2, 3, and 4 hold. Under the conditions of Theorem 1, suppose $\mathcal{A}(\cdot)$ satisfies the $(\delta, \kappa)$-robustness criterion with $\kappa = \mathcal{O}(\delta)$. The transient time of* BRED *in Algorithm 1 is*

$$
\tau = \mathcal{O}\left(\max\left\{\frac{n_h^3}{(1-\lambda)^2}, \frac{\delta^2 n_h^3}{(1-\lambda)^6}\right\}\right). \tag{113}
$$

*Proof.* - The algorithm achieves linear speedup when the higher-order terms in (112) are dominated by the $\mathcal{O}(1/\sqrt{n_h K})$ term. This requires each of the following conditions to hold:

$$
\frac{n_h \lambda^2 \sigma^2}{(1-\lambda)q\underline{\lambda}K} \lesssim \frac{1}{\sqrt{n_h K}}, \quad \frac{n_h \lambda^2 \sigma^2}{(1-\lambda)^3 q\underline{\lambda}K^2} \lesssim \frac{1}{\sqrt{n_h K}}, \quad \frac{n_h L^2 \lambda^2 \zeta^2}{(1-\lambda)^2 q\underline{\lambda}K^2} \lesssim \frac{1}{\sqrt{n_h K}}, \tag{114}
$$

$$
\frac{\kappa n_h \sigma^2}{(1-\lambda)^3 q\underline{\lambda}K} \lesssim \frac{1}{\sqrt{n_h K}}, \quad \frac{\kappa n_h \zeta^2}{(1-\lambda)^2 q\underline{\lambda}K} \lesssim \frac{1}{\sqrt{n_h K}}. \tag{115}
$$

Rearranging each condition in terms of $K$ yields the respective requirements:

$$
K \gtrsim \frac{n_h^3 \lambda^4}{(1-\lambda)^2 q^2 \underline{\lambda}^2}, \quad K \gtrsim \frac{n_h \lambda^{4/3}}{(1-\lambda)^2 q^{2/3} \underline{\lambda}^{2/3}}, \quad K \gtrsim \frac{n_h \lambda^{4/3}}{(1-\lambda)^{4/3} q^{2/3} \underline{\lambda}^{2/3}}, \tag{116}
$$

$$
K \gtrsim \frac{\kappa^2 n_h^3}{(1-\lambda)^6 q^2 \underline{\lambda}^2}, \quad K \gtrsim \frac{\kappa^2 n_h^3}{(1-\lambda)^4 q^2 \underline{\lambda}^2}. \tag{117}
$$

Taking the maximum over all conditions. Suppose $\mathcal{A}(\cdot)$ satisfies the $(\delta, \kappa)$-robustness criterion with $\kappa = \mathcal{O}(\delta)$. Then the transient time satisfies

$$
K \gtrsim \mathcal{O}\left(\max\left\{\frac{n_h^3}{(1-\lambda)^2}, \frac{\delta^2 n_h^3}{(1-\lambda)^6}\right\}\right), \tag{118}
$$

which completes the proof. $\square$

### D.3. Proof of the Supporting Lemmas

#### D.3.1. PROOF OF LEMMA 3

**Lemma 3.** *Suppose that Assumptions 2, 3, and 4 hold. For* BRED *in Algorithm 1, the following holds for all $k \in \{0, \cdots, K-1\}$:*

$$
\frac{1}{n_h} \sum_{k=0}^{K-1} \mathbb{E}\left[\left\|\boldsymbol{\xi}^k\right\|^2\right] \leq \sum_{k=0}^{K-1} \frac{15\kappa v_2^2 v_1^2}{\eta^2} \mathbb{E}\left[\left\|\hat{\mathbf{e}}^k\right\|^2\right] + 24\kappa K \sigma^2 + 12\kappa K \zeta^2. \tag{84}
$$

*Proof.* For each honest client $i \in \mathcal{H}$, the update rule in (81) yields

$$x_i^{k+\frac{1}{2}} = x_i^k + q\left(x_i^k - x_i^{k-1}\right) - \eta\left(g_i^k - qg_i^{k-1}\right). \tag{119}$$

Let $\bar{g}_i^k := \sum_{j \in \mathcal{H}_i} w_{ij} g_j^k$ denote the aggregated gradient over honest agents connected to agent $i$ (including $i$). By linearity, the averaged iterate satisfies

$$\bar{x}_i^{k+\frac{1}{2}} = \bar{x}_i^k + q\left(\bar{x}_i^k - \bar{x}_i^{k-1}\right) - \eta\left(\bar{g}_i^k - q\bar{g}_i^{k-1}\right).$$

For $k \in \{0, \cdots, K-1\}$, under Assumption 3 and Definition 2, the robust aggregation property implies

$$\mathbb{E}\left[\left\|R_i^{k+\frac{1}{2}} - \bar{x}_i^{k+\frac{1}{2}}\right\|^2\right] \leq \kappa \sum_{j \in \mathcal{H}_i} w_{ij} \mathbb{E}\left[\left\|x_j^{k+\frac{1}{2}} - \bar{x}_i^{k+\frac{1}{2}}\right\|^2\right], \tag{120}$$

where $R_i^{k+\frac{1}{2}} := \mathcal{A}_i\left(x_i^{k+\frac{1}{2}}, \{x_m^{k+\frac{1}{2}}\}_{m \in \mathcal{N}_i}\right)$ and $\bar{x}_i^{k+\frac{1}{2}} := \sum_{j \in \mathcal{H}_i} w_{ij} x_j^{k+\frac{1}{2}}$.

Substituting the update rule into (120) and applying the Cauchy–Schwarz inequality, we obtain

$$\mathbb{E}\left[\left\|R_i^{k+\frac{1}{2}} - \bar{x}_i^{k+\frac{1}{2}}\right\|^2\right] \leq \kappa \sum_{j \in \mathcal{H}_i} w_{ij} \mathbb{E}\left[\left\|x_j^k - \bar{x}_i^k + q(x_j^k - x_j^{k-1}) - q(\bar{x}_i^k - \bar{x}_i^{k-1}) - \eta\left(g_j^k - \bar{g}_i^k - qg_j^{k-1} + q\bar{g}_i^{k-1}\right)\right\|^2\right]$$

$$\leq 3\kappa \sum_{j \in \mathcal{H}_i} w_{ij}\left(\mathbb{E}\left[\left\|x_j^k - \bar{x}_i^k\right\|^2\right] + q^2\mathbb{E}\left[\left\|(x_j^k - x_j^{k-1}) - (\bar{x}_i^k - \bar{x}_i^{k-1})\right\|^2\right]\right.$$

$$\left. + \eta^2\mathbb{E}\left[\left\|g_j^k - \bar{g}_i^k - qg_j^{k-1} + q\bar{g}_i^{k-1}\right\|^2\right]\right). \tag{121}$$

For each $j \in \mathcal{H}_i$, using Assumption 3 and 4, we have

$$\sum_{j \in \mathcal{H}_i} w_{ij} \mathbb{E}\left[\left\|g_j^k - \bar{g}_j^k\right\|^2\right] = \sum_{j \in \mathcal{H}_i} w_{ij} \mathbb{E}\left[\left\|\nabla F_j(x_j^k; \varphi_j^k) - \sum_{c \in \mathcal{H}_i} w_{ic}\nabla F_c(x_c^k; \varphi_c^k)\right\|^2\right]$$

$$\leq \sum_{j \in \mathcal{H}_i} w_{ij}\left(\mathbb{E}\left[\left\|\nabla f_j(x_j^k) - \sum_{c \in \mathcal{H}_i} w_{ic}\nabla f_c(x_c^k)\right\|^2\right] + \mathbb{E}\left[\left\|\nabla f_j(x_j^k) - F_j(x_j^k; \varphi_j^k)\right\|^2\right]\right.$$

$$\left. + \mathbb{E}\left[\left\|\sum_{c \in \mathcal{H}_i} w_{ic}\nabla f_c(x_c^k) - \sum_{c \in \mathcal{H}_i} w_{ic}\nabla F_c(x_c^k; \varphi_c^k)\right\|^2\right]\right) \tag{122}$$

$$\leq \sum_{j \in \mathcal{H}_i} w_{ij} \mathbb{E}\left[\left\|\nabla f_j(x_j^k) - \sum_{c \in \mathcal{H}_i} w_{ic}\nabla f_c(x_c^k)\right\|^2\right] + 2\sigma^2$$

$$\leq \zeta^2 + 2\sigma^2.$$

Applying the Cauchy–Schwarz inequality again and using $0 < q < 1$, we have

$$\mathbb{E}\left[\left\|R_i^{k+\frac{1}{2}} - \bar{x}_i^{k+\frac{1}{2}}\right\|^2\right] \leq 9\kappa \sum_{j \in \mathcal{H}_i} w_{ij} \mathbb{E}\left[\left\|x_j^k - \bar{x}_i^k\right\|^2\right] + 6\kappa \sum_{j \in \mathcal{H}_i} w_{ij} \mathbb{E}\left[\left\|x_j^{k-1} - \bar{x}_i^{k-1}\right\|^2\right]$$

$$+ 24\eta^2\kappa\sigma^2 + 12\eta^2\kappa\zeta^2. \tag{123}$$

Summing (123) over $k = 0, \ldots, K-1$ and combining the shifted index terms yields

$$\sum_{k=0}^{K-1} \mathbb{E}\left[\left\|R_i^{k+\frac{1}{2}} - \bar{x}_i^{k+\frac{1}{2}}\right\|^2\right] \leq \sum_{k=0}^{K-1} 9\kappa \sum_{j \in \mathcal{H}_i} w_{ij} \mathbb{E}\left[\left\|x_j^k - \bar{x}_i^k\right\|^2\right] + \sum_{k=1}^{K-1} 6\kappa \sum_{j \in \mathcal{H}_i} w_{ij} \mathbb{E}\left[\left\|x_j^{k-1} - \bar{x}_i^{k-1}\right\|^2\right]$$

$$+ 24\eta^2\kappa K\sigma^2 + 12\eta^2\kappa K\zeta^2 \tag{124}$$

$$\leq \sum_{k=0}^{K-1} 15\kappa \sum_{j \in \mathcal{H}_i} w_{ij} \mathbb{E}\left[\left\|x_j^k - \bar{x}_i^k\right\|^2\right] + 24\eta^2\kappa K\sigma^2 + 12\eta^2\kappa K\zeta^2.$$

Under Assumption 1, the communication matrix $\mathbf{W}$ is doubly stochastic, we have

$$\sum_{i\in\mathcal{H}}\sum_{j\in\mathcal{H}_i} w_{ij}\mathbb{E}\left[\left\|x_j^k-\bar{x}_i^k\right\|^2\right] = \sum_{i\in\mathcal{H}}\sum_{j\in\mathcal{H}_i} w_{ij}\mathbb{E}\left[\left\|(x_j^k-\bar{x}^k)-(\bar{x}_i^k-\bar{x}^k)\right\|^2\right]$$
$$= \sum_{i\in\mathcal{H}}\sum_{j\in\mathcal{H}_i} w_{ij}\mathbb{E}\left[\left\|x_j^k-\bar{x}^k\right\|^2+\left\|\bar{x}_i^k-\bar{x}^k\right\|^2-2\left\langle x_j^k-\bar{x}^k,\bar{x}_i^k-\bar{x}^k\right\rangle\right]. \tag{125}$$

Note that $\sum_{j\in\mathcal{H}_i} w_{ij}x_j^k = \bar{x}_i^k$. Thus, the first part of the inner product becomes $(\bar{x}_i^k-\bar{x}^k)$. The cross term simplifies to $-2\left\|\bar{x}_i^k-\bar{x}^k\right\|^2$. Then, we obtain that

$$\sum_{i\in\mathcal{H}}\sum_{j\in\mathcal{H}_i} w_{ij}\mathbb{E}\left[\left\|x_j^k-\bar{x}_i^k\right\|^2\right] = \sum_{i\in\mathcal{H}}\mathbb{E}\left[\left(\sum_{j\in\mathcal{H}_i} w_{ij}\left\|x_j^k-\bar{x}^k\right\|^2\right)+\left\|\bar{x}_i^k-\bar{x}^k\right\|^2-2\left\|\bar{x}_i^k-\bar{x}^k\right\|^2\right]$$
$$= \sum_{i\in\mathcal{H}}\sum_{j\in\mathcal{H}_i} w_{ij}\mathbb{E}\left[\left\|x_j^k-\bar{x}^k\right\|^2\right]-\sum_{i\in\mathcal{H}}\mathbb{E}\left[\left\|\bar{x}_i^k-\bar{x}^k\right\|^2\right]$$
$$= \sum_{j\in\mathcal{H}}\mathbb{E}\left[\left\|x_j^k-\bar{x}^k\right\|^2\right]\underbrace{\left(\sum_{i\in\mathcal{H}_j} w_{ji}\right)}_{=1}-\sum_{i\in\mathcal{H}}\mathbb{E}\left[\left\|\bar{x}_i^k-\bar{x}^k\right\|^2\right] \tag{126}$$
$$= \sum_{i\in\mathcal{H}}\mathbb{E}\left[\left\|x_i^k-\bar{x}^k\right\|^2\right]-\underbrace{\sum_{i\in\mathcal{H}}\mathbb{E}\left[\left\|\bar{x}_i^k-\bar{x}^k\right\|^2\right]}_{\geq 0}\leq\sum_{i\in\mathcal{H}}\mathbb{E}\left[\left\|x_i^k-\bar{x}^k\right\|^2\right].$$

Since $\hat{\mathbf{U}}^\top\hat{\mathbf{U}}=\mathbf{I}$, we have

$$\left\|\hat{\mathbf{U}}^\top\mathbf{x}\right\|^2 = \mathbf{x}^\top\hat{\mathbf{U}}\hat{\mathbf{U}}^\top\hat{\mathbf{U}}\hat{\mathbf{U}}^\top\mathbf{x} = \left\|\hat{\mathbf{U}}\hat{\mathbf{U}}^\top\mathbf{x}^k\right\|^2 = \left\|\mathbf{x}^k-\bar{\mathbf{x}}^k\right\|^2. \tag{127}$$

By the definition of $\hat{\mathbf{e}}^k$, we obtain

$$\left\|v\hat{\mathbf{V}}\hat{\mathbf{e}}^k\right\|^2 = \left\|\hat{\mathbf{U}}^\top\mathbf{x}^k\right\|^2 + \left\|\hat{\mathbf{\Lambda}}_b^{-1}\hat{\mathbf{U}}^\top\mathbf{s}^k\right\|^2, \tag{128}$$

which implies $\sum_{i\in\mathcal{H}}\left\|x_i^k-\bar{x}^k\right\|^2 = \left\|\mathbf{x}^k-\bar{\mathbf{x}}^k\right\|^2 \leq v^2 v_1^2\left\|\hat{\mathbf{e}}^k\right\|^2$. Choosing $v=\sqrt{n_h}v_2$, combining (126), (128) with (124), and summing over all honest clients gives

$$\sum_{k=0}^{K-1}\frac{1}{|\mathcal{H}|}\sum_{i\in\mathcal{H}}\mathbb{E}\left[\left\|R_i^{k+\frac{1}{2}}-\bar{x}_i^{k+\frac{1}{2}}\right\|^2\right] \leq \sum_{k=0}^{K-1}15\kappa v_2^2 v_1^2\mathbb{E}\left[\left\|\hat{\mathbf{e}}^k\right\|^2\right] + 24\eta^2\kappa K\sigma^2 + 12\eta^2\kappa K\zeta^2. \tag{129}$$

Recalling the definition $\xi_i^k := \frac{1}{\eta}\left(R_i^{k+\frac{1}{2}}-\bar{x}_i^{k+\frac{1}{2}}\right)$ and $\boldsymbol{\xi}^k = col\{\xi_i^k\}_{i\in\mathcal{H}}$, we conclude that

$$\frac{1}{n_h}\sum_{k=0}^{K-1}\mathbb{E}\left[\left\|\boldsymbol{\xi}^k\right\|^2\right] \leq \sum_{k=0}^{K-1}\frac{15\kappa v_2^2 v_1^2}{\eta^2}\mathbb{E}\left[\left\|\hat{\mathbf{e}}^k\right\|^2\right] + 24\kappa K\sigma^2 + 12\kappa K\zeta^2. \tag{130}$$

This completes the proof. $\square$

### D.3.2. PROOF OF LEMMA 4

**Lemma 4.** *Suppose that Assumptions 1, 3, and 4 hold. For* BRED *in Algorithm 1, the following holds for all $k\geq 0$:*

$$\sum_{k=0}^{K-1}\mathbb{E}\left[\left\|\hat{\mathbf{e}}^{k+1}\right\|^2\right] \leq \sum_{k=0}^{K-1}\frac{\gamma+1}{2}\mathbb{E}\left[\left\|\hat{\mathbf{e}}^k\right\|^2\right] + \sum_{k=0}^{K-1}\frac{4\eta^4\lambda_a^2 L^2}{(1-\gamma)\underline{\lambda_b}^2}\mathbb{E}\left[\left\|\bar{v}^k\right\|^2\right] + \eta^2\lambda_a^2\sigma^2 K$$
$$+ \sum_{k=0}^{K-1}\frac{2\eta^2\lambda_a^2(1-q)^2}{(1-\gamma)\underline{\lambda_b}^2}\mathbb{E}\left[\left\|\nabla f_\mathcal{H}(\bar{x}^k)\right\|^2\right] + \frac{56\eta^2\kappa}{1-\gamma}\sigma^2 K + \frac{28\eta^2\kappa}{1-\gamma}\zeta^2 K, \tag{86}$$

*where $v_1 := \|\hat{\mathbf{V}}\|$, $v_2 := \left\|\hat{\mathbf{V}}^{-1}\right\|$, $\lambda_a := \|\mathbf{\Lambda}_a\|$, $\gamma := \|\mathbf{\Gamma}\| < 1$, and $\underline{\lambda_b} := \frac{1}{\|\mathbf{\Lambda}_b^{-1}\|}$.*

*Proof.* From the canonical recursion in (47), the consensus error evolves as

$$\hat{\mathbf{e}}^{k+1} = \boldsymbol{\Gamma}\hat{\mathbf{e}}^k - \frac{\eta}{v}\hat{\mathbf{V}}^{-1}\begin{bmatrix} \hat{\boldsymbol{\Lambda}}_a\hat{\mathbf{U}}^\top\left(\nabla\mathbf{f}_{\mathcal{H}}(\mathbf{x}^k) - \mathbf{N}^k + \mathbf{w}^k\right) + \hat{\mathbf{U}}^\top\boldsymbol{\xi}^k \\ \hat{\boldsymbol{\Lambda}}_b^{-1}\hat{\boldsymbol{\Lambda}}_a\hat{\mathbf{U}}^\top\left(\mathbf{N}^{k+1} - q\mathbf{N}^k\right) \end{bmatrix}, \tag{131}$$

where $\mathbf{N}^k := \nabla\mathbf{f}_{\mathcal{H}}(\bar{\mathbf{x}}^k)$ is the reference gradient and $\mathbf{w}^k := \mathbf{g}^k - \nabla\mathbf{f}_{\mathcal{H}}(\mathbf{x}^k)$ denotes the stochastic noise. Taking the conditional expectation and applying Assumption 3 to bound the noise term yields

$$\mathbb{E}\left[\left\|\hat{\mathbf{e}}^{k+1}\right\|^2\right] \leq \mathbb{E}\left[\left\|\boldsymbol{\Gamma}\hat{\mathbf{e}}^k - \frac{\eta}{v}\hat{\mathbf{V}}^{-1}\begin{bmatrix} \hat{\boldsymbol{\Lambda}}_a\hat{\mathbf{U}}^\top\left(\nabla\mathbf{f}_{\mathcal{H}}(\mathbf{x}^k) - \mathbf{N}^k\right) + \hat{\mathbf{U}}^\top\boldsymbol{\xi}^k \\ \hat{\boldsymbol{\Lambda}}_b^{-1}\hat{\boldsymbol{\Lambda}}_a\hat{\mathbf{U}}^\top\left(\mathbf{N}^{k+1} - q\mathbf{N}^k\right) \end{bmatrix}\right\|^2\right] + \frac{\eta^2 v_2^2\lambda_a^2\sigma^2 n_h}{v^2}. \tag{132}$$

To separate the contraction from the perturbation, we apply the weighted Young's inequality $\|a+b\|^2 \leq c^{-1}\|a\|^2 + (1-c)^{-1}\|b\|^2$ with $c = \gamma$, then we obtain that:

$$\mathbb{E}\left[\left\|\hat{\mathbf{e}}^{k+1}\right\|^2\right] \leq \gamma\mathbb{E}\left[\left\|\hat{\mathbf{e}}^k\right\|^2\right] + \frac{1}{1-\gamma}\cdot\frac{\eta^2 v_2^2}{v^2}\mathbb{E}\left[\left\|\begin{bmatrix} \hat{\boldsymbol{\Lambda}}_a\hat{\mathbf{U}}^\top\left(\nabla\mathbf{f}_{\mathcal{H}}(\mathbf{x}^k) - \mathbf{N}^k\right) + \hat{\mathbf{U}}^\top\boldsymbol{\xi}^k \\ \hat{\boldsymbol{\Lambda}}_b^{-1}\hat{\boldsymbol{\Lambda}}_a\hat{\mathbf{U}}^\top\left(\mathbf{N}^{k+1} - q\mathbf{N}^k\right) \end{bmatrix}\right\|^2\right] + \frac{\eta^2 v_2^2\lambda_a^2\sigma^2 n_h}{v^2}. \tag{133}$$

Expanding the squared norm and applying the Cauchy–Schwarz inequality, we obtain

$$\mathbb{E}\left[\left\|\hat{\mathbf{e}}^{k+1}\right\|^2\right] \leq \gamma\mathbb{E}\left[\left\|\hat{\mathbf{e}}^k\right\|^2\right] + \frac{2\eta^2 v_2^2}{(1-\gamma)v^2}\mathbb{E}\left[\left\|\boldsymbol{\xi}^k\right\|^2\right] + \frac{2\eta^2 v_2^2\lambda_a^2}{(1-\gamma)v^2}\underbrace{\mathbb{E}\left[\left\|\hat{\mathbf{U}}^\top\left(\nabla\mathbf{f}_{\mathcal{H}}(\mathbf{x}^k) - \mathbf{N}^k\right)\right\|^2\right]}_{\text{gradient residual}}$$

$$+ \frac{\eta^2 v_2^2\lambda_a^2}{(1-\gamma)v^2\underline{\lambda_b}^2}\underbrace{\mathbb{E}\left[\left\|\hat{\mathbf{U}}^\top\left(\mathbf{N}^{k+1} - q\mathbf{N}^k\right)\right\|^2\right]}_{\text{reference gradient difference}} + \frac{\eta^2 v_2^2\lambda_a^2\sigma^2 n_h}{v^2}. \tag{134}$$

We now bound the two gradient-related terms in (134). For the reference gradient difference, we decompose it as follows

$$\mathbf{N}^{k+1} - q\mathbf{N}^k = \left(\nabla\mathbf{f}_{\mathcal{H}}(\bar{\mathbf{x}}^{k+1}) - \nabla\mathbf{f}_{\mathcal{H}}(\bar{\mathbf{x}}^k)\right) + (1-q)\nabla\mathbf{f}_{\mathcal{H}}(\bar{\mathbf{x}}^k).$$

Applying the Cauchy–Schwarz inequality and Assumption 2 yields

$$\mathbb{E}\left[\left\|\hat{\mathbf{U}}^\top(\mathbf{N}^{k+1} - q\mathbf{N}^k)\right\|^2\right] \leq 2\mathbb{E}\left[\left\|\nabla\mathbf{f}_{\mathcal{H}}(\bar{\mathbf{x}}^{k+1}) - \nabla\mathbf{f}_{\mathcal{H}}(\bar{\mathbf{x}}^k)\right\|^2\right] + 2(1-q)^2\mathbb{E}\left[\left\|\nabla\mathbf{f}_{\mathcal{H}}(\bar{\mathbf{x}}^k)\right\|^2\right]$$

$$\leq 2L^2\mathbb{E}\left[\left\|\bar{\mathbf{x}}^{k+1} - \bar{\mathbf{x}}^k\right\|^2\right] + 2(1-q)^2\mathbb{E}\left[\left\|\nabla\mathbf{f}_{\mathcal{H}}(\bar{\mathbf{x}}^k)\right\|^2\right] \tag{135}$$

$$\leq 2\eta^2 L^2\mathbb{E}\left[\left\|\bar{\mathbf{v}}^k - \bar{\boldsymbol{\xi}}^k\right\|^2\right] + 2(1-q)^2\mathbb{E}\left[\left\|\nabla\mathbf{f}_{\mathcal{H}}(\bar{\mathbf{x}}^k)\right\|^2\right],$$

where we used $\bar{\mathbf{x}}^{k+1} - \bar{\mathbf{x}}^k = -\eta\bar{\mathbf{v}}^k - \eta\bar{\boldsymbol{\xi}}^k$ from (47).

For the gradient residual $\mathbf{R}^k := \nabla\mathbf{f}_{\mathcal{H}}(\mathbf{x}^k) - \mathbf{N}^k$, using Assumption 2 and the consensus error relation in (50), we obtain that

$$\mathbb{E}\left[\left\|\hat{\mathbf{U}}^\top\mathbf{R}^k\right\|^2\right] \leq \mathbb{E}\left[\left\|\nabla\mathbf{f}_{\mathcal{H}}(\mathbf{x}^k) - \nabla\mathbf{f}_{\mathcal{H}}(\bar{\mathbf{x}}^k)\right\|^2\right] \leq L^2\mathbb{E}\left[\left\|\mathbf{x}^k - \bar{\mathbf{x}}^k\right\|^2\right] \leq v^2 v_1^2 L^2\mathbb{E}\left[\left\|\hat{\mathbf{e}}^k\right\|^2\right]. \tag{136}$$

Substituting (135) and (136) into (134) and taking total expectations yields

$$\mathbb{E}\left[\left\|\hat{\mathbf{e}}^{k+1}\right\|^2\right] \leq \underbrace{\left(\gamma + \frac{2\eta^2 v_2^2 v_1^2 L^2\lambda_a^2}{1-\gamma}\right)}_{=:\rho}\mathbb{E}\left[\left\|\hat{\mathbf{e}}^k\right\|^2\right] + \frac{2\eta^2 v_2^2}{(1-\gamma)v^2}\mathbb{E}\left[\left\|\boldsymbol{\xi}^k\right\|^2\right] + \frac{2\eta^2 v_2^2\lambda_a^2(1-q)^2}{(1-\gamma)v^2\underline{\lambda_b}^2}\mathbb{E}\left[\left\|\nabla\mathbf{f}_{\mathcal{H}}(\bar{\mathbf{x}}^k)\right\|^2\right]$$

$$+ \frac{2\eta^4 v_2^2 \lambda_a^2 L^2}{(1-\gamma)v^2 \underline{\lambda_b}^2} \mathbb{E}\left[\left\|\bar{\mathbf{v}}^k - \bar{\boldsymbol{\xi}}^k\right\|^2\right] + \frac{\eta^2 v_2^2 \lambda_a^2 \sigma^2 n_h}{v^2}. \tag{137}$$

Summing (137) over $k = 0, \ldots, K-1$ and applying Cauchy-Schwarz inequality, we obtain

$$\sum_{k=0}^{K-1} \mathbb{E}\left[\left\|\hat{\mathbf{e}}^{k+1}\right\|^2\right] \le \sum_{k=0}^{K-1} \rho\, \mathbb{E}\left[\left\|\hat{\mathbf{e}}^k\right\|^2\right] + \frac{2\eta^2 v_2^2}{(1-\gamma)v^2} \sum_{k=0}^{K-1} \mathbb{E}\left[\left\|\boldsymbol{\xi}^k\right\|^2\right] + \frac{2\eta^2 v_2^2 \lambda_a^2 (1-q)^2}{(1-\gamma)v^2 \underline{\lambda_b}^2} \sum_{k=0}^{K-1} \mathbb{E}\left[\left\|\nabla \mathbf{f}_{\mathcal{H}}(\bar{\mathbf{x}}^k)\right\|^2\right]$$
$$+ \frac{4\eta^4 v_2^2 \lambda_a^2 L^2}{(1-\gamma)v^2 \underline{\lambda_b}^2} \sum_{k=0}^{K-1} \left(\mathbb{E}\left[\left\|\bar{\mathbf{v}}^k\right\|^2\right] + \mathbb{E}\left[\left\|\bar{\boldsymbol{\xi}}^k\right\|^2\right]\right) + \frac{\eta^2 v_2^2 \lambda_a^2 \sigma^2 n_h K}{v^2}. \tag{138}$$

By Jensen's inequality, we have

$$\mathbb{E}\left[\left\|\bar{\boldsymbol{\xi}}^k\right\|^2\right] = n_h \mathbb{E}\left[\left\|\bar{\xi}^k\right\|^2\right] = n_h \mathbb{E}\left[\left\|\frac{1}{n_h}\sum_{i\in\mathcal{H}}\xi_i^k\right\|^2\right] \le \sum_{i\in\mathcal{H}} \mathbb{E}\left[\left\|\xi_i^k\right\|^2\right] = \mathbb{E}\left[\left\|\boldsymbol{\xi}^k\right\|^2\right]. \tag{139}$$

Setting $\eta \le \frac{\underline{\lambda_b}}{4\lambda_a L}$, $v = \sqrt{n_h} v_2$ to simplify coefficients, we have

$$\sum_{k=0}^{K-1} \mathbb{E}\left[\left\|\hat{\mathbf{e}}^{k+1}\right\|^2\right] \le \sum_{k=0}^{K-1} \rho \mathbb{E}\left[\left\|\hat{\mathbf{e}}^k\right\|^2\right] + \frac{7\eta^2}{3(1-\gamma)n_h} \sum_{k=0}^{K-1} \mathbb{E}\left[\left\|\boldsymbol{\xi}^k\right\|^2\right] + \sum_{k=0}^{K-1} \frac{4\eta^4 \lambda_a^2 L^2}{(1-\gamma)n_h \underline{\lambda_b}^2} \mathbb{E}\left[\left\|\bar{\mathbf{v}}^k\right\|^2\right]$$
$$+ \sum_{k=0}^{K-1} \frac{2\eta^2 \lambda_a^2 (1-q)^2}{(1-\gamma)n_h \underline{\lambda_b}^2} \mathbb{E}\left[\left\|\nabla \mathbf{f}_{\mathcal{H}}(\bar{\mathbf{x}}^k)\right\|^2\right] + \eta^2 \lambda_a^2 \sigma^2 K + \frac{56\eta^2 \kappa K}{1-\gamma}\sigma^2 + \frac{28\eta^2 \kappa K}{1-\gamma}\zeta^2. \tag{140}$$

Invoking the aggregation error bound (130) in Lemma 3, we have

$$\sum_{k=0}^{K-1} \mathbb{E}\left[\left\|\hat{\mathbf{e}}^{k+1}\right\|^2\right] \le \sum_{k=0}^{K-1} \left(\gamma + \frac{35\kappa v_2^2 v_1^2}{1-\gamma} + \frac{2\eta^2 v_2^2 v_1^2 L^2 \lambda_a^2}{1-\gamma}\right) \mathbb{E}\left[\left\|\hat{\mathbf{e}}^k\right\|^2\right]$$
$$+ \sum_{k=0}^{K-1} \frac{4\eta^4 \lambda_a^2 L^2}{(1-\gamma)n_h \underline{\lambda_b}^2} \mathbb{E}\left[\left\|\bar{\mathbf{v}}^k\right\|^2\right] + \sum_{k=0}^{K-1} \frac{2\eta^2 \lambda_a^2 (1-q)^2}{(1-\gamma)n_h \underline{\lambda_b}^2} \mathbb{E}\left[\left\|\nabla \mathbf{f}_{\mathcal{H}}(\bar{\mathbf{x}}^k)\right\|^2\right]$$
$$+ \eta^2 \lambda_a^2 \sigma^2 K + \frac{56\eta^2 \kappa K}{1-\gamma}\sigma^2 + \frac{28\eta^2 \kappa K}{1-\gamma}\zeta^2. \tag{141}$$

To guarantee geometric contraction, the coefficient of the error term $\mathbb{E}\left[\left\|\hat{\mathbf{e}}^k\right\|^2\right]$ must satisfy the following stability condition:

$$\gamma + \frac{35\kappa v_2^2 v_1^2}{1-\gamma} + \frac{2\eta^2 v_2^2 v_1^2 L^2 \lambda_a^2}{1-\gamma} \le \frac{1+\gamma}{2}.$$

By substituting the bounds, we verify that this condition holds whenever the step size $\eta$ and the parameter $\kappa$ satisfy

$$\eta \le \frac{1-\gamma}{4Lv_1 v_2 \lambda_a} \quad \text{and} \quad \kappa \le \frac{(1-\gamma)^2}{140 v_1^2 v_2^2}. \tag{142}$$

Under these conditions, (141) simplifies to

$$\sum_{k=0}^{K-1} \mathbb{E}\left[\left\|\hat{\mathbf{e}}^{k+1}\right\|^2\right] \le \sum_{k=0}^{K-1} \frac{1+\gamma}{2}\mathbb{E}\left[\left\|\hat{\mathbf{e}}^k\right\|^2\right] + \sum_{k=0}^{K-1} \frac{4\eta^4 \lambda_a^2 L^2}{(1-\gamma)n_h \underline{\lambda_b}^2} \mathbb{E}\left[\left\|\bar{\mathbf{v}}^k\right\|^2\right] + \eta^2 \lambda_a^2 \sigma^2 K$$
$$+ \sum_{k=0}^{K-1} \frac{2\eta^2 \lambda_a^2 (1-q)^2}{(1-\gamma)n_h \underline{\lambda_b}^2} \mathbb{E}\left[\left\|\nabla \mathbf{f}_{\mathcal{H}}(\bar{\mathbf{x}}^k)\right\|^2\right] + \frac{56\eta^2 \kappa K}{1-\gamma}\sigma^2 + \frac{28\eta^2 \kappa K}{1-\gamma}\zeta^2. \tag{143}$$

This completes the proof. □

### D.3.3. PROOF OF LEMMA 5

**Lemma 5.** *Suppose that Assumptions 1 and 2 hold. For* BRED *in Algorithm 1, the following holds for all $k \geq 0$:*

$$
\begin{aligned}
f_{\mathcal{H}}(\bar{x}^K) - f_{\mathcal{H}}(\bar{x}^0) \leq & -\frac{\eta}{4} \sum_{k=0}^{K-1} \mathbb{E}\left[\|\nabla f_{\mathcal{H}}(\bar{x}^k)\|^2\right] + \sum_{k=0}^{K-1} \eta L^2 v_2^2 v_1^2 \mathbb{E}\left[\|\hat{\mathbf{e}}^k\|^2\right] + \frac{\eta^2 L \sigma^2 K}{n_h} \\
& + \sum_{k=0}^{K-1} \left(L\eta^2 - \frac{\eta}{2}\right) \mathbb{E}\left[\|\bar{\tilde{v}}^k\|^2\right] + \frac{\eta}{(1-q)^2} \sum_{k=0}^{K-1} \mathbb{E}\left[\|\bar{\xi}^k\|^2\right] + \sum_{k=0}^{K-1} \left(L\eta^2 + \eta\right) \mathbb{E}\left[\|\bar{\xi}^k\|^2\right].
\end{aligned}
\tag{89}
$$

*Proof.* The dynamics of the averaged iterate $\bar{x}^{k+1}$ are governed by the following recursive update rule

$$
\bar{x}^{k+1} = \bar{x}^k - \eta \bar{v}^k + \eta \bar{\xi}^k,
\tag{144}
$$

where we define the local update direction and its average as

$$
v_i^k := -\frac{q}{\eta}(x_i^k - x_i^{k-1}) + \left(g_i^k - q g_i^{k-1}\right), \quad \bar{v}^k := \frac{1}{|\mathcal{H}|} \sum_{i \in \mathcal{H}} v_i^k, \quad \bar{\xi}^k := \frac{1}{|\mathcal{H}|} \sum_{i \in \mathcal{H}} \xi_i^k.
\tag{145}
$$

We also define the noise-free counterpart of $v_i^k$ as follows

$$
\tilde{v}_i^k := -\frac{q}{\eta}(x_i^k - x_i^{k-1}) + \left(\nabla f_i(x_i^k) - q g_i^{k-1}\right), \quad \bar{\tilde{v}}^k := \frac{1}{|\mathcal{H}|} \sum_{i \in \mathcal{H}} \tilde{v}_i^k.
\tag{146}
$$

Therefore, using the definition of the averaged gradient and applying Assumption 3, we have

$$
\left\|\bar{v}^k - \bar{\tilde{v}}^k\right\|^2 = \left\|\frac{1}{|\mathcal{H}|} \sum_{i \in \mathcal{H}} \left(g_i^k - \nabla f_i(x_i)\right)\right\|^2 \leq \frac{\sigma^2}{n_h}.
\tag{147}
$$

By the $L$-smoothness of $f$ (Assumption 2), we have

$$
f_{\mathcal{H}}(\bar{x}^{k+1}) \leq f_{\mathcal{H}}(\bar{x}^k) + \mathbb{E}\left[\langle \nabla f_{\mathcal{H}}(\bar{x}^k), \bar{x}^{k+1} - \bar{x}^k \rangle\right] + \frac{L}{2} \mathbb{E}\left[\|\bar{x}^{k+1} - \bar{x}^k\|^2\right].
\tag{148}
$$

Substituting (144) into the above inequality yields

$$
f_{\mathcal{H}}(\bar{x}^{k+1}) \leq f_{\mathcal{H}}(\bar{x}^k) - \eta \mathbb{E}\left[\langle \nabla f_{\mathcal{H}}(\bar{x}^k), \bar{v}^k \rangle\right] + \eta \mathbb{E}\left[\langle \nabla f_{\mathcal{H}}(\bar{x}^k), \bar{\xi}^k \rangle\right] + \frac{L}{2} \mathbb{E}\left[\|-\eta \bar{v}^k + \eta \bar{\xi}^k\|^2\right].
\tag{149}
$$

Applying the identity $\langle a, b \rangle = \frac{1}{2}\|a\|^2 + \frac{1}{2}\|b\|^2 - \frac{1}{2}\|a - b\|^2$, we obtain

$$
-\eta \mathbb{E}\left[\langle \nabla f_{\mathcal{H}}(\bar{x}^k), \bar{v}^k \rangle\right] = -\eta \mathbb{E}\left[\langle \nabla f_{\mathcal{H}}(\bar{x}^k), \bar{\tilde{v}}^k \rangle\right] = \frac{\eta}{2} \mathbb{E}\left[\|\nabla f_{\mathcal{H}}(\bar{x}^k) - \bar{\tilde{v}}^k\|^2\right] - \frac{\eta}{2} \mathbb{E}\left[\|\nabla f_{\mathcal{H}}(\bar{x}^k)\|^2\right] - \frac{\eta}{2} \mathbb{E}\left[\|\bar{\tilde{v}}^k\|^2\right].
\tag{150}
$$

By Young's inequality $2\langle a, b \rangle \leq c\|a\|^2 + \frac{1}{c}\|b\|^2$, we have

$$
\mathbb{E}\left[\eta \langle \nabla f_{\mathcal{H}}(\bar{x}^k), \bar{\xi}^k \rangle\right] \leq \frac{1}{2}\left(\frac{\eta}{2} \mathbb{E}\left[\|\nabla f_{\mathcal{H}}(\bar{x}^k)\|^2\right] + 2\eta \mathbb{E}\left[\|\bar{\xi}^k\|^2\right]\right).
\tag{151}
$$

Substituting (150) and (151) into (149) gives

$$
f_{\mathcal{H}}(\bar{x}^{k+1}) \leq f_{\mathcal{H}}(\bar{x}^k) - \frac{\eta}{4} \mathbb{E}\left[\|\nabla f_{\mathcal{H}}(\bar{x}^k)\|^2\right] - \frac{\eta}{2} \mathbb{E}\left[\|\bar{\tilde{v}}^k\|^2\right] + \frac{\eta}{2} \mathbb{E}\left[\|\nabla f_{\mathcal{H}}(\bar{x}^k) - \bar{\tilde{v}}^k\|^2\right] + \eta \mathbb{E}\left[\|\bar{\xi}^k\|^2\right] + \frac{L\eta^2}{2} \mathbb{E}\left[\|\bar{v}^k - \bar{\xi}^k\|^2\right].
$$

Applying the Cauchy–Schwarz inequality and rearranging terms yields

$$
\begin{aligned}
f_{\mathcal{H}}(\bar{x}^{k+1}) \leq & f_{\mathcal{H}}(\bar{x}^k) - \frac{\eta}{4} \mathbb{E}\left[\|\nabla f_{\mathcal{H}}(\bar{x}^k)\|^2\right] + \frac{\eta}{2} \mathbb{E}\left[\|\nabla f_{\mathcal{H}}(\bar{x}^k) - \bar{\tilde{v}}^k\|^2\right] + L\eta^2 \mathbb{E}\left[\|\bar{v}^k\|^2\right] \\
& - \frac{\eta}{2} \mathbb{E}\left[\|\bar{\tilde{v}}^k\|^2\right] + \left(L\eta^2 + \eta\right) \mathbb{E}\left[\|\bar{\xi}^k\|^2\right].
\end{aligned}
\tag{152}
$$

Then, using Assumption 3, we obtain that

$$f_{\mathcal{H}}(\bar{x}^{k+1}) \leq f_{\mathcal{H}}(\bar{x}^k) - \frac{\eta}{4} \mathbb{E}\left[\left\|\nabla f_{\mathcal{H}}(\bar{x}^k)\right\|^2\right] + \frac{\eta}{2} \mathbb{E}\left[\left\|\nabla f_{\mathcal{H}}(\bar{x}^k) - \bar{\bar{v}}^k\right\|^2\right] + \left(L\eta^2 - \frac{\eta}{2}\right) \mathbb{E}\left[\left\|\bar{\bar{v}}^k\right\|^2\right]$$
$$+ \frac{L\eta^2\sigma^2}{n_h} + \left(L\eta^2 + \eta\right) \mathbb{E}\left[\left\|\bar{\xi}^k\right\|^2\right]. \tag{153}$$

Decomposing the bias term $\nabla f_{\mathcal{H}}(\bar{x}^k) - \bar{\bar{v}}^k$, and applying the Cauchy-Schwarz inequality, we have

$$f_{\mathcal{H}}(\bar{x}^{k+1}) \leq f_{\mathcal{H}}(\bar{x}^k) - \frac{\eta}{4} \mathbb{E}\left[\left\|\nabla f_{\mathcal{H}}(\bar{x}^k)\right\|^2\right] + \eta \mathbb{E}\left[\left\|\nabla f_{\mathcal{H}}(\bar{x}^k) - \overline{\nabla f_{\mathcal{H}}}(x^k)\right\|^2\right] + \eta \mathbb{E}\left[\left\|\overline{\nabla f_{\mathcal{H}}}(x^k) - \bar{\bar{v}}^k\right\|^2\right]$$
$$+ \left(L\eta^2 - \frac{\eta}{2}\right) \mathbb{E}\left[\left\|\bar{\bar{v}}^k\right\|^2\right] + \left(L\eta^2 + \eta\right) \mathbb{E}\left[\left\|\bar{\xi}^k\right\|^2\right] + \frac{L\eta^2\sigma^2}{n_h}. \tag{154}$$

By the $L$-smoothness of $f$ (Assumption 2) and Jensen's inequality, we have

$$\mathbb{E}\left[\left\|\nabla f_{\mathcal{H}}(\bar{x}^k) - \overline{\nabla f_{\mathcal{H}}}(x^k)\right\|^2\right] = \frac{1}{n_h^2} \mathbb{E}\left[\left\|\sum_{i\in\mathcal{H}} \left(\nabla f_i\left(\bar{x}^k\right) - \nabla f_i\left(x_i^k\right)\right)\right\|^2\right]$$
$$\leq \frac{1}{n_h} \sum_{i\in\mathcal{H}} \mathbb{E}\left[\left\|\nabla f_i\left(\bar{x}^k\right) - \nabla f_i\left(x_i^k\right)\right\|^2\right] \tag{155}$$
$$\leq \frac{L^2}{n_h} \sum_{i\in\mathcal{H}} \mathbb{E}\left[\left\|\bar{x}^k - x_i^k\right\|^2\right].$$

Summing (154) over $k = 0, \ldots, K-1$ and telescoping the left-hand side yields

$$f_{\mathcal{H}}(\bar{x}^K) - f_{\mathcal{H}}(\bar{x}^0) \leq -\frac{\eta}{4} \sum_{k=0}^{K-1} \mathbb{E}\left[\left\|\nabla f_{\mathcal{H}}(\bar{x}^k)\right\|^2\right] + \sum_{k=0}^{K-1} \frac{\eta L^2}{n_h} \sum_{i\in\mathcal{H}} \mathbb{E}\left[\left\|\bar{x}^k - x_i^k\right\|^2\right] + \eta \sum_{k=0}^{K-1} \mathbb{E}\left[\left\|\bar{\bar{v}}^k - \overline{\nabla f_{\mathcal{H}}}(x^k)\right\|^2\right]$$
$$+ \sum_{k=0}^{K-1} \left(L\eta^2 - \frac{\eta}{2}\right) \mathbb{E}\left[\left\|\bar{\bar{v}}^k\right\|^2\right] + \sum_{k=0}^{K-1} \left(L\eta^2 + \eta\right) \mathbb{E}\left[\left\|\bar{\xi}^k\right\|^2\right] + \frac{L\eta^2\sigma^2}{n_h}. \tag{156}$$

Since $\hat{\mathbf{U}}^\top \hat{\mathbf{U}} = \mathbf{I}$, we have

$$\left\|\hat{\mathbf{U}}^\top \mathbf{x}\right\|^2 = \mathbf{x}^\top \hat{\mathbf{U}} \hat{\mathbf{U}}^\top \hat{\mathbf{U}} \hat{\mathbf{U}}^\top \mathbf{x} = \left\|\hat{\mathbf{U}} \hat{\mathbf{U}}^\top \mathbf{x}^k\right\|^2 = \left\|\mathbf{x}^k - \bar{\mathbf{x}}^k\right\|^2. \tag{157}$$

By the definition of $\hat{\mathbf{e}}^k$, we obtain

$$\left\|v \hat{\mathbf{V}} \hat{\mathbf{e}}^k\right\|^2 = \left\|\hat{\mathbf{U}}^\top \mathbf{x}^k\right\|^2 + \left\|\hat{\mathbf{\Lambda}}_b^{-1} \hat{\mathbf{U}}^\top \mathbf{s}^k\right\|^2, \tag{158}$$

which implies $\mathbb{E}\left[\left\|\mathbf{x}^k - \bar{\mathbf{x}}^k\right\|^2\right] \leq v^2 v_1^2 \mathbb{E}\left[\left\|\hat{\mathbf{e}}^k\right\|^2\right]$. Choosing the scaling parameter $v = \sqrt{n_h} v_2$ and substituting this result into (156), we obtain

$$f_{\mathcal{H}}(\bar{x}^K) - f_{\mathcal{H}}(\bar{x}^0) \leq -\frac{\eta}{4} \sum_{k=0}^{K-1} \mathbb{E}\left[\left\|\nabla f_{\mathcal{H}}(\bar{x}^k)\right\|^2\right] + \sum_{k=0}^{K-1} \eta L^2 v_2^2 v_1^2 \mathbb{E}\left[\left\|\hat{\mathbf{e}}^k\right\|^2\right] + \frac{\eta^2 L\sigma^2 K}{n_h}$$
$$+ \eta \sum_{k=0}^{K-1} \mathbb{E}\left[\left\|\bar{\bar{v}}^k - \overline{\nabla f_{\mathcal{H}}}(x^k)\right\|^2\right] + \sum_{k=0}^{K-1} \left(L\eta^2 - \frac{\eta}{2}\right) \mathbb{E}\left[\left\|\bar{\bar{v}}^k\right\|^2\right] + \sum_{k=0}^{K-1} \left(L\eta^2 + \eta\right) \mathbb{E}\left[\left\|\bar{\xi}^k\right\|^2\right]. \tag{159}$$

Finally, invoking Lemma 6 to bound the aggregation bias term, we arrive at

$$f_{\mathcal{H}}(\bar{\mathbf{x}}^K) - f_{\mathcal{H}}(\bar{x}^0) \leq -\frac{\eta}{4} \sum_{k=0}^{K-1} \mathbb{E}\left[\left\|\nabla f_{\mathcal{H}}(\bar{x}^k)\right\|^2\right] + \sum_{k=0}^{K-1} \eta L^2 v_2^2 v_1^2 \mathbb{E}\left[\left\|\hat{\mathbf{e}}^k\right\|^2\right] + \frac{\eta^2 L\sigma^2 K}{n_h}$$
$$+ \sum_{k=0}^{K-1} \left(L\eta^2 - \frac{\eta}{2}\right) \mathbb{E}\left[\left\|\bar{\bar{v}}^k\right\|^2\right] + \frac{\eta}{(1-q)^2} \sum_{k=0}^{K-1} \mathbb{E}\left[\left\|\bar{\xi}^k\right\|^2\right] + \sum_{k=0}^{K-1} \left(L\eta^2 + \eta\right) \mathbb{E}\left[\left\|\bar{\xi}^k\right\|^2\right]. \tag{160}$$

This completes the proof. $\qquad\square$

D.3.4. Proof of Lemma 6

**Lemma 6.** *Suppose that Assumptions 1, 2, 3, and 4 hold. For* BRED *in Algorithm 1, the following holds for all $k \geq 0$:*

$$\sum_{k=0}^{K-1} \mathbb{E}\left[\left\|\bar{\tilde{v}}^k - \overline{\nabla f_{\mathcal{H}}}(x^k)\right\|^2\right] \leq \frac{1}{(1-q)^2} \sum_{k=0}^{K-1} \mathbb{E}\left[\left\|\bar{\xi}^k\right\|^2\right]. \tag{90}$$

*Proof.* By the definition of $\bar{\tilde{v}}^k$ and $\bar{x}^k = \bar{x}^{k-1} - \eta\bar{v}^{k-1} + \eta\bar{\xi}^{k-1}$, we obtain that

$$\mathbb{E}\left[\left\|\bar{\tilde{v}}^k - \overline{\nabla f_{\mathcal{H}}}(x^k)\right\|^2\right] = \mathbb{E}\left[\left\|-\frac{q}{\eta}\left(\bar{x}^k - \bar{x}^{k-1}\right) + \overline{\nabla f_{\mathcal{H}}}(x^k) - q\bar{g}^{k-1} - \overline{\nabla f_{\mathcal{H}}}(x^k)\right\|^2\right] \tag{161}$$

$$= \mathbb{E}\left[\left\|q\left(\bar{v}^{k-1} - \bar{\xi}^{k-1}\right) - q\bar{g}^{k-1}\right\|^2\right].$$

Similarly, by the recursive definition of $\bar{v}^k$, we obtain that

$$\bar{v}^k - \bar{g}^k = -\frac{q}{\eta}\left(\bar{x}^k - \bar{x}^{k-1}\right) - q\bar{g}^{k-1} = q(\bar{v}^{k-1} - \bar{g}^{k-1}) - q\bar{\xi}^{k-1}. \tag{162}$$

For $k \geq 1$, the aggregation bias can be expressed as

$$\mathbb{E}\left[\left\|\bar{\tilde{v}}^k - \overline{\nabla f_{\mathcal{H}}}(x^k)\right\|^2\right] = \mathbb{E}\left[\left\|q(\bar{v}^{k-1} - \bar{g}^{k-1}) - q\bar{\xi}^{k-1}\right\|^2\right]$$

$$= \mathbb{E}\left[\left\|q^2(\bar{v}^{k-2} - \bar{g}^{k-2}) - q^2\bar{\xi}^{k-2} - q\bar{\xi}^{k-1}\right\|^2\right] = \mathbb{E}\left[\left\|\sum_{t=0}^{k-1} q^{k-t}\bar{\xi}^t\right\|^2\right]. \tag{163}$$

Since $\sum_{t=0}^{k-1}(1-q)q^t \leq 1$ for $q \in (0,1)$, using Jensen's inequality, we have

$$\mathbb{E}\left[\left\|\bar{\tilde{v}}^k - \overline{\nabla f_{\mathcal{H}}}(x^k)\right\|^2\right] = \mathbb{E}\left[\left\|\sum_{t=0}^{k-1} q^{k-t}\bar{\xi}^t\right\|^2\right] \leq \frac{1}{1-q}\sum_{t=0}^{k-1} q^{k-t}\mathbb{E}\left[\left\|\bar{\xi}^t\right\|^2\right]. \tag{164}$$

Summing (290) over $k = 0, \ldots, K-1$ and exchanging the order of summation yields

$$\sum_{k=0}^{K-1} \mathbb{E}\left[\left\|\bar{\tilde{v}}^k - \overline{\nabla f_{\mathcal{H}}}(x^k)\right\|^2\right] = \sum_{k=1}^{K-1}\frac{1}{1-q}\sum_{t=0}^{k-1} q^{k-t}\mathbb{E}\left[\left\|\bar{\xi}^t\right\|^2\right] \leq \frac{1}{(1-q)^2}\sum_{k=0}^{K-1}\mathbb{E}\left[\left\|\bar{\xi}^k\right\|^2\right]. \tag{165}$$

This completes the proof. $\qquad\square$

# E. Convergence Analysis: Proof of Theorem 2 and Corollary 2

In this section, we establish the convergence guarantees for BRED-M (Algorithm 1 with momentum) by proving Theorem 2 and Corollary 2. We first present a proof outline in Section E.1, where we introduce four key lemmas: (i) an aggregation error bound, (ii) a consensus error bound, (iii) a momentum deviation bound capturing the discrepancy between the momentum buffer and the true gradient, and (iv) a descent lemma. In Section E.2, we combine these lemmas via a Lyapunov argument to prove the main convergence theorem and derive the transient time.

## E.1. Proof Outline

We begin by recalling the update rule of BRED-M in Algorithm 1. At each iteration $k$, every honest agent $i \in \mathcal{H}$ computes the momentum, performs a local update with scaled dual ascent, and then aggregates information from its neighbors using a robust aggregation rule:

$$m_i^k = \beta m_i^{k-1} + (1-\beta)\nabla F_i(x_i^k; \varphi_i^k),$$

$$x_i^{k+\frac{1}{2}} = x_i^k + q(x_i^k - x_i^{k-1}) - \eta(m_i^k - qm_i^{k-1}),$$
$$x_i^{k+1} = R_i^{k+\frac{1}{2}} = \mathcal{A}\big(x_i^{k+\frac{1}{2}}, \{x_m^{k+\frac{1}{2}}\}_{m \in \mathcal{N}_i}\big),$$

(166)

where $\beta \in (0, 1)$ is the momentum coefficient. Consistent with Subsection D.1, we define $\bar{x}_i^{k+\frac{1}{2}} := \sum_{j \in \mathcal{H}_i} w_{ij} x_j^{k+\frac{1}{2}}$.

With these definitions in place, the convergence analysis relies on bounding four key quantities: the *aggregation error* induced by robust aggregation, the *consensus error* measuring deviation from the global average, the *momentum deviation* quantifying the discrepancy between the momentum buffer $m_i^k$ and the true gradient, and the *descent* in objective value per iteration.

**Aggregation error.** Analogous to Lemma 3, we define the aggregation error as $\xi_i^k := \frac{1}{\eta}(R_i^{k+\frac{1}{2}} - \bar{x}_i^{k+\frac{1}{2}})$, and $\boldsymbol{\xi}^k = col\{\xi_i^k\}_{i \in \mathcal{H}}$ denotes the stacked vector of aggregation error across the honest clients. For the momentum variant, the bound involves an additional factor depending on $\beta$, with the proof deferred to Appendix E.3.2.

**Lemma 7.** *Suppose that Assumptions 2, 3, and 4 hold. For* BRED-M *in Algorithm 1, the following holds for all* $k \in \{0, \cdots, K-1\}$:

$$\frac{1}{n_h} \sum_{k=0}^{K-1} \mathbb{E}\left[\left\|\boldsymbol{\xi}^k\right\|^2\right] \le \sum_{k=0}^{K-1} \frac{15\kappa v_2^2 v_1^2}{\eta^2} \mathbb{E}\left[\left\|\hat{\mathbf{e}}^k\right\|^2\right] + 72\kappa K \left(\frac{1-\beta}{1+\beta}\right)\sigma^2 + 36\kappa K \zeta^2.$$

(167)

**Consensus error.** The consensus error analysis follows a similar approach as Lemma 4, where we show that bounding $\|\hat{\mathbf{e}}^k\|^2$ suffices to control the consensus error. For the momentum variant, the bound includes additional terms involving $\beta$, the detailed proof is provided in Appendix E.3.3.

**Lemma 8.** *Suppose that Assumptions 1, 3, and 4 hold. For* BRED-M *in Algorithm 1 with momentum, the following holds for all* $k \ge 0$:

$$\sum_{k=0}^{K-1} \mathbb{E}\left[\left\|\hat{\mathbf{e}}^{k+1}\right\|^2\right] \le \sum_{k=0}^{K-1} \frac{\gamma+1}{2} \mathbb{E}\left[\left\|\hat{\mathbf{e}}^k\right\|^2\right] + \sum_{k=0}^{K-1} \frac{4\eta^4 \lambda_a^2 L^2}{(1-\gamma)\underline{\lambda_b}^2} \mathbb{E}\left[\left\|\bar{v}^k\right\|^2\right] + \sum_{k=0}^{K-1} \frac{2\eta^2 \lambda_a^2 (1-q)^2}{(1-\gamma)\underline{\lambda_b}^2} \mathbb{E}\left[\left\|\nabla f_{\mathcal{H}}(\bar{x}^k)\right\|^2\right]$$
$$+ \left(\frac{1-\beta}{1+\beta}\right) \cdot \eta^2 \lambda_a^2 \sigma^2 K + \frac{2\eta^2 \lambda_a^2 \zeta^2}{(1-\gamma)\underline{\lambda_b}^2} + \frac{168\eta^2 \kappa}{1-\gamma}\left(\frac{1-\beta}{1+\beta}\right)\sigma^2 K + \frac{84\eta^2 \kappa}{1-\gamma}\zeta^2 K,$$

(168)

*where* $v_1 := \|\hat{\mathbf{V}}\|, v_2 := \left\|\hat{\mathbf{V}}^{-1}\right\|, \lambda_a := \|\boldsymbol{\Lambda}_a\|, \gamma := \|\boldsymbol{\Gamma}\| < 1$, *and* $\underline{\lambda_b} := \frac{1}{\|\boldsymbol{\Lambda}_b^{-1}\|}$.

**Momentum deviation.** A key distinction from the SGD case is the need to control the momentum deviation, which quantifies the discrepancy between the momentum buffer $m_i^k$ and the true gradient $\nabla f_i(x_i^k)$. This deviation arises because the momentum accumulates stale gradient information and can be amplified by Byzantine attacks. We define the averaged momentum deviation as

$$d^k := \bar{m}^k - \overline{\nabla f_{\mathcal{H}}}(x^k).$$

(169)

The following lemma bounds this quantity. The proof is provided in Appendix E.3.4.

**Lemma 9.** *Suppose that Assumptions 1, 2, 3, and 4 hold. For* BRED-M *in Algorithm 1, the following holds for all* $k \ge 0$:

$$\mathbb{E}\left[\left\|d^k\right\|^2\right] \le \beta^2 (1 + \eta L) \mathbb{E}\left[\left\|d^{k-1}\right\|^2\right] + (1 - \beta)^2 \frac{\sigma^2}{n_h} + 2\beta^2 \eta L (\eta L + 1) \left(\mathbb{E}\left[\left\|\bar{v}^{k-1}\right\|^2\right] + \mathbb{E}\left[\left\|\bar{\xi}^{k-1}\right\|^2\right]\right).$$

(170)

**Descent lemma.** Finally, we quantify the per-iteration progress in minimizing the objective $f_{\mathcal{H}}$. Based on (44c), the averaged iterate evolves according to $\bar{x}^{k+1} = \bar{x}^k - \eta \bar{v}^k + \eta \bar{\xi}^k$, where we define the auxiliary variables:

$$v_i^k := -\frac{q}{\eta}(x_i^k - x_i^{k-1}) + \big(m_i^k - qm_i^{k-1}\big), \quad \bar{v}^k := \frac{1}{n_h} \sum_{i \in \mathcal{H}} v_i^k, \quad \bar{\xi}^k := \frac{1}{n_h} \sum_{i \in \mathcal{H}} \xi_i^k.$$

(171)

Analogous to Lemma 5, this leads to a descent inequality that incorporates an additional term $\|d^k\|^2$ to account for the momentum deviation. The detailed proof is provided in Appendix E.3.5.

**Lemma 10.** *Suppose that Assumptions 1 and 2 hold. For* BRED-M *in Algorithm 1, the following holds for all $k \geq 0$:*

$$
\begin{aligned}
f_{\mathcal{H}}(\bar{x}^K) - f_{\mathcal{H}}(\bar{x}^0) \leq &-\frac{\eta}{4} \sum_{k=0}^{K-1} \mathbb{E}\left[\|\nabla f_{\mathcal{H}}(\bar{x}^k)\|^2\right] + \sum_{k=0}^{K-1} \eta L^2 v_2^2 v_1^2 \mathbb{E}\left[\|\hat{\mathbf{e}}^k\|^2\right] + 2\eta \sum_{k=0}^{K-1} \mathbb{E}\left[\|d^k\|^2\right] \\
&+ \sum_{k=0}^{K-1}\left(L\eta^2 - \frac{\eta}{2}\right)\mathbb{E}\left[\|\bar{v}^k\|^2\right] + \frac{2\eta}{(1-q)^2}\sum_{k=0}^{K-1}\mathbb{E}\left[\|\xi^k\|^2\right] + \sum_{k=0}^{K-1}\left(L\eta^2 + \eta\right)\mathbb{E}\left[\|\xi^k\|^2\right].
\end{aligned}
\tag{172}
$$

### E.2. Proof of Theorem 2

**Theorem 2.** *Suppose Assumptions 1–4 hold. Let the momentum coefficient satisfy $1 - \beta = \Theta(\eta L)$. For step size $\eta = \Theta(\sqrt{n_h/K})$,* BRED-M *in Algorithm 1 is $(\delta, \varepsilon_K)$-Byzantine-resilient with $\varepsilon_K$ as follows*

$$
\begin{aligned}
\frac{1}{K}\sum_{k=0}^{K-1}\mathbb{E}\left[\|\nabla f_{\mathcal{H}}(\bar{x}^k)\|^2\right] \leq &\underbrace{\mathcal{O}\left(\frac{f_{\mathcal{H}}(x^0) - f_{\mathcal{H}}^\star}{\sqrt{n_h K}} + \frac{\sigma^2}{\sqrt{n_h K}} + \frac{\kappa\sigma^2\sqrt{n_h}}{\sqrt{K}}\right)}_{\text{dominant terms}} \\
&+ \underbrace{\mathcal{O}\left(\frac{n_h^{3/2}\lambda^2\sigma^2}{(1-\lambda)q\underline{\lambda}K^{3/2}} + \frac{n_h\lambda^2\zeta^2}{(1-\lambda)^3 q\underline{\lambda}K^2} + \frac{n_h{}^2 L^4\lambda^2\zeta^2}{(1-\lambda)^2 q\underline{\lambda}K^3}\right)}_{\text{network topology}} \\
&+ \underbrace{\mathcal{O}\left(\frac{\kappa n_h^{3/2}\sigma^2}{(1-\lambda)^2 q\underline{\lambda}K^{3/2}} + \frac{\kappa n_h\zeta^2}{(1-\lambda)^2 q\underline{\lambda}K}\right)}_{\text{Byzantine-induced terms}} + \underbrace{\mathcal{O}\left(\frac{\kappa\zeta^2}{(1-q)^2}\right)}_{\text{asymptotic error}}.
\end{aligned}
\tag{173}
$$

*Proof.* Our convergence analysis proceeds via a Lyapunov argument. We define the Lyapunov function $\mathcal{L}^k$ as follows

$$
\mathcal{L}^k := \mathbb{E}\left[f_{\mathcal{H}}(\bar{x}^k) - f_{\mathcal{H}}^\star + z_3\mathbb{E}\left[\|d^k\|^2\right] + z_\eta\mathbb{E}\left[\|\hat{\mathbf{e}}^k\|^2\right]\right],
\tag{174}
$$

where $z_3 := \frac{1}{8L}$ controls the momentum deviation, and the composite term $z_\eta$ is given by

$$
z_\eta := z_1\eta + \frac{\kappa z_2}{\eta}, \quad z_1 := \frac{2L^2 v_2^2 v_1^2}{1-\gamma}, \quad z_2 := \frac{30 v_2^2 v_1^2 c_1}{1-\gamma}, \quad c_1 := 2 + \frac{2}{(1-q)^2}.
\tag{175}
$$

By applying Lemma 8, we bound the evolution of the weighted consensus error over $K$ iterations as:

$$
\begin{aligned}
\sum_{k=0}^{K-1}\mathbb{E}\left[z_\eta\|\hat{\mathbf{e}}^{k+1}\|^2 - z_\eta\|\hat{\mathbf{e}}^k\|^2\right] \leq &\sum_{k=0}^{K-1}\frac{z_\eta(\gamma-1)}{2}\mathbb{E}\left[\|\hat{\mathbf{e}}^k\|^2\right] + \sum_{k=0}^{K-1}\frac{4z_\eta\eta^4\lambda_a^2 L^2}{(1-\gamma)\underline{\lambda_b}^2}\mathbb{E}\left[\|\bar{v}^k\|^2\right] \\
&+ \sum_{k=0}^{K-1}\frac{2z_\eta\eta^2\lambda_a^2(1-q)^2}{(1-\gamma)\underline{\lambda_b}^2}\mathbb{E}\left[\|\nabla f_{\mathcal{H}}(\bar{x}^k)\|^2\right] + \left(\frac{1-\beta}{1+\beta}\right)\cdot z_\eta\eta^2\lambda_a^2\sigma^2 K \\
&+ \frac{2z_\eta\eta^2\lambda_a^2}{(1-\gamma)\underline{\lambda_b}^2}\zeta^2 + \frac{168 z_\eta\eta^2\kappa}{1-\gamma}\left(\frac{1-\beta}{1+\beta}\right)\sigma^2 K + \frac{84 z_\eta\eta^2\kappa}{1-\gamma}\zeta^2 K.
\end{aligned}
\tag{176}
$$

Applying Lemma 9, the momentum deviation term is bounded by:

$$
\begin{aligned}
\sum_{k=0}^{K-1}\frac{1}{8L}\left(\mathbb{E}\left[\|d^{k+1}\|^2\right] - \mathbb{E}\left[\|d^k\|^2\right]\right) \leq &\frac{\beta^2(1+\eta L)-1}{8L}\sum_{k=0}^{K-1}\mathbb{E}\left[\|d^k\|^2\right] + \frac{(1-\beta)^2\sigma^2 K}{8Ln_h} \\
&+ \frac{\beta^2\eta(\eta L+1)}{4}\sum_{k=0}^{K-1}\left(\mathbb{E}\left[\|\bar{v}^k\|^2\right] + \mathbb{E}\left[\|\bar{\xi}^k\|^2\right]\right).
\end{aligned}
\tag{177}
$$

Applying Lemma 10 yields the following descent inequality for the global objective:

$$f_{\mathcal{H}}(\bar{x}^K) - f_{\mathcal{H}}(\bar{x}^0) \leq -\frac{\eta}{4} \sum_{k=0}^{K-1} \mathbb{E}\left[\left\|\nabla f_{\mathcal{H}}(\bar{x}^k)\right\|^2\right] + \eta L^2 v_2^2 v_1^2 \sum_{k=0}^{K-1} \mathbb{E}\left[\left\|\hat{\mathbf{e}}^k\right\|^2\right] + 2\eta \sum_{k=0}^{K-1} \mathbb{E}\left[\left\|d^k\right\|^2\right]$$

$$+ \left(L\eta^2 - \frac{\eta}{2}\right) \sum_{k=0}^{K-1} \mathbb{E}\left[\left\|\bar{v}^k\right\|^2\right] + \left(L\eta^2 + \eta + \frac{2\eta}{(1-q)^2}\right) \sum_{k=0}^{K-1} \mathbb{E}\left[\left\|\bar{\xi}^k\right\|^2\right]. \tag{178}$$

Summing (177), (176), and (178) yields the following bound on the one-step drift of the Lyapunov function:

$$\sum_{k=0}^{K-1} \left(\mathcal{L}^{k+1} - \mathcal{L}^k\right) \leq \underbrace{\left(-\frac{\eta}{4} + \frac{2z_\eta \eta^2 \lambda_a^2 (1-q)^2}{(1-\gamma)\underline{\lambda_b}^2}\right)}_{=:\,\alpha_1} \sum_{k=0}^{K-1} \mathbb{E}\left[\left\|\nabla f_{\mathcal{H}}(\bar{x}^k)\right\|^2\right] + \underbrace{\left(\frac{z_\eta(\gamma-1)}{2} + \eta L^2 v_2^2 v_1^2\right)}_{=:\,\alpha_2} \sum_{k=0}^{K-1} \mathbb{E}\left[\left\|\hat{\mathbf{e}}^k\right\|^2\right]$$

$$+ \underbrace{\left(L\eta^2 - \frac{\eta}{2} + \frac{\eta}{4}(\eta L + 1) + \frac{4z_\eta \eta^4 \lambda_a^2 L^2}{(1-\gamma)\underline{\lambda_b}^2}\right)}_{=:\,\alpha_3} \sum_{k=0}^{K-1} \mathbb{E}\left[\left\|\bar{v}^k\right\|^2\right]$$

$$+ \underbrace{\left(L\eta^2 + \eta + \frac{\eta}{4}(\eta L + 1) + \frac{2\eta}{(1-q)^2}\right)}_{=:\,\alpha_4} \sum_{k=0}^{K-1} \mathbb{E}\left[\left\|\bar{\xi}^k\right\|^2\right]$$

$$+ \underbrace{\left(\frac{1}{8L}\left(16\eta L + \beta^2(1+\eta L) - 1\right)\right)}_{=:\,\alpha_5} \sum_{k=0}^{K-1} \mathbb{E}\left[\left\|d^k\right\|^2\right] + R_K^{(m)}, \tag{179}$$

where $R_K^{(m)}$ collects the residual terms:

$$R_K^{(m)} := \frac{(1-\beta)^2 \sigma^2 K}{8L n_h} + \left(\frac{1-\beta}{1+\beta}\right) z_\eta \eta^2 \lambda_a^2 \sigma^2 K + \frac{2z_\eta \eta^2 \lambda_a^2 \zeta^2}{(1-\gamma)\underline{\lambda_b}^2} + \frac{168 z_\eta \eta^2 \kappa}{1-\gamma}\left(\frac{1-\beta}{1+\beta}\right)\sigma^2 K + \frac{84 z_\eta \eta^2 \kappa}{1-\gamma}\zeta^2 K.$$

We now analyze the coefficients $\alpha_1, \alpha_3, \alpha_4, \alpha_5$ to ensure convergence. First, for the gradient coefficient $\alpha_1$, substituting $z_\eta = z_1 \eta + \kappa z_2 / \eta$ yields

$$\alpha_1 = -\frac{\eta}{4} + \frac{2\eta^3 z_1 \lambda_a^2 (1-q)^2}{(1-\gamma)\underline{\lambda_b}^2} + \frac{2\kappa z_2 \eta \lambda_a^2 (1-q)^2}{(1-\gamma)\underline{\lambda_b}^2}.$$

Under the step size condition $\eta \leq \frac{(1-\gamma)\underline{\lambda_b}}{8\lambda_a v_1 v_2 L(1-q)}$ and the constraint $\kappa \leq \frac{(1-\gamma)\underline{\lambda_b}^2}{32 z_2 \lambda_a^2 (1-q)^2}$, the positive terms are dominated by the negative drift

$$\alpha_1 \leq -\frac{\eta}{4} + \frac{\eta}{16} + \frac{\eta}{16} = -\frac{\eta}{8}. \tag{180}$$

Next, for the coefficient $\alpha_3$, substituting $z_\eta = z_1 \eta + \kappa z_2 / \eta$, we have

$$\alpha_3 = L\eta^2 - \frac{\eta}{2} + \frac{\eta}{4}(\eta L + 1) + \frac{4z_\eta \eta^4 \lambda_a^2 L^2}{(1-\gamma)\underline{\lambda_b}^2} = -\frac{\eta}{4} + \frac{5\eta^2 L}{4} + \frac{8\eta^5 v_1^2 v_2^2 \lambda_a^2 L^4}{(1-\gamma)^2 \underline{\lambda_b}^2} + \frac{4\eta^3 \kappa z_2 \lambda_a^2 L^2}{(1-\gamma)\underline{\lambda_b}^2}.$$

Provided that $\kappa \leq \frac{(1-\gamma)^2}{v_1^2 v_2^2}$ and the step size satisfies

$$\eta \leq \min\left\{\frac{1}{20L}, \frac{\sqrt{(1-\gamma)\underline{\lambda_b}}}{4L\sqrt{\lambda_a} v_1 v_2}, \frac{\underline{\lambda_b}}{48\lambda_a L \sqrt{c_1}}\right\},$$

we can bound the terms as follows:

$$\alpha_3 \leq -\frac{\eta}{4} + \frac{\eta}{16} + \frac{\eta}{16} + \frac{\eta}{8} = 0. \tag{181}$$

Then, we bound the aggregation error coefficient $\alpha_4$. Provided that $\eta \leq \frac{1}{20L}$, and defining the constant $c_1 := 2 + \frac{2}{(1-q)^2}$, we have

$$\alpha_4 = L\eta^2 + \eta + \frac{\eta}{4}(\eta L + 1) + \frac{2\eta}{(1-q)^2} \leq 2\eta + \frac{2\eta}{(1-q)^2} = c_1\eta. \tag{182}$$

Finally, for the momentum deviation coefficient $\alpha_5$, using $1 - \beta = 18\eta L$, we have

$$\alpha_5 = \frac{1}{8L}\left(16\eta L + \beta^2\eta L + \beta^2 - 1\right) = \frac{1}{8L}\left(16\eta L + \beta^2\eta L - 18\eta L\right) \leq -\frac{\eta}{8} \leq 0. \tag{183}$$

Since $\alpha_1 \leq -\eta/8$, $\alpha_3 \leq 0$, $\alpha_4 \leq c_1\eta$ and $\alpha_5 \leq 0$, substituting these bounds into (179) yields

$$\sum_{k=0}^{K-1} \mathbb{E}\left[\mathcal{L}^{k+1} - \mathcal{L}^k\right] \leq -\frac{\eta}{8}\sum_{k=0}^{K-1}\mathbb{E}\left[\left\|\nabla f_{\mathcal{H}}(\bar{x}^k)\right\|^2\right] + \alpha_2\sum_{k=0}^{K-1}\mathbb{E}\left[\left\|\hat{\mathbf{e}}^k\right\|^2\right] + c_1\eta\sum_{k=0}^{K-1}\mathbb{E}\left[\left\|\bar{\xi}^k\right\|^2\right] + R_K^{(m)}. \tag{184}$$

By Lemma 7, the aggregation error satisfies

$$\sum_{k=0}^{K-1}\mathbb{E}\left[\left\|\bar{\xi}^k\right\|^2\right] \leq \sum_{k=0}^{K-1}\frac{1}{n_h}\sum_{i\in\mathcal{H}}\mathbb{E}\left[\left\|\xi_i^k\right\|^2\right] \leq \sum_{k=0}^{K-1}\frac{15\kappa v_2^2 v_1^2}{\eta^2}\mathbb{E}\left[\left\|\hat{\mathbf{e}}^k\right\|^2\right] + 72\kappa K\left(\frac{1-\beta}{1+\beta}\right)\sigma^2 + 36\kappa K\zeta^2.$$

Substituting the aggregation error bound established in Lemma 7 into (184), we obtain that

$$\sum_{k=0}^{K-1}\left(\mathcal{L}^{k+1} - \mathcal{L}^k\right) \leq -\frac{\eta}{8}\sum_{k=0}^{K-1}\mathbb{E}\left[\left\|\nabla f_{\mathcal{H}}(\bar{x}^k)\right\|^2\right] + \underbrace{\left(\alpha_2 + \frac{15\kappa v_2^2 v_1^2 c_1}{\eta}\right)}_{=: \alpha_2'}\sum_{k=0}^{K-1}\mathbb{E}\left[\left\|\hat{\mathbf{e}}^k\right\|^2\right] + R_K'^{(m)}, \tag{185}$$

where $R_K'^{(m)}$ collects all residual terms depending on $\sigma^2$, $\zeta^2$, and $K$:

$$R_K'^{(m)} := R_K^{(m)} + 72c_1\eta\kappa K\left(\frac{1-\beta}{1+\beta}\right)\sigma^2 + 36c_1\eta\kappa K\zeta^2.$$

It remains to bound the consensus error coefficient $\alpha_2'$. Substituting the definition of $z_\eta$ into the expression for $\alpha_2'$, we obtain:

$$\alpha_2' = \frac{z_\eta(\gamma - 1)}{2} + \eta L^2 v_2^2 v_1^2 + \frac{15\kappa v_2^2 v_1^2 c_1}{\eta} = z_1\eta\frac{(\gamma - 1)}{2} + \eta L^2 v_2^2 v_1^2 \leq 0, \tag{186}$$

where the inequality follows from $\gamma < 1$. Since $\alpha_2' \leq 0$, we can drop the consensus error term. Telescoping the Lyapunov sum from $k = 0$ to $K - 1$ yields the convergence bound:

$$\sum_{k=0}^{K-1}\left(\mathcal{L}^{k+1} - \mathcal{L}^k\right) \leq -\frac{\eta}{8}\sum_{k=0}^{K-1}\mathbb{E}\left[\left\|\nabla f_{\mathcal{H}}(\bar{x}^k)\right\|^2\right] + R_K'^{(m)}. \tag{187}$$

Since $\mathcal{L}^K \geq 0$ (by the non-negativity of the potential function), rearranging gives

$$\frac{\eta}{8}\sum_{k=0}^{K-1}\mathbb{E}\left[\left\|\nabla f_{\mathcal{H}}(\bar{x}^k)\right\|^2\right] \leq \mathcal{L}^0 + R_K'^{(m)}. \tag{188}$$

It remains to bound the initial Lyapunov value $\mathcal{L}^0$. Unlike the standard SGD setting, the analysis requires an explicit bound on the initial momentum deviation. Given the initialization $m_i^0 = (1 - \beta)\nabla F_i(x_i^0; \varphi_i^0)$, we have

$$\mathbb{E}\left[\left\|d^0\right\|^2\right] = \mathbb{E}\left[\left\|(1 - \beta)\nabla F(\bar{x}^0; \xi^0) - \nabla f_{\mathcal{H}}(\bar{x}^0)\right\|^2\right] \leq (1 - \beta)^2\frac{\sigma^2}{n_h} + \beta^2\mathbb{E}\left[\left\|\nabla f_{\mathcal{H}}(\bar{x}^0)\right\|^2\right]. \tag{189}$$

Substituting the parameter choice $1 - \beta = 18\eta L$ and applying the standard smoothness property $\mathbb{E}[\|\nabla f_{\mathcal{H}}(\bar{x}^0)\|^2] \leq 2L(f_{\mathcal{H}}(\bar{x}^0) - f_{\mathcal{H}}^\star)$, we obtain the bound for the initial momentum deviation:

$$\mathbb{E}\left[\left\|d^0\right\|^2\right] \leq \frac{324\eta^2 L^2 \sigma^2}{n_h} + 2L(f_{\mathcal{H}}(\bar{x}^0) - f_{\mathcal{H}}^\star). \tag{190}$$

Assuming identical initialization across all agents, i.e., $\mathbf{x}^0 = \mathbf{1} \otimes x^0$, we have $\|\hat{\mathbf{U}}^\top \mathbf{x}^0\| = 0$ and $\mathbf{x}^0 = \bar{\mathbf{x}}^0$. Hence, for $\mathbf{A} = \mathbf{W}$, we have

$$
\left\|\hat{\mathbf{e}}^0\right\|^2 \leq \frac{1}{\underline{\lambda_b}^2 n_h} \left\|\hat{\mathbf{U}}^\top \mathbf{s}^0\right\|^2 = \frac{\eta^2}{\underline{\lambda_b}^2 n_h} \left\|\hat{\mathbf{U}}^\top \mathbf{W} \mathbf{N}^0\right\|^2 = \frac{\eta^2}{\underline{\lambda_b}^2 n_h} \left\|\hat{\mathbf{U}}^\top \mathbf{W}(1-\beta)\nabla\mathbf{f}\left(\mathbf{x}^0\right)\right\|^2
$$

$$
= \frac{\eta^2(1-\beta)^2}{\underline{\lambda_b}^2 n_h} \left\|\left(\mathbf{W} - \frac{1}{n_h}\mathbf{1}^\top\mathbf{1} \otimes I_d\right)\nabla\mathbf{f}\left(\mathbf{x}^0\right)\right\|^2 = \frac{\eta^2(1-\beta)^2\varsigma_0^2}{\underline{\lambda_b}^2} \leq \frac{324\eta^4\lambda^2 L^2\zeta^2}{\underline{\lambda_b}^2},
$$
(191)

where $\varsigma_0^2 := \frac{1}{n_h}\sum_{i\in\mathcal{H}}\|\sum_{j\in\mathcal{H}_i}w_{ij}\nabla f_j(x^0) - \nabla f_{\mathcal{H}}(x^0)\|^2 \leq \lambda^2\zeta^2$. Combining (191) and (188), we obtain

$$
\mathcal{L}^0 \leq \frac{5}{4}\left(f_{\mathcal{H}}(\bar{x}^0) - f_{\mathcal{H}}^\star\right) + \frac{81\eta^2 L\sigma^2}{2n_h} + \frac{648\eta^5 L^4 v_1^2 v_2^2\lambda^2\zeta^2}{\underline{\lambda_b}^2(1-\gamma)} + \frac{324\eta^3 L^2\kappa z_2\lambda^2\zeta^2}{\underline{\lambda_b}^2}.
$$
(192)

Combining (192) with the residual terms $R_K'^{(m)}$ and substituting into (188), we collect the auxiliary constants:

$$
a_1 := \frac{648 L^4 v_1^2 v_2^2\lambda^2}{\underline{\lambda_b}^2(1-\gamma)},\ a_2 := \frac{4\lambda_a^2 L^2 v_2^2 v_1^2}{(1-\gamma)^2\underline{\lambda_b}^2},\ a_3 := \frac{648 L^4 v_2^2 v_1^2\lambda^2}{(1-\gamma)\underline{\lambda_b}^2},
$$

$$
a_4 := 36c_1 + \frac{84\kappa z_2}{1-\gamma} + \frac{2z_2\lambda_a^2}{(1-\gamma)\underline{\lambda_b}^2 K},\ a_5 := \frac{3024\kappa z_2 L}{1-\gamma} + 1296c_1 L + \frac{18z_2\lambda_a^2 L}{1-\gamma},
$$
(193)

$$
a_6 := \frac{168 L^2 v_1^2 v_2^2}{(1-\gamma)^2},\ a_7 := \frac{6048 L^3 v_1^2 v_2^2}{(1-\gamma)^2},\ a_8 := \frac{36 v_1^2 v_2^2\lambda_a^2 L^3}{1-\gamma}.
$$

Using $1 - \beta \leq 18\eta L$, we obtain

$$
\frac{\eta}{8}\sum_{k=0}^{K-1}\mathbb{E}\left[\left\|\nabla f_{\mathcal{H}}(\bar{x}^k)\right\|^2\right] \leq \frac{5}{4}(f_{\mathcal{H}}(\bar{x}^0) - f_{\mathcal{H}}^\star) + \frac{81\eta^2 L\sigma^2}{2n_h} + \frac{81\eta^2 L\sigma^2 K}{2n_h} + a_1\eta^5\zeta^2 + a_2\zeta^2\eta^3 + a_3\kappa\eta^3\zeta^2
$$

$$
+ a_4\kappa\eta\zeta^2 K + a_5\kappa\sigma^2\eta^2 K + a_6\kappa\zeta^2\eta^3 K + a_7\kappa\eta^4\sigma^2 K + a_8\eta^4\sigma^2 K.
$$
(194)

Dividing both sides by $\eta K/8$ and rearranging yields

$$
\frac{1}{K}\sum_{k=0}^{K-1}\mathbb{E}\left[\left\|\nabla f_{\mathcal{H}}(\bar{x}^k)\right\|^2\right] \leq \frac{10(f_{\mathcal{H}}(\bar{x}^0) - f_{\mathcal{H}}^\star)}{\eta K} + \frac{324\eta L\sigma^2}{n_h} + \frac{324\eta L\sigma^2}{n_h K} + \frac{8a_1\eta^4\zeta^2}{K} + \frac{8a_2\eta^2\zeta^2}{K} + \frac{8a_3\kappa\eta^2\zeta^2}{K}
$$

$$
+ 8a_4\kappa\zeta^2 + 8a_5\kappa\eta\sigma^2 + 8a_6\kappa\eta^2\zeta^2 + 8a_7\kappa\eta^3\sigma^2 + 8a_8\eta^3\sigma^2.
$$
(195)

By Lemma 2, substituting the spectral quantities, choosing the learning rate to be $\eta = \Theta\left(\sqrt{n_h/K}\right)$, and assuming $K$ is large enough, we arrive at the final convergence bound:

$$
\frac{1}{K}\sum_{k=0}^{K-1}\mathbb{E}\left[\left\|\nabla f_{\mathcal{H}}(\bar{x}^k)\right\|^2\right] \leq \underbrace{\mathcal{O}\left(\frac{f_{\mathcal{H}}(x^0) - f_{\mathcal{H}}^\star}{\sqrt{n_h K}} + \frac{\sigma^2}{\sqrt{n_h K}} + \frac{\kappa\sigma^2\sqrt{n_h}}{\sqrt{K}}\right)}_{\text{dominant terms}}
$$

$$
+ \underbrace{\mathcal{O}\left(\frac{n_h^{3/2}\lambda^2\sigma^2}{(1-\lambda)q\underline{\lambda}K^{3/2}} + \frac{n_h\lambda^2\zeta^2}{(1-\lambda)^3 q\underline{\lambda}K^2} + \frac{n_h^2 L^4\lambda^2\zeta^2}{(1-\lambda)^2 q\underline{\lambda}K^3}\right)}_{\text{network topology}}
$$
(196)

$$
+ \underbrace{\mathcal{O}\left(\frac{\kappa n_h^{3/2}\sigma^2}{(1-\lambda)^2 q\underline{\lambda}K^{3/2}} + \frac{\kappa n_h\zeta^2}{(1-\lambda)^2 q\underline{\lambda}K}\right)}_{\text{Byzantine-induced terms}} + \underbrace{\mathcal{O}\left(\frac{\kappa\zeta^2}{(1-q)^2}\right)}_{\text{asymptotic error}}.
$$

This completes the proof. $\qquad\square$

**Corollary 2.** *Suppose that Assumptions 1, 2, 3, and 4 hold. Under the conditions of Theorem 2, suppose $\mathcal{A}(\cdot)$ satisfies the $(\delta, \kappa)$-robustness criterion with $\kappa = \mathcal{O}(\delta)$. Then the transient time of* BRED-M *in Algorithm 1 satisfies*

$$\tau = \mathcal{O}\left(\max\left\{\frac{n_h^2}{1-\lambda}\frac{n_h^2}{(1-\lambda)^2}, \frac{\delta^2 n_h^3}{(1-\lambda)^4}\right\}\right). \tag{197}$$

*Proof.* The proof follows the same approach as the standard SGD case (Corollary 1), which requires each of the following conditions to hold

$$\frac{n_h^{3/2}\lambda^2\sigma^2}{(1-\lambda)q\underline{\lambda}K^{3/2}} \lesssim \frac{1}{\sqrt{n_h K}}, \quad \frac{n_h\lambda^2\zeta^2}{(1-\lambda)^3 q\underline{\lambda}K^2} \lesssim \frac{1}{\sqrt{n_h K}}, \quad \frac{n_h^2 L^4\lambda^2\zeta^2}{(1-\lambda)^2 q\underline{\lambda}K^3} \lesssim \frac{1}{\sqrt{n_h K}}, \tag{198}$$

$$\frac{\kappa n_h^{3/2}\sigma^2}{(1-\lambda)^2 q\underline{\lambda}K^{3/2}} \lesssim \frac{1}{\sqrt{n_h K}}, \quad \frac{\kappa n_h\zeta^2}{(1-\lambda)^2 q\underline{\lambda}K} \lesssim \frac{1}{\sqrt{n_h K}}. \tag{199}$$

Rearranging each condition in terms of $K$ yields the respective requirements:

$$K \gtrsim \frac{n_h^2\lambda^2}{(1-\lambda)q\underline{\lambda}}, \quad K \gtrsim \frac{n_h\lambda^{4/3}}{(1-\lambda)^2 q^{2/3}\underline{\lambda}^{2/3}}, \quad K \gtrsim \frac{n_h\lambda^{4/5}}{(1-\lambda)^{4/5}q^{2/5}\underline{\lambda}^{2/5}}, \tag{200}$$

$$K \gtrsim \frac{\kappa n_h^2}{(1-\lambda)^2 q\underline{\lambda}}, \quad K \gtrsim \frac{\kappa^2 n_h^3}{(1-\lambda)^4 q^2\underline{\lambda}^2}. \tag{201}$$

Take the maximum over all conditions, and suppose $\mathcal{A}(\cdot)$ satisfies the $(\delta, \kappa)$-robustness criterion with $\kappa = \mathcal{O}(\delta)$. Then the transient time satisfies

$$K \gtrsim \mathcal{O}\left(\max\left\{\frac{n_h^2}{1-\lambda}, \frac{n_h}{(1-\lambda)^2}, \frac{\delta^2 n_h^3}{(1-\lambda)^4}\right\}\right). \tag{202}$$

$\square$

### E.3. Proof of Supporting Lemmas

E.3.1. TECHNICAL LEMMAS

**Lemma 11.** *Suppose that Assumptions 3 and 4 hold true. Consider Algorithm 1. For each $k \in [K-1]$, we have*

$$\sum_{j\in\mathcal{H}_i} w_{ij}\mathbb{E}\left[\left\|m_j^k - m_j^{k-1} - \bar{m}_i^k + \bar{m}_i^{k-1}\right\|^2\right] \le 24\left(\frac{1-\beta}{1+\beta}\right)\sigma^2 + 12\zeta^2, \tag{203}$$

*where $\bar{m}_i^k = \sum_{j\in\mathcal{H}_i} w_{ij}m_j^k$ is the average momentum update of honest nodes connected to agent $i$ (including $i$).*

*Proof.* Utilizing Cauchy-Schwarz inequality, we obtain that

$$\sum_{j\in\mathcal{H}_i} w_{ij}\mathbb{E}\left[\left\|m_j^k - m_j^{k-1} - \bar{m}_i^k + \bar{m}_i^{k-1}\right\|^2\right] \le \sum_{j\in\mathcal{H}_i} w_{ij}\left(2\mathbb{E}\left[\left\|m_j^k - \bar{m}_i^k\right\|^2\right] + 2\mathbb{E}\left[\left\|m_j^{k-1} - \bar{m}_i^{k-1}\right\|^2\right]\right). \tag{204}$$

We then analyze the momentum drift at step $k$ in the R.H.S. of (204). Recall from (166) that

$$m_i^k = \beta m_i^{k-1} + (1-\beta)\nabla F_i(x_i^k; \varphi_i^k), \quad \bar{m}_i^k = \sum_{j\in\mathcal{H}_i} w_{ij}m_j^k. \tag{205}$$

For an arbitrary $i \in \mathcal{H}$, we recall the following notation

$$g_i^k = \nabla F_i\left(x_i^k; \varphi_i^k\right), \quad \bar{g}_i^k := \sum_{j\in\mathcal{H}_i} w_{ij}g_j^k. \tag{206}$$

Then, we consider an arbitrary step $k \in [K-1]$. Expanding the sum in (205) we obtain that

$$m_i^k = (1-\beta) \sum_{t=0}^{k} \beta^{k-t} g_i^t. \tag{207}$$

Then, using the Cauchy-Schwarz inequality, we can divide our objective into three parts

$$
\begin{aligned}
\sum_{j \in \mathcal{H}_i} w_{ij} \mathbb{E} \left[ \left\| m_j^k - \bar{m}_i^k \right\|^2 \right] &= (1-\beta)^2 \sum_{j \in \mathcal{H}_i} w_{ij} \mathbb{E} \left[ \left\| \sum_{t=0}^{k} \beta^{k-t} \left( g_j^t - \bar{g}_i^t \right) \right\|^2 \right] \\
&\leq 3(1-\beta)^2 \sum_{j \in \mathcal{H}_i} w_{ij} \mathbb{E} \left[ \left\| \sum_{t=0}^{k} \beta^{k-t} \left( g_j^t - \nabla f_j(x_j^t) \right) \right\|^2 \right] \\
&+ 3(1-\beta)^2 \sum_{j \in \mathcal{H}_i} w_{ij} \mathbb{E} \left[ \left\| \sum_{t=0}^{k} \beta^{k-t} \left( \bar{g}_i^t - \sum_{c \in \mathcal{H}_i} w_{ic} \nabla f_c(x_c^t) \right) \right\|^2 \right] \\
&+ 3(1-\beta)^2 \sum_{j \in \mathcal{H}_i} w_{ij} \mathbb{E} \left[ \left\| \sum_{t=0}^{k} \beta^{k-t} \left( \nabla f_j(x_j^t) - \sum_{c \in \mathcal{H}_i} w_{ic} \nabla f_c(x_c^t) \right) \right\|^2 \right].
\end{aligned} \tag{208}
$$

Then, we analyze the three terms in the R.H.S. of (208), respectively.

For the first term in the R.H.S. of (208), we obtain that

$$
\begin{aligned}
\mathbb{E} \left[ \left\| \sum_{t=0}^{k} \beta^{k-t} \left( g_j^t - \nabla f_j(x_j^t) \right) \right\|^2 \right] &= \mathbb{E} \left[ \left\| \sum_{t=0}^{k-1} \beta^{k-t} \left( g_j^t - \nabla f_j(x_j^t) \right) + \left( g_j^t - \nabla f_j(x_j^t) \right) \right\|^2 \right] \\
&= \mathbb{E} \left[ \left\| \sum_{t=0}^{k-1} \beta^{k-t} \left( g_j^t - \nabla f_j(x_j^t) \right) \right\|^2 \right] + \mathbb{E} \left[ \left\| \left( g_j^t - \nabla f_j(x_j^t) \right) \right\|^2 \right] \\
&+ 2 \mathbb{E} \left[ \left\langle \sum_{t=0}^{k-1} \beta^{k-t} \left( g_j^t - \nabla f_i \left( x_j^t \right) \right), g_j^t - \nabla f_i \left( x_j^t \right) \right\rangle \right].
\end{aligned} \tag{209}
$$

Using Assumption 3 that $g_j^t = \nabla F_j(x_j^t; \varphi_j^t)$ is the unbiased estimation of $\nabla f_i \left( x_j^t \right)$, we obtain that

$$\mathbb{E} \left[ \left\langle \sum_{t=0}^{k} \beta^{k-t} \left( g_j^t - \nabla f_i \left( x_j^t \right) \right), g_j^t - \nabla f_i \left( x_j^t \right) \right\rangle \right] = 0. \tag{210}$$

Then, using Assumption 3, we have $\mathbb{E} \left[ \left\| \left( g_j^t - \nabla f_j(x_j^t) \right) \right\|^2 \right] \leq \sigma^2$. Combining the above results, we obtain that

$$
\begin{aligned}
\underbrace{\mathbb{E} \left[ \left\| \sum_{t=0}^{k} \beta^{k-t} \left( g_j^k - \nabla f_j(x_j^k) \right) \right\|^2 \right]}_{:=A^k} &= \mathbb{E} \left[ \left\| \sum_{t=0}^{k-1} \beta^{k-t} \left( g_j^t - \nabla f_j(x_j^t) \right) \right\|^2 \right] + \mathbb{E} \left[ \left\| \left( g_j^t - \nabla f_j(x_j^t) \right) \right\|^2 \right] \\
&\leq \mathbb{E} \left[ \left\| \sum_{t=0}^{k-1} \beta^{k-t} \left( g_j^t - \nabla f_j(x_j^t) \right) \right\|^2 \right] + \sigma^2 \\
&= \beta^2 \underbrace{\mathbb{E} \left[ \left\| \sum_{t=0}^{k-1} \beta^{k-1-t} \left( g_j^t - \nabla f_j(x_j^t) \right) \right\|^2 \right]}_{=A^{k-1}} + \sigma^2.
\end{aligned} \tag{211}
$$

Thus, we have

$$A^k \le \beta^2 A^{k-1} + \sigma^2. \tag{212}$$

By recursion, we obtain

$$A^k \le \beta^{2k} A^0 + \sigma^2 \sum_{t=0}^{k-1} \beta^{2t}. \tag{213}$$

Using the inequality that $A^0 = \mathbb{E}\left[\left\|g_j^0 - \nabla F_j(x_j^0; \varphi_j^0)\right\|^2\right] \le \sigma^2$, we obtain that

$$A^k \le \sigma^2 \sum_{t=0}^{k} \beta^{2t} \le \frac{\sigma^2}{1 - \beta^2}. \tag{214}$$

For the second term in the R.H.S. of (208), we denote

$$B^k := \mathbb{E}\left[\left\|\sum_{t=0}^{k} \beta^{k-t}\left(\bar{g}_i^t - \sum_{c \in \mathcal{H}_i} w_{ic} \nabla f_c(x_c^t)\right)\right\|^2\right]. \tag{215}$$

Using Assumption 3 and the mutual independence of stochastic gradients, we obtain that

$$\mathbb{E}\left[\left\|\bar{g}_i^k - \sum_{c \in \mathcal{H}} w_{ic} \nabla f_c(x_c^k)\right\|^2\right] = \sum_{c \in \mathcal{H}_i} w_{ic}^2 \mathbb{E}\left[\left\|g_c^k - \nabla f_c(x_c^k)\right\|^2\right] \le \sigma^2. \tag{216}$$

Thus, similar to the bound for $A^k$, we have

$$B^k \le \sigma^2 \sum_{t=0}^{k} \beta^{2t} \le \frac{\sigma^2}{(1 - \beta^2)}. \tag{217}$$

For the third term in the R.H.S. of(208), we denote

$$C^k := \sum_{j \in \mathcal{H}_i} w_{ij} \mathbb{E}\left[\left\|\sum_{t=0}^{k} \beta^{k-t}\left(\nabla f_j(x_j^t) - \sum_{c \in \mathcal{H}_i} w_{ic} \nabla f_c(x_c^t)\right)\right\|^2\right]. \tag{218}$$

Using Assumption 4 and the Cauchy-Schwarz inequality, we obtain that

$$\sum_{j \in \mathcal{H}_i} w_{ij} \mathbb{E}\left[\left\|\sum_{t=0}^{k} \beta^{k-t}\left(\nabla f_j(x_j^t) - \sum_{c \in \mathcal{H}_i} w_{ic} \nabla f_c(x_c^t)\right)\right\|^2\right]$$

$$\le \left(\sum_{t=0}^{k} \beta^{k-t}\right) \sum_{t=0}^{k} \beta^{k-t} \sum_{j \in \mathcal{H}_i} w_{ij} \mathbb{E}\left[\left\|\nabla f_j(x_j^t) - \sum_{c \in \mathcal{H}_i} w_{ic} \nabla f_c(x_c^t)\right\|^2\right]$$

$$\le \left(\sum_{t=0}^{k} \beta^{k-t}\right)^2 \zeta^2 \le \frac{\zeta^2}{(1-\beta)^2}. \tag{219}$$

Substituting from (214), (217) and (219) into (208), we obtain that

$$\sum_{j \in \mathcal{H}_i} w_{ij} \mathbb{E}\left[\left\|m_j^k - \bar{m}_i^k\right\|^2\right] \le 3(1-\beta)^2 \sum_{j \in \mathcal{H}_i} w_{ij}\left(\frac{\sigma^2}{1-\beta^2} + \frac{\sigma^2}{1-\beta^2} + \frac{\zeta^2}{(1-\beta)^2}\right)$$

$$= 6\frac{1-\beta}{1+\beta}\sigma^2 + 3\zeta^2. \tag{220}$$

Substituting the result in (220) into (204), we obtain that

$$\sum_{j \in \mathcal{H}_i} w_{ij} \mathbb{E}\left[\left\|m_j^k - m_j^{k-1} - \bar{m}_i^k + \bar{m}_i^{k-1}\right\|^2\right] \le 24\left(\frac{1-\beta}{1+\beta}\right)\sigma^2 + 12\zeta^2, \tag{221}$$

which finishes our proof. $\square$

E.3.2. PROOF OF LEMMA 7

**Lemma 7.** *Suppose that Assumptions 2, 3, and 4 hold. For* BRED-M *in Algorithm 1, the following holds for all* $k \in \{0, \cdots, K-1\}$:

$$\frac{1}{n_h} \sum_{k=0}^{K-1} \mathbb{E}\left[\left\|\boldsymbol{\xi}^k\right\|^2\right] \leq \sum_{k=0}^{K-1} \frac{15\kappa v_2^2 v_1^2}{\eta^2} \mathbb{E}\left[\left\|\hat{\mathbf{e}}^k\right\|^2\right] + 72\kappa K \left(\frac{1-\beta}{1+\beta}\right)\sigma^2 + 36\kappa K\zeta^2. \tag{167}$$

*Proof.* For every $i \in \mathcal{H}$, by the update rule of $x_i^{k+\frac{1}{2}}$ given in equation (166), we have

$$x_i^{k+\frac{1}{2}} = x_i^k + q\left(x_i^k - x_i^{k-1}\right) - \eta\left(m_i^k - qm_i^{k-1}\right). \tag{222}$$

We also denote the averaged momentum of client $i$ as $\bar{m}_i^k = \sum_{j \in \mathcal{H}_i} w_{ij} m_j^k$. Therefore, we have $\bar{x}_i^{k+\frac{1}{2}} = \bar{x}_i^k + q\left(\bar{x}_i^k - \bar{x}_i^{k-1}\right) - \eta\left(\bar{m}_i^k - q\bar{m}_i^{k-1}\right)$. Let $k \in \{0, \cdots, K-1\}$. Suppose that the Assumption 3 holds, and using the Definition 2, we have

$$\mathbb{E}\left[\left\|R_i^{k+\frac{1}{2}} - \bar{x}_i^{k+\frac{1}{2}}\right\|^2\right] \leq \kappa \sum_{j \in \mathcal{H}_i} w_{ij}\mathbb{E}\left[\left\|x_j^{k+\frac{1}{2}} - \bar{x}_i^{k+\frac{1}{2}}\right\|^2\right], \tag{223}$$

where $R_i^{k+\frac{1}{2}} = \mathcal{A}\left(x_i^{k+\frac{1}{2}}, \{x_m^{k+\frac{1}{2}}\}_{m \in \mathcal{N}_i}\right)$ and $\bar{x}_i^{k+\frac{1}{2}} = \sum_{j \in \mathcal{H}_i} w_{ij} x_j^{k+\frac{1}{2}}$, and $\mathcal{H}_i$ denotes honest agents connected to agent $i$ (including $i$).

As a result, using the Cauchy-Schwarz inequality, we can rewrite the aggregation error for every $i \in \mathcal{H}$ in ( 223) as follows

$$\mathbb{E}\left[\left\|R_i^{k+\frac{1}{2}} - \bar{x}_i^{k+\frac{1}{2}}\right\|^2\right] \leq \kappa \sum_{j \in \mathcal{H}_i} w_{ij}\mathbb{E}\left[\left\|x_j^k - \bar{x}_i^k + (x_j^k - x_j^{k-1}) - (\bar{x}_i^k - \bar{x}_i^{k-1}) - \eta\left(m_j^k - \bar{m}_i^k - m_j^{k-1} + \bar{m}_i^{k-1}\right)\right\|^2\right]$$

$$\leq 3\kappa \sum_{j \in \mathcal{H}_i} w_{ij}\left(\mathbb{E}\left[\left\|x_j^k - \bar{x}_i^k\right\|^2\right] + \mathbb{E}\left[\left\|(x_j^k - x_j^{k-1}) - (\bar{x}_i^k - \bar{x}_i^{k-1})\right\|^2\right]\right.$$

$$\left. + \eta^2\mathbb{E}\left[\left\|m_j^k - \bar{m}_i^k - m_j^{k-1} + \bar{m}_i^{k-1}\right\|^2\right]\right). \tag{224}$$

Using the result in Lemma 11, the Cauchy-Schwarz inequality, and the fact that $0 < q < 1$, we obtain the following result

$$\mathbb{E}\left[\left\|R_i^{k+\frac{1}{2}} - \bar{x}_i^{k+\frac{1}{2}}\right\|^2\right] \leq 9\kappa \sum_{j \in \mathcal{H}_i} w_{ij}\mathbb{E}\left[\left\|x_j^k - \bar{x}_i^k\right\|^2\right] + 6\kappa \sum_{j \in \mathcal{H}_i} w_{ij}\mathbb{E}\left[\left\|x_j^{k-1} - \bar{x}_i^{k-1}\right\|^2\right]$$

$$+ 72\eta^2\kappa\left(\frac{1-\beta}{1+\beta}\right)\sigma^2 + 36\eta^2\kappa\zeta^2. \tag{225}$$

Summing (225) from $k = 0$ to $K - 1$, we obtain that

$$\sum_{k=0}^{K-1}\mathbb{E}\left[\left\|R_i^{k+\frac{1}{2}} - \bar{x}_i^{k+\frac{1}{2}}\right\|^2\right] \leq \sum_{k=0}^{K-1} 9\kappa \sum_{j \in \mathcal{H}_i} w_{ij}\mathbb{E}\left[\left\|x_j^k - \bar{x}_i^k\right\|^2\right] + \sum_{k=1}^{K-1} 6\kappa \sum_{j \in \mathcal{H}_i} w_{ij}\mathbb{E}\left[\left\|x_j^{k-1} - \bar{x}_i^{k-1}\right\|^2\right]$$

$$+ 72\eta^2\kappa K\left(\frac{1-\beta}{1+\beta}\right)\sigma^2 + 36\eta^2\kappa K\zeta^2, \tag{226}$$

$$\leq \sum_{k=0}^{K-1} 15\kappa \sum_{j \in \mathcal{H}_i} w_{ij}\mathbb{E}\left[\left\|x_j^k - \bar{x}_i^k\right\|^2\right] + 72\eta^2\kappa K\left(\frac{1-\beta}{1+\beta}\right)\sigma^2 + 36\eta^2\kappa K\zeta^2.$$

Recall that $\hat{\mathbf{U}}^\top\mathbf{U} = \mathbf{I}$, we obtain that

$$\left\|\hat{\mathbf{U}}^\top\mathbf{x}\right\|^2 = \mathbf{x}^\top\hat{\mathbf{U}}\hat{\mathbf{U}}^\top\hat{\mathbf{U}}\hat{\mathbf{U}}^\top\mathbf{x} = \left\|\hat{\mathbf{U}}\hat{\mathbf{U}}^\top\mathbf{x}^k\right\|^2 = \left\|\mathbf{x}^k - \bar{\mathbf{x}}^k\right\|^2. \tag{227}$$

Using the definition for $\hat{\mathbf{e}}^k$, we obtain that

$$\left\| v\hat{\mathbf{V}}\hat{\mathbf{e}}^k \right\|^2 = \left\| \hat{\mathbf{U}}^\top \mathbf{x}^k \right\|^2 + \left\| \hat{\mathbf{\Lambda}}_b^{-1} \hat{\mathbf{U}}^\top \mathbf{s}^k \right\|^2 . \tag{228}$$

Thus, we have $\left\| \mathbf{x}^k - \bar{\mathbf{x}}^k \right\|^2 \leq v^2 v_1^2 \left\| \hat{\mathbf{e}}^k \right\|^2$. Combining with (226), we obtain that

$$\sum_{k=0}^{K-1} \frac{1}{|\mathcal{H}|} \sum_{i \in \mathcal{H}} \mathbb{E}\left[ \left\| R_i^{k+\frac{1}{2}} - \bar{x}_i^{k+\frac{1}{2}} \right\|^2 \right] \leq \sum_{k=0}^{K-1} \frac{15\kappa v^2 v_1^2}{n_h} \mathbb{E}\left[ \left\| \hat{\mathbf{e}}^k \right\|^2 \right] + 72\eta^2 \kappa K \left( \frac{1-\beta}{1+\beta} \right) \sigma^2 + 36\eta^2 \kappa K \zeta^2 . \tag{229}$$

Recalling the definition h $\xi_i^k := \frac{1}{\eta} \left( R_i^{k+\frac{1}{2}} - \bar{x}_i^{k+\frac{1}{2}} \right)$ and $\boldsymbol{\xi}^k = col\{\xi_i^k\}_{i \in \mathcal{H}}$, we conclude that

$$\frac{1}{n_h} \sum_{k=0}^{K-1} \mathbb{E}\left[ \left\| \boldsymbol{\xi}^k \right\|^2 \right] \leq \sum_{k=0}^{K-1} \frac{15\kappa v_2^2 v_1^2}{\eta^2} \mathbb{E}\left[ \left\| \hat{\mathbf{e}}^k \right\|^2 \right] + 72\kappa K \left( \frac{1-\beta}{1+\beta} \right) \sigma^2 + 36\kappa K \zeta^2 , \tag{230}$$

which finishes our proof.

$\square$

### E.3.3. PROOF OF LEMMA 8

**Lemma 8.** *Suppose that Assumptions 1, 3, and 4 hold. For* BRED-M *in Algorithm 1 with momentum, the following holds for all $k \geq 0$:*

$$\sum_{k=0}^{K-1} \mathbb{E}\left[ \left\| \hat{\mathbf{e}}^{k+1} \right\|^2 \right] \leq \sum_{k=0}^{K-1} \frac{\gamma+1}{2} \mathbb{E}\left[ \left\| \hat{\mathbf{e}}^k \right\|^2 \right] + \sum_{k=0}^{K-1} \frac{4\eta^4 \lambda_a^2 L^2}{(1-\gamma)\underline{\lambda_b}^2} \mathbb{E}\left[ \left\| \bar{v}^k \right\|^2 \right] + \sum_{k=0}^{K-1} \frac{2\eta^2 \lambda_a^2 (1-q)^2}{(1-\gamma)\underline{\lambda_b}^2} \mathbb{E}\left[ \left\| \nabla f_{\mathcal{H}}(\bar{x}^k) \right\|^2 \right]$$

$$+ \left( \frac{1-\beta}{1+\beta} \right) \cdot \eta^2 \lambda_a^2 \sigma^2 K + \frac{2\eta^2 \lambda_a^2 \zeta^2}{(1-\gamma)\underline{\lambda_b}^2} + \frac{168\eta^2 \kappa}{1-\gamma} \left( \frac{1-\beta}{1+\beta} \right) \sigma^2 K + \frac{84\eta^2 \kappa}{1-\gamma} \zeta^2 K , \tag{168}$$

*where* $v_1 := \|\hat{\mathbf{V}}\|, v_2 := \left\| \hat{\mathbf{V}}^{-1} \right\|, \lambda_a := \|\mathbf{\Lambda}_a\|, \gamma := \|\mathbf{\Gamma}\| < 1$, *and* $\underline{\lambda_b} := \frac{1}{\|\hat{\mathbf{\Lambda}}_b^{-1}\|}$.

*Proof.* Unlike the SGD case, the momentum estimator $\mathbf{m}^k = (1-\beta) \sum_{t=0}^k \beta^{k-t} \nabla \mathbf{F}_{\mathcal{H}}(\mathbf{x}^t; \boldsymbol{\varphi}^t)$ introduces correlated noise across iterations. Define the expected momentum $\tilde{\mathbf{m}}^k := \mathbb{E}[\mathbf{m}^k]$ and the noise term $\mathbf{w}^k := \tilde{\mathbf{m}}^k - \mathbf{m}^k$, we have

$$\mathbb{E}\left[ \left\| \mathbf{w}^k \right\|^2 \right] = (1-\beta)^2 \underbrace{\mathbb{E}\left[ \left\| \sum_{t=0}^k \beta^{k-t} \left( \nabla \mathbf{f}_{\mathcal{H}}(\mathbf{x}^t) - \nabla \mathbf{F}_{\mathcal{H}}(\mathbf{x}^t; \boldsymbol{\varphi}^t) \right) \right\|^2 \right]}_{=: A^k} . \tag{231}$$

By the independence of stochastic gradients across iterations and Assumption 3, $A^k$ satisfies the recursion $A^k \leq \beta^2 A^{k-1} + \sigma^2 n_h$, which yields

$$A^k \leq \sigma^2 \sum_{t=0}^k \beta^{2t} \leq \frac{\sigma^2 n_h}{1-\beta^2} . \tag{232}$$

Combining (231) and (232) gives the noise bound

$$\mathbb{E}\left[ \left\| \mathbf{w}^k \right\|^2 \right] \leq \frac{(1-\beta)^2 \sigma^2 n_h}{1-\beta^2} = \frac{1-\beta}{1+\beta} \cdot \sigma^2 n_h . \tag{233}$$

From the canonical recursion (47), the consensus error evolves as

$$\hat{\mathbf{e}}^{k+1} = \mathbf{\Gamma} \hat{\mathbf{e}}^k - \frac{\eta}{v} \mathbf{V}^{-1} \begin{bmatrix} \hat{\mathbf{\Lambda}}_a \hat{\mathbf{U}}^\top \left( \tilde{\mathbf{m}}^k - \mathbf{N}^k + \mathbf{w}^k \right) + \hat{\mathbf{U}}^\top \boldsymbol{\xi}^k \\ \hat{\mathbf{\Lambda}}_b^{-1} \hat{\mathbf{\Lambda}}_a \hat{\mathbf{U}}^\top \left( \mathbf{N}^{k+1} - q\mathbf{N}^k \right) \end{bmatrix} , \tag{234}$$

where $\mathbf{N}^k$ denotes the reference gradient (defined below). Taking the expectation and applying the noise bound from (234), we obtain

$$\mathbb{E}\left[\left\|\hat{\mathbf{e}}^{k+1}\right\|^2\right] \leq \mathbb{E}\left[\left\|\mathbf{\Gamma}\hat{\mathbf{e}}^k - \frac{\eta}{v}\mathbf{V}^{-1}\begin{bmatrix}\hat{\mathbf{\Lambda}}_a\hat{\mathbf{U}}^\top\left(\tilde{\mathbf{m}}^k - \mathbf{N}^k\right) + \hat{\mathbf{U}}^\top\boldsymbol{\xi}^k \\ \hat{\mathbf{\Lambda}}_b^{-1}\hat{\mathbf{\Lambda}}_a\hat{\mathbf{U}}^\top\left(\mathbf{N}^{k+1} - q\mathbf{N}^k\right)\end{bmatrix}\right\|^2\right] + \frac{1-\beta}{1+\beta}\cdot\frac{\eta^2 v_2^2\lambda_a^2\sigma^2 n_h}{v^2}. \tag{235}$$

To separate the contraction from the perturbation, we apply the weighted Young's inequality $\|a + b\|^2 \leq c^{-1}\|a\|^2 + (1 - c)^{-1}\|b\|^2$ with $c = \gamma$:

$$\mathbb{E}\left[\left\|\hat{\mathbf{e}}^{k+1}\right\|^2\right] \leq \gamma\mathbb{E}\left[\left\|\hat{\mathbf{e}}^k\right\|^2\right] + \frac{1}{1-\gamma}\cdot\frac{\eta^2 v_2^2}{v^2}\mathbb{E}\left[\left\|\begin{bmatrix}\hat{\mathbf{\Lambda}}_a\hat{\mathbf{U}}^\top\left(\tilde{\mathbf{m}}^k - \mathbf{N}^k\right) + \hat{\mathbf{U}}^\top\boldsymbol{\xi}^k \\ \hat{\mathbf{\Lambda}}_b^{-1}\hat{\mathbf{\Lambda}}_a\hat{\mathbf{U}}^\top\left(\mathbf{N}^{k+1} - q\mathbf{N}^k\right)\end{bmatrix}\right\|^2\right] + \frac{1-\beta}{1+\beta}\cdot\frac{\eta^2 v_2^2\lambda_a^2\sigma^2 n_h}{v^2}. \tag{236}$$

Expanding the squared norm and applying the Cauchy–Schwarz inequality, we obtain

$$\mathbb{E}\left[\left\|\hat{\mathbf{e}}^{k+1}\right\|^2\right] \leq \gamma\mathbb{E}\left[\left\|\hat{\mathbf{e}}^k\right\|^2\right] + \frac{2\eta^2 v_2^2}{(1-\gamma)v^2}\mathbb{E}\left[\left\|\boldsymbol{\xi}^k\right\|^2\right] + \frac{\eta^2 v_2^2\lambda_a^2}{(1-\gamma)v^2\underline{\lambda_b}^2}\underbrace{\mathbb{E}\left[\left\|\hat{\mathbf{U}}^\top\left(\mathbf{N}^{k+1} - q\mathbf{N}^k\right)\right\|^2\right]}_{\text{reference gradient}}$$
$$+ \frac{2\eta^2 v_2^2\lambda_a^2}{(1-\gamma)v^2}\underbrace{\mathbb{E}\left[\left\|\hat{\mathbf{U}}^\top\left(\tilde{\mathbf{m}}^k - \mathbf{N}^k\right)\right\|^2\right]}_{\text{momentum residual}} + \frac{1-\beta}{1+\beta}\cdot\frac{\eta^2 v_2^2\lambda_a^2\sigma^2 n_h}{v^2}. \tag{237}$$

We now bound the reference gradient term in (237). For the reference gradient, we define $\mathbf{N}^k$ as follows

$$\mathbf{N}^k := (1 - \beta)\sum_{t=0}^{k}\beta^{k-t}\nabla\mathbf{f}_{\mathcal{H}}(\bar{\mathbf{x}}^t). \tag{238}$$

By the Cauchy–Schwarz inequality, we obtain that

$$\mathbb{E}\left[\left\|\hat{\mathbf{U}}^\top\left(\mathbf{N}^{k+1} - q\mathbf{N}^k\right)\right\|^2\right] \leq \mathbb{E}\left[\left\|\hat{\mathbf{U}}^\top\left(\mathbf{N}^{k+1} - \mathbf{N}^k\right) + \hat{\mathbf{U}}^\top(1 - q)\mathbf{N}^k\right\|^2\right]$$
$$\leq 2\mathbb{E}\left[\left\|\hat{\mathbf{U}}^\top\left(\mathbf{N}^{k+1} - \mathbf{N}^k\right)\right\|^2\right] + 2\mathbb{E}\left[\left\|\hat{\mathbf{U}}^\top(1 - q)\mathbf{N}^k\right\|^2\right]. \tag{239}$$

Bounding the *first term* on the R.H.S of (239) using (238) and the inequality $(1 - \beta)\left(\sum_{t=0}^{k}\beta^{k-t} + \beta^{k+1}\right) \leq 1$, we obtain

$$\mathbb{E}\left[\left\|\hat{\mathbf{U}}^\top\left(\mathbf{N}^{k+1} - \mathbf{N}^k\right)\right\|^2\right] = \mathbb{E}\left[\left\|(1 - \beta)\left(\sum_{t=0}^{k}\beta^{k-t}\hat{\mathbf{U}}^\top\left(\nabla\mathbf{f}_{\mathcal{H}}(\bar{\mathbf{x}}^{t+1}) - \nabla\mathbf{f}_{\mathcal{H}}(\bar{\mathbf{x}}^t)\right) + \beta^{k+1}\hat{\mathbf{U}}^\top\nabla\mathbf{f}_{\mathcal{H}}(\bar{\mathbf{x}}^0)\right)\right\|^2\right]$$
$$\leq (1 - \beta)\sum_{t=0}^{k}\beta^{k-t}\mathbb{E}\left[\left\|\nabla\mathbf{f}_{\mathcal{H}}(\bar{\mathbf{x}}^{t+1}) - \nabla\mathbf{f}_{\mathcal{H}}(\bar{\mathbf{x}}^t)\right\|^2\right] + (1 - \beta)\beta^{k+1}\mathbb{E}\left[\left\|\hat{\mathbf{U}}^\top\nabla\mathbf{f}_{\mathcal{H}}(\bar{\mathbf{x}}^0)\right\|^2\right]. \tag{240}$$

Using the $\hat{\mathbf{U}}^\top\hat{\mathbf{U}} = \mathbf{I}$, we obtain that

$$\left\|\hat{\mathbf{U}}^\top\nabla\mathbf{f}_{\mathcal{H}}(\bar{\mathbf{x}}^0)\right\|^2 = (\nabla\mathbf{f}_{\mathcal{H}}(\bar{\mathbf{x}}^0))^\top\hat{\mathbf{U}}\hat{\mathbf{U}}^\top\hat{\mathbf{U}}\hat{\mathbf{U}}^\top\nabla\mathbf{f}_{\mathcal{H}}(\bar{\mathbf{x}}^0) = \left\|\hat{\mathbf{U}}\hat{\mathbf{U}}^\top\nabla\mathbf{f}_{\mathcal{H}}(\bar{\mathbf{x}}^0)\right\|^2 = \left\|\nabla\mathbf{f}_{\mathcal{H}}(\bar{\mathbf{x}}^0) - \overline{\nabla\mathbf{f}_{\mathcal{H}}}(\bar{\mathbf{x}}^0)\right\|^2. \tag{241}$$

Then, using the Assumption 4, we obtain that

$$\left\|\hat{\mathbf{U}}^\top\nabla\mathbf{f}_{\mathcal{H}}(\bar{\mathbf{x}}^0)\right\|^2 = \left\|\nabla\mathbf{f}_{\mathcal{H}}(\bar{\mathbf{x}}^0) - \overline{\nabla\mathbf{f}_{\mathcal{H}}}(\bar{\mathbf{x}}^0)\right\|^2 \leq n_h\zeta^2. \tag{242}$$

Combining (240) and (242), and using the L-smooth Assumption (Assumption 2) and Cauchy-Schwarz inequality, we have

$$\mathbb{E}\left[\left\|\hat{\mathbf{U}}^\top\left(\mathbf{N}^{k+1}-\mathbf{N}^k\right)\right\|^2\right] \leq \eta^2 L^2 \sum_{t=0}^k \beta^{k-t}(1-\beta)\mathbb{E}\left[\left\|\bar{\mathbf{v}}^t - \bar{\boldsymbol{\xi}}^t\right\|^2\right] + (1-\beta)\beta^{k+1}\mathbb{E}\left[\left\|\nabla\mathbf{f}_\mathcal{H}(\bar{\mathbf{x}}^0) - \overline{\nabla\mathbf{f}_\mathcal{H}}(\bar{\mathbf{x}}^0)\right\|^2\right]$$

$$\leq \eta^2 L^2 \sum_{t=0}^k \beta^{k-t}(1-\beta)\mathbb{E}\left[\left\|\bar{\mathbf{v}}^t - \bar{\boldsymbol{\xi}}^t\right\|^2\right] + (1-\beta)\beta^{k+1}n_h\zeta^2. \tag{243}$$

For the *second term* on the R.H.S of (239), using the Cauchy-Schwarz inequality, we obtain

$$\mathbb{E}\left[\left\|\hat{\mathbf{U}}^\top\mathbf{N}^k\right\|^2\right] \leq \mathbb{E}\left[\left\|(1-\beta)\sum_{t=0}^k \beta^{k-t}\nabla\mathbf{f}_\mathcal{H}(\bar{\mathbf{x}}^t)\right\|^2\right]$$

$$\leq \sum_{t=0}^k \beta^{k-t}(1-\beta)\mathbb{E}\left[\left\|\nabla\mathbf{f}_\mathcal{H}(\bar{\mathbf{x}}^t)\right\|^2\right]. \tag{244}$$

Substituting (243) and (244) into (239) and applying Cauchy-Schwarz inequality, we obtain

$$\mathbb{E}\left[\left\|\hat{\mathbf{U}}^\top\left(\mathbf{N}^{k+1}-q\mathbf{N}^k\right)\right\|^2\right] \leq 2\eta^2 L^2\boldsymbol{\varphi}^k + 2(1-\beta)\beta^{k+1}n_h\zeta^2 + 2\boldsymbol{\vartheta}^k, \tag{245}$$

where we define

$$\boldsymbol{\varphi}^k := \sum_{t=0}^k \beta^{k-t}(1-\beta)\mathbb{E}\left[\left\|\bar{\mathbf{v}}^t - \bar{\boldsymbol{\xi}}^t\right\|^2\right], \quad \boldsymbol{\vartheta}^k := (1-q)^2\sum_{t=0}^k \beta^{k-t}(1-\beta)\mathbb{E}\left[\left\|\nabla\mathbf{f}_\mathcal{H}(\bar{\mathbf{x}}^t)\right\|^2\right].$$

For the momentum residual in (237), define $\mathbf{R}^k := \tilde{\mathbf{m}}^k - \mathbf{N}^k$, which satisfies the recursion

$$\mathbf{R}^{k+1} = \beta\mathbf{R}^k + (1-\beta)\left(\nabla\mathbf{f}_\mathcal{H}(\mathbf{x}^{k+1}) - \nabla\mathbf{f}_\mathcal{H}(\bar{\mathbf{x}}^{k+1})\right). \tag{246}$$

By Assumption 2 and the Cauchy–Schwarz inequality, we obtain that

$$\mathbb{E}\left[\left\|\hat{\mathbf{U}}^\top\mathbf{R}^k\right\|^2\right] = \mathbb{E}\left[\left\|\hat{\mathbf{U}}^\top\left(\beta\mathbf{R}^{k-1} + (1-\beta)(\nabla\mathbf{f}_\mathcal{H}(\mathbf{x}^k) - \nabla\mathbf{f}_\mathcal{H}(\bar{\mathbf{x}}^k))\right)\right\|^2\right]$$

$$\leq \beta\mathbb{E}\left[\left\|\hat{\mathbf{U}}^\top\mathbf{R}^{k-1}\right\|^2\right] + (1-\beta)L^2\mathbb{E}\left[\left\|\mathbf{x}^k - \bar{\mathbf{x}}^k\right\|^2\right] \tag{247}$$

$$\leq \beta\mathbb{E}\left[\left\|\hat{\mathbf{U}}^\top\mathbf{R}^{k-1}\right\|^2\right] + (1-\beta)v^2v_1^2L^2\mathbb{E}\left[\left\|\hat{\mathbf{e}}^k\right\|^2\right].$$

By recursion, we obtain that

$$\mathbb{E}\left[\left\|\hat{\mathbf{U}}^\top\mathbf{R}^k\right\|^2\right] \leq v^2v_1^2L^2\sum_{t=0}^k \beta^{k-t}(1-\beta)\mathbb{E}\left[\left\|\hat{\mathbf{e}}^t\right\|^2\right] = v^2v_1^2L^2\boldsymbol{\psi}^k, \tag{248}$$

where we define $\boldsymbol{\psi}^k := \sum_{t=0}^k \beta^{k-t}(1-\beta)\mathbb{E}\left[\left\|\hat{\mathbf{e}}^t\right\|^2\right]$.

Substituting (248) and (245) into (237), we obtain the per-iteration bound:

$$\mathbb{E}\left[\left\|\hat{\mathbf{e}}^{k+1}\right\|^2\right] \leq \gamma\mathbb{E}\left[\left\|\hat{\mathbf{e}}^k\right\|^2\right] + \frac{2\eta^2 v_2^2}{(1-\gamma)v^2}\mathbb{E}\left[\left\|\boldsymbol{\xi}^k\right\|^2\right] + \frac{2\eta^4 v_2^2\lambda_a^2 L^2}{(1-\gamma)v^2\underline{\lambda_b}^2}\boldsymbol{\varphi}^k + \frac{2\eta^2 v_2^2 v_1^2 L^2\lambda_a^2}{1-\gamma}\boldsymbol{\psi}^k$$

$$+ \frac{2\eta^2 v_2^2\lambda_a^2}{(1-\gamma)v^2\underline{\lambda_b}^2}\boldsymbol{\vartheta}^k + \frac{2n_h\eta^2 v_2^2\lambda_a^2}{(1-\gamma)v^2\underline{\lambda_b}^2}(1-\beta)\beta^{k+1}\zeta^2 + \frac{1-\beta}{1+\beta}\cdot\frac{\eta^2 v_2^2\lambda_a^2\sigma^2 n_h}{v^2}. \tag{249}$$

Summing (250) over $k = 0, \dots, K-1$, we have

$$\sum_{k=0}^{K-1} \mathbb{E}\left[\left\|\hat{\mathbf{e}}^{k+1}\right\|^2\right] \leq \gamma \sum_{k=0}^{K-1} \mathbb{E}\left[\left\|\hat{\mathbf{e}}^k\right\|^2\right] + \frac{2\eta^2 v_2^2}{(1-\gamma)v^2} \sum_{k=0}^{K-1} \mathbb{E}\left[\left\|\boldsymbol{\xi}^k\right\|^2\right] + \frac{2\eta^4 v_2^2 \lambda_a^2 L^2}{(1-\gamma)v^2 \underline{\lambda_b}^2} \sum_{k=0}^{K-1} \varphi^k + \frac{2\eta^2 v_2^2 v_1^2 L^2 \lambda_a^2}{1-\gamma} \sum_{k=0}^{K-1} \psi^k$$

$$+ \frac{2\eta^2 v_2^2 \lambda_a^2}{(1-\gamma)v^2 \underline{\lambda_b}^2} \sum_{k=0}^{K-1} \vartheta^k + \frac{2n_h \eta^2 v_2^2 \lambda_a^2}{(1-\gamma)v^2 \underline{\lambda_b}^2} \sum_{k=0}^{K-1} (1-\beta)\beta^{k+1}\zeta^2 + \frac{1-\beta}{1+\beta} \cdot \frac{\eta^2 v_2^2 \lambda_a^2 \sigma^2 n_h K}{v^2}.$$

(250)

Then, we need to bound the auxiliary sums $\sum_{k=0}^{K-1} \varphi^k$, $\sum_{k=0}^{K-1} \psi^k$, $\sum_{k=0}^{K-1} \vartheta^k$, and $\sum_{k=0}^{K-1}(1-\beta)\beta^{k+1}\zeta^2$. Using the definition of $\varphi^k$ and exchanging the order of summation, we obtain

$$\sum_{k=0}^{K-1} \varphi^k = \sum_{k=0}^{K-1} \sum_{t=0}^{k} \beta^{k-t}(1-\beta)\mathbb{E}\left[\left\|\bar{\mathbf{v}}^t - \bar{\boldsymbol{\xi}}^t\right\|^2\right]$$

$$= \sum_{t=0}^{K-1} \mathbb{E}\left[\left\|\bar{\mathbf{v}}^t - \bar{\boldsymbol{\xi}}^t\right\|^2\right] \underbrace{\left(\sum_{k=t}^{K-1} \beta^{k-t}(1-\beta)\right)}_{\leq 1}$$

$$\leq \sum_{k=0}^{K-1} \mathbb{E}\left[\left\|\bar{\mathbf{v}}^k - \bar{\boldsymbol{\xi}}^k\right\|^2\right] \leq 2 \sum_{k=0}^{K-1} \mathbb{E}\left[\left\|\bar{\mathbf{v}}^k\right\|^2\right] + 2 \sum_{k=0}^{K-1} \mathbb{E}\left[\left\|\bar{\boldsymbol{\xi}}^k\right\|^2\right],$$

where the first inequality uses $\sum_{k=t}^{K-1} \beta^{k-t}(1-\beta) \leq 1$, and the second uses the Cauchy–Schwarz inequality.

Similarly, for $\psi^k = \sum_{t=0}^{k} \beta^{k-t}(1-\beta)\mathbb{E}\left[\|\hat{\mathbf{e}}^t\|^2\right]$, we exchange the order of summation:

$$\sum_{k=0}^{K-1} \psi^k = \sum_{k=0}^{K-1} \sum_{t=0}^{k} \beta^{k-t}(1-\beta)\mathbb{E}\left[\|\hat{\mathbf{e}}^t\|^2\right] = \sum_{t=0}^{K-1} \mathbb{E}\left[\|\hat{\mathbf{e}}^t\|^2\right] \sum_{k=t}^{K-1} \beta^{k-t}(1-\beta) \leq \sum_{k=0}^{K-1} \mathbb{E}\left[\|\hat{\mathbf{e}}^k\|^2\right], \qquad (251)$$

where we used $\sum_{k=t}^{K-1} \beta^{k-t}(1-\beta) = (1-\beta)\sum_{j=0}^{K-1-t} \beta^j \leq 1$.

For $\vartheta^k = (1-q)^2 \sum_{t=0}^{k} \beta^{k-t}(1-\beta)\mathbb{E}\left[\|\nabla\mathbf{f}_{\mathcal{H}}(\bar{\mathbf{x}}^t)\|^2\right]$, the same argument yields

$$\sum_{k=0}^{K-1} \vartheta^k \leq (1-q)^2 \sum_{k=0}^{K-1} \mathbb{E}\left[\|\nabla\mathbf{f}_{\mathcal{H}}(\bar{\mathbf{x}}^k)\|^2\right]. \qquad (252)$$

Thus, the auxiliary sums satisfy

$$\sum_{k=0}^{K-1} \psi^k \leq \sum_{k=0}^{K-1} \mathbb{E}\left[\|\hat{\mathbf{e}}^k\|^2\right], \quad \sum_{k=0}^{K-1} \vartheta^k \leq (1-q)^2 \sum_{k=0}^{K-1} \mathbb{E}\left[\|\nabla\mathbf{f}_{\mathcal{H}}(\bar{\mathbf{x}}^k)\|^2\right]. \qquad (253)$$

For the term $\sum_{k=0}^{K-1}(1-\beta)\beta^{k+1}\zeta^2$ in (250), using $\sum_{k=0}^{K-1} \beta^{k+1} = \frac{\beta(1-\beta^K)}{1-\beta}$, we have

$$\sum_{k=0}^{K-1}(1-\beta)\beta^{k+1}\zeta^2 = \beta(1-\beta^K)\zeta^2 \leq \zeta^2. \qquad (254)$$

Substituting the above result into (250), applying Jensen's inequality (139) (specifically $\left\|\bar{\boldsymbol{\xi}}^k\right\|^2 \leq \left\|\boldsymbol{\xi}^k\right\|^2$), and setting

$v = \sqrt{n_h}v_2$, we obtain

$$\sum_{k=0}^{K-1} \mathbb{E}\left[\left\|\hat{\mathbf{e}}^{k+1}\right\|^2\right] \leq \sum_{k=0}^{K-1} \left(\gamma + \frac{2\eta^2 v_2^2 v_1^2 L^2 \lambda_a^2}{1-\gamma}\right) \mathbb{E}\left[\left\|\hat{\mathbf{e}}^k\right\|^2\right] + \left(\frac{2\eta^2}{(1-\gamma)n_h} + \frac{4\eta^4\lambda_a^2 L^2}{(1-\gamma)n_h\underline{\lambda_b}^2}\right) \sum_{k=0}^{K-1} \mathbb{E}\left[\left\|\boldsymbol{\xi}^k\right\|^2\right]$$

$$+ \frac{4\eta^4\lambda_a^2 L^2}{(1-\gamma)n_h\underline{\lambda_b}^2} \sum_{k=0}^{K-1} \mathbb{E}\left[\left\|\bar{\mathbf{v}}^k\right\|^2\right] + \frac{2\eta^2\lambda_a^2(1-q)^2}{(1-\gamma)n_h\underline{\lambda_b}^2} \sum_{k=0}^{K-1} \mathbb{E}\left[\left\|\nabla\mathbf{f}_{\mathcal{H}}(\bar{\mathbf{x}}^k)\right\|^2\right] \tag{255}$$

$$+ \frac{1-\beta}{1+\beta}\eta^2\lambda_a^2\sigma^2 K + \frac{2\eta^2\lambda_a^2\zeta^2}{(1-\gamma)\underline{\lambda_b}^2}.$$

Using the learning rate condition $\eta \leq \frac{\lambda_b}{4\lambda_a L}$ to absorb the $\boldsymbol{\xi}^k$ terms, we obtain

$$\sum_{k=0}^{K-1} \mathbb{E}\left[\left\|\hat{\mathbf{e}}^{k+1}\right\|^2\right] \leq \sum_{k=0}^{K-1} \left(\gamma + \frac{2\eta^2 v_2^2 v_1^2 L^2 \lambda_a^2}{1-\gamma}\right) \mathbb{E}\left[\left\|\hat{\mathbf{e}}^k\right\|^2\right] + \frac{7\eta^2}{3(1-\gamma)n_h} \sum_{k=0}^{K-1} \mathbb{E}\left[\left\|\boldsymbol{\xi}^k\right\|^2\right]$$

$$+ \frac{4\eta^4\lambda_a^2 L^2}{(1-\gamma)n_h\underline{\lambda_b}^2} \sum_{k=0}^{K-1} \mathbb{E}\left[\left\|\bar{\mathbf{v}}^k\right\|^2\right] + \frac{2\eta^2\lambda_a^2(1-q)^2}{(1-\gamma)n_h\underline{\lambda_b}^2} \sum_{k=0}^{K-1} \mathbb{E}\left[\left\|\nabla\mathbf{f}_{\mathcal{H}}(\bar{\mathbf{x}}^k)\right\|^2\right] \tag{256}$$

$$+ \frac{1-\beta}{1+\beta}\eta^2\lambda_a^2\sigma^2 K + \frac{2\eta^2\lambda_a^2\zeta^2}{(1-\gamma)\underline{\lambda_b}^2}.$$

Invoking the aggregation error bound from Lemma 7, we obtain

$$\sum_{k=0}^{K-1} \mathbb{E}\left[\left\|\hat{\mathbf{e}}^{k+1}\right\|^2\right] \leq \underbrace{\left(\gamma + \frac{35\kappa v_2^2 v_1^2}{1-\gamma} + \frac{2\eta^2 v_2^2 v_1^2 L^2 \lambda_a^2}{1-\gamma}\right)}_{=:\rho} \sum_{k=0}^{K-1} \mathbb{E}\left[\left\|\hat{\mathbf{e}}^k\right\|^2\right]$$

$$+ \frac{4\eta^4\lambda_a^2 L^2}{(1-\gamma)n_h\underline{\lambda_b}^2} \sum_{k=0}^{K-1} \mathbb{E}\left[\left\|\bar{\mathbf{v}}^k\right\|^2\right] + \frac{2\eta^2\lambda_a^2(1-q)^2}{(1-\gamma)n_h\underline{\lambda_b}^2} \sum_{k=0}^{K-1} \mathbb{E}\left[\left\|\nabla\mathbf{f}_{\mathcal{H}}(\bar{\mathbf{x}}^k)\right\|^2\right] \tag{257}$$

$$+ \eta^2\lambda_a^2\left(\frac{1-\beta}{1+\beta}\right)\sigma^2 K + \frac{2\eta^2\lambda_a^2\zeta^2}{(1-\gamma)\underline{\lambda_b}^2} + \frac{168\eta^2\kappa}{1-\gamma}\left(\frac{1-\beta}{1+\beta}\right)\sigma^2 K + \frac{84\eta^2\kappa}{1-\gamma}\zeta^2 K.$$

To guarantee geometric contraction, we require $\rho$ to satisfy $\rho \leq \frac{1+\gamma}{2}$. This condition is satisfied provided that the step size $\eta$ and the robustness parameter $\kappa$ meet the following bounds

$$\eta \leq \frac{1-\gamma}{4Lv_1 v_2 \lambda_a} \quad \text{and} \quad \kappa \leq \frac{(1-\gamma)^2}{140 v_1^2 v_2^2}. \tag{258}$$

Under these conditions, (257) simplifies to

$$\sum_{k=0}^{K-1} \mathbb{E}\left[\left\|\hat{\mathbf{e}}^{k+1}\right\|^2\right] \leq \frac{1+\gamma}{2} \sum_{k=0}^{K-1} \mathbb{E}\left[\left\|\hat{\mathbf{e}}^k\right\|^2\right] + \frac{4\eta^4\lambda_a^2 L^2}{(1-\gamma)n_h\underline{\lambda_b}^2} \sum_{k=0}^{K-1} \mathbb{E}\left[\left\|\bar{\mathbf{v}}^k\right\|^2\right] + \frac{2\eta^2\lambda_a^2(1-q)^2}{(1-\gamma)n_h\underline{\lambda_b}^2} \sum_{k=0}^{K-1} \mathbb{E}\left[\left\|\nabla\mathbf{f}_{\mathcal{H}}(\bar{\mathbf{x}}^k)\right\|^2\right]$$

$$+ \eta^2\lambda_a^2\left(\frac{1-\beta}{1+\beta}\right)\sigma^2 K + \frac{2\eta^2\lambda_a^2\zeta^2}{(1-\gamma)\underline{\lambda_b}^2} + \frac{168\eta^2\kappa}{1-\gamma}\left(\frac{1-\beta}{1+\beta}\right)\sigma^2 K + \frac{84\eta^2\kappa}{1-\gamma}\zeta^2 K.$$

$$\tag{259}$$

This completes the proof. $\qquad\qquad\qquad\qquad\qquad\qquad\qquad\qquad\qquad\qquad\qquad\qquad\qquad\qquad\qquad\qquad\qquad\square$

### E.3.4. PROOF OF LEMMA 9

**Lemma 9.** *Suppose that Assumptions 1, 2, 3, and 4 hold. For* BRED-M *in Algorithm 1, the following holds for all $k \geq 0$:*

$$\mathbb{E}\left[\left\|d^k\right\|^2\right] \leq \beta^2(1+\eta L)\mathbb{E}\left[\left\|d^{k-1}\right\|^2\right] + (1-\beta)^2\frac{\sigma^2}{n_h} + 2\beta^2\eta L\left(\eta L + 1\right)\left(\mathbb{E}\left[\left\|\bar{v}^{k-1}\right\|^2\right] + \mathbb{E}\left[\left\|\bar{\xi}^{k-1}\right\|^2\right]\right). \tag{170}$$

*Proof.* For simplicity, we define the momentum deviation as follows

$$d^k := \bar{m}^k - \overline{\nabla f_{\mathcal{H}}}(x^k), \quad \bar{g}^k = \frac{1}{n_h} \sum_{i \in \mathcal{H}} g_i^k. \tag{260}$$

Consider an arbitrary step $k > 1$. Following the update of momentum in (166), we obtain that

$$d^k = \beta \bar{m}^{k-1} + (1 - \beta)\bar{g}^k - \overline{\nabla f_{\mathcal{H}}}(x^k). \tag{261}$$

Upon adding and subtracting $\beta \overline{\nabla f_{\mathcal{H}}}(x^{k-1})$ and $\beta \bar{g}^k$ on the R.H.S. of (261) we obtain that

$$\begin{aligned}
d^k &= \beta \bar{m}^{k-1} - \beta \overline{\nabla f_{\mathcal{H}}}(x^{k-1}) + (1-\beta)\bar{g}^k - \overline{\nabla f_{\mathcal{H}}}(x^k) + \beta \overline{\nabla f_{\mathcal{H}}}(x^k) + \beta \left( \overline{\nabla f_{\mathcal{H}}}(x^{k-1}) - \overline{\nabla f_{\mathcal{H}}}(x^k) \right) \\
&= \beta(\bar{m}^{k-1} - \overline{\nabla f_{\mathcal{H}}}(x^{k-1})) + (1-\beta)\bar{g}^k - (1-\beta)\overline{\nabla f_{\mathcal{H}}}(x^k) + \beta \left( \overline{\nabla f_{\mathcal{H}}}(x^{k-1}) - \overline{\nabla f_{\mathcal{H}}}(x^k) \right).
\end{aligned} \tag{262}$$

Given $m^{k-1} - \overline{\nabla f_{\mathcal{H}}}(x^{k-1}) = d^{k-1}$, thus we obtain that

$$d^k = \beta d^{k-1} + (1 - \beta) \left( \bar{g}^k - \overline{\nabla f_{\mathcal{H}}}(x^k) \right) + \beta \left( \overline{\nabla f_{\mathcal{H}}}(x^k) - \overline{\nabla f_{\mathcal{H}}}(x^{k-1}) \right). \tag{263}$$

Taking the expectation on both sides, we obtain that

$$\begin{aligned}
\mathbb{E}\left[ \|d^k\|^2 \right] &= \beta^2 \mathbb{E}\left[ \|d^{k-1}\|^2 \right] + (1-\beta)^2 \mathbb{E}\left[ \|\bar{g}^k - \overline{\nabla f_{\mathcal{H}}}(x^k)\|^2 \right] + \beta^2 \mathbb{E}\left[ \|\overline{\nabla f_{\mathcal{H}}}(x^k) - \overline{\nabla f_{\mathcal{H}}}(x^{k-1})\|^2 \right] \\
&\quad + 2\beta(1-\beta)\mathbb{E}\left[ \langle d^{k-1}, \nabla \bar{g}^k - \overline{\nabla f_{\mathcal{H}}}(x^k) \rangle \right] + 2\beta^2 \mathbb{E}\left[ \langle d^{k-1}, \overline{\nabla f_{\mathcal{H}}}(x^k) - \overline{\nabla f_{\mathcal{H}}}(x^{k-1}) \rangle \right] \\
&\quad + 2\beta(1-\beta)\mathbb{E}\left[ \langle \bar{g}^k - \overline{\nabla f_{\mathcal{H}}}(x^k), \overline{\nabla f_{\mathcal{H}}}(x^k) - \overline{\nabla f_{\mathcal{H}}}(x^{k-1}) \rangle \right].
\end{aligned} \tag{264}$$

Recall that using Assumption 3, we have $\mathbb{E}\left[ \bar{g}^k \right] = \overline{\nabla f_{\mathcal{H}}}(x^k)$. Thus, we obtain that

$$\begin{aligned}
\mathbb{E}\left[ \|d^k\|^2 \right] &= \beta^2 \mathbb{E}\left[ \|d^{k-1}\|^2 \right] + (1-\beta)^2 \mathbb{E}\left[ \|\bar{g}^k - \overline{\nabla f_{\mathcal{H}}}(x^k)\|^2 \right] + \beta^2 \mathbb{E}\left[ \|\overline{\nabla f_{\mathcal{H}}}(x^k) - \overline{\nabla f_{\mathcal{H}}}(x^{k-1}))\|^2 \right] \\
&\quad + 2\beta^2 \mathbb{E}\left[ \langle d^{k-1}, \overline{\nabla f_{\mathcal{H}}}(x^k) - \overline{\nabla f_{\mathcal{H}}}(x^{k-1}) \rangle \right].
\end{aligned} \tag{265}$$

By Assumption 3, we have $\mathbb{E}\left[ \|\bar{g}^k - \overline{\nabla f_{\mathcal{H}}}(x^k)\|^2 \right] \leq \frac{\sigma^2}{n_h}$, thus we obtain that

$$\begin{aligned}
\mathbb{E}\left[ \|d^k\|^2 \right] &= \beta^2 \mathbb{E}\left[ \|d^{k-1}\|^2 \right] + (1-\beta)^2 \frac{\sigma^2}{n_h} + \beta^2 \mathbb{E}\left[ \|\overline{\nabla f_{\mathcal{H}}}(x^k) - \overline{\nabla f_{\mathcal{H}}}(x^{k-1}))\|^2 \right] \\
&\quad + 2\beta^2 \mathbb{E}\left[ \langle d^{k-1}, \overline{\nabla f_{\mathcal{H}}}(x^k) - \overline{\nabla f_{\mathcal{H}}}(x^{k-1}) \rangle \right].
\end{aligned} \tag{266}$$

Using the Cauchy-Schwarz inequality, we obtain that

$$\begin{aligned}
\mathbb{E}\left[ \|d^k\|^2 \right] &= \beta^2 \mathbb{E}\left[ \|d^{k-1}\|^2 \right] + (1-\beta)^2 \frac{\sigma^2}{n_h} + \beta^2 \mathbb{E}\left[ \|\overline{\nabla f_{\mathcal{H}}}(x^k) - \overline{\nabla f_{\mathcal{H}}}(x^{k-1}))\|^2 \right] \\
&\quad + 2\beta^2 \mathbb{E}\left[ \|d^{k-1}\| \cdot \|\overline{\nabla f_{\mathcal{H}}}(x^k) - \overline{\nabla f_{\mathcal{H}}}(x^{k-1})\| \right].
\end{aligned} \tag{267}$$

Then, using the Assumption 2 that $\|\overline{\nabla f_{\mathcal{H}}}(x^k) - \overline{\nabla f_{\mathcal{H}}}(x^{k-1})\| \leq L\|\bar{x}^k - \bar{x}^{k-1}\| \leq L\eta\|\bar{v}^{k-1} - \bar{\xi}^{k-1}\|$, thus we obtain that

$$\mathbb{E}\left[ \|d^k\|^2 \right] = \beta^2 \mathbb{E}\left[ \|d^{k-1}\|^2 \right] + (1-\beta)^2 \frac{\sigma^2}{n_h} + \beta^2 \eta^2 L^2 \mathbb{E}\left[ \|\bar{v}^{k-1} - \bar{\xi}^{k-1}\|^2 \right] + 2\beta^2 \eta L \mathbb{E}\left[ \|d^{k-1}\| \cdot \|\bar{v}^{k-1} - \bar{\xi}^{k-1}\| \right]. \tag{268}$$

Using the fact that $2ab \leq a^2 + b^2$, we have

$$\begin{aligned}
\mathbb{E}\left[ \|d^k\|^2 \right] &\leq \beta^2 \mathbb{E}\left[ \|d^{k-1}\|^2 \right] + (1-\beta)^2 \frac{\sigma^2}{n_h} + \beta^2 \eta^2 L^2 \mathbb{E}\left[ \|\bar{v}^{k-1} - \bar{\xi}^{k-1}\|^2 \right] + \beta^2 \eta L \left( \mathbb{E}\left[ \|d^{k-1}\|^2 \right] + \mathbb{E}\left[ \|\bar{v}^{k-1} - \bar{\xi}^{k-1}\|^2 \right] \right) \\
&= \beta^2 (1 + \eta L) \mathbb{E}\left[ \|d^{k-1}\|^2 \right] + (1-\beta)^2 \frac{\sigma^2}{n_h} + \beta^2 \eta L (\eta L + 1) \mathbb{E}\left[ \|\bar{v}^{k-1} - \bar{\xi}^{k-1}\|^2 \right].
\end{aligned} \tag{269}$$

Using the Cauchy-Schwarz inequality, we obtain that

$$\mathbb{E}\left[ \|d^k\|^2 \right] \leq \beta^2 (1 + \eta L) \mathbb{E}\left[ \|d^{k-1}\|^2 \right] + (1-\beta)^2 \frac{\sigma^2}{n_h} + 2\beta^2 \eta L (\eta L + 1) \left( \mathbb{E}\left[ \|\bar{v}^{k-1}\|^2 \right] + \mathbb{E}\left[ \|\bar{\xi}^{k-1}\|^2 \right] \right). \tag{270}$$

Thus, the proof is finished. $\qquad \square$

E.3.5. PROOF OF LEMMA 10

**Lemma 10.** *Suppose that Assumptions 1 and 2 hold. For* BRED-M *in Algorithm 1, the following holds for all $k \geq 0$:*

$$
\begin{aligned}
f_{\mathcal{H}}(\bar{x}^K) - f_{\mathcal{H}}(\bar{x}^0) \leq &-\frac{\eta}{4} \sum_{k=0}^{K-1} \mathbb{E}\left[\left\|\nabla f_{\mathcal{H}}(\bar{x}^k)\right\|^2\right] + \sum_{k=0}^{K-1} \eta L^2 v_2^2 v_1^2 \mathbb{E}\left[\left\|\hat{\mathbf{e}}^k\right\|^2\right] + 2\eta \sum_{k=0}^{K-1} \mathbb{E}\left[\left\|d^k\right\|^2\right] \\
&+ \sum_{k=0}^{K-1} \left(L\eta^2 - \frac{\eta}{2}\right) \mathbb{E}\left[\left\|\bar{v}^k\right\|^2\right] + \frac{2\eta}{(1-q)^2} \sum_{k=0}^{K-1} \mathbb{E}\left[\left\|\xi^k\right\|^2\right] + \sum_{k=0}^{K-1} \left(L\eta^2 + \eta\right) \mathbb{E}\left[\left\|\xi^k\right\|^2\right].
\end{aligned}
\tag{172}
$$

*Proof.* Recall that at step $k$, for each honest work, the model updates:

$$
\bar{x}^{k+1} = \bar{x}^k - \eta \bar{v}^k + \eta \bar{\xi}^k,
\tag{271}
$$

and recall the definition that

$$
v_i^k := -\frac{q}{\eta}(x_i^k - x_i^{k-1}) + \left(m_i^k - q m_i^{k-1}\right), \quad \bar{v}^k := \frac{1}{|\mathcal{H}|} \sum_{i \in \mathcal{H}} v_i^k, \quad \bar{\xi}^k := \frac{1}{|\mathcal{H}|} \sum_{i \in \mathcal{H}} \xi_i^k.
\tag{272}
$$

According to the $L$-smoothness of $f$ in Assumption 2, we obtain that

$$
f_{\mathcal{H}}(\bar{x}^{k+1}) \leq f_{\mathcal{H}}(\bar{x}^k) + \mathbb{E}\left[\left\langle \nabla f_{\mathcal{H}}(\bar{x}^k), \bar{x}^{k+1} - \bar{x}^k \right\rangle\right] + \frac{L}{2}\mathbb{E}\left[\left\|\bar{x}^{k+1} - \bar{x}^k\right\|^2\right].
\tag{273}
$$

Then, invoking (271), we can obtain that

$$
f_{\mathcal{H}}(\bar{x}^{k+1}) \leq f_{\mathcal{H}}(\bar{x}^k) - \mathbb{E}\left[\eta \left\langle \nabla f_{\mathcal{H}}(\bar{x}^k), \bar{v}^k \right\rangle\right] + \mathbb{E}\left[\eta \left\langle \nabla f_{\mathcal{H}}(\bar{x}^k), \bar{\xi}^k \right\rangle\right] + \frac{L}{2}\mathbb{E}\left[\left\|-\eta \bar{v}^k + \eta \bar{\xi}^k\right\|^2\right].
\tag{274}
$$

Using the equation that $ab \leq \frac{a^2}{2} + \frac{b^2}{2} - \frac{(a-b)^2}{2}$, we obtain that

$$
\mathbb{E}\left[-\eta \left\langle \nabla f_{\mathcal{H}}(\bar{x}^k), \bar{v}^k \right\rangle\right] = \frac{\eta \mathbb{E}\left[\left\|\nabla f_{\mathcal{H}}(\bar{x}^k) - \bar{v}^k\right\|^2\right]}{2} - \frac{\eta \mathbb{E}\left[\left\|\nabla f_{\mathcal{H}}(\bar{x}^k)\right\|^2\right]}{2} - \frac{\eta \mathbb{E}\left[\left\|\bar{v}^k\right\|^2\right]}{2}.
\tag{275}
$$

Then, using the inequality that $2ab \leq ca^2 + \frac{1}{c}b^2$, we obtain that

$$
\mathbb{E}\left[\eta \left\langle \nabla f_{\mathcal{H}}(\bar{x}^k), \bar{\xi}^k \right\rangle\right] \leq \frac{1}{2}\left(\frac{\eta \mathbb{E}\left[\left\|\nabla f_{\mathcal{H}}(\bar{x}^k)\right\|^2\right]}{2} + 2\eta \mathbb{E}\left[\left\|\bar{\xi}^k\right\|^2\right]\right).
\tag{276}
$$

Invoking (275) and (276) into (274), we obtain that

$$
f_{\mathcal{H}}(\bar{x}^{k+1}) \leq f_{\mathcal{H}}(\bar{x}^k) - \frac{\eta}{4}\mathbb{E}\left[\left\|\nabla f_{\mathcal{H}}(\bar{x}^k)\right\|^2\right] - \frac{\eta}{2}\mathbb{E}\left[\left\|\bar{v}^k\right\|^2\right] + \frac{\eta}{2}\mathbb{E}\left[\left\|\nabla f_{\mathcal{H}}(\bar{x}^k) - \bar{v}^k\right\|^2\right] + \eta \mathbb{E}\left[\left\|\bar{\xi}^k\right\|^2\right] + \frac{L\eta^2}{2}\mathbb{E}\left[\left\|\bar{v}^k - \bar{\xi}^k\right\|^2\right].
\tag{277}
$$

Using the Cauchy-Schwarz inequality, we obtain that

$$
\begin{aligned}
f_{\mathcal{H}}(\bar{x}^{k+1}) \leq &\ f_{\mathcal{H}}(\bar{x}^k) - \frac{\eta}{4}\mathbb{E}\left[\left\|\nabla f_{\mathcal{H}}(\bar{x}^k)\right\|^2\right] + \frac{\eta}{2}\mathbb{E}\left[\left\|\nabla f_{\mathcal{H}}(\bar{x}^k) - \bar{v}^k\right\|^2\right] + \left(L\eta^2 - \frac{\eta}{2}\right)\mathbb{E}\left[\left\|\bar{v}^k\right\|^2\right] + \left(L\eta^2 + \eta\right)\mathbb{E}\left[\left\|\bar{\xi}^k\right\|^2\right] \\
\leq &\ f_{\mathcal{H}}(\bar{x}^k) - \frac{\eta}{4}\mathbb{E}\left[\left\|\nabla f_{\mathcal{H}}(\bar{x}^k)\right\|^2\right] + \eta \mathbb{E}\left[\left\|\nabla f_{\mathcal{H}}(\bar{x}^k) - \overline{\nabla f_{\mathcal{H}}}(x^k)\right\|^2\right] + \eta \mathbb{E}\left[\left\|\overline{\nabla f_{\mathcal{H}}}(x^k) - \bar{v}^k\right\|^2\right] \\
&+ \left(L\eta^2 - \frac{\eta}{2}\right)\mathbb{E}\left[\left\|\bar{v}^k\right\|^2\right] + \left(L\eta^2 + \eta\right)\mathbb{E}\left[\left\|\bar{\xi}^k\right\|^2\right].
\end{aligned}
\tag{278}
$$

Using the $L$-smoothness of $f$ as in Assumption 2, we obtain that

$$
\begin{aligned}
\mathbb{E}\left[\left\|\nabla f_{\mathcal{H}}(\bar{x}^k) - \overline{\nabla f_{\mathcal{H}}}(x^k)\right\|^2\right] &= \frac{1}{n_h^2}\mathbb{E}\left[\left\|\sum_{i\in\mathcal{H}}\left(\nabla f_i(\bar{x}^k) - \nabla f_i(x_i^k)\right)\right\|^2\right] \\
&\leq \frac{1}{n_h}\sum_{i\in\mathcal{H}}\mathbb{E}\left[\left\|\nabla f_i(\bar{x}^k) - \nabla f_i(x_i^k)\right\|^2\right] \\
&\leq \frac{L^2}{n_h}\sum_{i\in\mathcal{H}}\mathbb{E}\left[\left\|\bar{x}^k - x_i^k\right\|^2\right].
\end{aligned}
\tag{279}
$$

Thus, invoking (279) into (278) and using Cauchy-Schwarz inequality, we can obtain

$$
f_{\mathcal{H}}(\bar{x}^{k+1}) \leq f_{\mathcal{H}}(\bar{x}^k) - \frac{\eta}{4}\mathbb{E}\left[\left\|\nabla f_{\mathcal{H}}(\bar{x}^k)\right\|^2\right] + \frac{\eta L^2}{n_h}\mathbb{E}\left[\left\|\bar{\mathbf{x}}^k - \mathbf{x}^k\right\|^2\right] + 2\eta\mathbb{E}\left[\left\|\overline{\nabla f_{\mathcal{H}}}(x^k) - \bar{m}^k\right\|^2\right] + 2\eta\mathbb{E}\left[\left\|\bar{v}^k - \bar{m}^k\right\|^2\right]
$$
$$
+ \left(L\eta^2 - \frac{\eta}{2}\right)\mathbb{E}\left[\left\|\bar{v}^k\right\|^2\right] + \left(L\eta^2 + \eta\right)\mathbb{E}\left[\left\|\bar{\xi}^k\right\|^2\right].
\tag{280}
$$

Summing (280) from $k=0$ to $K-1$, we obtain that

$$
\begin{aligned}
f_{\mathcal{H}}(\bar{x}^K) - f_{\mathcal{H}}(\bar{x}^0) &\leq -\frac{\eta}{4}\sum_{k=0}^{K-1}\mathbb{E}\left[\left\|\nabla f_{\mathcal{H}}(\bar{x}^k)\right\|^2\right] + \sum_{k=0}^{K-1}\frac{\eta L^2}{n_h}\mathbb{E}\left[\left\|\bar{\mathbf{x}}^k - \mathbf{x}^k\right\|^2\right] + 2\eta\sum_{k=0}^{K-1}\mathbb{E}\left[\left\|\overline{\nabla f_{\mathcal{H}}}(x^k) - \bar{m}^k\right\|^2\right] \\
&\quad + 2\eta\sum_{k=0}^{K-1}\mathbb{E}\left[\left\|\bar{v}^k - \bar{m}^k\right\|^2\right] + \sum_{k=0}^{K-1}\left(L\eta^2 - \frac{\eta}{2}\right)\mathbb{E}\left[\left\|\bar{v}^k\right\|^2\right] + \sum_{k=0}^{K-1}\left(L\eta^2 + \eta\right)\mathbb{E}\left[\left\|\bar{\xi}^k\right\|^2\right].
\end{aligned}
\tag{281}
$$

Recall that $\hat{\mathbf{U}}^\top\mathbf{U} = \mathbf{I}$, we obtain that

$$
\left\|\hat{\mathbf{U}}^\top\mathbf{x}\right\|^2 = \mathbf{x}^\top\hat{\mathbf{U}}\hat{\mathbf{U}}^\top\hat{\mathbf{U}}\hat{\mathbf{U}}^\top\mathbf{x} = \left\|\hat{\mathbf{U}}\hat{\mathbf{U}}^\top\mathbf{x}^k\right\|^2 = \left\|\mathbf{x}^k - \bar{\mathbf{x}}^k\right\|^2.
\tag{282}
$$

Using the definition for $\hat{\mathbf{e}}^k$, we have

$$
\left\|v\hat{\mathbf{V}}\hat{\mathbf{e}}^k\right\|^2 = \left\|\hat{\mathbf{U}}^\top\mathbf{x}^k\right\|^2 + \left\|\hat{\mathbf{\Lambda}}_b^{-1}\hat{\mathbf{U}}^\top\mathbf{s}^k\right\|^2.
\tag{283}
$$

Thus, we have $\mathbb{E}\left[\left\|\mathbf{x}^k - \bar{\mathbf{x}}^k\right\|^2\right] \leq v^2 v_1^2\mathbb{E}\left[\left\|\hat{\mathbf{e}}^k\right\|^2\right]$. Choosing the scaling parameter $v = \sqrt{n_h}v_2$ and substituting this result into (281), we obtain

$$
\begin{aligned}
f_{\mathcal{H}}(\bar{x}^K) - f_{\mathcal{H}}(\bar{x}^0) &\leq -\frac{\eta}{4}\sum_{k=0}^{K-1}\mathbb{E}\left[\left\|\nabla f_{\mathcal{H}}(\bar{x}^k)\right\|^2\right] + \sum_{k=0}^{K-1}\eta L^2 v_2^2 v_1^2\mathbb{E}\left[\left\|\hat{\mathbf{e}}^k\right\|^2\right] + 2\eta\sum_{k=0}^{K-1}\mathbb{E}\left[\left\|d^k\right\|^2\right] \\
&\quad + 2\eta\sum_{k=0}^{K-1}\mathbb{E}\left[\left\|\bar{v}^k - \bar{m}^k\right\|^2\right] + \sum_{k=0}^{K-1}\left(L\eta^2 - \frac{\eta}{2}\right)\mathbb{E}\left[\left\|\bar{v}^k\right\|^2\right] + \sum_{k=0}^{K-1}\left(L\eta^2 + \eta\right)\mathbb{E}\left[\left\|\bar{\xi}^k\right\|^2\right].
\end{aligned}
\tag{284}
$$

Combining (284) with Lemma 12, we obtain that

$$
\begin{aligned}
f_{\mathcal{H}}(\bar{x}^K) - f_{\mathcal{H}}(\bar{x}^0) &\leq -\frac{\eta}{4}\sum_{k=0}^{K-1}\mathbb{E}\left[\left\|\nabla f_{\mathcal{H}}(\bar{x}^k)\right\|^2\right] + \sum_{k=0}^{K-1}\eta L^2 v_2^2 v_1^2\mathbb{E}\left[\left\|\hat{\mathbf{e}}^k\right\|^2\right] + 2\eta\sum_{k=0}^{K-1}\mathbb{E}\left[\left\|d^k\right\|^2\right] \\
&\quad + \sum_{k=0}^{K-1}\left(L\eta^2 - \frac{\eta}{2}\right)\mathbb{E}\left[\left\|\bar{v}^k\right\|^2\right] + \frac{2\eta}{(1-q)^2}\sum_{k=0}^{K-1}\mathbb{E}\left[\left\|\xi^k\right\|^2\right] + \sum_{k=0}^{K-1}\left(L\eta^2 + \eta\right)\mathbb{E}\left[\left\|\bar{\xi}^k\right\|^2\right].
\end{aligned}
\tag{285}
$$

Thus, the proof is finished.

$\square$

E.3.6. PROOF OF LEMMA 12

**Lemma 12.** *Suppose that Assumptions 1, 2, 3, and 4 hold. For Algorithm 1, the following holds for all $k \geq 0$:*

$$\sum_{k=0}^{K-1} \mathbb{E}\left[\left\|\bar{v}^k - \bar{m}^k\right\|^2\right] \leq \frac{1}{(1-q)^2} \sum_{k=0}^{K-1} \mathbb{E}\left[\left\|\bar{\xi}^k\right\|^2\right]. \tag{286}$$

*Proof.* By the definition of $\bar{v}^k = -\frac{q}{\eta}(\bar{x}^k - \bar{x}^{k-1}) + \bar{m}^k - q\bar{m}^{k-1}$ and $\bar{x}^k = \bar{x}^{k-1} - \eta\bar{v}^{k-1} + \eta\bar{\xi}^{k-1}$, we obtain that

$$\begin{aligned}
\mathbb{E}\left[\left\|\bar{v}^k - \bar{m}^k\right\|^2\right] &= \mathbb{E}\left[\left\|-\frac{q}{\eta}(\bar{x}^k - \bar{x}^{k-1}) + \bar{m}^k - q\bar{m}^{k-1} - \bar{m}^k\right\|^2\right] \\
&= \mathbb{E}\left[\left\|q\left(\bar{v}^{k-1} - \bar{\xi}^{k-1}\right) - q\bar{m}^{k-1}\right\|^2\right] = \mathbb{E}\left[\left\|q(\bar{v}^{k-1} - \bar{m}^{k-1}) - q\bar{\xi}^{k-1}\right\|^2\right].
\end{aligned} \tag{287}$$

Then, by the recursive definition of $\bar{v}^k$, we have

$$\bar{v}^k - \bar{m}^k = -\frac{q}{\eta}\left(\bar{x}^k - \bar{x}^{k-1}\right) - q\bar{m}^{k-1} = q(\bar{v}^{k-1} - \bar{m}^{k-1}) - q\bar{\xi}^{k-1}. \tag{288}$$

For $k \geq 1$, then the aggregation bias can be expressed as

$$\mathbb{E}\left[\left\|\bar{v}^k - \bar{m}^k\right\|^2\right] = \mathbb{E}\left[\left\|q^2(\bar{v}^{k-2} - \bar{m}^{k-2}) - q^2\bar{\xi}^{k-2} - q\bar{\xi}^{k-1}\right\|^2\right] = \mathbb{E}\left[\left\|\sum_{t=0}^{k-1} q^{k-t}\bar{\xi}^t\right\|^2\right]. \tag{289}$$

Since $\sum_{t=0}^{k-1}(1-q)q^t \leq 1$ for $q \in (0,1)$, we have

$$\mathbb{E}\left[\left\|\bar{v}^k - \bar{m}^k\right\|^2\right] = \mathbb{E}\left[\left\|\sum_{t=0}^{k-1} q^{k-t}\bar{\xi}^t\right\|^2\right] \leq \frac{1}{1-q}\sum_{t=0}^{k-1} q^{k-t}\mathbb{E}\left[\left\|\bar{\xi}^t\right\|^2\right]. \tag{290}$$

Summing (290) over $k = 0, \ldots, K-1$ and exchanging the order of summation yields

$$\sum_{k=0}^{K-1} \mathbb{E}\left[\left\|\bar{v}^k - \bar{m}^k\right\|^2\right] = \sum_{k=1}^{K-1} \frac{1}{1-q}\sum_{t=0}^{k-1} q^{k-t}\mathbb{E}\left[\left\|\bar{\xi}^t\right\|^2\right] \leq \frac{1}{(1-q)^2}\sum_{k=0}^{K-1} \mathbb{E}\left[\left\|\bar{\xi}^k\right\|^2\right]. \tag{291}$$

This completes the proof. $\qquad\square$

# F. Proof of Corollary 3 and Corollary 4

**Corollary 3.** *Suppose Assumptions 1–4 hold. Let $\delta < \frac{1}{2}$ and let $\mathcal{A}(\cdot)$ be $(\delta, \kappa)$-robust with $\kappa = \mathcal{O}(\delta)$. When $q = 1$ (no scaling), the asymptotic error of* BRED *satisfies*

$$\varepsilon_{K\to\infty} = \mathcal{O}\left(K^2\delta(\sigma^2 + \zeta^2)\right). \tag{19}$$

*Proof.* The proof follows a similar structure as that of Lemma 6. By the recursive definition of $\bar{v}^k$, the aggregation bias satisfies

$$\mathbb{E}\left[\left\|\overline{\nabla f_{\mathcal{H}}}(x^k) - \bar{v}^k\right\|^2\right] = \mathbb{E}\left[\left\|(\bar{v}^{k-1} - \bar{g}^{k-1}) - \bar{\xi}^{k-1}\right\|^2\right]. \tag{292}$$

Similarly, by the recursive definition of $\bar{v}^k$, we obtain that

$$\bar{v}^k - \bar{g}^k = -\frac{q}{\eta}\left(\bar{x}^k - \bar{x}^{k-1}\right) - \bar{g}^{k-1} = \bar{v}^{k-1} - \bar{g}^{k-1} - \bar{\xi}^{k-1}. \tag{293}$$

For $k \geq 1$, the aggregation bias can be expressed as

$$\mathbb{E}\left[\left\|\overline{\nabla f_{\mathcal{H}}}(x^k) - \bar{\tilde{v}}^k\right\|^2\right] = \mathbb{E}\left[\left\|(\bar{v}^{k-2} - \bar{g}^{k-2}) - \bar{\xi}^{k-2} - \bar{\xi}^{k-1}\right\|^2\right]$$

$$= \mathbb{E}\left[\left\|\sum_{t=0}^{k-1} \bar{\xi}^t\right\|^2\right] \leq k \sum_{t=0}^{k-1} \mathbb{E}\left[\left\|\bar{\xi}^t\right\|^2\right]. \quad (294)$$

Summing (290) over $k = 0, \ldots, K - 1$ and exchanging the order of summation, we obtain

$$\sum_{k=0}^{K-1} \mathbb{E}\left[\left\|\bar{\tilde{v}}^k - \overline{\nabla f_{\mathcal{H}}}(x^k)\right\|^2\right] = \left(\sum_{k=1}^{K-1} k \sum_{t=0}^{k-1} \mathbb{E}\left[\left\|\bar{\xi}^t\right\|^2\right]\right) \leq \frac{K^2}{2} \sum_{k=0}^{K-1} \mathbb{E}\left[\left\|\bar{\xi}^k\right\|^2\right]. \quad (295)$$

Replacing (90) in Lemma 6 with (295), and following the same reasoning as in the proof of Theorem 1, we conclude that

$$\varepsilon_{K \to \infty} = \mathcal{O}\left(K^2 \delta(\sigma^2 + \zeta^2)\right). \quad (296)$$

This completes the proof of Corollary 3. $\square$

**Corollary 4.** *Suppose Assumptions 1–4 hold. Let $\delta < \frac{1}{2}$ and let $\mathcal{A}(\cdot)$ be $(\delta, \kappa)$-robust with $\kappa = \mathcal{O}(\delta)$. When $q \in (0, 1)$ (with scaling), the asymptotic error of* BRED *satisfies*

$$\varepsilon_{K \to \infty} = \mathcal{O}\left(\frac{\delta(\sigma^2 + \zeta^2)}{(1-q)^2}\right). \quad (20)$$

*Proof.* The proof of Corollary 4 follows directly by taking $K \to \infty$ in Theorem 1.

$\square$

