# OpenReview forum: "Improving the Robustness-Utility Trade-off in Decentralized Learning over Sparse Networks"
_ICML.cc/2026/Conference — ICML 2026 regular_

### Official Review · Reviewer_p3Ru · 2026-03-09

**Soundness:** 3
**Presentation:** 3
**Significance:** 2
**Originality:** 2
**Overall Recommendation:** 4
**Confidence:** 3

**Summary:**

In Improving the Robustness-Utility Trade-off in Decentralized Learning over Sparse Networks, the authors propose a method for solving decentralized optimization problems in the presence of Byzantine attackers. The proposed approach does not rely on decentralized stochastic gradient descent (DSGD), which is commonly used in Byzantine-robust decentralized learning due to its simplicity and robustness. Instead, the method is derived from a primal-dual formulation, which can be reformulated as a gradient-tracking type algorithm.

Gradient-tracking and dual-variable methods have typically been avoided in Byzantine settings because their additional memory variables can accumulate adversarial errors, making them sensitive to attacks. The authors address this issue by introducing a scaled dual-ascent mechanism that stabilizes the dual dynamics and prevents error accumulation.

The advantage of incorporating such a correction mechanism is that, under heterogeneous data, it improves the dependence of the transient time on the network topology. In particular, the transient complexity is reduced from $\mathcal{O}((1-\lambda)^{-6}),$ typical of DSGD-based Byzantine-robust methods, to $\mathcal{O}((1-\lambda)^{-2}).$ This improvement can be significant in sparse graphs, such as cycle networks, where the spectral gap $1-\lambda$ is small.

**Compliance With Llm Reviewing Policy:**

Affirmed.

**Final Justification:**

The authors addressed all my concerns except for the concerns regarding novelty. I update my score accordingly.

**Key Questions For Authors:**

1) The novelty of the paper is moderate, at least in terms of algorithm derivation and result. If there is novelty in the proof, I would suggest to highlight the main idea in the body of the paper.
2) I do not think most of the community is familiar with the type of robust aggregation rules involved in this work. It would be good to provide a couple of examples in the paper to make it more self contained.
3) Heterogeneity: usually, the motivation to employ GT/primal-dual schemes it have a better handle on data heterogeneity. In general, there is no need to bound data heterogeneity for such schemes (Theorem 2-Koloskova, Lin, Stich, Neurips 2021), but is typically assumed for DGD. why is this assumption introduced here? Why its it necessary?

**Strengths And Weaknesses:**

Soundness: The theoretical results appear to be technically sound and the conclusions seem consistent with the assumptions made in the analysis.

Presentation: The paper is clearly written, and the motivation for the proposed method is well explained. The structure of the presentation makes it easy to follow the main ideas and results.

Significance: The problem addressed may be of moderate significance to a sub-community within decentralized optimization, particularly for work focusing on Byzantine-robust decentralized learning over networks. The improvement mainly concerns the dependence on the spectral gap in the transient regime rather than the fundamental stochastic convergence rate or the handling of data heterogeneity.

Originality: The main insight of the paper is that the accumulation of errors in the dual variables can destabilize bias-correction methods under Byzantine attacks. To mitigate this issue, the authors introduce a rescaling of the dual ascent step that keeps the magnitude of the dual variables under control. While this stabilization mechanism is reasonable, the overall conceptual novelty appears moderate, as the approach mainly adapts existing gradient-tracking and robust aggregation techniques to coexist.

---

> ### Author Rebuttal · Authors · 2026-03-30
>
> > **Q1:  Novelty and contribution of our work.**
> >
>
> We thank the reviewer for raising this important concern. We respectfully emphasize that the novelty of this work extends well beyond algorithmic derivation. Our main contributions are summarized as follows.
>
> **(1) Identification of a previously overlooked fundamental challenge.**
> This is the first work to show that directly combining bias-correction methods (e.g., ED, EXTRA, and GT) with robust aggregators leads to an asymptotic error that grows linearly with the number of iterations $K$. The underlying reason is that the dual variable behaves as an integrator, accumulating Byzantine-induced aggregation errors over time. This negative result reveals a fundamental incompatibility between existing bias-correction techniques and Byzantine-robust aggregation, and helps explain why prior bias-correction methods have not been successfully extended to Byzantine settings.
>
> **(2) Augmented Lagrangian formulation with scaled dual ascent.**
> To overcome this fundamental difficulty, we propose scaled dual ascent (SDA) within an augmented Lagrangian framework. SDA introduces a fractional scaling factor $q\in(0,1)$ into the dual update. By explicitly contracting the dual dynamics, we show that SDA removes the dependence of the asymptotic error on the iteration horizon $K$. Importantly, SDA is not a heuristic modification; rather, it is derived in a principled manner from a modified Lagrangian with dual regularization.
>
> **(3) Theoretical contributions.**
>
> - **Improved transient iteration complexity.** Beyond resolving the error accumulation issue, our method also enjoys improved transient guarantees. In particular, BRED achieves linear speedup with respect to the number of honest clients $n_h$ (Table 1), with transient iteration complexity $ \mathcal{O}\bigl(\max\{n_h^3/(1-\lambda)^2,\delta^2 n_h^3/(1-\lambda)^6\}\bigr) $. When $\delta$ is small, the first term dominates, yielding more favorable scaling than DSGD-based Byzantine methods, whose transient complexity is $ \mathcal{O}\bigl(n_h^3/(1-\lambda-8\delta\sqrt{n_h})^6\bigr) $. Moreover, when $\delta=0$, our bound reduces to $ \mathcal{O}(n^3/(1-\lambda)^2) $, matching Exact Diffusion. The momentum variant BRED-M further improves this result to $ \mathcal{O}\bigl(\max\{n_h^2/(1-\lambda)^2,\delta^2 n_h^3/(1-\lambda)^4\}\bigr) $, improving both the topology-dependent term and the Byzantine-dependent term.
>
> - **Order-optimal Byzantine resilience in asymptotic error.** In addition to the improved transient behavior, BRED-M attains an asymptotic error that matches the lower bound $ \Omega(\delta\zeta^2) $ established in [1]. Therefore, BRED-M is order-optimal in terms of Byzantine resilience under this lower-bound criterion.
> ---
> > **Q2:  Clarification and examples of robust aggregation rules.**
> >
> We sincerely thank the reviewer for this constructive suggestion. We will add a more accessible description of robust aggregation rules in the revised manuscript. At a high level, a robust aggregator aims to produce a reliable estimate of the true average gradient while mitigating the influence of potentially malicious updates. Existing methods fall into two broad categories:
>
> 1. **Distance-based selection:** These methods identify trustworthy updates based on geometric proximity. For example, Krum selects the single update whose sum of distances to its nearest neighbors is minimized, and Median computes the coordinate-wise median of all received updates.
> 2. **Outlier filtering:** These methods explicitly remove suspicious updates before averaging. For example, Trimmed Mean discards the largest and smallest values in each coordinate, and IOS (Iterative Outlier Scissor) iteratively filters gradients identified as outliers.
>
> ---
> > **Q3:  Additional assumption on data heterogeneity.**
> >
>
> We thank the reviewer for this insightful question. The reviewer is correct that in standard, fault-free decentralized learning, bias-corrected methods (like GT and ED) do not require  the bounded data heterogeneity assumption and instead depend only on the initial gradient dissimilarity $\zeta_0$. The key reason our setting requires the assumption is that, in the presence of Byzantine attackers, the robust aggregator introduces an estimation error in **every** communication round (not just at initialization). Because this per-round error appears throughout training and its lower bound is $\Omega(\delta\zeta^2)$ [1], the analysis cannot be closed by bounding the initial dissimilarity $\zeta_0$ alone. We note that this is not a limitation specific to our work; it is an inherent requirement of the Byzantine-robust setting. Indeed, when there are no attackers ($\delta=0$), our proof degenerates to the standard case and the bounded gradient dissimilarity assumption can be removed.
>
> [1] Karimireddy, Sai Praneeth, Lie He, and Martin Jaggi. "Byzantine-robust learning on heterogeneous datasets via bucketing." *arXiv preprint arXiv:2006.09365* (2020).

---

> > ### Author Rebuttal · Reviewer_p3Ru · 2026-04-03
> >
> > The authors addressed all my concerns except for the concerns regarding novelty. I update my score accordingly.

---

> > > ### Author Response · Authors · 2026-04-04
> > >
> > > Thank you for your feedback. We appreciate your comments and will further clarify the novelty and contributions of our work in the revision.

---

### Official Review · Reviewer_KyuG · 2026-03-10

**Soundness:** 3
**Presentation:** 2
**Significance:** 2
**Originality:** 3
**Overall Recommendation:** 4
**Confidence:** 3

**Summary:**

This paper studies Byzantine-resilient decentralized learning over sparse networks. Within an augmented Lagrangian framework, the authors propose a scaled dual ascent mechanism, where a scaling factor $q$ explicitly contracts the dual updates to suppress error accumulation, and develop BRED and its momentum variant BRED-M. The paper establishes convergence guarantees, proves linear speed-up with respect to the number of honest clients $n_h$, and shows improved dependence of the transient time on $1-\lambda$ compared with existing methods. Numerical experiments are provided to support the theoretical findings.

**Compliance With Llm Reviewing Policy:**

Affirmed.

**Final Justification:**

The rebuttal addressed my main concerns and clarified the key technical points I raised. In particular, the explanations on the validity of Eq. (4), the roles of $\hat{B}$ and $\hat{D}$, and the transient complexity comparison made the contribution much clearer to me. For this reason, I am raising my score from 3 to 4. That said, I still think the writing can be improved in the revision, especially in the presentation of some assumptions and technical statements.

**Key Questions For Authors:**

1. Why is it valid to introduce a communication matrix defined on $\mathbb{R}^{n_h\times n_h}$? Without assuming that the subgraph induced by the honest nodes is connected, why does the equivalence in Eq. (4) hold? If my understanding is correct, Eq. (4) requires connectivity of the honest subgraph. Why is this assumption reasonable in practice?

2. The notation in Eq. (4) is confusing. According to the definitions, $\hat{B}$ and $\hat{D}$ appear to be identical. Is there any substantive difference between them, or are they only notationally different?

3. Based on Table 1, when $\delta = 0$, the transient time of BRED-M should be $O(\max\{ n^2(1-\lambda)^{-1}, n(1-\lambda)^{-2}\})$, which is upper bounded by $O(n^2(1-\lambda)^{-2})$. However, the contributions (page 2) state that “our results recover the transient complexity of Exact Diffusion $O(n^3(1-\lambda)^{-2})$". Could the authors clarify this point? If the correct dependence is $O(n^2)$, please explain why this yields an $O(n)$ improvement in transient time.

I may consider increasing my score if all of my questions are addressed.

**Limitations:**

yes

**Strengths And Weaknesses:**

**Strengths**

1. The motivation of the proposed algorithms is clearly presented.

2. The paper provides improved convergence guarantees compared with prior work, particularly in terms of transient time.

3. The paper includes extensive numerical experiments to demonstrate the effectiveness of the proposed methods and their behavior under different parameter settings.

**Weaknesses**

The writing is not sufficiently rigorous in several places and may be confusing. See **Questions**.

Minor Comments

1. The font size of the caption of Table 1 is too small.

2. Some citations should use \citet rather than \cite. For example, in the first paragraph of Related Work, most sentences would be more appropriate with \citet.

---

> ### Author Rebuttal · Authors · 2026-03-30
>
> > **Minor Comments on font size and citation.**
> >
>
> We thank the reviewer for pointing this out. We will address the font size and citation formatting issues in the revised manuscript.
>
> ---
> > **Q1:  Clarification on the honest subgraph connectivity and the communication matrix.**
> >
>
> Thank you for raising this important concern. The equivalence in Eq. (4) relies on the assumption that the subgraph induced by the honest nodes remains connected. Theoretically, if this subgraph were disconnected into isolated cliques, reaching global consensus among honest nodes would be impossible. Under such conditions, the spectral gap of the virtual mixing matrix would strictly be zero, invalidating the contraction properties required for Eq. (4).
>
> In real-world decentralized applications (e.g., P2P architectures or sensor networks), communication topologies network are intentionally designed with high topological redundancy and algebraic connectivity (e.g., $k$-connected networks) to ensure robust fault tolerance. Accordingly, as long as the fraction of Byzantine nodes is bounded, conceptually removing these attackers and their incident edges will not destroy the global connectivity of the remaining honest network. Therefore, assuming a connected honest subgraph is both practically reasonable and a standard prerequisite in Byzantine-resilient literature.
>
> ---
> > **Q2: Clarification on the identical notations $\widehat{B}$ and $\widehat{D}$ in Eq. (4).**
> >
>
> Thank you for your comment. Although $\widehat{B}$ and $\widehat{D}$ share identical consensus properties, they are intentionally defined as distinct matrices to guarantee the broad generality of the unified framework. Specifically, the framework utilizes the transformations $B = \sqrt{\eta \rho} \widehat{B}$, and $C = {I} - \eta\rho \widehat{D}^2$. If we strictly enforce $\widehat{B} = \widehat{D}$, the matrix $C$ would be rigidly coupled to $B$ (imposing a fixed structural dependency like $C = {I} - B^2$). This restriction would prevent the framework from subsuming broader classes of decentralized algorithms. For instance, Gradient Tracking (GT) algorithms fundamentally require decoupled parameterizations of $B=I-W$ and $C=W^2$. By keeping $\widehat{B}$ and $\widehat{D}$ notationally separate, the framework allows the effective mixing matrices $B$ and $C$ to be parameterized and selected independently.
>
> ---
> > **Q3: Clarification on the transient complexity of BRED vs. BRED-M.**
> >
>
> Thank you for raising this important point. The statement on page 2 regarding the recovery of Exact Diffusion's transient complexity specifically refers to the **BRED** algorithm (without momentum). When $\delta=0$, BRED exactly recovers the $\mathcal{O}(n^3(1-\lambda)^{-2})$ complexity of standard Exact Diffusion. In contrast, the $\mathcal{O}(\max\\{n^2(1-\lambda)^{-1}, n(1-\lambda)^{-2}\\}) \le \mathcal{O}(n^2(1-\lambda)^{-2})$ bound in Table 1 applies to **BRED-M**. By incorporating momentum, BRED-M explicitly achieves improvement in transient time over the **BRED**.
>
> The mechanism behind this improvement can be traced through the convergence rate. Without momentum, the variance term in the convergence bound is $\mathcal{O}(\sigma^2)$. Momentum suppresses this to $\mathcal{O}((1-\beta)\sigma^2)$, where $\beta$ is the momentum coefficient. By further setting $1-\beta = L_{\beta}\eta$, the variance contribution can be refined to $\mathcal{O}(\eta\sigma^2)$. Under the stepsize choice $\eta = \mathcal{O}(1/\sqrt{n_h K})$, this leads to a improved convergence guarantee. Moreover, since the transient time is closely related to the convergence rate, the improved bound also implies a shorter transient phase.

---

> > ### Author Rebuttal · Reviewer_KyuG · 2026-04-03
> >
> > Thank you for your detailed response addressing my main concerns. I will raise my score to 4. However, I still suggest that the authors make the presentation more rigorous in the revision, e.g., the honest subgraph connectivity assumption, and add more background for readers who are less familiar with the Byzantine setting.

---

> > > ### Author Response · Authors · 2026-04-04
> > >
> > > We sincerely thank the reviewer for the valuable suggestions. We will add more related works and carefully address these points in the revision.

---

### Official Review · Reviewer_YPag · 2026-03-12

**Soundness:** 3
**Presentation:** 3
**Significance:** 3
**Originality:** 2
**Overall Recommendation:** 4
**Confidence:** 3

**Summary:**

This paper studies decentralized learning over sparse networks in the presence of Byzantine attacks. The paper argues that existing DSGD-based robust methods suffer from high transient complexity, while directly combining bias-correction techniques such as Exact Diffusion with robust aggregation may lead to error accumulation in the dual variables.
To address this issue, the authors propose a scaled dual ascent (SDA) mechanism within an augmented Lagrangian framework. The key idea is to introduce a scaling factor in the dual updates so as to control the growth of the dual variables and mitigate error accumulation. Based on this idea, the paper develops the Byzantine-Resilient Exact Diffusion algorithm (BRED) and its momentum variant BRED-M.
On the theory side, the authors show that BRED achieves linear speedup together with a bounded asymptotic error, and that BRED-M further improves the transient complexity while matching the claimed optimal asymptotic error lower bound. The empirical results also suggest that the proposed methods perform favorably across several benchmark datasets and network topologies.

**Compliance With Llm Reviewing Policy:**

Affirmed.

**Final Justification:**

Please refer to my Rebuttal Acknowledgement.

**Key Questions For Authors:**

1. The transient-time order in Corollary 2 seems inconsistent with the summary in Remark 5. Specifically, Corollary 2 gives $\tau = O\biggl(\max\biggl(\frac{n_h^2}{1-\lambda}, \frac{n_h}{(1-\lambda)^2}, \frac{\delta^2 n_h^3}{(1-\lambda)^4}\biggr)\biggr)$, while Remark 5 summarizes the topology dependence as $O\left(\frac{n_h^2}{(1-\lambda)^2}\right)$. Is the latter intended as a simplified upper bound of the former? If so, it may be better to present these two statements in a more consistent way.

2. The last inequality in (294) may be problematic. The proof appears to bound $\mathbb{E}\Vert\sum_{t=0}^{k-1} \xi_t \Vert^2$ by $\sum_{t=0}^{k-1}\mathbb{E}\Vert\bar{\xi}_t\Vert^2$, which would in general ignore the cross terms. Please check this step carefully and clarify whether an additional argument is needed or whether this part should be corrected.

3. It would be helpful if the authors could provide more guidance on how to choose the scaling parameter $q$ from a theoretical perspective. Based on both the analysis and the experiments, the best choice of $q$ seems to depend on several factors, including the network topology, the attack intensity, and the degree of data heterogeneity. In the current version, however, it remains unclear whether there is any principled theoretical recommendation for selecting $q$, beyond empirical tuning. In addition, the statement at the bottom of page 18 that “larger $q$ values achieve better performance” seems inconsistent with the plots and the subsequent discussion.

4. I would appreciate some clarification regarding the robustness assumption in Definition 2. In particular, how restrictive is this condition in practice, and is it expected to be easy to satisfy under the decentralized setting considered in this paper? The paper mentions several robust aggregation methods, but it is still unclear whether these methods can indeed satisfy Definition 2 under the specific conditions studied here.

**Limitations:**

Yes.

**Strengths And Weaknesses:**

**Strengths:**

By incorporating the Polyak momentum mechanism, the BRED-M algorithm is shown to achieve the theoretical lower bound of asymptotic error, $\Omega(\delta\zeta^2)$, under specific conditions. This result helps establish a strong theoretical property of the proposed framework and suggests that, under the stated assumptions, the method can eliminate the influence of stochastic noise on the steady-state error.

The paper provides comprehensive numerical experiments to evaluate the proposed algorithms across diverse network topologies and attack scenarios. These empirical results provide solid support for the theoretical claims.

**Weakness:**

The proposed method appears to be developed by incorporating a dual scaling mechanism into an existing decentralized primal-dual framework, together with a momentum-based extension for further performance improvement. While the resulting method is reasonable and practically motivated, the overall novelty appears somewhat limited, and the methodological contribution could be further strengthened.

Moreover, some parts of the paper would benefit from clearer explanation, particularly regarding the theoretical presentation and the interpretation of the experimental results; see the questions for details.

---

> ### Author Rebuttal · Authors · 2026-03-30
>
> > **Q1:  Transient time simplification.**
> >
>
> We thank the reviewer for raising this point. The overall transient time in Corollary 2 consists of both Byzantine-irrelevant and Byzantine-relevant components. The purpose of Remark 5 is to isolate the Byzantine-irrelevant part so as to clearly demonstrate the acceleration effect of momentum. To this end, we apply the standard upper bound $\max\\{ n_h^2(1-\lambda)^{-1}, n_h(1-\lambda)^{-2} \\} \le n_h^2(1-\lambda)^{-2}$, which yields a unified $\mathcal{O}(n_h^2(1-\lambda)^{-2})$ expression. This simplification allows a direct comparison of the topological dependence between the momentum and non-momentum variants (Corollary 2 vs. Corollary 1).
>
> ---
> > **Q2:   Problem of inequality (294).**
> >
>
> We sincerely thank the reviewer for the careful verification. The reviewer is correct that the cross terms in Eq. (294) cannot be dropped directly, and we appreciate the opportunity to clarify this point. We will fix this derivation step in the appendix. Importantly, as we detail below, the correction preserves all of our main results.
>
> Specifically, we apply the Cauchy–Schwarz inequality to obtain the proper bound: $\mathbb{E}[\\| \sum_{t=0}^{k-1} \bar{\xi}^t \\|^2] \leq k \sum_{t=0}^{k-1} \mathbb{E}[\\| \bar{\xi}^t \\|^2]$. Summing over $K$ iterations then yields $\sum_{k=1}^{K-1} k \sum_{t=0}^{k-1} \mathbb{E}[\\| \bar{\xi}^t \\|^2]\le \frac{K^2}{2} \sum_{k=0}^{K-1} \mathbb{E}[\\| \bar{\xi}^k \\|^2]$, which leads to the asymptotic error $\varepsilon_{K \to \infty} = \mathcal{O}\left(K^2 \delta (\sigma^2 + \zeta^2)\right)$.
>
> We would like to emphasize that this corrected bound fully preserves the main conclusion: without the proposed scaled dual ascent (i.e., $q=1$), the adversarial error grows with the number of training iterations $K$. This theoretical prediction is also well supported by our experimental observations. Specifically, setting $q=1$ leads to significant performance degradation under attack, which is consistent with the quadratic accumulation in $K$.
>
> ---
> > **Q3: Theoretical guidance for selecting $q$.**
> >
>
> Thank you for your comment. Both Theorems 1 and 2 provide principled theoretical guidance for selecting the scaling parameter $q$. Using Theorem 1 as an example, the optimal $q$ requires balancing the consensus error against the adversarial perturbation across several factors:
> - **Network Sparsity ($p = 1-\lambda$):** In sparser networks, $p$ becomes smaller. To minimize the dominant consensus error term $\mathcal{O}((pq)^{-1})$, a **larger** $q$ is required.
>
> - **Attack Intensity ($\kappa$):** As the number of attackers increases, the adversarial error term $\mathcal{O}(\kappa(\sigma^2+\zeta^2)(1-q)^{-2})$ dominates. To suppress this perturbation, a **smaller** $q$ is preferred.
>
> - **Data Heterogeneity ($\zeta^2$):** Higher heterogeneity introduces the terms $\mathcal{O}(\zeta^2 q^{-1})$ and $\mathcal{O}(\kappa \zeta^2 (1-q)^{-2})$. If the fraction of attackers ($\kappa$) is relatively small, the first term dominates, suggesting a **larger** $q$ to mitigate the influence of non-IID data.
>
> Regarding the experimental validation: the ablation on page 18 uses a relatively simple setting with sythetic data, making the sensitivity to $q$ less visually distinguishable. To provide clearer empirical evidence, we conduct additional $q$-sensitivity experiments on MNIST (ALIE attack, $k \in \{4,8,12\}$, $\delta \in \{1/8, 1/4\}$, Trimmed Mean and Median). The results are reported in Figures 1–4 of link https://anonymous.4open.science/r/ICML_fig-EBAD/fig.md.
>
> The trends match our theory: fixing $\delta$ and increasing $k$ (left → right in each figure), $q^\star$ shifts upward (sparser networks need larger $q$); fixing $k$ and increasing $\delta$ (Figure 1 vs. 2, or 3 vs. 4), $q^\star$ shifts downward (stronger attacks favor smaller $q$).
>
> ---
> > **Q4: More clarification for the Definition 2.**
> >
>
> Thank you for your concern. Definition 2 ($(\delta,\kappa)$-robustness) is one of the most widely adopted characterization of robust aggregators in the Byzantine-robust learning literature, and most commonly used aggregation rules provably satisfy it. Concretely, Krum satisfies Definition 2 with $\kappa = 1 + \delta/(1-2\delta)$, and Median with $\kappa = (1 + \delta/(1-2\delta))^2$. Moreover, when these aggregators are combined with pre-aggregation techniques such as NNM [1], the robustness parameter can be further tightened; for example, Krum + NNM achieves $\kappa = \mathcal{O}(\delta)$. We note that Definition 2 naturally extends the robustness characterization from the distributed (server-based) setting in [1] to our decentralized setting, where each client independently applies a robust aggregator to the updates received from its neighbors.
>
> [1] Allouah, Youssef, et al. "Fixing by mixing: A recipe for optimal byzantine ml under heterogeneity." *International Conference on Artificial Intelligence and Statistics*. PMLR, 2023.

---

> > ### Author Rebuttal · Reviewer_YPag · 2026-04-03
> >
> > Thanks to the authors for the detailed rebuttal and clarifications.
> >
> > The response has addressed most of my concerns and improved my understanding of the paper. However, based on the weaknesses I raised, I still believe the paper is better characterized as an incremental contribution rather than a fundamentally new one.
> >
> > For this reason, I will maintain my current score.

---

> > > ### Author Response · Authors · 2026-04-04
> > >
> > > We thank the reviewer for the positive feedback and for acknowledging that our rebuttal addressed most of their concerns. Regarding the assessment of novelty, we respectfully emphasize that while our approach builds upon existing frameworks, the resulting contributions offer substantial and non-trivial value to the field. Our contributions mainly lie in two areas:
> > >
> > > **Methodological:** We demonstrate that directly combining Byzantine-resilient learning with bias-correcting methods is highly non-trivial and, in fact, infeasible without the careful, specific design introduced in our work. Specifically, we identify a previously overlooked error-accumulation issue in naive combinations and resolve it through our proposed Scaled Dual Ascent (SDA).
> > >
> > > **Theoretical:** We establish improved transient complexity and demonstrate the mathematical benefits of incorporating momentum in this setting, ultimately proving that our approach achieves order-optimal Byzantine resilience.
> > >
> > > We believe these insights provide a meaningful step forward. We will gladly expand upon this discussion in the final version to better highlight these non-trivial aspects.
> > >
> > > Again, we sincerely thank the reviewer for their time and constructive suggestions.

---

### Official Review · Reviewer_B3eG · 2026-03-13

**Soundness:** 3
**Presentation:** 3
**Significance:** 3
**Originality:** 3
**Overall Recommendation:** 5
**Confidence:** 5

**Summary:**

This paper studies the trade-off between Byzantine robustness and convergence efficiency over sparse communication networks. The core motivation is that existing Byzantine-resilient distributed gradient descent methods are highly sensitive to the spectral gap on sparse graphs, whereas bias-correction methods such as ED, EXTRA, and GT may inadvertently absorb attack-induced bias under Byzantine adversaries and thus become vulnerable to corrupted information. From an augmented-Lagrangian primal–dual perspective, the paper argues that historical errors accumulate in the dual variables, and to mitigate this issue, it introduces a scaled dual ascent mechanism, derives a unified update framework, and instantiates it with Exact Diffusion to obtain BRED and its momentum variant BRED-M. The paper claims two main theoretical advantages: compared with BR-DSGD, the proposed method achieves improved transient complexity, and for BRED-M, the asymptotic error is claimed to match the known lower-bound order $\Omega(\delta \zeta^2)$. Empirically, the paper evaluates the proposed approach on MNIST and CIFAR-10 under multiple communication topologies, Byzantine ratios, attack models, and robust aggregation rules, and the results suggest that, on average, the proposed method achieves higher accuracy than competing frameworks across a range of Byzantine settings. The paper also includes a sensitivity analysis of the scaling parameter q and provides practical guidance on its selection.

**Compliance With Llm Reviewing Policy:**

Affirmed.

**Final Justification:**

My concerns  have been addressed, I consider to raise my score accordingly.

**Key Questions For Authors:**

1. The paper appears to contain several potentially unclear or problematic statements. For example, Section 3.1 mentions “reaching consensus across all participants.” Does this consensus refer to all clients, or only to the honest clients? In addition, how is the bound $O(n_h^2/(1−\lambda)^2)$ stated in Remark 5 derived from Corollary 2?
2. Could the authors provide a more concise summary table covering the average performance across all attack/aggregator combinations reported in the paper, rather than presenting in the main text only the ALIE + Median setting?
3. For the additional experiments in the appendix, such as the local-epoch extension study in B.4.3, how is the degree of heterogeneity specified? Is it the same as in the other experiments, e.g., α=0.3?

**Limitations:**

yes

**Strengths And Weaknesses:**

Soundness:  The authors first identify the weakness of existing DSGD-based methods in terms of spectral-gap dependence, while noting that bias-correction methods such as ED, GT, and EXTRA are more favorable on sparse networks, thereby motivating their integration. They then further analyze the additional vulnerability of bias-correction methods under Byzantine attacks, and on this basis introduce SDA and instantiate it as BRED and BRED-M, yielding a coherent technical development. The assumptions used are generally reasonable, and the paper provides both convergence analysis and extensive experimental evidence, which together make the approach convincing.

The paper also has several limitations. Although the authors consider four attack models and four defense mechanisms in the experiments, the main text only reports one representative setting, namely ALIE + Median, while the other cases are relegated to appendix line plots. As a result, the main paper lacks a global summary, such as an average-performance comparison across different frameworks and datasets, which weakens the completeness of experimental presentation. In addition, the theoretical part does not prove that $(\delta,\kappa)$-robustness holds, but instead assumes it directly, which somewhat weakens the rigor of the theory.

Presentation: There remain several issues that could be improved. For example, in Appendix lines 983–989, the statement “In sparse networks (k=4), convergence is more sensitive to the choice of q, with larger q values achieving better performance due to enhanced global information exchange” is not fully consistent with the later sentence “These observations align with the theoretical predictions that sparser networks necessitate smaller q to maintain convergence guarantees,” which may confuse readers. In addition, the main text consistently uses the “BR”-prefixed naming convention, whereas the appendix switches back to the bare framework names; similarly, the proposed method is placed at the end of the method list in the main text but appears in the middle in the appendix, which affects readability. While such issues may not directly cause theoretical errors, they do raise the barrier to understanding and reduce the overall clarity of presentation.

Significance: The problem addressed in this paper is meaningful and relevant. In decentralized federated systems, sparse communication networks are often more realistic than fully connected ones, so it is important to study robustness and spectral-gap dependence within a unified framework. The paper clearly argues that if Byzantine-resilient methods only work well under fully connected or otherwise idealized topologies, their practical value would be limited.

Originality:  Rather than simply combining an existing decentralized framework with variance-reduction or bias-correction techniques, it explicitly analyzes why such techniques can fail in Byzantine settings due to the accumulation of historical errors, and argues that this issue should be addressed by introducing explicit contraction into the dual update through SDA.

---

> ### Author Rebuttal · Authors · 2026-03-30
>
> > **Q1: Clarification on the consensus definition and the derivation in Remark 5.**
> >
>
> We thank the reviewer for the careful reading and the opportunity to clarify.
>
> **1. Definition of "Consensus":** In Section 3.1, "consensus" strictly refers to the agreement among the *honest* clients only. Malicious participants are inherently excluded from this set because they do not follow the prescribed protocol. We will update the manuscript to explicitly state this and remove any ambiguity.
>
> **2. Derivation in Remark 5:** The bound $O(n_h^2 / (1-\lambda)^2)$ characterizes the "Byzantine-irrelevant" transient time, which is utilized to demonstrate the improvement achieved by introducing momentum. This bound is derived directly from the Byzantine-irrelevant terms in Corollary 2. Specifically, it follows from the inequality: $\max\\{ n_h^2(1-\lambda)^{-1}, n_h(1-\lambda)^{-2} \\} \leq n_h^2(1-\lambda)^{-2}$.
>
> ---
> > **Q2: Request for summary tables of performance across attack/aggregator combinations.**
> >
>
> Thank you for your concern. While the comprehensive training curves for all combinations of attack and robust aggregator combinations remain available in the experimental appendix, we agree that adding concise summary tables to the main text will significantly strengthen the paper. Accordingly, we  will incorporate two such summary tables into the manuscript. The complete per-setting breakdowns underlying Tables 1 and 2 are provided in our extended supplementary tables: https://anonymous.4open.science/r/ICML_Table-4505/table.md.  For the reviewer's convenience, we include them below:
>
> **1. Worst-Case Robustness Across Attacks (Table 1):** This table fixes the robust aggregation rule (e.g., Median) and reports the *minimum* test accuracy across four distinct attacks (ALIE, FOE, LF, SF). This highlights the worst-case resilience of each algorithm against varying adversarial strategies.
>
> | Algorithm | setting 1 | setting 2 | setting 3 | setting 4 | setting 5 | setting 6 | setting 7 | setting 8 |
> |-----------|---------|---------|---------|---------|---------|---------|---------|---------|
> | BR-DSGD | 0.8812 | 0.4469 | 0.6855 | 0.4906 | 0.8875 | 0.7094 | 0.8845 | 0.6596 |
> | BR-DSGD-M | 0.8794 | 0.3997 | 0.6695 | 0.4535 | 0.8844 | 0.7040 | 0.8826 | 0.6534 |
> | BRGT | 0.4358 | 0.2626 | 0.2778 | 0.3107 | 0.5237 | 0.2891 | 0.5129 | 0.0961 |
> | BRGT-M | 0.4307 | 0.3655 | 0.2516 | 0.2839 | 0.3244 | 0.1900 | 0.3373 | 0.0985 |
> | BR-EXTRA | 0.4611 | 0.4121 | 0.0982 | 0.1003 | 0.4368 | 0.0928 | 0.4310 | 0.4726 |
> | BR-EXTRA-M | 0.7711 | 0.6457 | 0.0951 | 0.1006 | 0.7323 | 0.0940 | 0.7321 | 0.6261 |
> | BRED | 0.8944 | 0.8188 | 0.8492 | 0.5769 | 0.8975 | 0.8555 | 0.8943 | 0.8323 |
> | BRED-M | 0.8958 | 0.7284 | 0.8489 | 0.5992 | 0.8989 | 0.8716 | 0.8983 | 0.8559 |
>
> Table 1: Worst-Case Test Accuracy Across Attacks (Fixed Aggregator = Median)
>
> **2. Average Performance Across Aggregators (Table 2):** This table fixes the attack (e.g., ALIE) and reports the *mean* test accuracy across four different robust aggregation rules (IOS, Krum, Median, Trimmed Mean). This demonstrates the generalizability and average-case performance of the algorithms when the defense mechanism changes.
>
> | Algorithm | setting 1 | setting 2 | setting 3 | setting 4 | setting 5 | setting 6 | setting 7 | setting 8 |
> |-----------|---------|---------|---------|---------|---------|---------|---------|---------|
> | BR-DSGD | 0.8686 | 0.7602 | 0.6621 | 0.5994 | 0.8721 | 0.7425 | 0.8924 | 0.7848 |
> | BR-DSGD-M | 0.8204 | 0.7475 | 0.6640 | 0.5999 | 0.8680 | 0.7435 | 0.8918 | 0.7819 |
> | BRGT | 0.6459 | 0.4137 | 0.5599 | 0.3866 | 0.6967 | 0.6144 | 0.7082 | 0.6172 |
> | BRGT-M | 0.4027 | 0.4638 | 0.2700 | 0.4383 | 0.2971 | 0.3642 | 0.6620 | 0.6603 |
> | BR-EXTRA | 0.5254 | 0.5340 | 0.5213 | 0.4301 | 0.5659 | 0.5439 | 0.6044 | 0.5648 |
> | BR-EXTRA-M | 0.7518 | 0.6717 | 0.6811 | 0.5732 | 0.7204 | 0.7618 | 0.7868 | 0.7745 |
> | BRED | 0.8717 | 0.8720 | 0.8185 | 0.7606 | 0.8892 | 0.8450 | 0.8989 | 0.8661 |
> | BRED-M | 0.8467 | 0.8455 | 0.8165 | 0.7543 | 0.8790 | 0.8647 | 0.9006 | 0.8705 |
>
> Table 2: Average Test Accuracy Across Robust Aggregators (Fixed Attack = ALIE)
>
>
>
> ---
> > **Q3: Question on the experiment setting.**
>
> Yes, for the ablation studies, we use the same heterogeneity setting as in the main experiments. All other hyperparameters for the ablation studies are kept unchanged, except for the factor being varied.
>
> ---
> > **Weakness: The holdness of $(\delta,\kappa)$-robustness.**
>
> We thank the reviewer for raising this concern. We address the generality of the $(\delta,\kappa)$-robustness assumption in detail in our response to Q4 of Reviewer YPag. In short, this is the one of the most widely adopted characterizations in the Byzantine-robust learning literature, and most commonly used aggregation rules satisfy it.

---

> > ### Author Rebuttal · Reviewer_B3eG · 2026-04-03
> >
> > Thank you for the response, my concerns have been addressed, I consider to raise my score accordingly.

---

> > > ### Author Response · Authors · 2026-04-04
> > >
> > > Thanks for the recognition and for raising the score. We appreciate your time and effort in reviewing our paper.

---

### Decision · Program_Chairs · 2026-04-30

**Decision:**

Accept (regular)

**Comment:**

This paper studies Byzantine-robust decentralized learning over sparse networks and proposes a scaled dual ascent (SDA) framework with Byzantine-robust Exact Diffusion instantiations (BRED and BRED-M). The main concerns were addressed during the rebuttal phase, and there is now clear consensus among reviewers in favor of acceptance. The work is technically solid, with strong theoretical contributions (notably the identification of error accumulation in bias-correction under Byzantine settings, the SDA correction mechanism, and improved transient complexity guarantees) supported by extensive empirical evaluation. While some reviewers note that the methodological novelty is moderate and largely builds on existing primal-dual and gradient-tracking ideas, they also agree that the technical analysis is careful and the resulting improvements are meaningful.

I therefore recommend to accept this paper. I ask the authors to address the remaining presentation-related suggestions, and to cite and discuss the following recent work on Byzantine-robust gossip:
- Gaucher et al. Byzantine-Robust Gossip: Insights from a Dual Approach. TMLR 2025
- Gaucher et al. Unified Breakdown Analysis for Byzantine Robust Gossip. ICML 2025